# Provable In-Context Vector Arithmetic via Retrieving Task Concepts

**Dake Bu**[† 1 2]  **Wei Huang**[2]  **Andi Han**[2 3]  **Atsushi Nitanda**[4 5 6]  **Qingfu Zhang**[1]  **Hau-San Wong**[1]  **Taiji Suzuki**[7 2]

## Abstract

In-context learning (ICL) has garnered significant attention for its ability to grasp functions/tasks from demonstrations. Recent studies suggest the presence of a latent **task/function vector** in LLMs during ICL. Merullo et al. (2024) showed that LLMs leverage this vector alongside the residual stream for Word2Vec-like vector arithmetic, solving factual-recall ICL tasks. Additionally, recent work empirically highlighted the key role of Question-Answer data in enhancing factual-recall capabilities. Despite these insights, a theoretical explanation remains elusive. To move one step forward, we propose a theoretical framework building on empirically grounded *hierarchical* concept modeling. We develop an optimization theory, showing how nonlinear residual transformers trained via gradient descent on cross-entropy loss perform factual-recall ICL tasks via vector arithmetic. We prove 0-1 loss convergence and show the strong generalization, including robustness to concept recombination and distribution shifts. These results elucidate the advantages of transformers over static embedding predecessors. Empirical simulations corroborate our theoretical insights.

## 1. Introduction

Transformer-based Large Language Models (LLMs) (Vaswani et al., 2017) have ushered in a new era of foundation models. A growing academic perspective highlights that the core strength of LLMs lies in their remarkable In-Context Learning (ICL) capability (Lu et al., 2024), enabling them to infer underlying tasks or functions from input demonstration pairs (Minegishi et al., 2025b)—akin to the concept of *function references* in programming (Bernays, 1936). For example, given the prompt `Japan Tokyo France Paris China`, the expected output "Beijing" reflects the underlying function "Country's Capital" applied to the final query.

**Role of Task Vector**. A growing body of research seeks to uncover the internal mechanisms underpinning ICL, shedding light on how LLMs retrieve function/task from demonstration. Recent studies employing advanced probing techniques have identified the emergence of a **task vector** (Hendel et al., 2023; Merullo et al., 2024; Yang et al., 2025) (or **function vector** (Todd et al., 2024; Kahardipraja et al., 2025)) within the latent representations of LLMs during ICL deduction. This vector appears around the 15-19th layers in models like GPT-J-6B (Wang & Komatsuzaki, 2021). Formally, Hendel et al. (2023); Merullo et al. (2024) propose that, given the prompt $\mathbf{T} = [\mathbf{x}_1, f(\mathbf{x}_1), \mathbf{x}_2, f(\mathbf{x}_2), \cdots, \mathbf{x}_{\text{query}}]$, the LLM $\boldsymbol{\theta}$ constructs the **task vector** in the earlier layers, denoted as $\mathbf{a}_{\boldsymbol{\theta}}^f(\mathbf{T})$, based on its understanding of the task/function message in $\mathbf{T}$. Furthermore, Merullo et al. (2024) revealed that, certain factual-recall ICL tasks indeed correspond to some latent representation $f(\mathbf{x}_{\text{query}}) = \mathbf{a}_{\boldsymbol{\theta}}^f(\mathbf{T}) + \mathbf{b}_{\boldsymbol{\theta}}^{\text{query}}(\mathbf{x}_{\text{query}})$, where $\mathbf{b}_{\boldsymbol{\theta}}^{\text{query}}(\mathbf{x}_{\text{query}})$ is the **query-encoded residual stream** in deeper layers. That is, the model $\boldsymbol{\theta}$ can execute the ICL task through a simple **vector addition**:

$$
\begin{aligned}
p_{\boldsymbol{\theta}}(f(\mathbf{x}_{\text{query}}) = \cdot \mid \mathbf{T}) = \int p_{\boldsymbol{\theta}}(\mathbf{a}_{\boldsymbol{\theta}} + \mathbf{b}_{\boldsymbol{\theta}}^{\text{query}} \mid \mathbf{a}_{\boldsymbol{\theta}}, \mathbf{T}) \\
\cdot p_{\boldsymbol{\theta}}(\mathbf{a}_{\boldsymbol{\theta}} \mid \mathbf{T}) d(\mathbf{a}_{\boldsymbol{\theta}}),
\end{aligned} \tag{1}
$$

where $p_{\boldsymbol{\theta}}(\mathbf{a}_{\boldsymbol{\theta}} \mid \mathbf{T})$ denotes the model's confidence in recognizing $\mathbf{a}_{\boldsymbol{\theta}}$ as the task vector $\mathbf{a}_{\boldsymbol{\theta}}^f(\mathbf{T})$. The formulation in Eq. (1) highlights the pivotal role of residual streams. However, existing theories on ICL either overlook the residual term entirely (Tian et al., 2023; Kim & Suzuki, 2024; Bu et al., 2024a) or handle it in an unnatural manner (Nichani et al., 2024). Additionally, while some studies suggest that Question-Answer (QA) training data are crucial for enabling LLMs to retrieve factual knowledge (Allen-Zhu &

---

[†]Work completed during internship at RIKEN. [1]Department of Computer Science, City University of Hong Kong, Hong Kong SAR [2]Center for Advanced Intelligence Project, RIKEN, Japan [3]School of Mathematics and Statistics, The University of Sydney, Australia [4]Institute of High Performance Computing (IHPC), Agency for Science, Technology and Research (A⋆STAR), Singapore [5]Centre for Frontier AI Research (CFAR), Agency for Science, Technology and Research (A⋆STAR), Singapore [6]College of Computing and Data Science, Nanyang Technological University, Singapore [7]Department of Mathematical Informatics, The University of Tokyo, Japan. Correspondence to: Wei Huang <wei.huang.vr@riken.jp>, Hau-San Wong <cshswong@cityu.edu.hk>.

*Proceedings of the 42ⁿᵈ International Conference on Machine Learning*, Vancouver, Canada. PMLR 267, 2025. Copyright 2025 by the author(s).

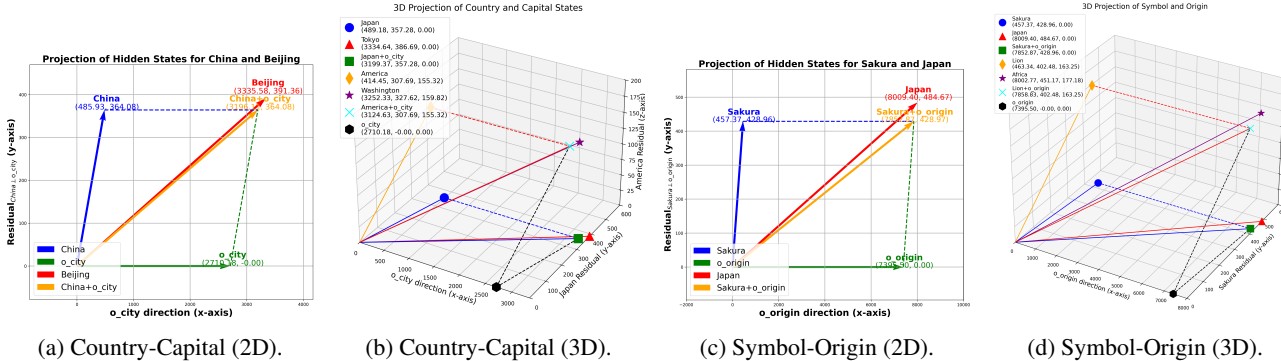

(a) Country-Capital (2D).  (b) Country-Capital (3D).  (c) Symbol-Origin (2D).  (d) Symbol-Origin (3D).

*Figure 1.* Visualization of task vector and word embedding. The 22nd layer's vector embeddings of GPT2-medium in the 2-D and 3-D projection spaces are shown, where the task vector ('o_task') is extracted from some internal layer, following Merullo et al. (2024). For the task of retrieving a country's capital (a-b) and the task of retrieving a symbolic creature's origin (c-d), the latent embeddings of the LLM demonstrate the approximate relationship 'x + o_task = y', with 'x' containing little components aligned with 'o_task'.

Li, 2024; Zhang et al., 2025a), there is currently no theoretical framework to substantiate this claim, nor an explanation of how it facilitates factual-recall ICL. These gaps naturally raise the following research question.

---
**Research Questions 1**

How does a non-linear residual transformer, trained via gradient methods with a realistic cross-entropy loss on QA data, naturally perform factual-recall ICL in the vector arithmetic style described by Eq. (1)?

---

**Analogy to Word2Vec.** As noted by (Merullo et al., 2024), the vector arithmetic in Eq. (1) mirrors that of Word2Vec, a static word embedding model (Mikolov et al., 2013). For instance, the representation arithmetic "France − Paris + Poland = Warsaw" can be interpreted as "France − Paris" representing the task vector $\mathbf{a}_{\boldsymbol{\theta}}^f$, which performs the function "get_capital($\cdot$)," while representations of countries like "Poland" or "Japan" act as $\mathbf{b}_{\boldsymbol{\theta}}^{query}$.

Recent work by Wibisono & Wang (2023) examined the relationship between one-layer bidirectional attention optimized via Masked Language Model loss through reparameterization and the Word2Vec method. However, they show that such BERT-like models function as an inexact approximation of the Word2Vec method, exhibiting non-trivial errors. Their analysis failed to delve into the the optimization process, recognize the task vector mechanisms, or clarify the comparative advantages of transformer over Word2Vec. This raises the following question for further investigation.

---
**Research Question 2**

In the context of Eq.(1), what are the primary strengths of transformers over their Word2Vec predecessors?

---

To address these questions theoretically, we first observe that the latent geometry of LLMs exhibits intriguing prop-erties, as illustrated in Figure 1. In factual-recall tasks, answer/label vectors $\boldsymbol{y}$ tend to align more strongly with the task vector $\mathbf{a}_{\boldsymbol{\theta}}^f$ than query word vectors $\boldsymbol{x}$, satisfying $\boldsymbol{x} + \mathbf{a}_{\boldsymbol{\theta}}^f \approx \boldsymbol{y}$. Here, $\mathbf{a}_{\boldsymbol{\theta}}^f$ can be interpreted as the representation of high-level task concepts (aligned with the x-axis in Figure 1), while the components of $\boldsymbol{x}$ orthogonal to $\mathbf{a}_{\boldsymbol{\theta}}^f$, represent low-level concepts (aligned with the y/z-axes in Figure 1). This interpretation is consistent with Park et al. (2025), which demonstrates that LLMs exhibit hierarchical linear concept geometry after generative pretraining, with word representations residing within polytopes. These insights inspire the modeling approaches in Section 2.1.

Our contributions are summarized as follows:

1. We develop an optimization theory demonstrating that transformers with nonlinear softmax attention, MLP, layer normalization, and residual connections—trained via Gradient Descent (GD) with cross-entropy loss—can effectively perform factual-recall ICL in a vector arithmetic manner, grounded in empirically motivated data modeling. Our analysis shows that the transformer retrieves the high-level task/function concept through attention-MLP, which, when combined with *any embedded query vector within the same high-level task concept*, yields the correct corresponding answer vector. Despite the inherent non-convexity of the learning problem, we establish the asymptotic behaviors of the optimization dynamics and prove the convergence of ICL test losses.

2. We demonstrate a clear learning scenario separation between training on QA data and on Word-Label demo-pair ICL data, both framed within hierarchical concept modeling. While prior theoretical works assume both training and testing on Word-Label pair ICL data (Zhang et al., 2024; Kim & Suzuki, 2024; Chen et al., 2024a; Bu et al., 2024a), we show that this approach

fails to retrieve the high-level task vector and instead leads to harmful memorization of low-level features. In contrast, we prove that training on QA data enables the model to effectively learn the task vector, achieving arbitrarily small error when tested on either Word-Label pair ICL or QA-ICL distributions.

3. We provide theoretical guarantees for compositional generalization in in-context learning, focusing on task vector emergence and OOD robustness. Specifically: (i) we show that transformers can regress task vectors directly from demonstration pairs, without requiring explicit query, and can apply these vectors compositionally at test time via arithmetic manipulation; (ii) we prove that models trained on structured QA data can generalize to unseen task prompts or dictionaries by leveraging conic combinations of learned high-level task vectors and novel orthogonal low-level and task-irrelevant features; and (iii) we establish the model's adaptability to distribution shifts in prompt content and length. These bonuses shows transformer's superiority over traditional methods.

*Remark* 1.1. **Humble Remark**. We humbly acknowledge that the scope of this study is confined to illustrating the merits of task vector arithmetic mechanisms observed in some single-token factual-recall ICL tasks, modeled based on prior empirical and theoretical observations. We do not claim this as a universal explanation for how LLMs handle complex multi-token or factual tasks. Nevertheless, to the best of our knowledge, this is the first theoretical work to **explicitly** consider the pivotal role of the residual stream and layer-wise normalization in ICL, contributing to the growing body of research on transformers and ICL. These nonlinearities, alongside softmax attention and cross-entropy loss, introduce significant challenges in analyzing optimization dynamics.

### 1.1. Related Work

**Task Vector Mechanism**. This line of research empirically validated the emergence of task vectors (Hendel et al., 2023; Todd et al., 2024; Liu et al., 2024; Yang et al., 2025). Especially, Merullo et al. (2024) revealed that large language models (LLMs) implement certain single-token factual-recall tasks through vector arithmetic. However, no existing study explains why and how gradient-update transformer models naturally implement this mechanism.

**Storage and Retrieval of Factual Knowledge**. Recent research has explored how LLMs perform factual recall or associative memory tasks (Nichani et al., 2025; Cabannes et al., 2024; Allen-Zhu & Li, 2025). Allen-Zhu & Li (2024); Zhang et al. (2025a) empirically show the importance of QA data to improve LLM's capability to retrieve factual knowledge. Additionally, Park et al. (2024); Jiang

et al. (2024b); Marks & Tegmark (2024) provided both theoretical and empirical evidence that LLMs tend to represent independent concepts and facts in a linear manner.

**In-Context Learning Theory.** Recent work has studied how transformers perform in-context learning across various settings (Zhang et al., 2024; Chen et al., 2024b; Kim & Suzuki, 2024; Nichani et al., 2024; Chen et al., 2024a). However, these analyses often overlook the crucial role of the residual stream and the phenomenon of in-context vector arithmetic. Moreover, they typically rely on unrealistic assumptions, such as linearized or combined QK attention (Kim & Suzuki, 2024; Nichani et al., 2024), or on impractical loss functions like squared or hinge loss (Chen et al., 2024a).

Additional Related Work could be found in Appendix A.

## 2. Problem Design and Intuition

**Notation**. We denote the Bernoulli distribution with parameter $p$, which represents a discrete distribution over $\{0, 1\}$, as $\mathrm{Ber}(p)$. We use $\mathbb{R}^d, \mathbb{R}^{d \times d}$ to denote vector and matrix space, and utilize $\mathbb{I} \in \mathbb{R}^{d \times d}$ to denote identity matrix. For two sequences $a_n$ and $b_n$, $a_n = O(b_n)$ indicates that there exist constants $C > 0$ and $N > 0$ such that $|a_n| \leq C|b_n|$ for all $n \geq N$. Similarly, $a_n = \Omega(b_n)$ means $b_n = O(a_n)$, and $a_n = \Theta(b_n)$ signifies both $a_n = O(b_n)$ and $a_n = \Omega(b_n)$. We define $\mathrm{span}(v_1, v_2, \ldots, v_k)$ as the linear space spanned by vectors $v_1, v_2, \ldots, v_k$, and $\mathrm{conic}(v_1, v_2, \ldots, v_k)$ as the conic hull, which includes all non-negative linear combinations of these vectors. We use $\| \cdot \|$ for the $l_2$ norm and $\| \cdot \|_F$ for the Frobenius norm.

### 2.1. Hierarchical Data Modeling

In this section, we present our data modeling based on the observations of task vector arithmetic in factual-recall ICL illustrated in Figure 1. We found the near-orthogonal properties in Figure 1 coincide with Park et al. (2025), which suggests that LLMs encode high- and low-level concepts in an approximately orthogonal manner. Specifically, we treat the task vector as a high-level concept representation, while orthogonal components represent task-specific low-level concepts. Details are delayed to Appendix C.

**High-Level Task Concept Vector**. There are $K$ high-level binary concepts, each denoted by $z_k \in \{0, 1\}$. We define the steering vectors $\boldsymbol{a}_k \in \mathbb{R}^d$ to represent $z_k = 1$ over $z_k = 0$, where all $K$ vectors are mutually orthogonal. This orthogonality ensures that for any distinct $k_1, k_2, k_3, k_4 \in [K]$, $\boldsymbol{a}_{k_1} - \boldsymbol{a}_{k_2} \perp \boldsymbol{a}_{k_3} - \boldsymbol{a}_{k_4}$, signifying the independence between concepts, consistent with findings in recent work (Park et al., 2024; 2025; Jiang et al., 2024b; Marconato et al., 2025; Liu et al., 2025). For example, the concept of "capital" can be considered independent of "gender".

**Low-Level Task-Specific Concept Vector**. For each high-

level concept $z_k, k \in [K]$, there is an associated low-level binary concept $w_k \in \{0, 1\}$, represented by two semantically opposite vectors $-\boldsymbol{b}_k$ and $\boldsymbol{b}_k \in \mathbb{R}^d$, where $w_k = 0$ corresponds to $-\boldsymbol{b}_k$ and $w_k = 1$ to $\boldsymbol{b}_k$. As noted in (Park et al., 2025) in the latent representation of LLM, the high-level task concept vectors are orthogonal to the low-level ones, i.e., $\boldsymbol{a}_{k_1} \perp \boldsymbol{b}_{k_2}$ for all $k_1, k_2 \in [K]$, and $\boldsymbol{b}_k$ are mutual-orthogonal.

**Word-Label Pair ICL Prompt Distribution** ($\mathcal{P}_{\mathbf{T}}$). The prompt $\mathbf{T} := [\mathbf{x}_1, \mathbf{y}_1, \cdots, \mathbf{x}_J, \mathbf{y}_J, \mathbf{x}_{J+1}] \in \mathbb{R}^{d \times (2J+1)}$ and label $\mathbf{y}_{J+1}$ are generated as follows. Each prompt's co-task concept, which is shared within the demo-pairs within the prompt, is sampled as $k_{\mathbf{T}} \sim \text{Unif}[K]$, with label indicator $y_{k_{\mathbf{T}},l} \sim \text{Unif}\{\pm 1\}$ and noise terms $\boldsymbol{\xi}_{l,\mathbf{x}}, \boldsymbol{\xi}_{l,\mathbf{y}} \sim \mathcal{N}(\mathbf{0}, \sigma_p^2 \mathbb{I})$. Word-label pairs in demonstrations and queries follow:

$$\begin{aligned} \mathbf{x}_l &:= \sum_{k \in \mathcal{X}_{\mathbf{T},l}} (x_a \cdot \boldsymbol{a}_k + y_{k,l} \cdot \boldsymbol{b}_k) + \boldsymbol{\xi}_{l,\mathbf{x}}, \\ \mathbf{y}_l &:= \sum_{k \in \mathcal{Y}_{\mathbf{T},l}} (\boldsymbol{a}_k + y_{k,l} \cdot \boldsymbol{b}_k) + \boldsymbol{\xi}_{l,\mathbf{y}}, \end{aligned} \quad (2)$$

for $l \in [J]$. To model word polysemy, we define $\mathcal{X}_{\mathbf{T},l}, \mathcal{Y}_{\mathbf{T},l} \subset [K]$ as the sets of latent concepts associated with $\mathbf{x}_l$ and $\mathbf{y}_l$, respectively; each set includes the shared co-task concept $k_{\mathbf{T}}$ along with possibly task-specific but contextually irrelevant concepts. The noise term $\boldsymbol{\xi}_l$ captures semantic variation, and the task anchor $x_a \cdot \boldsymbol{a}_k$ facilitates concept retrieval, as illustrated in Figure 1, with $x_a$ set to 0.1 (see Appendix C). The query word is defined as $\mathbf{x}_{J+1} = \sum_{k \in \mathcal{X}_{\mathbf{T},J+1}} (x_a \cdot \boldsymbol{a}_k + y_{k,J+1} \cdot \boldsymbol{b}_k) + \boldsymbol{\xi}_{J+1,\mathbf{x}}$, following Eq. (2). The expected label, by contrast, is given by $\mathbf{y}_{J+1} = \boldsymbol{a}_{k_{\mathbf{T}}} + y_{k_{\mathbf{T}},J+1} \cdot \boldsymbol{b}_{k_{\mathbf{T}}}$, since only semantics tied to the shared co-task concept $k_{\mathbf{T}}$ contribute to prediction under task $k_{\mathbf{T}}$. An illustrative example with $J = 2$ to show our modeling intuition is

$$\underset{\mathbf{x}_1}{\text{Japan}} \quad \underset{\mathbf{y}_1}{\text{Sakura}} \quad \underset{\mathbf{x}_2}{\text{France}} \quad \underset{\mathbf{y}_2}{\text{Rooster}} \quad \underset{\mathbf{x}_3}{\text{China}}$$

where the shared co-task concept $k_{\mathbf{T}}$ in this context is "National Symbol", and the expected label vector is $\mathbf{y}_3 =$ Panda—the symbol associated with China under the task $k_{\mathbf{T}}$. While the words "China" and "Panda" may carry semantics relevant to other tasks (e.g., country capitals or animal categories), only the "National Symbol" meaning is expected to contribute to $\mathbf{y}_3$, illustrating how the model selectively attends to task-relevant semantics in the presence of polysemy. If the prompt is modified to

$$\underset{\mathbf{x}_1}{\text{Japan}} \quad \underset{\mathbf{y}_1}{\text{Sakura}} \quad \underset{\mathbf{x}_2}{\text{France}} \quad \underset{\mathbf{y}_2'}{\text{Iris}} \quad \underset{\mathbf{x}_3}{\text{China}}$$

then the expected label vector becomes $\mathbf{y}_3' =$ Peony, as the shared co-task concept $k_{\mathbf{T}}'$ now corresponds to "National Flower". This illustrates how the shared co-concept steers correctness during inference.

**ICL vs. QA Training**. Indeed, training and testing solely on the ICL data is unrealistic, despite its popularity in theoretical studies (Zhang et al., 2024; Kim & Suzuki, 2024; Chen et al., 2024a; Bu et al., 2024a). A more practical approach involves generative pretraining or Question-Answer (QA) pretraining or fine-tuning (Allen-Zhu & Li, 2024; Zhang et al., 2025a). Notably, our concept data modeling **naturally arises** from generative pretraining (Park et al., 2024; Jiang et al., 2024b; Park et al., 2025). Furthermore, as highlighted by Allen-Zhu & Li (2024); Zhang et al. (2025a), QA data plays a crucial role in **enhancing** a transformer's ability to retrieve relevant factual knowledge. Additionally, Merullo et al. (2024) leverage QA data to extract task vectors. Motivated by these insights, we incorporate QA data into our training framework, which we formalize as follows.

**QA Sentence Distribution**[1] ($\mathcal{P}_{\text{QA}}$). We model our QA-type sentence $\mathbf{S} := [\mathbf{x}^{\text{QA}}, \mathbf{y}]$ as follows. Each sentence is associated with a task concept $k_{\mathbf{S}} \sim \text{Unif}[K]$. The word $\mathbf{x}$ and label vector $\mathbf{y}$ are constructed similarly to the word-label pair distribution $\mathcal{P}_{\mathbf{T}}$. The QA prefix $\mathbf{x}^{\text{QA}}$ is created by combining common tokens $\boldsymbol{\nu}_{n,1:M}$ ($M$ task-irrelevant common tokens) and the task vector $\boldsymbol{a}_{k_{\mathbf{S}}}$, which can appear anywhere before $\mathbf{x}$ as the last column of $\mathbf{x}^{\text{QA}}$. To capture semantic variability, noise vectors $\boldsymbol{\xi}_{1:M}, \boldsymbol{\xi}_{\mathbf{x}} \sim \mathcal{N}(\mathbf{0}, \sigma_p^2 \mathbb{I})$ are added to the common tokens. Formal details are provided in Appendix C. An illustrative example with $M = 5$ to show our modeling intuition on concept $k$ is:

$$\underset{\substack{\boldsymbol{\nu}_{n,1} \\ +\boldsymbol{\xi}_1}}{\text{What}} \quad \underset{\substack{\boldsymbol{\nu}_{n,2} \\ +\boldsymbol{\xi}_2}}{\text{is}} \quad \underset{\substack{\boldsymbol{\nu}_{n,3} \\ +\boldsymbol{\xi}_3}}{\text{the}} \quad \underset{\substack{\boldsymbol{a}_{k_{\mathbf{S}}} \\ +\boldsymbol{\xi}_4}}{\text{capital}} \quad \underset{\substack{\boldsymbol{\nu}_{n,5} \\ +\boldsymbol{\xi}_5}}{\text{of}} \quad \underset{\substack{x_a \cdot \boldsymbol{a}_{k_{\mathbf{S}}} + e\boldsymbol{b}_{k_{\mathbf{S}}} \\ +\boldsymbol{\xi}_{\mathbf{x}} + \cdots}}{\mathbf{x}} \quad \underset{\boldsymbol{a}_{k_{\mathbf{S}}} + e\boldsymbol{b}_{k_{\mathbf{S}}}}{\mathbf{y}}$$

Here, we assume that a specific position $m_{\mathbf{S}} = 4 \in [M]$ encodes task vector $\boldsymbol{a}_{k_{\mathbf{S}}}$ encoding the high-level task message. This formulism is inspired by the empirical findings in Allen-Zhu & Li (2024) and serves a role similar to the relation token described in Nichani et al. (2025), but ours differs in its empirically-supported concept modeling.

**QA-ICL Prompt Distribution**. ($\mathcal{P}_{\text{QA}}^{\mathbf{T}}$). A natural extension of the above two distributions is that we can replace the word-based demonstration by QA-based demonstration in the task-specific ICL prompt, namely $\mathbf{T}_{\text{QA}} := [\mathbf{x}_1^{\text{QA}}, \mathbf{y}_1, \cdots, \mathbf{x}_J^{\text{QA}}, \mathbf{y}_J, \mathbf{x}_{J+1}^{\text{QA}}] \in \mathbb{R}^{d \times (J+1)(M+2)} \sim \mathcal{P}_{\mathbf{T}_{\text{QA}}}$.

### 2.2. Residual-Layernom Transformer Model

The structured "up-word-down-label" embedding $\mathbf{E}(\mathbf{T}) = \begin{pmatrix} \boldsymbol{x}_1, & \cdots & \boldsymbol{x}_J, & \boldsymbol{x}_{\text{query}} \\ \boldsymbol{y}_1, & \cdots & \boldsymbol{y}_J, & \mathbf{0} \end{pmatrix}$ proposed in prior theoretical work (Bai et al., 2023; Zhang et al., 2024; Huang et al., 2024a; Kim & Suzuki, 2024; Chen et al., 2024a; Bu et al., 2024a) is designed for simplified, structured, and residual-free transformers, *where words are only available to at-*

---

[1]Indeed, the formula of the sentence can be a factual statement (e.g. $\mathbf{S} = [\mathbf{x}^{\text{QA}}, \mathbf{y}] = [\underset{\boldsymbol{\nu}_1}{\text{The}}, \boldsymbol{a}_k, \underset{\boldsymbol{\nu}_4}{\text{of}}, \mathbf{x}, \mathbf{y}]$) other than a QA.

*tention matrices and labels are restricted to value/combine matrices*. This residual-free setup deviates from real-world scenarios. In contrast, we consider the case where the prompt $\mathbf{T} = [\boldsymbol{x}_1, \boldsymbol{y}_1, \cdots, \boldsymbol{y}_J, \boldsymbol{x}_{\text{query}}] = [\mathbf{T}_1, \cdots \mathbf{T}_L] \in \mathbb{R}^{d \times L}$ ($L = 2J + 1$) is *directly processed* by a non-linear transformer with a residual stream as follows.

$$\mathbf{h}_{\boldsymbol{\theta},0}(\mathbf{T}) = \sum_{l=1}^{L-1} \mathbf{W}_V \mathbf{T}_l \sigma_S\left( (\mathbf{W}_K \mathbf{T}_l)^\top (\mathbf{W}_Q \mathbf{T}_L) \right) \in \mathbb{R}^d,$$

$$\mathbf{h}_{\boldsymbol{\theta}} = \mathbf{W}_O \text{LN}(\mathbf{h}_{\boldsymbol{\theta},0}(\mathbf{T})) + \mathbf{T}_L,$$

where $\sigma_S(\cdot)$ denotes the column-wise softmax operation, $\mathbf{W}_O := \mathbb{I}_{d \times d}$, $\text{LN}(\mathbf{z}) := \mathbf{z}/\|\mathbf{z}\|_2$ is the $l_2$ layer-wise normalization, and $\mathbf{h}_{\boldsymbol{\theta}}$ is the vector output of transformer.

**Connection to Word2Vec Arithmetic**. When $\|\boldsymbol{a}_k\| = \|\boldsymbol{b}_{k'}\|$ for all $k, k' \in [K]$, and both the cardinalities of $\mathcal{X}_{\mathbf{T},l}, \mathcal{Y}_{\mathbf{T},l}$ and the noise magnitude are sufficiently bounded, we have an approximate identity:

$$\mathbf{y}_{J+1} \approx \boldsymbol{a}_{k_{\mathbf{T}}} + \mathbf{x}_{J+1}, \tag{3}$$

where components unrelated to the current co-concept $k_{\mathbf{T}}$ are negligible compared to the dominant term $1.1\boldsymbol{a}_{k_{\mathbf{T}}} \pm \boldsymbol{b}_{k_{\mathbf{T}}}$, which governs the logits when the model attends to the vocabulary dictionary. Therefore, if the transformer $\boldsymbol{\theta}$ can extract the high-level task vector $\mathbf{h}_{\boldsymbol{\theta},0}(\mathbf{T})$, then adding it to **any word vector within the same task concept** should yield the task-specific label vector in the context of argmax sampling. This aligns with empirical findings by Merullo et al. (2024), which show that adding various embedded query words, such as "Poland" or "China", to the vector $\vec{o}_{city}$—which captures the function `get_capital(·)` in the latent space—produces the correct capital city.

**Training Setups**. Define $\boldsymbol{\theta} := \{\mathbf{W}_K, \mathbf{W}_Q, \mathbf{W}_V\}$. We minimize the $L_2$-regularized cross-entropy loss by gradient descent (GD) at each time step

$$L_{\mathcal{P}^{\text{tr}}}(\boldsymbol{\theta}) = -\mathbb{E}_{\mathcal{P}^{\text{tr}}}\left[ \log\left( \frac{\exp(\mathbf{u}_{k_{\mathbf{y}}}^\top \mathbf{h}_{\boldsymbol{\theta}})}{\sum_{k \in [7K + K']} \exp(\mathbf{u}_k^\top \mathbf{h}_{\boldsymbol{\theta}})} \right) \right], \tag{4}$$

where $K$ denotes the number of tasks, $K'$ the number of task-irrelevant tokens, and $\mathbf{u}_{k_{\mathbf{y}}}$ refers to the vector corresponding to $\mathbf{y}$ in the token embedding matrix $\mathbf{U}$, which contains $K$ both high- and bi-label low-level concepts, noise-free and single-task-specific word and label tokens, as well as $K'$ irrelevant tokens–the number "$7K + K'$" is thus by $K$ sets of $\{\boldsymbol{a}_k \pm \boldsymbol{b}_k, 0.1\boldsymbol{a}_k \pm \boldsymbol{b}_k, \pm \boldsymbol{b}_k, \boldsymbol{a}_k\}$ as well as $K'$ irrelevant tokens. Notably, our analysis focuses exclusively on the semantics associated with the prompt's shared co-task encoded in a given word or label token.[2] Normalization ensures that each token $\mathbf{u}_k$ has the same length, enabling fair comparisons during sampling. For simplicity,

---

[2] While it is tractable to assume that all dictionary entries follow the structure in Eq. (2), doing so would require additional assumptions and care regarding how fixed polysemous words relate to their corresponding labels.

we assume $\|\boldsymbol{a}_k\| = \|\boldsymbol{a}\| = \|\boldsymbol{b}_k\| = \|\boldsymbol{b}\| = \|\boldsymbol{\nu}_m\| = 1$, a common setup for theoretical studies (Tian et al., 2023; 2024). Formal details of $\mathbf{U}$ are provided in Appendix C.

The training data $\mathcal{P}^{\text{tr}}$ is sampled from training distribution $\mathcal{P}$, with a sample size of $N$. To address the scale difference between the gradients of attention and MLP, we adopt a smaller learning step for MLP characterized by a scale factor $q_V$. The initial weight matrices $\mathbf{W}_Q^{(0)}$ and $\mathbf{W}_K^{(0)}$ are sampled independently from a Gaussian distribution $\mathcal{N}(\mathbf{0}, \sigma_0^2 \cdot \mathbb{I})$, a more realistic choice than scaled identity or overly constrained initializations used in prior work (Li et al., 2023b; Bu et al., 2024a; Chen et al., 2024a). Similarly, $\mathbf{W}_V^{(0)}$ is initialized as $\mathcal{N}(\mathbf{0}, \sigma_1^2 \cdot \mathbb{I})$, consistent with standard practice in recent theoretical analyses of Transformers (Tian et al., 2023; Jiang et al., 2024a; Li et al., 2025a; Yang et al., 2024b).

**Test Setup**. During testing, we examine the probability that the label vector is the most-likely token to be selected from the disrupted token dictionary on the test prompt distribution $\mathcal{P}^\star$ where the noises are sampled from $\mathcal{N}(\mathbf{0}, \sigma_p^{\star 2} \cdot \mathbb{I})$ (either word-based prompt or QA sentence-based prompt)

$$L_{\mathcal{P}^\star} = \mathbb{E}_{\mathcal{P}^\star}[\mathbf{1}(k_{\mathbf{y}} \neq \operatorname*{argmax}_k (\frac{\exp(\mathbf{u}_k^\top \mathbf{h}_{\boldsymbol{\theta}})}{\sum_{k \in [7K + K']} \exp(\mathbf{u}_k^\top \mathbf{h}_{\boldsymbol{\theta}})}))]$$
$$= \mathbb{E}_{\mathcal{P}^\star}[\mathbf{1}(k_{\mathbf{y}} \neq \operatorname*{argmax}_k \mathbf{u}_k^\top \mathbf{h}_{\boldsymbol{\theta}})], \tag{5}$$

where $k_{\mathbf{y}}$ is the index of $\mathbf{y}$ in the total dictionary $\mathbf{U}$. The whole procedure is in Algorithm 1.

---

**Algorithm 1** Training algorithm

**Input:** Training distribution $\mathcal{P}$, Test distribution $\mathcal{P}^\star$, Training size $N$, step size $\eta q_V$, scaled parameter $q_V$, stopping criterion $\varepsilon$ and total epochs $T$.
Initialize the model $\boldsymbol{\theta}^{(0)} = \{\mathbf{W}_V^{(0)}, \mathbf{W}_K^{(0)}, \mathbf{W}_Q^{(0)}\}$.
Sample training data $\mathcal{P}^{\text{tr}} \sim \mathcal{P}$.
**for** $t = 0, 1, \ldots, T - 1$ **do**
  If $L_{\mathcal{P}^\star}^{0-1}(\boldsymbol{\theta}^{(t)}) \leq \varepsilon$ stop else continue.
  Update model parameters:
  $\mathbf{W}_V^{(t+1)} = \mathbf{W}_V^{(t)} - \eta q_V \nabla_{\mathbf{W}_V^{(t)}} L_{\mathcal{B}_t}(\boldsymbol{\theta}^{(t)})$,
  $\mathbf{W}_K^{(t+1)} = \mathbf{W}_K^{(t)} - \eta \nabla_{\mathbf{W}_K^{(t)}} L_{\mathcal{B}_t}(\boldsymbol{\theta}^{(t)})$,
  $\mathbf{W}_Q^{(t+1)} = \mathbf{W}_Q^{(t)} - \eta \nabla_{\mathbf{W}_Q^{(t)}} L_{\mathcal{B}_t}(\boldsymbol{\theta}^{(t)})$
**end for**

---

## 3. Theoretical Results

In this section, we present our main theoretical results.

**Condition 3.1.** Suppose the following holds for some sufficiently large constant $C > 0$:

1. Dimension $d$ satisfies $d \geq CM^2 \log(\frac{K'^2 N^2 M^2}{\delta})$.

2. Training sample size $N$ is sufficiently large $N \geq C \max\{K \log(\frac{1}{\delta}), \frac{KK' \log(\frac{1}{\delta})}{M}\}$.

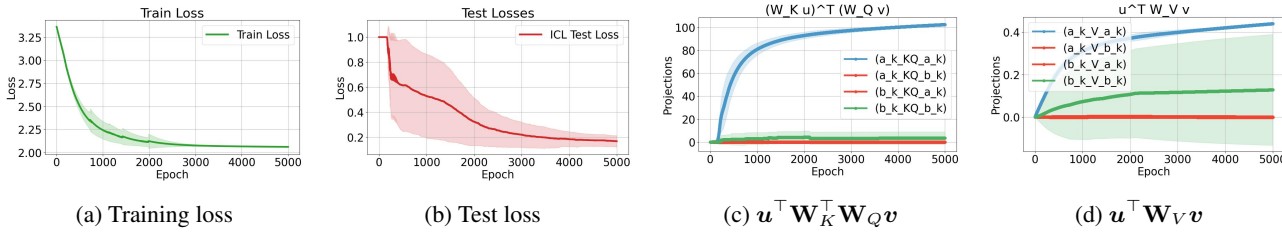

*Figure 2.* Training dynamics over the ICL prompt training distribution. (a–b) Training and test losses; (c–d) projection values of key-query and value matrices. In (d), $\mathbf{W}_V$ overfits to low-level features $\boldsymbol{b}_k$, resulting in persistently constant (0.2) test error shown in (b).

3. Dictionary size $K, K'$ satisfies $K \geq C \log(\frac{1}{\delta})$, $K' \geq C \max\{M, K\}$.

4. The standard deviations of Gaussian initializations satisfies $\sigma_1 \leq \min\{\frac{d^{-\frac{1}{2}}}{C}, \frac{\sqrt{q_V (\log(\frac{K'^2}{\delta}))}}{C}\}$, and $\sigma_0 \leq \min\{\frac{d^{-\frac{1}{4}}}{C} (\log(\frac{(K')^2}{\delta}))^{-\frac{1}{4}} (\log(M))^{\frac{1}{2}}, \frac{d^{-\frac{1}{2}}}{C}\}$.

5. The noise level $\sigma_p$ satisfies $\sigma_p \leq \frac{d^{-\frac{1}{2}}}{C}$.

6. Scaled parameter $q_V$ and learning rate $\eta$ satisfy $\frac{C \sigma_1^2 d}{\log(\sigma_0^{-2} d^{-1} \log(\frac{M-1}{0.06}))} \leq q_V \leq \frac{\sigma_1^2 d}{C \log(d^{-\frac{1}{2}} \sqrt{\log(\frac{K'^2}{\delta})})}$, $\eta \leq \min\{\frac{\sigma_1^2 d^{\frac{1}{2}} K \sqrt{\log(\frac{K'^2}{\delta})}}{q_V C}, \frac{\sigma_1 d^{\frac{1}{2}} M^4}{C(M-1)^2}\}$.

The conditions on $d, N, K$ ensure that certain concentration inequalities hold and that the learning problem is adequately overparameterized (Chatterji & Long, 2021; Frei et al., 2022; Cao et al., 2022; Kou et al., 2023). The condition on $K'$ controls the impact of contributions from answer-irrelevant dictionary tokens, though it can be relaxed at the cost of a more intricate analysis. The conditions on $\sigma_0$ and $\sigma_1$ regulate the model's initial bias and ensure that gradient descent updates the model effectively. The condition on $\sigma_p$ guarantees that the gradient flow is only mildly affected by noise, which is reasonable given the typically high signal-to-noise ratio in language data. Finally, the conditions on $\eta$ and $q_V$ are technical assumptions necessary for the optimization analysis.

The followings present our primary results regarding the retrieval of task vectors and the convergence of test loss.

**Theorem 3.2 (Task Vector Retrieval).** *Under Condition 3.1, let test ICL prompt distributions include Word-Label ICL Prompt $\mathcal{P}_{\mathbf{T}}^{\star}$ and QA-ICL Prompt $\mathcal{P}_{QA}^{\mathbf{T}}{}^{\star}$. Then, for the gradient descent iterates in Algorithm 1, with probability at least $1-\delta$, there exists $t = \Omega((\eta q_V)^{-1} \sigma_1^2 dK$, such that:*

- ***Training on $\mathcal{P}_{\mathbf{T}}$ or $\mathcal{P}_{QA}^{\mathbf{T}}$:*** *When testing on a sample with task concept $k^{\star} \in [K]$, before adding the residual stream, the model generates a hybrid vector with both high- and low-level components:*

$$\cos\langle \mathbf{h}_{\boldsymbol{\theta}^{(t)},0}, \boldsymbol{a}_{k^{\star}} \rangle = \Theta(1), \quad \cos\langle \mathbf{h}_{\boldsymbol{\theta}^{(t)},0}, \boldsymbol{b}_{k^{\star}} \rangle = \Theta(1). \quad (6)$$

- ***Training on $\mathcal{P}_{QA}$:*** *When testing on a sample with task concept $k^{\star} \in [K]$, the model approximately retrieves the appropriate task vector before adding the residual stream:*

$$\cos\langle \mathbf{h}_{\boldsymbol{\theta}^{(t)},0}, \boldsymbol{a}_{k^{\star}} \rangle = \Theta(1), \quad \cos\langle \mathbf{h}_{\boldsymbol{\theta}^{(t)},0}, \boldsymbol{u} \rangle = o(1), \quad (7)$$

*for all $\boldsymbol{u} \in \{\boldsymbol{a}_s\}_{s \neq k^{\star} \in [K]} \cup \{\boldsymbol{b}_s\}_{s \in [K]} \cup \{\boldsymbol{\nu}_{k'}\}_{k' \in [K']}$.*

The first result shows that under our setting, when training and testing on ICL-type data, the transformer fails to reliably retrieve high-level co-task information from ICL or QA-ICL prompts. Consequently, the resulting $\mathbf{h}_{\boldsymbol{\theta}^{(t)}}$ becomes a hybrid vector, contaminated by low-level features that *disrupt* task vector addition with residuals, as validated in Figure 2(d). Unlike *harmful overfitting* in vision models (Frei et al., 2022; Cao et al., 2022), which stems from *noise memorization*, the harmful overfitting here arises from *low-level feature memorization*, due to their unintended co-occurrence in ICL-style data. In contrast, the second result shows that training on QA data enables more accurate retrieval of co-task vectors, which is inherently due to the absence of $\boldsymbol{b}_k$ in $\mathbf{x}_{QA}$, as illustrated in Figure 3.

The following proposition further validates the consequences of harmful vs. benign task vector retrieval.

**Proposition 3.3 (Test Losses Disparity).** *For $\forall \varepsilon > 0$, under Condition 3.1, for any test distribution $\mathcal{P}^{\star} \in \{\mathcal{P}_{\mathbf{T}}^{\star}, \mathcal{P}_{QA}^{\mathbf{T}}{}^{\star}\}$, with the prompt length[3] satisfying $J^{\star} = \Omega(\frac{\log(1/\varepsilon)}{2 \log(K)})$ and the noise level $\sigma_p^{\star} = O((K \log(\frac{L}{\varepsilon}))^{-\frac{1}{2}})$, with probability at least $1 - \delta$, there exists $t = O(\eta^{-1} q_V^{-1} \sigma_1^2 d^2 KM \log(\frac{1}{\varepsilon}))$, it holds that*

- ***Training on $\mathcal{P}_{\mathbf{T}}$ or $\mathcal{P}_{QA}^{\mathbf{T}}$:*** *The test loss satisfies $L_{\mathcal{P}^{\star}}(\boldsymbol{\theta}^{(t)}) = \Theta(1)$.*

- ***Training on $\mathcal{P}_{QA}$:*** *The test loss satisfies $L_{\mathcal{P}^{\star}}(\boldsymbol{\theta}^{(t)}) \leq \varepsilon$. Furthermore, we have two bonuses*

  - ***Task Vector Regression from Demonstrations.*** *The transformer could approximately retrieve the task*

---

[3]This condition ensures that other task concepts within the prompt do not interfere with recognizing the prompt's co-task. It can be relaxed by assuming a lower probability of words being associated with task concepts outside the prompt's co-task concept or by limiting the number of task concepts per word.

vector (i.e. satisfies Eq.(7)) solely from demo-pairs, without requiring a query word.

- **Task Vector Arithmetic in Factual Recall**. *Given a test sample with co-concept $k^\star$, the intermediate output $\mathbf{h}_{\boldsymbol{\theta}^{(t)},0}$ can be added to any query with co-concept $k^\star$, which also leads to $L_{\mathcal{P}^\star}(\boldsymbol{\theta}^{(t)}) \leq \varepsilon$.*

The first bonus parallels findings in Hendel et al. (2023); Yang et al. (2025), demonstrating that *transformers can regressively infer task vectors from demo-pairs*. The second aligns with empirical studies Hendel et al. (2023); Merullo et al. (2024), where Merullo et al. (2024) add extracted task vectors from current prompt to queries outside the prompt and still obtain accurate predictions.

The following proposition examines the model's capability to address OOD unseen ICL tasks.

**Proposition 3.4** (**OOD Generalization**). *Under the conditions of Theorem 3.2, suppose the transformer is trained on $\mathcal{P}_{QA}$. At test time, for any distribution $\mathcal{P}^\star \in \{\mathcal{P}_{\mathbf{T}}^\star, \mathcal{P}_{QA}^{\mathbf{T}\,\star}\}$, the model satisfies:*

1. **Dictionary Shift Adaptability**. *The model generalizes to unseen dictionaries $\mathbf{U}^{new}$ with high-level ($K_a^\star \leq K$), low-level ($K_b^\star \leq d - K_a^\star$), and irrelevant ($K_\nu^\star \leq d - K_a^\star - K_b^\star$) concepts, where $\boldsymbol{a}_{k_a}^\star \in \mathrm{conic}(\{\boldsymbol{a}_k\}_{k \in [K]})$, and $\|\boldsymbol{a}_{k_a}^\star\| = \|\boldsymbol{b}_{k_b}^\star\| = \|\boldsymbol{\nu}_{k_\nu}^\star\| = 1$ for all $k_a \in K_a^\star$, $k_b \in K_b^\star$, and $k_\nu \in K_\nu^\star$, with $\{\boldsymbol{a}_{k_a}^\star\}$, $\{\boldsymbol{b}_{k_b}^\star\}$, and $\{\boldsymbol{\nu}_{k_\nu}^\star\}$ mutually orthogonal.*

2. **Distribution Shift Adaptability**. *The model accommodates test prompts containing multiple co-task concepts $\mathcal{K} \subset [K]$ with $|\mathcal{K}| < 4$. The test prompt length $J^\star$ and the distributions over $\mathcal{X}_{\mathbf{T},l}$ and $\mathcal{Y}_{\mathbf{T},l}$ for all $l \in [J+1]$ are arbitrary, provided that $\sum_l |\mathcal{X}_{\mathbf{T},l}|, \sum_l |\mathcal{Y}_{\mathbf{T},l}| = o(J^\star/10)$.*

This proposition contributes to the ongoing discussion of LLMs' linear latent geometry and OOD extrapolation, particularly in relation to Question 5.1.4 of Reizinger et al. (2024), which ask whether the linear latent geometry aids the OOD extrapolation, superior to static embedding methods. Prior works (Li et al., 2023b; Bu et al., 2024a) explore OOD benefits in multi-task classification but assume new label features share a consistent signal (e.g., $\pm 1$). In contrast, our framework allows diverse low-level features (e.g., $\boldsymbol{b}_{k_b}^\star = \boldsymbol{b}_{k_2^\star} - \boldsymbol{b}_{k_3^\star}$), highlighting the role of task vector arithmetic in *retrieving task message rather than memorizing label-level features*, consistent with Todd et al. (2024) on compositional task vectors. Under a Dictionary Shift scenario where $\boldsymbol{b}^\star \perp \mathbf{U}$, word-label pairs involving unseen low-level task concepts (i.e., $\mathbf{x} = x_a \boldsymbol{a}_k + \boldsymbol{b}^\star + \boldsymbol{\xi}_{\mathbf{x}}$ and $\mathbf{y} = \boldsymbol{a}_k + \boldsymbol{b}^\star + \boldsymbol{\xi}_{\mathbf{y}}$) benefit from the model's high-level task retrieval capability. This phenomenon resonates with the "celebrity helps minority" effect in Allen-Zhu & Li (2024), where the presence of high-frequency entities enables the

model's *fact retrieval capability* to generalize to rare entries. While their settings emphasize *fact retrieval*, our framework centers on the retrieval of *task vectors*, a single token subcase of theirs. Indeed, despite constrained on factual-recall task, the results offer a foundational perspective on why simple task vector arithmetic holds potential for a range of downstream applications, including concept erasure (Ilharco et al., 2023), mitigation of forgetting (Jiang et al., 2025), model editing (Li et al., 2025b), and model merging (Wortsman et al., 2022; Matena & Raffel, 2022; Yang et al., 2024a; Lee et al., 2025; Yoshida et al., 2025; Cao et al., 2025; He et al., 2025).

For Distribution Shift, when multiple co-task concepts appear in a prompt, the transformer would likely form a hybrid task vector:

$$\mathbf{h}_{\boldsymbol{\theta}^{(t)},0} \approx \sum_{k \in \mathcal{K}} w_{\boldsymbol{\theta},k} \boldsymbol{a}_k, \quad w_{\boldsymbol{\theta},k} > 0, \quad \sum_{k \in \mathcal{K}} w_{\boldsymbol{\theta},k} = 1. \quad (8)$$

This suggests a soft, weighted integration of concepts, where $w_{\boldsymbol{\theta},k}$ denotes the model's inferred likelihood of task $k$ by softmax operation. Such behavior closely resembles Bayesian Model Averaging (BMA) in topic models–an interpretation that recent work has proposed as an underlying mechanism of ICL in LLMs (Blei et al., 2001; Xie et al., 2022; Wang et al., 2023):

$$p_{\boldsymbol{\theta}}(f_{\boldsymbol{\theta}}(\mathbf{x}_{\mathrm{query}}) = \cdot \mid \mathbf{T}) = \int_{\mathbf{z}} p_{\boldsymbol{\theta}}(\cdot \mid \mathbf{z}, \mathbf{T}) \, p_{\boldsymbol{\theta}}(\mathbf{z} \mid \mathbf{T}) \, d\mathbf{z}. \quad (9)$$

Here, $\mathbf{z}$ represents the latent concept variables inferred by the model, linking Eq. (9) to Eq. (1) in the factual-recall ICL setting. The integral over $\mathbf{z}$ corresponds to a conic combination of task vectors $\{\boldsymbol{a}_k\}_{k \in \mathcal{K}}$ as described in Eq. (8), computed by the transformer. Moreover, $f_{\boldsymbol{\theta}}(\mathbf{x}_{\mathrm{query}})$ aligns with $\mathbf{h}_{\boldsymbol{\theta}^{(t)},0} + \mathbf{x}_{\mathrm{query}}$ in our framework.

## 4. Proof Idea

The high-level intuition of proof is as follows:

- For any $\varepsilon > 0$, to establish the convergence of the test loss defined by argmax sampling in Eq. (5), the primary task is to prove Eq. (6) and Eq. (7) in terms of constant disparity. Additionally, the required successful cosine similarity (i.e., how well $\mathbf{h}_{\boldsymbol{\theta}^{(t)},0}$ approximates the task vector) must scale with $\sigma_p^\star \sqrt{\log(\varepsilon^{-1})}$ to mitigate noise disruption during testing.

- The cosine similarity in Eq. (7) is closely tied to the learning dynamics of matrix projections along the directions of task vector $\boldsymbol{a}_k$ for $k \in [K]$. For QA training, the remaining task is to characterize the learning dynamics along these directions (specifically, $\boldsymbol{a}_k^\top \mathbf{W}_V^{(t)} \boldsymbol{a}_k$ and $(\mathbf{W}_Q^{(t)} \boldsymbol{a}_k)^\top (\mathbf{W}_K^{(t)} \boldsymbol{a}_k)$) and the negligible updates of other projections.

- For ICL-type training, we show that there is only one key difference: the unexpected co-occurrence-induced non-negligible growth of some $|\boldsymbol{b}_k^\top \mathbf{W}_V^{(T_1)} \boldsymbol{b}_k|$.

This drives MLP falsely memorize low-level features, which results in imperfect task vector retrieval and is responsible for the constant-level test error.

**Main Challenges**. We begin by observing that the gradients of $\mathbf{W}_K$, $\mathbf{W}_Q$, and $\mathbf{W}_V$ can be computed as follows:

$$
\mathbb{E}_{\mathcal{P}_{QA}^{tr}}[\partial_{\mathbf{W}_K} L_{\mathcal{B}}(\boldsymbol{\theta})] = -\mathbb{E}_{\mathcal{P}_{QA}^{tr}}\left[\sum_{j=1}^{M}\iota_{\mathbf{u_{k_y}},j}\cdot(\mathbf{W}_Q\mathbf{S}e_L)\cdot(\mathbf{S}(e_j-\boldsymbol{\pi}))^{\top}\right]
$$

$$
\mathbb{E}_{\mathcal{P}_{QA}^{tr}}[\partial_{\mathbf{W}_Q} L_{\mathcal{B}}(\boldsymbol{\theta})] = -\mathbb{E}_{\mathcal{P}_{QA}^{tr}}\left[\sum_{j=1}^{M}\iota_{\mathbf{u_{k_y}},j}\cdot\mathbf{W}_K\mathbf{S}(e_j-\boldsymbol{\pi})\cdot(\mathbf{S}e_L)^{\top}\right]
$$

$$
\mathbb{E}_{\mathcal{P}_{QA}^{tr}}[\partial_{\mathbf{W}_V} L_{\mathcal{B}}(\boldsymbol{\theta})] = -\mathbb{E}_{\mathcal{P}_{QA}^{tr}}\left[\frac{\mathbf{\Pi}^{\mathbf{u_{k_y}}-\sum_{k\in[7K+K']}\omega_k\mathbf{u}_k}_{(\mathbf{W}_V\mathbf{S}\boldsymbol{\pi})^{\perp}}(\mathbf{S}\boldsymbol{\pi})^{\top}}{\|\mathbf{W}_V\mathbf{S}\boldsymbol{\pi}\|}\right]
$$

(10)

where $\mathbf{\Pi}^{\boldsymbol{v}}_{\boldsymbol{u}_{\perp}} = \boldsymbol{v} - \left(\boldsymbol{v}^{\top}\frac{\boldsymbol{u}}{\|\boldsymbol{u}\|}\right)\cdot\frac{\boldsymbol{u}}{\|\boldsymbol{u}\|}$, $\iota_{\mathbf{u_{k_y}},j}$ is a coefficient defined in Eq. (30), $\boldsymbol{\pi}$ represents the concatenated softmax weights for positions $1,\ldots,L-1$ in the sentence, and $\omega_k$ denotes the likelihood of the model's output matching the $k$-th dictionary token $\mathbf{u}_k$ (See Eq. (30)). The primary challenge in our analysis stems from the fact that many components in the gradient computation, such as $\iota_{\mathbf{u_{k_y}},j}$ and $\|\mathbf{W}_V\mathbf{S}\boldsymbol{\pi}\|$, exhibit *intricate monotonicity properties* and varying rates of evolution across different phases. This complexity arises inherently from the *non-linear interactions* between the softmax operation, layer normalization, residual connections, and the cross-entropy loss, making it challenging to precisely characterize the dynamics. To address this, we derive separate upper and lower bounds for the discrete gradients of both the attention and MLP components in each phase. Specifically, we introduce a set of *six differential equations* (see Appendix D.1) as continuous surrogates, which effectively capture the upper and lower bounds on the evolution rates of the attention and MLP components across different phases.

## 4.1. Phase 1: Linear Growth of MLP and Accelerating Growth of Attention

In this phase, we analyze the regime where $a_k^{\top}\mathbf{W}_V^{(t)}a_k = o(\|\mathbf{W}_V\mathbf{S}\boldsymbol{\pi}\|)$ and $\|\mathbf{W}_V\mathbf{S}\boldsymbol{\pi}\| = \Theta(\sigma_1 d)$. Given the known order of the denominator in the update rule for $\mathbf{W}_V$ (Eq. 10), the update of $a_k^{\top}\mathbf{W}_V^{(t)}a_k$ can be *linearly* bounded by update $a_{t+1} = a_t + b$, where $b$ is constant. The accumulated update is further upper bounded by the integral of its continuous-time counterpart, as characterized in Lemma D.1. When $\mathcal{P}^{tr} \sim \mathcal{P}_{QA}$, the terms $|b_k^{\top}\mathbf{W}_V^{(T_1)}b_k|$ remain negligible, since $\mathbf{x}^{QA}$ contains no $b_k$. In contrast, under $\mathcal{P}^{tr} \sim \mathcal{P}_{\mathbf{T}}$ or $\mathcal{P}_{QA}^{\mathbf{T}}$, co-occurrence asymmetries between demo-pairs and query inputs with respect to some $\pm b_k$ induce nontrivial growth of some $|b_k^{\top}\mathbf{W}_V^{(T_1)}b_k|$. These behaviors are formally summarized below.

**Lemma 4.1.** *Under Condition 3.1, during $t \leq T_1 \leq t_2^{\star}$, where $t_2$ is defined in Lemma F.5. Consider $\mathcal{P}^{tr}$ sampled*

from either $\mathcal{P}_{QA}$, $\mathcal{P}_{\mathbf{T}}$ or $\mathcal{P}_{QA}^{\mathbf{T}}$. Then for $\forall k \in [K]$ and $n \in \mathcal{N}_k$, there exists some $C_{1-2}, C'_{1-2}, \hat{C}_{1-2} > 0$ such that

$$
a_k^{\top}\mathbf{W}_V^{(t)}a_k \geq \frac{C_1\eta q_V t}{\sigma_1 d^{1/2}MK} - \sqrt{2\log(\tfrac{8(2K+K')^2}{\delta})}\sigma_1,
$$
$$
a_k^{\top}\mathbf{W}_V^{(t)}a_k \leq \frac{C_2\eta q_V t}{\sigma_1 d^{1/2}K} + \sqrt{2\log(\tfrac{8(2K+K')^2}{\delta})}\sigma_1.
$$

(11)

*Furthermore, when $\mathcal{P}^{tr} \sim \mathcal{P}_{\mathbf{T}}$ or $\mathcal{P}^{tr} \sim \mathcal{P}_{QA}^{\mathbf{T}}$, for $\forall y \in \{\pm 1\}$, there exists $k_y \in [K]$ and some constant $\underline{C}_4, \underline{C}_5 > 0$ such that*

$$
|b_{k_+}^{\top}\mathbf{W}_V^{(T_1)}b_{k_+}| \geq \frac{\underline{C}_4\sigma_1 d^{1/2}}{M} - \sqrt{2\log(\tfrac{8(2K+K')^2}{\delta})}\sigma_1,
$$
$$
|b_{k_-}^{\top}\mathbf{W}_V^{(T_1)}b_{k_-}| \geq \frac{\underline{C}_5\sigma_1 d^{1/2}}{M} - \sqrt{2\log(\tfrac{8(2K+K')^2}{\delta})}\sigma_1.
$$

(12)

Additionally, Lemma F.1 ensures that other types of projections remain constrained near their initialization scale. Also, the update of attention projections along key directions satisfies the recurrence order $f_{t+1} = f_t + a(t-t_3)f_t$, where $a$ is a constant. The accumulated outcome of such updates can be upper bounded by an exponential function and lower bounded by a quadratic function, as demonstrated via integration of its continuous-time analogue in Lemma D.5. The precise statement is provided below.

**Lemma 4.2.** *Under Condition 3.1, suppose Eq. (49) holds at iteration $t \leq T_1$. Consider $\mathcal{P}^{tr}$ sampled from either $\mathcal{P}_{QA}$, $\mathcal{P}_{\mathbf{T}}$ or $\mathcal{P}_{QA}^{\mathbf{T}}$. Then for $\forall k \in [K]$, there exist some positive constants $C_3, C_4$, during $[t_4, T_1]$, where $t_4$ is defined in Lemma F.7, it holds that*

$$
(\mathbf{W}_Q^{(t)}a_k)^{\top}(\mathbf{W}_K^{(t)}a_k) \geq \frac{C_3\eta^2 q_V\sigma_0^2[(t-t_4-1)^2-1]}{\sigma_1^2 M^2 K^2}
$$
$$
- 4\sqrt{\log(16(2K+K')^2/\delta)}\sigma_0^2 d^{1/2},
$$
$$
(\mathbf{W}_Q^{(t)}a_k)^{\top}(\mathbf{W}_K^{(t)}a_k) \leq C_4((4\sqrt{\log(16(2K+K')^2/\delta)}\sigma_0^2 d^{1/2}
$$
$$
+ \frac{3\sigma_0^2 d}{2}) + \frac{3\sigma_0^2 d}{2}\exp(\frac{\hat{C}_2 C_2\eta^2 q_V t^2}{\sigma_1^2 dK^2})).
$$

## 4.2. Phase 2: Decelerating Growth of MLP and Attention

In this phase, $a_k^{\top}\mathbf{W}_V^{(t)}a_k$ determines the order of $\|\mathbf{W}_V\mathbf{S}\boldsymbol{\pi}\|$, which, as the denominator in the gradient, leads to a deceleration in the updates of both $a_k^{\top}\mathbf{W}_V^{(t)}a_k$ and the attention projections. As in the previous phase, there is nontrivial growth in some $|b_k^{\top}\mathbf{W}_V^{(T_1)}b_k|$ when trained on ICL-type data (i.e., $\mathcal{P}^{tr} \sim \mathcal{P}_{\mathbf{T}}$ or $\mathcal{P}_{QA}^{\mathbf{T}}$) due to co-occurrence asymmetries. The MLP dynamics slow to a sublinear rate, with their analogue captured in Lemma D.3. In contrast, the evolution of the attention projections is bounded between $e^{-1/t}$ and $e^t$ as in Lemma D.2 and Lemma D.6. The formal statement is as below.

**Lemma 4.3.** *Under Condition 3.1, during $t_1 \leq t \leq T^{\star} = \Omega(\eta^{-1}q_V^{-1}\sigma_1^2 KMd^2\log(\tfrac{1}{\epsilon}))$ where $t_1 \leq T_1$. Consider $\mathcal{P}^{tr}$*

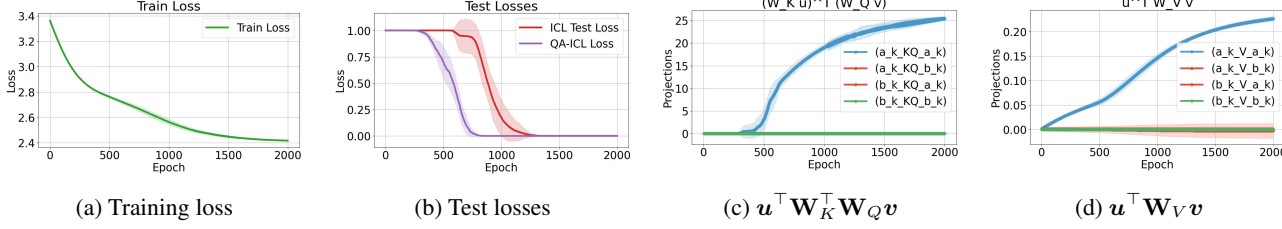

*Figure 3.* Training dynamics over the QA sentence training distribution. Legends are consistent with those in Figure 2. Compared to the ICL-trained model in Figure 2, $\mathbf{W}_V$ trained on QA data does not overfit to low-level features $\boldsymbol{b}_k$ (see (d)), leading to improved generalization and reduced test errors in (b).

sampled from either $\mathcal{P}_{QA}$, $\mathcal{P}_{\mathbf{T}}$ or $\mathcal{P}_{QA}^{\mathbf{T}}$. Then for $\forall k \in [K]$, there exists some $C_{7-10}$ such that

$$\boldsymbol{a}_k^\top \mathbf{W}_V^{(t)} \boldsymbol{a}_k \geq \sqrt{C_7 \frac{\eta q_V (t-t_1)}{MK} + (\boldsymbol{a}_k^\top \mathbf{W}_V^{(t_1)} \boldsymbol{a}_k)^2},$$

$$\boldsymbol{a}_k^\top \mathbf{W}_V^{(t)} \boldsymbol{a}_k \leq \sqrt{C_8 \frac{\eta q_V (t-t_1)}{K} + (\boldsymbol{a}_k^\top \mathbf{W}_V^{(t_1)} \boldsymbol{a}_k)^2} + \frac{C_8 \eta q_V}{K(\boldsymbol{a}_k^\top \mathbf{W}_V^{(t_1)} \boldsymbol{a}_k)},$$

$$(\mathbf{W}_Q^{(t)} \boldsymbol{a}_{\hat{k}})^\top (\mathbf{W}_K^{(t)} \boldsymbol{a}_{\hat{k}}) \leq C_9 \sigma_0^2 d e^{q_V^{-1} \sigma_1^2 d} \left[\frac{\eta q_V t}{\sigma_1^2 dKM}\right]^{\frac{M}{16 q_V}},$$

$$(\mathbf{W}_Q^{(t)} \boldsymbol{a}_{\hat{k}})^\top (\mathbf{W}_K^{(t)} \boldsymbol{a}_{\hat{k}}) \geq C_{10} \frac{\sigma_0^2 \sigma_1^2 d^2}{M^2 q_V} e^{2 q_V^{-1} K \triangle^2 (1-\triangle)^2 (\frac{-1}{\frac{\eta q_V t}{2 \sigma_1^2 dK}+1}+1)}$$
$$- \frac{C_3 \eta^2 q_V \sigma_0^2}{\sigma_1^2 M^2 K^2} - 4 C_3 \sqrt{\log(16(2K+K')^2/\delta)} \sigma_0^2 d^{1/2},$$

where $\triangle$ is defined in Lemma F.9. Furthermore, when $\mathcal{P}^{tr} \sim \mathcal{P}_{\mathbf{T}}$ or $\mathcal{P}^{tr} \sim \mathcal{P}_{QA}^{\mathbf{T}}$, for $\forall y \in \{\pm 1\}$, there exists $k_y \in [K]$, for $\forall k \in [K]$ we have

$$|\boldsymbol{b}_{k_+}^\top \mathbf{W}_V^{(t)} \boldsymbol{b}_k|, \ |\boldsymbol{b}_{k_-}^\top \mathbf{W}_V^{(t)} \boldsymbol{b}_{k_-}| = \Theta(\boldsymbol{a}_k^\top \mathbf{W}_V^{(t)} \boldsymbol{a}_k). \quad (13)$$

After a sufficient number of epochs beyond $T^\star$, for models trained on $\mathcal{P}_{QA}$, the task vector $\boldsymbol{a}_{k^\star}$ becomes the dominant direction in $\mathbf{h}_{\boldsymbol{\theta}^{(t)},0} = \text{LN}(\mathbf{W}_V^{(t)} \mathbf{T} \boldsymbol{\pi}^{(t)})$ during testing when encountering ICL prompts with $k^\star$ as the co-task concept. In contrast, for models trained on $\mathcal{P}^{tr} \sim \mathcal{P}_{\mathbf{T}}$ and $\mathcal{P}^{tr} \sim \mathcal{P}_{QA}^{\mathbf{T}}$, the components $\pm \boldsymbol{b}_{k_y}$ remain negligible, which supports Theorem 3.2. Building on these learned projections, the proofs for the propositions follow.

## 5. Experiments

In this section, we present simulations of Algorithm 1. For comparison, we use the same parameters for both ICL-trained and QA-trained models and plot the 1-sigma error dynamics, as illustrated in Figures 2 and 3: $K = 2$, $K' = 100$, $d = 3000$, $n = 200$, $M = 30$, $L = L^\star = 30$, $\eta = 5$, $q_V = 10^{-5}$, $\sigma_0 = 10^{-3}$, $\sigma_1 = 5 \times 10^{-3}$, $\sigma_p = \sigma_p^\star = 10^{-2}$. The QA-trained model is trained for $T = 2000$ epochs, while the ICL-trained model undergoes a longer training process with $T = 5000$ epochs. Despite the extended number of iterations, the ICL-trained and QA-ICL-trained model (Figure 2 and Figure 4 in Appendix B) maintain a constant test error, whereas the QA-trained model (Figure 3) converges to zero rapidly. These findings validate our main theorems.

## 6. Conclusion, Limitations, and Future Work

This work presents an optimization theory for non-linear residual transformer's task vector arithmetic mechanisms under empirically-motivated hierarchical modelings. Our analysis framework provides notable merits for such mechanisms, including flexible out-of-distribution generalization and the ability to handle multi-concept words, superior to traditional static Word2Vec predecessors lacking the flexible generalization potential.

However, our framework is currently limited to single-token settings, whereas real-world factual retrieval often requires multi-token reasoning, demanding more sophisticated probing techniques to uncover internal mechanisms *beyond simple geometric relationships* (Allen-Zhu & Li, 2024; Zhang et al., 2025a). Moreover, while our analysis is grounded in empirical modeling, it remains idealized and does not investigate how task vectors naturally emerge in the deeper layers of transformer models (Knittel et al., 2024; Yang et al., 2025). In practice, the representation of high-level concepts is not strictly equivalent to the corresponding task vectors—e.g., the $\vec{o}_{city}$ in Merullo et al. (2024) does not exactly match the latent representation of "Capital". Additionally, real-world vocabulary dictionaries consist of output embeddings of natural language tokens (e.g., words or labels) that are inherently polysemous and evolve over the course of training, rather than remaining static. This dynamic evolution of semantic representations likely co-occurs with the emergence of retrieval capabilities, potentially in a self-reinforcing manner.

Besides, LLMs often rely on more sophisticated or unexplainable operations on task vectors for complex tasks, beyond simple vector arithmetic (Hendel et al., 2023; Todd et al., 2024; Zhang et al., 2025a). While the ground-truth task-vector-leveraging function $f(x, \boldsymbol{a}_{k^\star})$ (Hendel et al., 2023) may lack a closed-form, the *softmax-layernorm-residual* architecture in transformers may implicitly approximate such a function. Future work could explore how these components enable LLMs to interact with task vectors across broader tasks.

## Impact Statement

This paper presents work whose goal is to advance the field of Machine Learning. There are many potential societal consequences of our work, none which we feel must be specifically highlighted here.

## Acknowledgements

We thank the anonymous reviewers for their instrumental comments. DB and HW are supported in part by the Research Grants Council of the Hong Kong Special Administrative Region (Project No. CityU 11206622). WH was supported by JSPS KAKENHI Grant Number 24K20848. TS was partially supported by JSPS KAKENHI (24K02905) and JST CREST (JPMJCR2115). This research is supported by the National Research Foundation, Singapore, Infocomm Media Development Authority under its Trust Tech Funding Initiative, and the Ministry of Digital Development and Information under the AI Visiting Professorship Programme (award number AIVP-2024-004). Any opinions, findings and conclusions or recommendations expressed in this material are those of the author(s) and do not reflect the views of National Research Foundation, Singapore, Infocomm Media Development Authority, and the Ministry of Digital Development and Information.

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

# A. Additional Related Work

**Task Vector Mechanism**. This line of research investigates the emergence of task vectors through controlled experiments. Hendel et al. (2023) and Todd et al. (2024) independently observed the emergence of task or function vectors in tasks such as translation and factual recall. Knittel et al. (2024) demonstrated that GPT-2 effectively learns mechanisms analogous to vector symbolic architectures, where different blocks communicate by writing and reading nearly orthogonal vectors from the residual stream. Liu et al. (2024) proposed a framework that recasts in-context learning as in-context vector manipulation. Yang et al. (2025) explored the emergence and benefits of task vectors in in-context learning (ICL). Finally, Merullo et al. (2024) revealed that large language models (LLMs) implement certain single-token factual-recall tasks through vector arithmetic. However, no existing study explains why and how gradient-update transformer models naturally implement this mechanism.

**Applications of Task Vectors.** Owing to the inherent sparsity in LLM representations, task vector arithmetic has emerged as an efficient and interpretable method for model merging (Lee et al., 2025; Yoshida et al., 2025; Cao et al., 2025), explicitly via arithmetic mean (Wortsman et al., 2022; Ilharco et al., 2023), Fisher information weighting (Matena & Raffel, 2022), regression-based mean (Jin et al., 2023), or learned merging weights (Yang et al., 2024a). Recently, He et al. (2025) introduced a localized merging approach to better exploit task arithmetic. Beyond merging, Li et al. (2025b) demonstrate task vectors' effectiveness for model editing, and Jiang et al. (2025) reveal their promise in mitigating catastrophic forgetting.

**Storage and Retrieval of Factual Knowledge**. Recent research has explored how large language models (LLMs) perform factual recall or associative memory tasks (Meng et al., 2022; Nichani et al., 2025; Cabannes et al., 2024; Ghosal et al., 2024; Allen-Zhu & Li, 2024; Marks & Tegmark, 2024). These studies span a wide range of topics, including the expressiveness of LLMs (Bietti et al., 2023), the optimality of memory storage and optimization dynamics (Cabannes et al., 2024; Nichani et al., 2025), and the empirical mechanisms by which LLMs encode factual knowledge and their inherent limitations. Allen-Zhu & Li (2025) empirically demonstrated the memorization capacity of transformer language models of varying sizes trained on synthetic factual recall tasks, observing near-linear scaling with the number of parameters. (Allen-Zhu & Li, 2024) found that QA data is essential for learning to retrieve factual knowledge in biography data. Subsequently, (Zhang et al., 2025a) further showed that training on QA data, when mixed with Bio data, would increase both quantity and importance of shared parameters compared to training upon biography and QA data in a sequential manner, and (Kahardipraja et al., 2025) revealed that different attention heads play distinct roles–in context heads comprehend instructions and retrieve relevant contextual information, while parametric heads store relational knowledge. Additionally, Park et al. (2024; 2025); Jiang et al. (2024b); Marks & Tegmark (2024); Marconato et al. (2025); Liu et al. (2025) provided both theoretical and empirical evidence that LLMs tend to represent independent concepts, words, and factual statements in a linear manner. Minegishi et al. (2025a) showed the Sparse autoencoders (SAEs) have their limits for representing polysemous contexts, while deeper layers of LLM contributes to distinguishing the polysemy. These observations inspire further investigation into the underlying mechanisms of factual knowledge retrieval.

**Empirical Investigations of ICL Mechanisms.** Garg et al. (2022) showed that Transformer-based in-context learning (ICL) is robust to distribution shifts, motivating a series of studies on out-of-distribution (OOD) generalization (Yadlowsky et al., 2023; Raventós et al., 2023; Ahuja & Lopez-Paz, 2023; Kossen et al., 2024; Pan et al., 2023; Fan et al., 2024; Wang et al., 2025). Yadlowsky et al. (2023) and Wang et al. (2025) analyze mixtures of function classes and show that Transformers struggle to generalize to unseen ones; Kwon et al. (2025) further extend this to mixtures over multiple distributions and provide optimal parameterization for regression. However, these works largely overlook ICL's core capability: inferring the underlying function/task behind demonstrations (i.e., *function reference*). This aligns with the perspective that identifying high-level latent concepts corresponds to task inference, a view supported by empirical findings (Wang et al., 2023). While BMA-based models (Zhang et al., 2025b; Ye et al., 2024) offer a theoretical lens, they rely on unrealistic assumptions that distort Transformer architectures for kernel regression. Empirical work has instead uncovered that LLMs encode task identity via function or task vectors during ICL (Hendel et al., 2023; Todd et al., 2024; Yang et al., 2025; Merullo et al., 2024), and that multi-head attention-only models exhibit multi-phases in forming task circuits (Minegishi et al., 2025b). Further, Kahardipraja et al. (2025) identify distinct roles of in-context and parametric heads in function comprehension and knowledge storage. Yet, these works do not systematically connect task vector mechanisms with BMA-style inference or with OOD generalization in ICL, leaving a gap that we aim to fill.

**Theoretical Investigations of ICL Mechanisms.** A recent line of theoretical research investigates how transformers learn in-context in various scenarios (Zhang et al., 2024; Tian et al., 2023; Nichani et al., 2024; Chen et al., 2024b; Liang et al.,

2025; Shen et al., 2024; Huang & Ge, 2024; Chen et al., 2024a). However, these studies often rely on idealized assumptions, such as linearized or query-key (QK)-combined attention mechanisms (Zhang et al., 2024; Tian et al., 2023; Nichani et al., 2024; Chen et al., 2024b; Oko et al., 2024; Zhang et al., 2025b), or employ simplified loss functions like squared or hinge loss (Chen et al., 2024a; Huang et al., 2024b; Li et al., 2023a; 2024; Chang et al., 2025). Furthermore, rather than investigating how transformers inherently process task demonstrations $\mathbf{T}$, these studies adopt specialized embedding schemes, such as the "up-word-down-label" approach:

$$\mathbf{E}(\mathbf{T}) = \begin{pmatrix} \boldsymbol{x}_1, & \cdots & \boldsymbol{x}_J, & \boldsymbol{x}_{\text{query}} \\ \boldsymbol{y}_1, & \cdots & \boldsymbol{y}_J, & \mathbf{0} \end{pmatrix},$$

where words and labels are processed separately by attention and MLP layers (Bai et al., 2023; Li et al., 2023a; 2024; Bu et al., 2024a; Chang et al., 2025). In contrast, we consider the case where the prompt $\mathbf{T} = [\boldsymbol{x}_1, \boldsymbol{y}_1, \cdots, \boldsymbol{y}_J, \boldsymbol{x}_{\text{query}}] = [\mathbf{T}_1, \cdots \mathbf{T}_L] \in \mathbb{R}^{d \times L}$ $(L = 2J + 1)$ is *directly processed* by a non-linear transformer with a residual stream. Besides, in these work's construction, there must be similar label-related (low-level) patterns in the demonstration pairs, and essentially not showed the *function reference* of transformer models during ICL. For example, given the prompt `Japan Tokyo France Paris China`, the expected output "`Beijing`" reflects the underlying function "`Country's Capital`" applied to the final query, not dependent on prior low-level label semantics in "`Japan`" or "`France`". In addition, none of these theoretical works consider the real-world latent geometric relationship between words and labels depicted in Figure 1, nor do they explain the observed in-context vector arithmetic.

# B. Additional Experiments

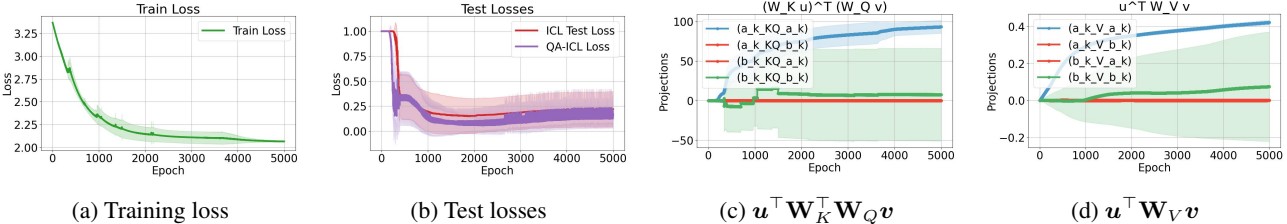

(a) Training loss      (b) Test losses      (c) $\boldsymbol{u}^\top \mathbf{W}_K^\top \mathbf{W}_Q \boldsymbol{v}$      (d) $\boldsymbol{u}^\top \mathbf{W}_V \boldsymbol{v}$

*Figure 4.* Training Dynamics over the QA-ICL Training Distribution. The definitions of legends follows Figure 2. In (d), $\mathbf{W}_V$ overfits to low-level features $\boldsymbol{b}_k$, resulting in persistently constant (0.2) test error shown in (b).

We also conducted experiments on the scenario when the training is conducted on QA-ICL data. For comparison, we adopt the similar parameter configuration : $K = 2$, $K' = 100$, $d = 3000$, $n = 200$, $M = 30$, $L = L^\star = 30$, $\eta = 5$, $q_V = 1e^{-5}$, $\sigma_0 = 1e^{-3}$, $\sigma_1 = 5 \times 1e^{-3}$, $\sigma_p = \sigma_p^\star = 1e^{-2}$, and $T = 5000$.

# C. Model Details

In this section, we present the missing details of our learning problem.

## C.1. Training Algorithm

Our training algorithms for transformer is presented in Algorithm 1.

## C.2. Hierarchical Data Modeling

To model the empirically observed task vector arithmetic mechanism in factual-recall ICL as illustrated in Figure 1, where the answer/label vectors $\boldsymbol{y}$ tend to align more strongly with the task vector $\mathbf{a}_\theta^f$ than with query word vectors $\boldsymbol{x}$, we draw inspiration from recent work Park et al. (2025) on hierarchical concept geometry. This work observes that LLMs tend to encode high- and low-level vectors in an approximately orthogonal manner within their latent space, and that concept vectors at the same level, which are semantically independent, also exhibit mutual orthogonality—an observation that is further supported by both theoretical and empirical studies Jiang et al. (2024b); Yamagiwa et al. (2023); Park et al. (2024). Furthermore, Figure 1 in Park et al. (2025) suggests that word representations reside within polytopes, aligning with the

empirical illustration in Figure 1. Motivated by these findings, a natural theoretical modeling approach is to treat the task vector as a high-level concept vector, while the components orthogonal to the task vector correspond to task-specific low-level concept vectors, which we define as follows.

**Definition C.1. Word-Label Pair ICL Prompt Distribution** ($\mathcal{P}_{\mathbf{T}}$). The prompt $\mathbf{T} := [\mathbf{x}_1, \mathbf{y}_1, \cdots, \mathbf{x}_J, \mathbf{y}_J, \mathbf{x}_{J+1}] \in \mathbb{R}^{d \times (2J+1)}$ and $\mathbf{y}_{J+1} \in \mathbb{R}^d$ is generated as follows. Each word-label pair in $\mathbf{T}$ shares a high-level task concept with uniform probability, and each pair has an equal chance of possessing a positive or negative low-level label feature. Specifically, for all $l \in [J+1]$, the word and label vectors are defined as:

$$
\begin{aligned}
& k_{\mathbf{T}} \sim \mathrm{Unif}[K], \ y_{k_{\mathbf{T}},l} \sim \mathrm{Unif}[\pm 1], \ \boldsymbol{\xi}_{l,\mathbf{x}}, \boldsymbol{\xi}_{l,\mathbf{y}} \sim \mathcal{N}(\mathbf{0}, \sigma_p^2 \mathbb{I}), \\
& \mathcal{X}_{\mathbf{T},l} = \{k_{\mathbf{T}}\} \cup \{i \in [K] \setminus \{k_{\mathbf{T}}\} : B_i \sim \mathrm{Ber}(K^{-1}), B_i = 1\}. \\
& \mathcal{Y}_{\mathbf{T},l} = \{k_{\mathbf{T}}\} \cup \{i \in [K] \setminus \{k_{\mathbf{T}}\} : B_i \sim \mathrm{Ber}(K^{-1}), B_i = 1\}. \\
& y_{k_l,l} \sim \mathrm{Unif}[\pm 1], \quad \forall k_l \in \mathcal{X}_{\mathbf{T},l} \setminus \{k_{\mathbf{T}}\}, \\
& \mathbf{x}_l := \Big( \sum_{k_l \in \mathcal{X}_{\mathbf{T},l}} x_a \cdot \boldsymbol{a}_{k_l} + y_{k_l,l} \cdot \boldsymbol{b}_{k_l} \Big) + \boldsymbol{\xi}_{l,\mathbf{x}}, \\
& \mathbf{y}_{l_y} := \Big( \sum_{k_{l_y} \in \mathcal{Y}_{\mathbf{T},l_y}} \boldsymbol{a}_{k_{l_y}} + y_{k_{l_y},l_y} \cdot \boldsymbol{b}_{k_{l_y}} \Big) + \boldsymbol{\xi}_{l_y,\mathbf{y}}, \ \forall l_y < J+1 \\
& \mathbf{y}_{J+1} := \boldsymbol{a}_{k_{\mathbf{T}}} + y_{k_{\mathbf{T}},J+1} \cdot \boldsymbol{b}_{k_{\mathbf{T}}},
\end{aligned}
\tag{14}
$$

where the noise vector $\boldsymbol{\xi}_l$ accounts for the inherent semantic inexactness of latent representation for language. The expected label vector of the prompt, $\mathbf{y}_{J+1}$, is constrained by the co-task concept $k_{\mathbf{T}}$ of the prompt and remains noiseless to ensure an accurate semantic representation of the concept-specific answer. The term $x_a \cdot \boldsymbol{a}_k$ acts as an anchor, guiding the transformer to retrieve high-level task concepts, with $x_a \|\boldsymbol{a}_k\| / \|\boldsymbol{b}_k\| \le o(1)$. For simplicity, we set $x_a := 0.1$.

*Remark* C.2. To model *polysemy*, each word vector $\mathbf{x}_{l_x}$ ($l_x \in [J+1]$) and label vector $\mathbf{y}_{l_y}$ ($l_y \in [J]$) is associated with multiple task concepts. However, all tokens in a prompt share the co-task concept $k_{\mathbf{T}}$. For example, "*Apple*" may relate to both "*Color*" and "*Species*" as task concepts, with labels "*Red*" and "*Fruit*", respectively. Likewise, "*Fruit*" may indicate "*Outcome*" in contexts unrelated to "*Color*". To reduce the freedom of our learning problem, we assume the ground truth label follows a single-concept noiseless form:

$$
\mathbf{y}_{J+1} = \boldsymbol{a}_{k_{\mathbf{T}}} + y_{k_{\mathbf{T}},J+1} \cdot \boldsymbol{b}_{k_{\mathbf{T}}}.
$$

This is reasonable, as $k_{\mathbf{T}}$ naturally disambiguates label meanings. For instance, while "*Fruit*" could mean "*Outcome*", under the "*Species*" task, its meaning is unambiguous.

**Failure of ICL-trained Transformer**. Under the above definitions, we conduct experiments across various parameter settings in the procedure of Algorithm 1. However, as illustrated in Figure 2, the test loss always fails to converge, remaining at a constant level. Moreover, the projection dynamics indicate that the value matrix $\mathbf{W}_V$ learns low-level concepts in a disorganized manner, suggesting that $\mathbf{h}_{\boldsymbol{\theta},0}(\mathbf{T})$ becomes a hybrid vector containing both $\boldsymbol{a}_{k_{\mathbf{T}}}$ and $\pm \boldsymbol{b}_{k_{\mathbf{T}}}$. Consequently, this compromises its role as approximating a task vector: the presence of low-level concept components $e\boldsymbol{b}_{k_{\mathbf{T}}}, e \in \{\pm 1\}$, within $\mathbf{h}_{\boldsymbol{\theta},0}(\mathbf{T})$ disrupts the vector arithmetic in Eq. (3), biasing it toward a specific $e\boldsymbol{b}_{k_{\mathbf{T}}}$. This leads to degraded predictions, particularly when $\mathbf{x}_{J+1}$ contains $-e\boldsymbol{b}_{k_{\mathbf{T}}}$.

**Reason of Failure: ICL Data vs. QA Data**. At a high level, the failure arises from unintended co-occurrences between low-level features in the demonstration pairs and those in the query words. In contrast, QA data contains only the task vector and irrelevant tokens in $\mathbf{x}_{\mathrm{QA}}$, avoiding the possibility of such co-occurrences. According to Yang et al. (2024b), transformers tend to encode fixed co-occurrence patterns via gradient descent. Moreover, training and testing solely on the ICL data distribution—though common in theoretical studies (Zhang et al., 2024; Kim & Suzuki, 2024; Chen et al., 2024a; Bu et al., 2024a)—is unrealistic in real-world settings. A more practical training paradigm involves generative pretraining or QA-style pretraining/fine-tuning (Allen-Zhu & Li, 2024; Zhang et al., 2025a). Notably, our concept data modeling **naturally arises** from generative pretraining (Park et al., 2024; Jiang et al., 2024b; Park et al., 2025). Furthermore, QA data has been shown to **enhance** transformers' factual retrieval capabilities (Allen-Zhu & Li, 2024; Zhang et al., 2025a), and has been effectively leveraged to extract task vectors (Merullo et al., 2024). Motivated by these observations, we incorporate QA data into our training framework, which we formalize as follows.

**Definition C.3. QA Sentence Distribution**[4] ($\mathcal{P}_{\text{QA}}$). Building on the findings of Allen-Zhu & Li (2024); Nichani et al. (2025), which imply that incorporating high-level task messages or relation tokens within questions significantly enhances a transformer's ability to *retrieve* relevant factual knowledge, we model our QA-type sentence $\mathbf{S}_n := [\mathbf{x}_n^{\text{QA}}, \mathbf{y}_n] \in \mathbb{R}^{d \times (M+2)}$ from distribution $\mathcal{P}_{\text{QA}}$ as follows.

$$
\begin{aligned}
&k_{\mathbf{S}_n} \sim \text{Unif}[K], \; y_{k_{\mathbf{S}_n}} = y_n \sim \text{Unif}[\pm 1], \;\; \boldsymbol{\xi}_{n,1:M}, \boldsymbol{\xi}_{n,\mathbf{x}} \sim \mathcal{N}(\mathbf{0}, \sigma_p^2 \mathbb{I}), \\
&\mathcal{X}_{\mathbf{S}_n} = \{k_{\mathbf{S}}\} \cup \{k \in [K] \setminus \{k_{\mathbf{S}_n}\} : B_k \sim \text{Ber}(K^{-1}), B_k = 1\}. \\
&y_{k_n} \sim \text{Unif}[\pm 1], \forall k_l \in \mathcal{X}_{\mathbf{S}_n} \setminus \{k_{\mathbf{S}_n}\}, \\
&\mathbf{x}_n := \Big( \sum_{k_n \in \mathcal{X}_{\mathbf{S}_n}} x_a \cdot \boldsymbol{a}_{k_n} + y_{k_n} \cdot \boldsymbol{b}_{k_n} \Big) + \boldsymbol{\xi}_{n,\mathbf{x}}, \\
&\mathbf{y}_n := \boldsymbol{a}_{k_{\mathbf{S}}} + y_n \cdot \boldsymbol{b}_{k_{\mathbf{S}}}, \\
&m_{\mathbf{S}_n} \sim \text{Unif}[M], \; i_{1:M} \sim \text{Unif}[K'], \; \boldsymbol{\nu}_{n,1:M} := \boldsymbol{\nu}_{i_{1:M}}, \\
&\mathbf{x}_n^{\text{QA}} := [\boldsymbol{\nu}_{n,1} + \boldsymbol{\xi}_{n,1}, \cdots, \boldsymbol{\nu}_{n,m_{\mathbf{S}_n}-1} + \boldsymbol{\xi}_{n,m_{\mathbf{S}_n}-1}, \; \boldsymbol{a}_{k_{\mathbf{S}_n}} + \boldsymbol{\xi}_{n,m_{\mathbf{S}_n}}, \\
&\qquad\quad \boldsymbol{\nu}_{n,m_{\mathbf{S}_n}+1} + \boldsymbol{\xi}_{n,m_{\mathbf{S}_n}+1} \cdots, \boldsymbol{\nu}_{n,M} + \boldsymbol{\xi}_{n,M}, \; \mathbf{x}] \in \mathbb{R}^{d \times (M+2)},
\end{aligned}
\tag{15}
$$

where $K'$ denotes the number of task-irrelevant tokens, $\boldsymbol{\nu}_{n,1:M}$ denotes common tokens (e.g., task-irrelevant tokens) that appear uniformly across different tasks $k \in [K]$, and the position of $\boldsymbol{a}_k$ can be arbitrary before $\mathbf{x}$.

An illustrative example with $M = 4$ on concept $k$ is:

$$
\underset{\boldsymbol{\nu}_{n,1}+\boldsymbol{\xi}_1}{\text{What}} \; \underset{\boldsymbol{\nu}_{n,2}+\boldsymbol{\xi}_2}{\text{is}} \; \underset{\boldsymbol{\nu}_{n,3}+\boldsymbol{\xi}_3}{\text{the}} \; \underset{\boldsymbol{\nu}_{n,4}+\boldsymbol{\xi}_4}{\boldsymbol{a}_k} \; \underset{x_a \cdot \boldsymbol{a}_k + e\boldsymbol{b}_k + \boldsymbol{\xi}_{n,M}}{\text{of}} \underset{\substack{x_a \cdot \boldsymbol{a}_k + e\boldsymbol{b}_k + \boldsymbol{\xi}_{n,M} \\ +\cdots}}{\mathbf{x}} \; \underset{\boldsymbol{a}_k + e\boldsymbol{b}_k}{\mathbf{y}} \; .
$$

The index $(n, m)$, where $n \in [N]$ and $m \in [M]$, denotes the sample index ($n$-th) and the position index ($m$) for the noise vector $\boldsymbol{\xi}_{n,m}$. **Complexity of QA and ICL Tasks**. In our setting involving retrieval over a hierarchical concept graph, one natural way to measure task complexity is by the model's difficulty in confidently producing its argmax prediction. A simple metric is $C(\mathbf{T}) = 1/\max_y p_{\boldsymbol{\theta}}(y|\mathbf{T})$, where a higher $C(\mathbf{T})$ indicates greater complexity. QA tasks tend to be simpler: a cue word (e.g., "capital" in "What is the capital of Japan?") guides $\boldsymbol{\theta}$ to retrieve the relevant task in collaboration with the query word, often resulting in lower $C(\mathbf{T})$. In contrast, Word-Label tasks lack such explicit cues, requiring $\boldsymbol{\theta}$ to infer the task underlying the word-label pair. For instance, given the prompt $\mathbf{T} = $ [Japan, Sakura, China], the model may assign non-trivial confidence to both "Panda" and "Peony" (e.g., struggling in choosing "National Symbol" or "National Flower" as the tasks). This reflects the polysemy-induced challenge posed by the hierarchical conceptual knowledge embedded in natural tokens.

**QA-ICL Prompt Distribution**. ($\mathcal{P}_{\text{QA}}^{\mathbf{T}}$). A natural extension of the above two distributions is that we can replace the word-based demonstration by QA-based demonstration in the task-specific prompt, namely $\mathbf{T}_{\text{QA}} := [\mathbf{x}_1^{\text{QA}}, \mathbf{y}_1, \cdots, \mathbf{x}_J^{\text{QA}}, \mathbf{y}_J, \mathbf{x}_{J+1}^{\text{QA}}] \in \mathbb{R}^{d \times (J+1)(M+2)} \sim \mathcal{P}_{\mathbf{T}_{\text{QA}}}$.

**Success of QA-Trained Transformers**. Unlike ICL-trained scenarios, we observe the success of QA-trained dynamics, as shown in Figure 3, where the column and row projections of the value matrix $\mathbf{W}_V$ primarily align with the task vector directions. As a result, $\mathbf{h}_{\boldsymbol{\theta},0}(\mathbf{T})$ closely approximates the task concept vector, leading to an arbitrarily small test loss for both ICL and QA-ICL data after sufficient training epochs. Interestingly, as depicted in Figure 4, training a transformer with QA-ICL data induces harmful hybrid task vector generation, yet its test error remains at a similar level to that observed when training on ICL data. These findings motivate a rigorous theoretical analysis of the underlying optimization dynamics and the test convergence guarantees, which will be explored in detail in the following section.

**ICL and QA Tasks**. Given a prompt $\mathbf{T}$ or an QA sentence $\mathbf{S}$, where we denote the last token as the residual query vector: $\mathbf{b}_{\boldsymbol{\theta}}^{\text{query}}$ with some $y \in [\pm 1], k \in [K]$. We expect the transformer model can return the original low-level semantic label vector $y \cdot \boldsymbol{b}_k$ by argmax sampling from the undisturbed normalized token dictionary

$$
\begin{aligned}
\mathbf{U} := \Big[ & \frac{\boldsymbol{a}_1 + \boldsymbol{b}_1}{\|\boldsymbol{a}_1 + \boldsymbol{b}_1\|}, \frac{\boldsymbol{a}_1 - \boldsymbol{b}_1}{\|\boldsymbol{a}_1 - \boldsymbol{b}_1\|}, \frac{x_a \cdot \boldsymbol{a}_1 + \boldsymbol{b}_1}{\|x_a \cdot \boldsymbol{a}_1 + \boldsymbol{b}_1\|}, \frac{x_a \cdot \boldsymbol{a}_1 - \boldsymbol{b}_1}{\|x_a \cdot \boldsymbol{a}_1 - \boldsymbol{b}_1\|}, \frac{\boldsymbol{a}_1}{\|\boldsymbol{a}_1\|}, \frac{\boldsymbol{b}_1}{\|\boldsymbol{b}_1\|}, \frac{-\boldsymbol{b}_1}{\|\boldsymbol{b}_1\|}, \cdots, \\
& \frac{\boldsymbol{b}_K}{\|\boldsymbol{b}_K\|}, \frac{-\boldsymbol{b}_K}{\|\boldsymbol{b}_K\|}, \cdots, \frac{\boldsymbol{\nu}_1}{\|\boldsymbol{\nu}_1\|}, \cdots, \frac{\boldsymbol{\nu}_{K'}}{\|\boldsymbol{\nu}_{K'}\|} \Big] = [\mathbf{u}_1, \cdots, \mathbf{u}_{7K+K'}] \in \mathbb{R}^{d \times (7K+K')}
\end{aligned}
\tag{16}
$$

---

[4]Indeed, the formula of the sentence is unnecessarily a QA, but can be a factual statement (e.g. $\mathbf{S} = [\mathbf{x}^{\text{QA}}, \mathbf{y}] = [\underset{\boldsymbol{\nu}_1}{\text{The}}, \boldsymbol{a}_k, \underset{\boldsymbol{\nu}_4}{\text{of}}, \mathbf{x}, \mathbf{y}]$).

The normalization of the noiseless token in the dictionary is to ensure that every token $\mathbf{u}_k, \forall k \in [K]$ enjoys the same length fairly. Similar empirically driven dictionary approaches have also been adopted in recent theoretical work (Tian et al., 2023; 2024). For simplicity here we let $\|\boldsymbol{a}_k\| = \|\mathbf{a}\| = \|\boldsymbol{b}_k\| = \|\mathbf{b}\| = \|\boldsymbol{\nu}_m\| = 1$. Given a label vector $\mathbf{y} = \boldsymbol{a}_k + y \cdot \boldsymbol{b}_k + \boldsymbol{\xi}$, its corresponding dictionary vector is defined as:

$$\mathbf{u}_{k_\mathbf{y}} = \mathrm{LN}\left(\boldsymbol{a}_{\left\lceil \frac{k_\mathbf{y}}{7} \right\rceil} + \left(2\left(\frac{k_\mathbf{y}}{7} - \left\lceil \frac{k_\mathbf{y}}{7} \right\rceil\right) - 1\right) \boldsymbol{b}_{\left\lceil \frac{k_\mathbf{y}}{7} \right\rceil}\right),$$

where the dictionary index is determined by:

$$k_\mathbf{y} = 7k + \frac{y+1}{2}.$$

**Connection to Word-2-Vector Arithmetic**. In the context above, when $|\mathcal{X}_{\mathbf{T},l}|, |\mathcal{Y}_{\mathbf{T},l}|$ and the noise length are limited, we approximately have:

$$\mathbf{y}_{J+1} \approx \boldsymbol{a}_{k_\mathbf{T}} + \mathbf{x}_{J+1} \tag{17}$$

Therefore, if the transformer $\boldsymbol{\theta}$ can extract the high-level task vector $\mathbf{h}_{\boldsymbol{\theta},0}(\mathbf{T})$, then adding it to **any word vector within the same task concept** should yield the task-specific label vector in the context of argmax sampling. This aligns with empirical findings by Merullo et al. (2024), which show that adding various embedded query words, such as "Poland" or "China," to the vector $\vec{o}_{city}$—which captures the function `get_capital(·)` in the latent space—produces the correct capital city as the answer.

## D. Preliminary Lemmas

### D.1. Continuous Flows

In this section, we present several lemmas whose continuous flows serve as surrogate processes to bound the discrete training dynamics. While similar but simpler approaches have been explored in Tian et al. (2023); Meng et al. (2024); Bu et al. (2024a), our analysis considers a significantly larger number of more complex flows. This extension is necessitated by the inherent complexity of our model, which incorporates softmax attention, layer-wise normalization, and residual stream structures—features that make our setting more realistic than prior work.

The key idea of using surrogate continuous flows is to leverage the monotonicity of differential equation derivatives. By comparing earlier derivatives with their integrals over a time interval, we establish upper and lower bounds for the discrete process via its continuous counterpart.

**Lemma D.1.** *Suppose that a sequence $a_t, t_0 \leq t \leq t_1$ follows the iterative formula*

$$a_{t+1} = a_t + b,$$

*for some constant $b > 0$. Then it holds that*

$$a_t = bt + a_0.$$

*for all $t_0 \leq t \leq t_1$.*

*Proof.* The proof is direct by the monotonicity of the linear function and Comparison Theorem. $\square$

**Lemma D.2.** *Suppose that a positive sequence $b_t > 0, t_1 \leq t \leq t_2$ follows the iterative formula*

$$b_{t+1} = b_t + \frac{ab_t}{ct+d},$$

*for some constant $a > c > 0, d > 0$. Then it holds that*

$$b_t \leq \left(\frac{b_{t_1}}{(t_1 + \frac{d}{c})^{\frac{a}{c}}}\right)\left(t + \frac{d}{c}\right)^{\frac{a}{c}}.$$

*for all $t_1 \leq t \leq t_2$.*

*Proof.* Consider a continuous-time sequence $x_t, t > t_1$ defined by the integral equation

$$x_t = x_{t_1} + a \int_{t_1}^{t_2} \frac{x_\tau}{c\tau + d} d\tau, \ x_{t_1} = b_{t_1}. \tag{18}$$

Here, $x_t$ is an increasing function over $t$, such that

$$\frac{dx_t}{dt} = \frac{ax_\tau}{c\tau + d}, \ x_{t_1} = b_{t_1}. \tag{19}$$

To solve this ODE system, by separation of variables we first have

$$d \log(x_t) = d \log((t + \frac{d}{c})^{\frac{a}{c}}),$$

which then leads to

$$\log(x_t) = \log((t + \frac{d}{c})^{\frac{a}{c}}) + \text{constant}.$$

To finalize the constant, we extend $x_{t_1} = b_{t_1}$ into the formula, and have

$$x_t = (\frac{b_{t_1}}{(t_1 + \frac{d}{c})^{\frac{a}{c}}})(t + \frac{d}{c})^{\frac{a}{c}},$$

which is unique by the monotonicity of log function and the positiveness of $a, c, d$. Note that $a > c$, define $f(t) = \frac{\left(t + \frac{d}{c}\right)^{a/c}}{ct + d}$.
To determine whether $f(t)$ is increasing, we compute its derivative $f'(t)$ and analyze its sign. We use the quotient rule:

$$f'(t) = \frac{g'(t)(ct + d) - g(t)c}{(ct + d)^2},$$

where $g(t) = \left(t + \frac{d}{c}\right)^{a/c}$. Thus,

$$g'(t) = \frac{a}{c}\left(t + \frac{d}{c}\right)^{\frac{a}{c} - 1}.$$

Substituting $g(t)$ and $g'(t)$, we obtain:

$$f'(t) = \frac{\frac{a}{c}\left(t + \frac{d}{c}\right)^{\frac{a}{c} - 1}(ct + d) - c\left(t + \frac{d}{c}\right)^{a/c}}{(ct + d)^2}.$$

We can simplify the numerator:

$$f'(t) = \frac{\left(t + \frac{d}{c}\right)^{\frac{a}{c} - 1}}{(ct + d)^2} \cdot \left[\frac{a}{c}(ct + d) - c\left(t + \frac{d}{c}\right)\right].$$

Focus on the bracketed term:

$$\Delta = \frac{a}{c}(ct + d) - c\left(t + \frac{d}{c}\right).$$

Expanding $\Delta$:

$$\Delta = \frac{a}{c}ct + \frac{a}{c}d - ct - \frac{cd}{c}.$$

Simplify:

$$\Delta = (a - c)t + \frac{ad}{c} - d.$$

Therefore, given $a > c$: 1. $(a - c)t > 0$, since $t > 0$. 2. $\frac{ad}{c} - d > 0$, as $a > c$ ensures $\frac{a}{c} > 1$.

Thus, $\Delta > 0$, implying $f'(t) > 0$, and $f(t)$ is strictly increasing.

$$x_{t+1} = x_t + \int_t^{t+1} \frac{ax_\tau}{c\tau + d} d\tau \geq x_t + \int_t^{t+1} \frac{ax_t}{ct + d} d\tau,$$

where the inequality is by examine the monotonicity of $f(t) = \frac{\left(t + \frac{d}{c}\right)^{a/c}}{ct + d}$. By Comparison Theorem of two positive series, we obtain $b_t \leq x_t = \left(\frac{b_{t_1}}{(t_1 + \frac{d}{c})^{\frac{a}{c}}}\right)(t + \frac{d}{c})^{\frac{a}{c}}$. $\qquad\square$

**Lemma D.3.** *Suppose that a sequence* $c_t \geq 0, t_1 \leq t \leq t_2$ *follows the iterative formula*

$$c_{t+1} = c_t + \frac{d}{c_t},$$

*for some constant* $d > 0$*. Then it holds that*

$$x_t \leq c_t \leq \frac{d}{c_{t_1}} + x_t$$

*for all* $t_1 \leq t \leq t_2$*. Here,* $x_t = \sqrt{d(t - t_1) + c_{t_1}^2}$*.*

*Proof.* The proof strategy follows Lemma C.1 in Meng et al. (2024) despite the consideration of different ODE processes. The key is the decaying and positive nature of $g(x) = 1/x$ as well as the examination of integration. Consider a continuous-time sequence $x_t, t > t_1$ defined by the integral equation

$$x_t = x_{t_1} + d\int_{t_1}^{t_2} \frac{d\tau}{x_\tau}, x_{t_1} = c_{t_1} \tag{20}$$

Note that $x_t$ is an increasing function of $t$, satisfying

$$\frac{dx_t}{dt} = \frac{d}{x_t}, x_{t_1} = c_{t_1}. \tag{21}$$

By solving this ODE system, we have

$$x_t = \sqrt{d(t - t_1) + c_{t_1}^2},$$

which is unique. We first show the lower bound of $c_t$, by Eq.(20) we have

$$x_{t+1} = x_t + d\int_t^{t+1} \frac{d\tau}{x_\tau}$$

$$\leq x_t + d\int_t^{t+1} \frac{d\tau}{x_t} = x_t + \frac{d}{x_t}.$$

where the inequality is by the increasing nature of $x_t$ and the decreasing nature of $g(x) = 1/x$. Then by Comparison Theorem we have $x_t \leq a_t$. On the other side, we have

$$c_t = c_{t_1} + \sum_{\tau = t_1}^{t} \frac{d}{c_\tau},$$

$$\leq c_{t_1} + \sum_{\tau = t_1}^{t} \frac{d}{x_\tau}$$

$$= c_{t_1} + \frac{d}{c_{t_1}} + \sum_{\tau = t_1 + 1}^{t} \frac{d}{x_\tau}$$

$$\leq c_{t_1} + \frac{d}{c_{t_1}} + d\int_{t_1}^{t} \frac{d\tau}{x_\tau}$$

$$= c_{t_1} + \frac{d}{c_{t_1}} + \int_{t_1}^{t} dx_\tau = c_{t_1} + \frac{d}{c_{t_1}} + x_t - x_{t_1}$$

$$= \frac{d}{c_{t_1}} + x_t.$$

Here, the first inequality is by $c_t \geq x_t$; the second inequality is by the definition of integration as well as the decreasing nature of $g(x) = 1/x$, the third equality is by Eq.(21). The proof is completed. □

**Lemma D.4.** *Suppose that a sequence $e_t, e_0 > 0, 0 \leq t \leq t_3$ follows the iterative formula*

$$e_{t+1} \geq e_t - be_t,$$

*for some constant $b > 0$. Then it holds that*

$$e_t \geq -be_0 t_3 + e_0$$

*for all $0 \leq t \leq t_3$.*

*Proof.* It is obvious that $e_t$ is decreasing, and since $e_0 > 0$, the decreasing speed during $t \leq t_3$ would not exceed $-be_0$. By the monotonicity of the linear function, the result follows directly from the Comparison Theorem. □

**Lemma D.5.** *Suppose that a sequence $f_t, f_{t_3} > 0, t_3 \leq t \leq t_4$ follows the iterative formula*

$$f_{t+1} = f_t + a(t - t_3)f_t,$$

*for some constant $a > 0$. Then it holds that*

$$f_{t_3} + \frac{af_{t_3}[(t-1-t_3)^2 - 1]}{2} \leq f_t \leq f_{t_3} \exp(at^2/2)$$

*for all $t_3 \leq t \leq t_4$.*

*Proof.* Obviously $f_t$ is an increasing sequence. First we can set $s = t + t_3$ by Substitution Method. Similar to Lemma D.3, by the definition of integral, the monotonicity of $g(x) = \exp(x)$ and Comparison Theorem, the continuous flow $x_s$ satisfying

$$\frac{dx_s}{ds} = ax_s, \ x_0 = f_{t_3}, \ 0 \leq s \leq t_4 - t_3$$

would be the upper bound of $f_s$. By solving the ODE and consider the monotonicity of $g(x) = \exp(x)$, the unique upper bound result follows.

For the lower bound, observing that

$$
\begin{aligned}
f_t &= f_{t_3} + \sum_{\tau=t_3}^{t} a(\tau - t_3)f_\tau \\
&\geq f_{t_3} + \sum_{\tau=t_3}^{t} a(\tau - t_3)f_{t_3} \\
&\geq f_{t_3} + \int_{t_3-1}^{t-1} a(\tau - t_3)f_{t_3}d\tau \\
&= f_{t_3} + \frac{af_{t_3}[(t-1-t_3)^2 - 1]}{2}.
\end{aligned}
\tag{22}
$$

Here, the first inequality is by the increasing nature of $f_t$; the second inequality is by the definition of integral and the increasing nature of $g(x) = ax$; the last equality is by the integration of linear function. The proof is complete. □

**Lemma D.6.** *Suppose that a positive sequence $g_t > 0, t_1 \leq t \leq t_2$ follows the iterative formula*

$$g_{t+1} = g_t + \frac{ag_t}{(bt+c)^2},$$

*for some constant $a, b, c > 0$ and satisfies $a < 2c$. Then it holds that*

$$g_t \geq g_{t_1} e^{-\frac{a}{b(bt+c)} + \frac{a}{b(bt_1+c)}}.$$

*for all $t_1 \leq t \leq t_2$.*

*Proof.* Consider a continuous-time sequence $x_t, t > t_1$ defined by the integral equation

$$x_t = x_{t_1} + \int_{t_1}^{t_2} \frac{ax_\tau}{(b\tau + c)^2} d\tau, \; x_{t_1} = g_{t_1}. \tag{23}$$

Here, $x_t$ is an increasing function over $t$, such that

$$\frac{dx_t}{dt} = \frac{ax_t}{(bt + c)^2}, \; x_{t_1} = g_{t_1}. \tag{24}$$

Then, by separating the variables we have

$$d\log(x_t) = \frac{-a}{b} d\frac{1}{bt + c}.$$

Therefore, it holds that

$$\log(x_t) = \frac{-a}{b(bt + c)} + \text{constant}.$$

To settle the unique constant, we extend $x_{t_1} = g_{t_1}$ into the formula, and have

$$x_t = g_{t_1} e^{-\frac{a}{b(bt+c)} + \frac{a}{b(bt_1+c)}},$$

which is unique by the monotonicity of exponential function and the positiveness of $a, c, d$. Note that $a < 2c$, define $f(t) = \frac{e^{-\frac{a}{b(bt+c)}}}{(bt+c)^2}$. To determine whether $f(t)$ is increasing, we compute its derivative $f'(t)$ and analyze its sign. We use the quotient rule and have:

$$f'(t) = \left(-2 + \frac{a}{(bt+c)}\right) \frac{e^{-\frac{a}{b(bt+c)}}}{(bt+c)^2} < 0.$$

As such, it holds that

$$x_{t+1} = x_t + \int_t^{t+1} \frac{ax_\tau}{(b\tau + c)^2} d\tau \leq x_t + \int_t^{t+1} \frac{ax_t}{(bt+c)^2} d\tau,$$

where the inequality is by examine the declining monotonicity of $f(t) = \frac{e^{-\frac{a}{b(bt+c)}}}{(bt+c)^2}$. By Comparison Theorem of two positive series, $g_t \geq g_{t_1} e^{-\frac{a}{b(bt+c)} + \frac{a}{b(bt_1+c)}}$ immediately holds. $\qquad \square$

### D.2. Concentration Inequaities

**Lemma D.7.** *Suppose that $\delta > 0$ and $d = \Omega(\log((2K + K')^2 N^2 M^2/\delta))$. Then with probability at least $1 - \delta$, for $\forall \xi_l, \xi'_l \in \{\xi_{n,m}\}_{n\in[N],m\in[M]}, l \in [M], u \in \{a_s\}_{s\in[K]} \cup \{b_s\}_{s\in[K]} \cup \{\nu_{k'}\}_{k'\in[K']}.$*

$$\frac{\sigma_p^2 d}{2} \leq \|\xi_l\|_2^2 \leq 3\frac{\sigma_p^2 d}{2},$$

$$|\langle \xi_i, \xi_{l'} \rangle| \leq 2\sigma_p^2 \cdot \sqrt{d \log\left(\frac{7(NM)^2}{\delta}\right)},$$

$$|\langle \xi_i, u \rangle| \leq \|u\|_2 \sigma_p \cdot \sqrt{2\log(\frac{3(2K + K')NM}{\delta})}.$$

*Proof.* See Lemma B.2 in Cao et al. (2022) for a proof. $\qquad \square$

**Lemma D.8.** *Suppose that $\delta > 0$ and $d = \Omega(\log((2K + K')^2 N^2 M^2/\delta))$. Then with probability at least $1 - \delta$, for*

$\forall \boldsymbol{u}, \boldsymbol{v} \in \{\boldsymbol{a}_s\}_{s \in [K]} \cup \{\boldsymbol{b}_s\}_{s \in [K]} \cup \{\boldsymbol{\nu}_{k'}\}_{k' \in [K']}, \boldsymbol{\xi}_l, \boldsymbol{\xi}_l' \in \{\boldsymbol{\xi}_{n,m}\}_{n \in [N], m \in [M]}$

$$\|\mathbf{W}_V^{(0)}\|_F^2 \leq 2d^2 \sigma_1^2,$$

$$|\boldsymbol{v}^\top \mathbf{W}_V^{(0)\top} \boldsymbol{u}| \leq \sqrt{2 \log(\frac{8(2K+K')^2}{\delta})} \sigma_1 \|\boldsymbol{u}\| \|\boldsymbol{v}\|,$$

$$0.99 {\sigma_1}^2 \|\boldsymbol{u}\|^2 d \leq \|\mathbf{W}_V^{(0)} \boldsymbol{u}\|^2 \leq 1.01 {\sigma_1}^2 \|\boldsymbol{u}\|^2 d,$$

$$|\boldsymbol{v}^\top \mathbf{W}_V^{(0)} \boldsymbol{u}| / \|\mathbf{W}_V^{(0)} \boldsymbol{u}\| \leq 4\sqrt{2 \log(\frac{8(2K+K')^2}{\delta})} \|\boldsymbol{v}\| / \sqrt{d},$$

$$|\boldsymbol{v}^\top \mathbf{W}_V^{(0)\top} \mathbf{W}_V^{(0)} \boldsymbol{u}| \leq 4\sqrt{\log(16(2K+K')^2/\delta)} \sigma_1^2 d^{1/2} \|\boldsymbol{u}\| \|\boldsymbol{v}\|,$$

$$|\boldsymbol{v}^\top \mathbf{W}_V^{(0)\top} \boldsymbol{\xi}_l| \leq 2\sqrt{2 \log(16(2K+K')NM/\delta)} \sigma_1 \sigma_p d^{1/2} \|\boldsymbol{u}\|,$$

$$|\boldsymbol{\xi}_l'^\top \mathbf{W}_V^{(0)\top} \boldsymbol{\xi}_l| \leq 4\sqrt{2 \log(16(2K+K')NM/\delta)} \sigma_1 \sigma_p^2 d,$$

$${\sigma_1}^2 {\sigma_p}^2 d^2 \leq \|\mathbf{W}_V^{(0)} \boldsymbol{\xi}_l\|^2 \leq 3{\sigma_1}^2 {\sigma_p}^2 d^2,$$

$$|\boldsymbol{v}^\top \mathbf{W}_V^{(0)} \boldsymbol{\xi}_l| / \|\mathbf{W}_V^{(0)} \boldsymbol{\xi}_l\| \leq 8\sqrt{2 \log(16(2K+K')^2/\delta)} \|\boldsymbol{v}\| / \sqrt{d},$$

$$|\boldsymbol{\xi}_l'^\top \mathbf{W}_V^{(0)} \boldsymbol{\xi}_l| / \|\mathbf{W}_V^{(0)} \boldsymbol{\xi}_l\| \leq 8\sqrt{2 \log(16(2K+K')^2/\delta)} \sigma_p,$$

$$|\boldsymbol{\xi}_l^\top \mathbf{W}_V^{(0)\top} \mathbf{W}_V^{(0)} \boldsymbol{\xi}_l'| \leq 16\sqrt{\log(16(2K+K')^2/\delta)} \sigma_1^2 \sigma_p^4 d^{3/2}$$

*Proof.* The proof strategies follow Lemma B.3 in Cao et al. (2022).

We analyze the Frobenius norm squared of $\mathbf{W}_V^{(0)}$, where $\mathbf{W}_V^{(0)}$ is a $d \times d$ matrix with entries independently sampled from $N(0, \sigma_1^2)$. The Frobenius norm squared is:

$$\|\mathbf{W}_V^{(0)}\|_F^2 = \sum_{i,j=1}^d W_{i,j}^2.$$

Each $W_{i,j}^2$ is an independent chi-squared random variable with mean $\sigma_1^2$ and variance $2\sigma_1^4$. The sum $\|\mathbf{W}_V^{(0)}\|_F^2$ is thus a sum of $d^2$ independent chi-squared variables. By Bernstein-type bounds for sub-exponential random variables, when $d = \Omega\left(\sqrt{\log(1/\delta)}\right)$, we have:

$$\|\mathbf{W}_V^{(0)}\|_F^2 \leq 2d^2 \sigma_1^2,$$

with probability at least $1 - \delta/8$. For $\forall \boldsymbol{u}, \boldsymbol{v} \in \{\boldsymbol{a}_s\}_{s \in [K]} \cup \{\boldsymbol{b}_s\}_{s \in [K]} \cup \{\boldsymbol{\nu}_{k'}\}_{k' \in [K']}$, it holds that $\boldsymbol{v}^\top \mathbf{W}_V^{(0)\top} \boldsymbol{u} \sim \mathcal{N}(0, \sigma_1^2 \|\boldsymbol{u}\|^2 \|\boldsymbol{v}\|^2)$. Then by Gaussian tail bound and union bound, we have

$$|\boldsymbol{v}^\top \mathbf{W}_V^{(0)\top} \boldsymbol{u}| \leq \sqrt{2 \log(\frac{8(2K+K')^2}{\delta})} \sigma_1 \|\boldsymbol{u}\| \|\boldsymbol{v}\| \leq \sqrt{2 \log(\frac{8(2K+K')^2}{\delta})} \sigma_1 \|\boldsymbol{u}\| \|\boldsymbol{v}\|$$

with probability at least $1 - \delta/8$. Besides, for $\forall \boldsymbol{u} \in \{\boldsymbol{a}_s\}_{s \in [K]} \cup \{\boldsymbol{b}_s\}_{s \in [K]}$, we notice that $\mathbf{W}_V^{(0)} \boldsymbol{u} \sim \mathcal{N}(\boldsymbol{0}, \sigma_1^2 \|\boldsymbol{u}\|^2 \mathbf{I}_{d \times d})$. Then it holds that $\boldsymbol{u}^\top \mathbf{W}_V^{(0)\top} \mathbf{W}_V^{(0)} \boldsymbol{u} \sim \sigma_1^2 \|\boldsymbol{u}\|^2 \chi_d^2$, where $\chi_d^2$ is a chi-square distribution with $d$ degrees of freedom. By Bernstein tail bounds (for an appropriately large $d$) and union bound, we have

$$|\boldsymbol{u}^\top \mathbf{W}_V^{(0)\top} \mathbf{W}_V^{(0)} \boldsymbol{u} - {\sigma_1}^2 \|\boldsymbol{u}\|^2 d| \leq O(2\sigma_1^2 \|\boldsymbol{u}\|^2 \sqrt{d \log(16(2K+K')/\delta)})$$

with probability $1 - \delta/8$. By appropriately configuring $d = \Omega(\log((2K+K')^2/\delta))$, we have that with probability at least $1 - \delta/8$ we have

$$0.99 {\sigma_1}^2 \|\boldsymbol{u}\|^2 d \leq \|\mathbf{W}_V^{(0)} \boldsymbol{u}\|^2 \leq 1.01 {\sigma_1}^2 \|\boldsymbol{u}\|^2 d.$$

Combining the obtained results we obtain the following bound

$$|\boldsymbol{v}^\top \mathbf{W}_V^{(0)} \boldsymbol{u} / \|\mathbf{W}_V^{(0)} \boldsymbol{u}\|| \leq 2\sqrt{2 \log(\frac{8(2K+K')^2}{\delta})} \|\boldsymbol{v}\| / \sqrt{d}$$

On the other hand, notice that for $\forall \boldsymbol{u} \neq \boldsymbol{v} \in \{\boldsymbol{a}_s\}_{s\in[K]} \cup \{\boldsymbol{b}_s\}_{s\in[K]}$, $\boldsymbol{v}^\top \mathbf{W}_V^{(0)}{}^\top \mathbf{W}_V^{(0)} \boldsymbol{u}$ is a sub-exponential variable with mean 0. Given that

$$\text{Cov}(\mathbf{W}_V^{(0)}\boldsymbol{v}, \mathbf{W}_V^{(0)}\boldsymbol{u}) = \mathbb{E}[(\mathbf{W}_V^{(0)}\boldsymbol{v})(\mathbf{W}_V^{(0)}\boldsymbol{u})^\top] = \mathbf{0},$$

thus

$$\begin{aligned}
\text{Var}(\boldsymbol{u}^\top \mathbf{W}_V^{(0)}{}^\top \mathbf{W}_V^{(0)} \boldsymbol{v}) &= \mathbb{E}[(\mathbf{W}_V^{(0)}\boldsymbol{u})^\top (\mathbf{W}_V^{(0)}\boldsymbol{v})(\mathbf{W}_V^{(0)}\boldsymbol{u})^\top (\mathbf{W}_V^{(0)}\boldsymbol{v})] \\
&= \sum_{i,j\in[d]} \mathbb{E}[(\mathbf{W}_V^{(0)}\boldsymbol{u})_i(\mathbf{W}_V^{(0)}\boldsymbol{u})_j]\mathbb{E}[(\mathbf{W}_V^{(0)}\boldsymbol{v})_i(\mathbf{W}_V^{(0)}\boldsymbol{v})_j] \\
&= \text{Tr}(\Sigma_{\mathbf{W}_V^{(0)}\boldsymbol{u}} \cdot \Sigma_{\mathbf{W}_V^{(0)}\boldsymbol{v}}) = \sigma_1^4\|\boldsymbol{u}\|^2\|\boldsymbol{v}\|^2 d.
\end{aligned}$$

Therefore, for $\forall \boldsymbol{v} \neq \boldsymbol{w} \in \{\boldsymbol{a}_s, \boldsymbol{b}_s\}_{s\in[K]}$, by Bernstein inequality and union bound, with an appropriately large $d$, we have

$$\begin{aligned}
|\boldsymbol{v}^\top \mathbf{W}_V^{(0)}{}^\top \mathbf{W}_V^{(0)} \boldsymbol{u} - 0| &\leq O(2\sqrt{d\log(16(2K+K')(2K+K'-1)/\delta)}\sigma_1^2\|\boldsymbol{u}\|\|\boldsymbol{v}\|) \\
&\leq 4\sqrt{d\log(16(2K+K')^2/\delta)}\sigma_1^2\|\boldsymbol{u}\|\|\boldsymbol{v}\|
\end{aligned}$$

with probability at least $1 - \delta/8$. Finally, given that $\sigma_p^2 d/2 \leq \|\boldsymbol{\xi}_l\|_2^2 \leq 3\sigma_p^2 d/2$ in Lemma D.7, the rest can be proved using the same strategy.

By the union bound the proof is completed. $\qquad\square$

**Lemma D.9.** *Suppose that $\delta > 0$ and $d = \Omega(\log(2K + K')^2 N^2 M^2/\delta)$. Then with probability at least $1 - \delta$, for $\forall X \in \{Q, K\}, \boldsymbol{u}, \boldsymbol{v} \in \{\boldsymbol{a}_s\}_{s\in[K]} \cup \{\boldsymbol{b}_s\}_{s\in[K]} \cup \{\boldsymbol{\nu}_{k'}\}_{k'\in[K']}$*

$$\|\mathbf{W}_X^{(0)}\|_F^2 \leq 2d^2\sigma_1^2,$$

$$|\boldsymbol{v}^\top \mathbf{W}_X^{(0)}{}^\top \boldsymbol{u}| \leq \sqrt{2\log(\frac{8(2K+K')^2}{\delta})}\sigma_0\|\boldsymbol{u}\|\|\boldsymbol{v}\|,$$

$$\sigma_0{}^2\|\boldsymbol{u}\|^2 d/2 \leq \|\mathbf{W}_X^{(0)}\boldsymbol{u}\|^2 \leq 3\sigma_0{}^2\|\boldsymbol{u}\|^2 d/2,$$

$$|\boldsymbol{v}^\top \mathbf{W}_X^{(0)}{}^\top \mathbf{W}_{\neg X}^{(0)}\boldsymbol{u}| \leq 4\sqrt{\log(16(2K+K')^2/\delta)}\sigma_0^2 d^{1/2}\|\boldsymbol{u}\|\|\boldsymbol{v}\|,$$

$$|\boldsymbol{v}^\top \mathbf{W}_X^{(0)}{}^\top \mathbf{W}_{\neg X}^{(0)}\boldsymbol{\xi}_l| \leq 4\sqrt{\log(16(2K+K')^2/\delta)}\sigma_0^2 d^{1/2}\|\boldsymbol{u}\|\|\boldsymbol{v}\|,$$

$$|\boldsymbol{v}^\top \mathbf{W}_X^{(0)}{}^\top \boldsymbol{\xi}_l| \leq 2\sqrt{2\log(16(2K+K')NM)/\delta}\sigma_0\sigma_p d^{1/2}\|\boldsymbol{u}\|,$$

$$\sigma_0{}^2\sigma_p{}^2 d^2 \leq \|\mathbf{W}_X^{(0)}\boldsymbol{\xi}_l\|^2 \leq 3\sigma_0{}^2\sigma_p{}^2 d^2,$$

$$|\boldsymbol{\xi}_l^\top \mathbf{W}_X^{(0)}{}^\top \mathbf{W}_{\neg X}^{(0)}\boldsymbol{\xi}_l'| \leq 16\sqrt{\log(16(2K+K')^2/\delta)}\sigma_0^2\sigma_p^4 d^{3/2},$$

*Proof.* The proof strategies follow Lemma D.8. $\qquad\square$

**Lemma D.10.** *Suppose that $\delta > 0$, and $N = \Omega(K\log(1/\delta))$, $K = \Omega(\log(1/\delta))$. Let $\mathcal{N}_k^y$ denote the index set of sampled QA data points in $\mathcal{P}_{QA}^{tr}$ (or ICL-type data points $\mathcal{P}_{\mathbf{T}}^{tr}$) where the high-level task concept is $k_\mathbf{S} = k \in [K]$ and the task-specific low-level semantic real-valued label is $y_{k_\mathbf{S}} = y \in \{\pm 1\}$. Then with probability at least $1 - \delta$, we have*

$$\frac{(1-10^{-2})N}{2K} \leq |\mathcal{N}_k^y| \leq \frac{(1+10^{-2})N}{2K}, \tag{25}$$

*for $\forall k \in [K], y \in \{\pm 1\}$. This would further suggest*

$$\frac{(1-2\cdot 10^{-2})N}{K} \leq |\mathcal{N}_k| \leq \frac{(1+2\cdot 10^{-2})N}{K}, \tag{26}$$

*where $|\mathcal{N}_k| = |\mathcal{N}_k|^+ \cap |\mathcal{N}_k|^-$ represents the index set of sampled QA data $\mathcal{P}_{QA}^{tr}$ with their high-level task concepts as $k_\mathbf{S} = k \in [K]$. Moreover, for $\forall y \in \{\pm 1\}$, there exists $k \in [K]$, such that*

$$|\mathcal{N}_k^y - N/(2K)| \geq 5 \cdot 10^{-3} N/(2K). \tag{27}$$

*Proof.* The proof strategy follows Lemma G.3 in Bu et al. (2024b). See $|\mathcal{N}_k^y|$ as a binomial random variable with probability $(2K)^{-1}$ and number of experiments $N$:

$$|\mathcal{N}_k^y| \sim \text{Bin}(N, (2K)^{-1}).$$

The expectation and standard deviation are given by:

$$\mathbb{E}[|\mathcal{N}_k^y|] = \frac{N}{2K}, \quad \sigma_k = \sqrt{N \cdot \frac{1}{2K} \cdot \left(1 - \frac{1}{2K}\right)}.$$

By Exercise 2.9.(a) and (b) in Wainwright (2019) and $(2K)^{-1} < 1/2$, with probability at least $1 - \delta$, we can directly have $|\mathcal{N}_k^y - N/(2K)| \leq 0.001$ by Hoeffding Inequality as well as an appropriately large $N$.

On the other hand, define the event:

$$A_k = \left\{|\mathcal{N}_k^y - N/(2K)| \leq 10^{-3}N/(2K)\right\}.$$

Observing that the probability of $A_k, k \in [K]$ occurring is lower bounded by an absolute constant , denoted as $c$. Therefore, considering $K$ trials and with the condition $K = \Omega(\log(1/\delta))$, by the linearity (independence of different tasks), we have:

$$\mathbb{P}\left(\bigcup_{k=1}^{K} A_k^c\right) = 1 - \mathbb{P}\left(\bigcap_{k=1}^{K} A_k\right)$$
$$\geq 1 - c^K \geq 1 - \delta.$$

Thus, with probability at least $1 - \delta$, there exists some $k \in [K]$ such that:

$$|\mathcal{N}_k^y - N/(2K)| \geq 5 \cdot 10^{-3}N/(2K).$$

$\square$

**Lemma D.11.** *Suppose that $\delta > 0$, and $N = \Omega(KK'\log(1/\delta)/M)$. For $\forall k \in [K], k' \in [K']$, denote $\mathcal{V}_{\mathcal{N}_k,k'}$ as the number of common token $\boldsymbol{\nu}_{k'}$ appearing in sample set $\mathcal{N}_k \subset \mathcal{P}_{QA}^{tr}$ (or $\mathcal{P}_{\mathbf{T}}^{tr}$). Then with probability at least $1 - \delta$, we have*

$$\frac{0.99NM}{KK'} \leq |\mathcal{V}_{\mathcal{N}_k,k'}| \leq \frac{1.01NM}{KK'} \tag{28}$$

*Proof.* It is worth noting that $|\mathcal{V}_{\mathcal{N}_k,k'}|$ can be viewed as a binomial random variable with probability $(K')^{-1}$ and the number of trials $|\mathcal{N}_k|$. Using the bounds $\frac{0.998N}{K} \leq |\mathcal{N}_k| \leq \frac{1.002N}{K}$ from Lemma D.10 and applying the same proof strategy, the result follows. $\square$

# E. Detailed Gradients

For clarity, in this section we omit the notation of sample index $n$ and iteration index $(t)$.

**Lemma E.1.** *The gradients of $\mathbf{W}_K, \mathbf{W}_Q, \mathbf{W}_V$ can be computed as follow.*

$$\mathbb{E}_{\mathcal{P}_{QA}^{tr}}[\partial_{\mathbf{W}_K} L_{\mathcal{B}}(\boldsymbol{\theta})] = -\mathbb{E}_{\mathcal{P}_{QA}^{tr}}[\sum_{j=1}^{M} \iota_{\mathbf{u}_{k_{\mathbf{y}}},j} \cdot (\mathbf{W}_Q \mathbf{S} e_L) \cdot (\mathbf{S}(e_j - \boldsymbol{\pi}))^{\top}]$$

$$\mathbb{E}_{\mathcal{P}_{QA}^{tr}}[\partial_{\mathbf{W}_Q} L_{\mathcal{B}}(\boldsymbol{\theta})] = -\mathbb{E}_{\mathcal{P}_{QA}^{tr}}[\sum_{j=1}^{M} \iota_{\mathbf{u}_{k_{\mathbf{y}}},j} \cdot \mathbf{W}_K \mathbf{S}(e_j - \boldsymbol{\pi}) \cdot (\mathbf{S} e_L)^{\top}] \tag{29}$$

$$\mathbb{E}_{\mathcal{P}_{QA}^{tr}}[\partial_{\mathbf{W}_V} L_{\mathcal{B}}(\boldsymbol{\theta})] = -\mathbb{E}_{\mathcal{P}_{QA}^{tr}}[\frac{\mathbf{\Pi}_{(\mathbf{W}_V \mathbf{S}\boldsymbol{\pi})^{\perp}}^{\mathbf{u}_{k_{\mathbf{y}}} - \sum_{k\in[7K+K']} \omega_k \mathbf{u}_k} (\mathbf{S}\boldsymbol{\pi})^{\top}}{\|\mathbf{W}_V \mathbf{S}\boldsymbol{\pi}\|}]$$

*where $\boldsymbol{\pi} = [\pi_1, \cdots, \pi_{L-1}, 0]^\top \in \mathbb{R}^L$, $\boldsymbol{\omega} = [\omega_1, \cdots, \omega_{5K}]^\top$. Additionally, for $\forall j \in [L-1]$, $k \in [7K + K']$ as well as any vector $\mathbf{y}, \mathbf{u}, \mathbf{v} \in \mathbb{R}^d$, the coefficients $\pi_j, \omega_k, \iota_{\mathbf{u}_{k_\mathbf{y}}, j}$, and operator $\mathbf{\Pi}_{\mathbf{u}^\perp}^{\mathbf{v}}$ are defined as*

$$
\begin{aligned}
\pi_j &:= \mathrm{softmax}((\mathbf{W}_K \mathbf{S} \mathbf{e}_l)^T (\mathbf{W}_Q \mathbf{S} \mathbf{e}_L)) = \frac{\exp(\mathbf{e}_L^\top \mathbf{S}^\top \mathbf{W}_Q^\top \mathbf{W}_K \mathbf{S} \mathbf{e}_j)}{\sum_{i \in [L-1]} \exp(\mathbf{e}_L^\top \mathbf{S}^\top \mathbf{W}_Q^\top \mathbf{W}_K \mathbf{S} \mathbf{e}_i)}, \\
\omega_k &:= \frac{\exp(\mathbf{u}_k^\top (\mathbf{x}) + \mathbf{u}_k^\top LN(\sum_{j=1}^M \pi_j \mathbf{W}_V \mathbf{S} \mathbf{e}_j))}{\sum_{k \in [7K+K']} \exp(\mathbf{u}_k^\top (\mathbf{x}) + \mathbf{u}_k^\top LN(\sum_{j=1}^M \pi_j \mathbf{W}_V \mathbf{S} \mathbf{e}_j))}, \\
\iota_{\mathbf{u}_{k_\mathbf{y}}, j} &:= \frac{\{(\mathbf{u}_{k_\mathbf{y}} - \sum_{k \in [7K+K']} \omega_k \mathbf{u}_k)^\top (\mathbf{W}_V \mathbf{S} \pi_j \mathbf{e}_j)\} \cdot \{(\mathbf{W}_V \mathbf{S} \boldsymbol{\pi})^\top (\mathbf{W}_V \mathbf{S} (\boldsymbol{\pi} - \pi_j \mathbf{e}_j))\}}{\|\mathbf{W}_V \mathbf{S} \boldsymbol{\pi}\|^3}, \\
\mathbf{\Pi}_{\mathbf{u}^\perp}^{\mathbf{v}} &:= \mathbf{v} - \mathbf{v}^\top (\frac{\mathbf{u}}{\|\mathbf{u}\|}) \cdot \frac{\mathbf{u}}{\|\mathbf{u}\|}.
\end{aligned}
\tag{30}
$$

*Proof.* For $\mathbf{W} \in \{\mathbf{W}_K, \mathbf{W}_Q, \mathbf{W}_V\}$, we have

$$
\begin{aligned}
\partial_{\mathbf{W}} L_{\mathcal{B}}(\boldsymbol{\theta}) &= -\partial_{\mathbf{W}} \mathbb{E}_{\mathcal{P}_{\mathrm{QA}}^{\mathrm{tr}}} [\mathbf{u}_{k_\mathbf{y}}^\top \mathbf{h}_{\boldsymbol{\theta}} - \log(\sum_{k \in [7K+K']} \exp(\mathbf{u}_k^\top \mathbf{h}_{\boldsymbol{\theta}}))] \\
&= -\partial_{\mathbf{W}} \mathbb{E}_{\mathcal{P}_{\mathrm{QA}}^{\mathrm{tr}}} [\mathbf{u}_{k_\mathbf{y}}^\top ((\mathbf{x}) + \mathrm{LN}(\sum_{j=1}^M \pi_j \mathbf{W}_V \mathbf{S} \mathbf{e}_j)) \\
&\quad - \log(\sum_{k \in [7K+K']} \exp(\mathbf{u}_k^\top ((\mathbf{x}) + \mathrm{LN}(\sum_{j=1}^M \pi_j \mathbf{W}_V \mathbf{S} \mathbf{e}_j))))] \\
&= -\mathbb{E}_{\mathcal{P}_{\mathrm{QA}}^{\mathrm{tr}}} [\partial_{\mathbf{W}} \mathbf{u}_{k_\mathbf{y}}^\top \mathrm{LN}(\sum_{j=1}^M \pi_j \mathbf{W}_V \mathbf{S} \mathbf{e}_j) \\
&\quad - \partial_{\mathbf{W}} \log(\sum_{k \in [7K+K']} \exp(\mathbf{u}_k^\top ((\mathbf{x}) + \mathrm{LN}(\sum_{j=1}^M \pi_j \mathbf{W}_V \mathbf{S} \mathbf{e}_j))))],
\end{aligned}
\tag{31}
$$

where

$$
\pi_j := \mathrm{softmax}((\mathbf{W}_K \mathbf{S} \mathbf{e}_l)^T (\mathbf{W}_Q \mathbf{S} \mathbf{e}_L)) = \frac{\exp(\mathbf{e}_L^\top \mathbf{S}^\top \mathbf{W}_Q^\top \mathbf{W}_K \mathbf{S} \mathbf{e}_j)}{\sum_{i \in [L-1]} \exp(\mathbf{e}_L^\top \mathbf{S}^\top \mathbf{W}_Q^\top \mathbf{W}_K \mathbf{S} \mathbf{e}_i)}.
\tag{32}
$$

We compute $\partial_{\mathbf{W}} \mathbf{u}_{k_\mathbf{y}}^\top \mathrm{LN}(\sum_{j=1}^M \pi_j \mathbf{W}_V \mathbf{S} \mathbf{e}_j)$ as follows. By the fact that the trace of a scalar is the scalar itself, we directly have

$$
\mathrm{d}(\mathbf{u}_{k_\mathbf{y}}^\top \mathrm{LN}(\sum_{j=1}^M \pi_j \mathbf{W}_V \mathbf{S} \mathbf{e}_j)) = \mathrm{tr}(\mathrm{d}(\mathbf{u}_{k_\mathbf{y}}^\top \mathrm{LN}(\sum_{j=1}^M \pi_j \mathbf{W}_V \mathbf{S} \mathbf{e}_j))).
$$

Then we see

$$
\begin{aligned}
\mathrm{d}(\mathbf{u}_{k_{\mathbf{y}}}^{\top}\mathrm{LN}(\sum_{j=1}^{M}\pi_j\mathbf{W}_V\mathbf{S}e_j)) &= \mathrm{tr}(\partial_{\mathbf{W}_K}{}^{\top}(\mathbf{u}_{k_{\mathbf{y}}}^{\top}\mathrm{LN}(\sum_{j=1}^{M}\pi_j\mathbf{W}_V\mathbf{S}e_j))\mathrm{d}\mathbf{W}_K) \\
&+ \mathrm{tr}(\partial_{\mathbf{W}_Q}{}^{\top}(\mathbf{u}_{k_{\mathbf{y}}}^{\top}\mathrm{LN}(\sum_{j=1}^{M}\pi_j\mathbf{W}_V\mathbf{S}e_j))\mathrm{d}\mathbf{W}_Q) \\
&+ \mathrm{tr}((\partial_{\mathbf{W}_V}{}^{\top}(\mathbf{u}_{k_{\mathbf{y}}}^{\top}\mathrm{LN}(\sum_{j=1}^{M}\pi_j\mathbf{W}_V\mathbf{S}e_j)))\mathrm{d}\mathbf{W}_V) \\
&= \mathrm{tr}((\partial_{\mathbf{W}_K}(\mathbf{u}_{k_{\mathbf{y}}}^{\top}\mathrm{LN}(\sum_{j=1}^{M}\pi_j\mathbf{W}_V\mathbf{S}e_j)))^{\top}\mathrm{d}\mathbf{W}_K) \\
&+ \mathrm{tr}(\partial_{\mathbf{W}_Q}(\mathbf{u}_{k_{\mathbf{y}}}^{\top}\mathrm{LN}(\sum_{j=1}^{M}\pi_j\mathbf{W}_V\mathbf{S}e_j))\mathrm{d}\mathbf{W}_Q^{\top}) \\
&+ \mathrm{tr}((\partial_{\mathbf{W}_V}(\mathbf{u}_{k_{\mathbf{y}}}^{\top}\mathrm{LN}(\sum_{j=1}^{M}\pi_j\mathbf{W}_V\mathbf{S}e_j)))^{\top}\mathrm{d}\mathbf{W}_V)
\end{aligned}
\tag{33}
$$

As such, we can compute

$$
\begin{aligned}
\mathrm{tr}(\mathrm{d}(\mathbf{u}_{k_{\mathbf{y}}}^{\top}\mathrm{LN}(\sum_{j=1}^{M}\pi_j\mathbf{W}_V\mathbf{S}e_j))) &= \mathrm{tr}(\mathbf{u}_{k_{\mathbf{y}}}^{\top}\mathrm{d}(\mathrm{LN}(\sum_{j=1}^{M}\pi_j\mathbf{W}_V\mathbf{S}e_j))) \\
&= \mathrm{tr}(\mathbf{u}_{k_{\mathbf{y}}}^{\top}\mathrm{d}(\frac{\sum_{j=1}^{M}\pi_j\mathbf{W}_V\mathbf{S}e_j}{\|\sum_{j=1}^{M}\pi_j\mathbf{W}_V\mathbf{S}e_j\|})) = \sum_{j=1}^{M}\mathrm{tr}(\mathbf{u}_{k_{\mathbf{y}}}^{\top}\mathrm{d}(\frac{\pi_j\mathbf{W}_V\mathbf{S}e_j}{\|\sum_{j=1}^{M}\pi_j\mathbf{W}_V\mathbf{S}e_j\|}))
\end{aligned}
\tag{34}
$$

Then we have

$$
\begin{aligned}
\sum_{j=1}^{M} \mathrm{tr}(\mathbf{u}_{k_\mathbf{y}}^\top \mathrm{d}(\frac{\pi_j \mathbf{W}_V \mathbf{S} e_j}{\| \sum_{i=1}^{L-1} \pi_i \mathbf{W}_V \mathbf{S} e_i \|})) &= \sum_{j=1}^{M} \mathrm{tr}(\mathbf{u}_{k_\mathbf{y}}^\top \mathrm{d}(\frac{\pi_j \mathbf{W}_V \mathbf{S} e_j}{\| \pi_j \mathbf{W}_V \mathbf{S} e_j + \sum_{i \neq j} \pi_i \mathbf{W}_V \mathbf{S} e_i \|})) \\
&= \sum_{j=1}^{M} \mathrm{tr}(\mathbf{u}_{k_\mathbf{y}}^\top (\frac{\mathrm{d}(\pi_j \mathbf{W}_V) \mathbf{S} e_j \| \sum_{i=1}^{L-1} \pi_i \mathbf{W}_V \mathbf{S} e_i \|}{\| \sum_{i=1}^{L-1} \pi_i \mathbf{W}_V \mathbf{S} e_i \|^2})) \\
&\quad - \sum_{j=1}^{M} \mathrm{tr}(\mathbf{u}_{k_\mathbf{y}}^\top (\frac{\pi_j \mathbf{W}_V \mathbf{S} e_j \mathrm{d}(\| \sum_{i=1}^{L-1} \pi_i \mathbf{W}_V \mathbf{S} e_i \|)}{\| \sum_{i=1}^{L-1} \pi_i \mathbf{W}_V \mathbf{S} e_i \|^2})) \\
&= \sum_{j=1}^{M} \mathrm{tr}(\frac{(\mathbf{u}_{k_\mathbf{y}}^\top \mathbf{W}_V \mathbf{S} e_j \mathrm{d}(\pi_j) + \pi_j \mathbf{u}_{k_\mathbf{y}}^\top \mathrm{d}(\mathbf{W}_V) \mathbf{S} e_j)}{\| \sum_{i=1}^{L-1} \pi_i \mathbf{W}_V \mathbf{S} e_i \|}) \\
&\quad - \sum_{j=1}^{M} \mathrm{tr}(\frac{\pi_j \mathbf{u}_{k_\mathbf{y}}^\top \mathbf{W}_V \mathbf{S} e_j \sum_{i,l \in [L-1]}(e_i^\top \mathbf{S}^\top \mathrm{d}(\mathbf{W}_V^\top \pi_i \pi_l \mathbf{W}_V) \mathbf{S} e_l)}{2\| \sum_{i=1}^{L-1} \pi_i \mathbf{W}_V \mathbf{S} e_i \|^3}) \\
&= \sum_{j=1}^{M} \mathrm{tr}(\frac{(\mathbf{u}_{k_\mathbf{y}}^\top \mathbf{W}_V \mathbf{S} e_j \mathrm{d}(\pi_j) + \pi_j \mathbf{u}_{k_\mathbf{y}}^\top \mathrm{d}(\mathbf{W}_V) \mathbf{S} e_j)}{\| \sum_{i=1}^{L-1} \pi_i \mathbf{W}_V \mathbf{S} e_i \|}) \\
&\quad - \sum_{j=1}^{M} \mathrm{tr}(\sum_{l=1}^{2J} \frac{\pi_j \mathbf{u}_{k_\mathbf{y}}^\top \mathbf{W}_V \mathbf{S} e_j \sum_{i \in [L-1]} \pi_i (e_i^\top \mathbf{S}^\top \mathbf{W}_V^\top \mathbf{W}_V \mathbf{S} e_l)}{\| \sum_{i=1}^{L-1} \pi_i \mathbf{W}_V \mathbf{S} e_i \|^3} \mathrm{d}(\pi_l)) \\
&\quad - \sum_{j=1}^{M} \mathrm{tr}(\frac{\pi_j \sum_{i,l \in [L-1]} \mathbf{S} e_l \mathbf{u}_{k_\mathbf{y}}^\top \mathbf{W}_V \mathbf{S} e_j \pi_i \pi_l (e_i^\top \mathbf{S}^\top (\mathrm{d}(\mathbf{W}_V^\top) \mathbf{W}_V + \mathbf{W}_V^\top \mathrm{d}(\mathbf{W}_V)))}{2\| \sum_{j=1}^{M} \pi_j \mathbf{W}_V \mathbf{S} e_j \|^3}) \\
&= \sum_{j=1}^{M} (\mathrm{tr}(\zeta_{\mathbf{u}_{k_\mathbf{y}}}^j (\mathrm{d}\pi_j)) + \mathrm{tr}(\Upsilon_{\mathbf{u}_{k_\mathbf{y}}}^j \mathrm{d}\mathbf{W}_V))
\end{aligned}
\tag{35}
$$

where

$$
\begin{aligned}
\zeta_{\mathbf{u}_{k_\mathbf{y}}}^j &:= (\mathbf{u}_{k_\mathbf{y}}^\top \mathbf{W}_V \mathbf{S} e_j)(\frac{1}{\| \sum_{i=1}^{L-1} \pi_i \mathbf{W}_V \mathbf{S} e_i \|} - \frac{\sum_{i \in [L-1]} \pi_i (e_i^\top \mathbf{S}^\top \mathbf{W}_V^\top \mathbf{W}_V \mathbf{S} \pi_j e_j)}{\| \sum_{i=1}^{L-1} \pi_i \mathbf{W}_V \mathbf{S} e_i \|^3}) \\
&= (\mathbf{u}_{k_\mathbf{y}}^\top \mathbf{W}_V \mathbf{S} e_j)(\frac{\sum_{i \neq j}(\mathbf{W}_V \mathbf{S} \pi_i e_i)^\top \mathbf{W}_V \mathbf{S} \pi_j e_j}{\| \sum_{i=1}^{L-1} \pi_i \mathbf{W}_V \mathbf{S} e_i \|^3}) \\
\Upsilon_{\mathbf{u}_{k_\mathbf{y}}}^j &:= \frac{\pi_j \mathbf{S} e_j \mathbf{u}_{k_\mathbf{y}}^\top - \dfrac{\pi_j \sum_{i,l \in [L-1]} \mathbf{S} e_l \mathbf{u}_{k_\mathbf{y}}^\top \mathbf{W}_V \mathbf{S} e_j \pi_i \pi_l e_i^\top \mathbf{S}^\top \mathbf{W}_V^\top}{2\| \sum_{i=1}^{L-1} \pi_i \mathbf{W}_V \mathbf{S} e_i \|^2}}{\| \sum_{i=1}^{L-1} \pi_i \mathbf{W}_V \mathbf{S} e_i \|} \\
&\quad - \frac{\dfrac{(\pi_j \sum_{i,l \in [L-1]} \mathbf{W}_V \mathbf{S} e_l \mathbf{u}_{k_\mathbf{y}}^\top \mathbf{W}_V \mathbf{S} e_j \pi_i \pi_l e_i^\top \mathbf{S}^\top)^\top}{2\| \sum_{i=1}^{L-1} \pi_i \mathbf{W}_V \mathbf{S} e_i \|^2}}{\| \sum_{i=1}^{L-1} \pi_i \mathbf{W}_V \mathbf{S} e_i \|} \\
&= \frac{\pi_j (\mathbf{y} e_j^\top \mathbf{S}^\top)^\top - \dfrac{\pi_j \sum_{i,l \in [L-1]} \pi_i \pi_l (\mathbf{W}_V \mathbf{S} e_i e_j^\top \mathbf{S}^\top \mathbf{W}_V^\top \mathbf{y} e_l^\top \mathbf{S}^\top)^\top}{2\| \sum_{i=1}^{L-1} \pi_i \mathbf{W}_V \mathbf{S} e_i \|^2}}{\| \sum_{i=1}^{L-1} \pi_i \mathbf{W}_V \mathbf{S} e_i \|} \\
&\quad - \frac{\dfrac{\pi_j \sum_{i,l \in [L-1]} \pi_i \pi_l (\mathbf{W}_V \mathbf{S} e_l \mathbf{u}_{k_\mathbf{y}}^\top \mathbf{W}_V \mathbf{S} e_j e_i^\top \mathbf{S}^\top)^\top}{2\| \sum_{i=1}^{L-1} \pi_i \mathbf{W}_V \mathbf{S} e_i \|^2}}{\| \sum_{i=1}^{L-1} \pi_i \mathbf{W}_V \mathbf{S} e_i \|}
\end{aligned}
\tag{36}
$$

Then for $\forall j \in [L-1]$, we investigate $\mathrm{d}(\pi_j)$:

$$
\begin{aligned}
\mathrm{d}(\pi_j) &= \mathrm{d}\frac{\exp(\boldsymbol{e}_L^\top \mathbf{S}^\top \mathbf{W}_Q^\top \mathbf{W}_K \mathbf{S}\boldsymbol{e}_j)}{\sum_{i\in[L-1]}\exp(\boldsymbol{e}_L^\top \mathbf{S}^\top \mathbf{W}_Q^\top \mathbf{W}_K \mathbf{S}\boldsymbol{e}_i)} \\
&= \pi_j \boldsymbol{e}_L^\top \mathbf{S}^\top \mathrm{d}(\mathbf{W}_Q^\top \mathbf{W}_K)\mathbf{S}\boldsymbol{e}_j - \\
&\quad \frac{\pi_j \sum_{i\in[L-1]}(\exp(\boldsymbol{e}_L^\top \mathbf{S}^\top \mathbf{W}_Q^\top \mathbf{W}_K \mathbf{S}\boldsymbol{e}_i))\boldsymbol{e}_L^\top \mathbf{S}^\top \mathrm{d}(\mathbf{W}_Q^\top \mathbf{W}_K)\mathbf{S}\boldsymbol{e}_i}{(\sum_{i\in[L-1]}\exp(\boldsymbol{e}_L^\top \mathbf{S}^\top \mathbf{W}_Q^\top \mathbf{W}_K \mathbf{S}\boldsymbol{e}_i))} \\
&= \pi_j \boldsymbol{e}_L^\top \mathbf{S}^\top(\mathrm{d}(\mathbf{W}_Q^\top \mathbf{W}_K)\mathbf{S}\boldsymbol{e}_j - \sum_{i\in[L-1]}\pi_i \mathrm{d}(\mathbf{W}_Q^\top \mathbf{W}_K)\mathbf{S}\boldsymbol{e}_i)
\end{aligned}
\tag{37}
$$

By Eq.(34), (35),(37) we have

$$
\begin{aligned}
\mathrm{d}(\mathbf{u}_{k_\mathbf{y}}^\top \mathrm{LN}(\sum_{j=1}^M \pi_j \mathbf{W}_V \mathbf{S}\boldsymbol{e}_j)) &= \mathrm{tr}(\sum_{j=1}^M \zeta_{\mathbf{u}_{k_\mathbf{y}}}^j \pi_j \boldsymbol{e}_L^\top \mathbf{S}^\top(\mathrm{d}(\mathbf{W}_Q^\top \mathbf{W}_K)\mathbf{S}\boldsymbol{e}_j - \sum_{i\in[L-1]}\pi_i \mathrm{d}(\mathbf{W}_Q^\top \mathbf{W}_K)\mathbf{S}\boldsymbol{e}_i)) \\
&\quad + \mathrm{tr}(\sum_{j=1}^M \boldsymbol{\Upsilon}_{\mathbf{u}_{k_\mathbf{y}}}^j \mathrm{d}\mathbf{W}_V) \\
&= \mathrm{tr}(\sum_{j=1}^M \zeta_{\mathbf{u}_{k_\mathbf{y}}}^j \pi_j \mathbf{S}(\boldsymbol{e}_j - \sum_{i=1}^{L-1}\pi_i \boldsymbol{e}_i)\boldsymbol{e}_L^\top \mathbf{S}^\top(\mathrm{d}(\mathbf{W}_Q^\top \mathbf{W}_K))) \\
&\quad + \mathrm{tr}(\sum_{j=1}^M \boldsymbol{\Upsilon}_{\mathbf{u}_{k_\mathbf{y}}}^j \mathrm{d}\mathbf{W}_V)
\end{aligned}
\tag{38}
$$

As we see that

$$
\mathrm{d}(\mathbf{W}_Q^\top \mathbf{W}_K) = \mathrm{d}(\mathbf{W}_Q^\top)\mathbf{W}_K + \mathbf{W}_Q^\top \mathrm{d}(\mathbf{W}_K).
$$

Therefore, by the Cyclic Invariance Property of operator trace and Eq.(38),(31), we have

$$
(\partial_{\mathbf{W}_K}(\mathbf{u}_{k_\mathbf{y}}^\top \mathrm{LN}(\sum_{j=1}^M \pi_j \mathbf{W}_V \mathbf{S}\boldsymbol{e}_j)))^\top = \sum_{j=1}^M \zeta_{\mathbf{u}_{k_\mathbf{y}}}^j \pi_j \mathbf{S}(\boldsymbol{e}_j - \sum_{i=1}^{L-1}\pi_i \boldsymbol{e}_i)\boldsymbol{e}_L^\top \mathbf{S}^\top \mathbf{W}_Q^\top.
$$

That is,

$$
\partial_{\mathbf{W}_K}(\mathbf{u}_{k_\mathbf{y}}^\top \mathrm{LN}(\sum_{j=1}^M \pi_j \mathbf{W}_V \mathbf{S}\boldsymbol{e}_j)) = \sum_{j=1}^M \zeta_{\mathbf{u}_{k_\mathbf{y}}}^j \pi_j \mathbf{W}_Q \mathbf{S}\boldsymbol{e}_L(\boldsymbol{e}_j - \sum_{i=1}^{L-1}\pi_i \boldsymbol{e}_i)^\top \mathbf{S}^\top.
\tag{39}
$$

Similarly, we have

$$
\partial_{\mathbf{W}_Q}(\mathbf{u}_{k_\mathbf{y}}^\top \mathrm{LN}(\sum_{j=1}^M \pi_j \mathbf{W}_V \mathbf{S}\boldsymbol{e}_j)) = \sum_{j=1}^M \zeta_{\mathbf{u}_{k_\mathbf{y}}}^j \pi_j \mathbf{W}_K \mathbf{S}(\boldsymbol{e}_j - \sum_{i=1}^{L-1}\pi_i \boldsymbol{e}_i)\boldsymbol{e}_L^\top \mathbf{S}^\top.
\tag{40}
$$

By Eq.(33, 35), we can also obtain

$$
\partial_{\mathbf{W}_V}(\mathbf{u}_{k_\mathbf{y}}^\top \mathrm{LN}(\sum_{j=1}^M \pi_j \mathbf{W}_V \mathbf{S}\boldsymbol{e}_j)) = \sum_{j=1}^M (\boldsymbol{\Upsilon}_{\mathbf{u}_{k_\mathbf{y}}}^j)^\top
\tag{41}
$$

Similarly, to compure $\partial_\mathbf{W} \log(\sum_{k\in[7K+K']} \exp(\mathbf{u}_k^\top((\mathbf{x}) + \mathrm{LN}(\sum_{j=1}^M \pi_j \mathbf{W}_V \mathbf{S}\boldsymbol{e}_j))))$, we investigate

$$
\mathrm{d}\log(\sum_{k\in[7K+K']} \exp(\mathbf{u}_k^\top(\mathbf{x}) + \mathbf{u}_k^\top \mathrm{LN}(\sum_{j=1}^M \pi_j \mathbf{W}_V \mathbf{S}\boldsymbol{e}_j))),
$$

which can be calculated as

$$
\sum_{k \in [7K+K']} \frac{\exp(\mathbf{u}_k^\top(\mathbf{x}) + \mathbf{u}_k^\top \mathrm{LN}(\sum_{j=1}^M \pi_j \mathbf{W}_V \mathbf{S} e_j))(\mathrm{d}(\mathbf{u}_k^\top \mathrm{LN}(\sum_{j=1}^M \pi_j \mathbf{W}_V \mathbf{S} e_j)))}{\sum_{k \in [7K+K']} \exp(\mathbf{u}_k^\top(\mathbf{x}) + \mathbf{u}_k^\top \mathrm{LN}(\sum_{j=1}^M \pi_j \mathbf{W}_V \mathbf{S} e_j))}
$$

$$
= \sum_{k \in [7K+K']} \omega_k \mathrm{d}(\mathbf{u}_k^\top \mathrm{LN}(\sum_{j=1}^M \pi_j \mathbf{W}_V \mathbf{S} e_j)),
$$

(42)

where

$$
\omega_k := \frac{\exp(\mathbf{u}_k^\top(\mathbf{x}) + \mathbf{u}_k^\top \mathrm{LN}(\sum_{j=1}^M \pi_j \mathbf{W}_V \mathbf{S} e_j))}{\sum_{k \in [7K+K']} \exp(\mathbf{u}_k^\top(\mathbf{x}) + \mathbf{u}_k^\top \mathrm{LN}(\sum_{j=1}^M \pi_j \mathbf{W}_V \mathbf{S} e_j))}, \quad \sum_{k \in [7K+K']} \omega_k = 1.
$$

(43)

On the other hand, by the derivative results we got from $\mathrm{d}(\mathbf{u}_{k_\mathbf{y}}^\top \mathrm{LN}(\sum_{j=1}^M \pi_j \mathbf{W}_V \mathbf{S} e_j))$, we directly have similar outcome for $\mathbf{u}_k, k \in [7K + K']$:

$$
\partial_{\mathbf{W}_K}(\mathbf{u}_k^\top \mathrm{LN}(\sum_{j=1}^M \pi_j \mathbf{W}_V \mathbf{S} e_j)) = \sum_{j=1}^M \zeta_{\mathbf{u}_k}^j \pi_j \mathbf{W}_Q \mathbf{S} e_L (e_j - \sum_{i=1}^{L-1} \pi_i e_i)^\top \mathbf{S}^\top
$$

$$
\partial_{\mathbf{W}_Q}(\mathbf{u}_k^\top \mathrm{LN}(\sum_{j=1}^M \pi_j \mathbf{W}_V \mathbf{S} e_j)) = \sum_{j=1}^M \zeta_{\mathbf{u}_k}^j \pi_j \mathbf{W}_K \mathbf{S} (e_j - \sum_{i=1}^{L-1} \pi_i e_i) e_L^\top \mathbf{S}^\top
$$

$$
\partial_{\mathbf{W}_V}(\mathbf{u}_k^\top \mathrm{LN}(\sum_{j=1}^M \pi_j \mathbf{W}_V \mathbf{S} e_j)) = \sum_{j=1}^M (\Upsilon_{\mathbf{u}_k}^j)^\top
$$

(44)

$$
= \sum_{j=1}^M \frac{\pi_j \mathbf{u}_k e_j^\top \mathbf{S}^\top - \dfrac{\pi_j \sum_{i,l \in [L-1]} \pi_i \pi_l \mathbf{W}_V \mathbf{S} e_i e_j^\top \mathbf{S}^\top \mathbf{W}_V^\top \mathbf{u}_k e_l^\top \mathbf{S}^\top}{2\|\sum_{i=1}^{L-1} \pi_i \mathbf{W}_V \mathbf{S} e_i\|^2}}{\|\sum_{i=1}^{L-1} \pi_i \mathbf{W}_V \mathbf{S} e_i\|}
$$

$$
- \frac{\dfrac{\pi_j \sum_{i,l \in [L-1]} \pi_i \pi_l \mathbf{W}_V \mathbf{S} e_l \mathbf{u}_k^\top \mathbf{W}_V \mathbf{S} e_j e_i^\top \mathbf{S}^\top}{2\|\sum_{i=1}^{L-1} \pi_i \mathbf{W}_V \mathbf{S} e_i\|^2}}{\|\sum_{i=1}^{L-1} \pi_i \mathbf{W}_V \mathbf{S} e_i\|}
$$

As such, by Eq.(31, 43) we have

$$
\mathbb{E}_{\mathcal{P}_{\mathrm{QA}}^{\mathrm{tr}}}[\partial_{\mathbf{W}_K} L_{\mathcal{B}}(\boldsymbol{\theta})] = -\mathbb{E}_{\mathcal{P}_{\mathrm{QA}}^{\mathrm{tr}}}[(1 - \omega_{k_\mathbf{y}}) \sum_{j=1}^M \zeta_{\mathbf{u}_{k_\mathbf{y}}}^j \pi_j \mathbf{W}_Q \mathbf{S} e_L (e_j - \sum_{i=1}^{L-1} \pi_i e_i)^\top \mathbf{S}^\top
$$

$$
- \sum_{k \neq k_\mathbf{y}} \omega_k \sum_{j=1}^M \zeta_{\mathbf{u}_k}^j \pi_j \mathbf{W}_Q \mathbf{S} e_L (e_j - \sum_{i=1}^{L-1} \pi_i e_i)^\top \mathbf{S}^\top]
$$

$$
\mathbb{E}_{\mathcal{P}_{\mathrm{QA}}^{\mathrm{tr}}}[\partial_{\mathbf{W}_Q} L_{\mathcal{B}}(\boldsymbol{\theta})] = -\mathbb{E}_{\mathcal{P}_{\mathrm{QA}}^{\mathrm{tr}}}[(1 - \omega_{k_\mathbf{y}}) \sum_{j=1}^M \zeta_{\mathbf{u}_{k_\mathbf{y}}}^j \pi_j \mathbf{W}_K \mathbf{S} (e_j - \sum_{i=1}^{L-1} \pi_i e_i) e_L^\top \mathbf{S}^\top
$$

(45)

$$
- \sum_{k \neq k_\mathbf{y}} \omega_k \sum_{j=1}^M \zeta_{\mathbf{u}_k}^j \pi_j \mathbf{W}_K \mathbf{S} (e_j - \sum_{i=1}^{L-1} \pi_i e_i) e_L^\top \mathbf{S}^\top]
$$

$$
= -\mathbb{E}_{\mathcal{P}_{\mathrm{QA}}^{\mathrm{tr}}}[\sum_{j=1}^M \sum_{k \neq k_\mathbf{y}} \pi_j \omega_k (\zeta_{\mathbf{u}_{k_\mathbf{y}}}^j - \zeta_{\mathbf{u}_k}^j) \mathbf{W}_K \mathbf{S} (e_j - \sum_{i=1}^{L-1} \pi_i e_i) e_L^\top \mathbf{S}^\top].
$$

Define $\boldsymbol{\pi} = [\pi_1, \cdots, \pi_{L-1}, 0]^\top \in \mathbb{R}^L$ and $\boldsymbol{\omega} = [\omega_1, \cdots, \omega_{5K}]^\top$. By the definitions in Eq.(36), we have

$$
\begin{aligned}
\mathbb{E}_{\mathcal{P}_{\mathrm{QA}}^{\mathrm{tr}}}[\partial_{\mathbf{W}_K} L_{\mathcal{B}}(\boldsymbol{\theta})] = \mathbb{E}_{\mathcal{P}_{\mathrm{QA}}^{\mathrm{tr}}}[(\mathbf{W}_Q \mathbf{S} \boldsymbol{e}_L) \cdot \\
\sum_{j=1}^{M} \frac{(((1-\omega_{k_{\mathbf{y}}})\mathbf{u}_{k_{\mathbf{y}}} - \sum_{k \neq k_{\mathbf{y}}} \omega_k \mathbf{u}_k)^\top (\mathbf{W}_V \mathbf{S} \pi_j \boldsymbol{e}_j))}{(\frac{(\mathbf{W}_V \mathbf{S} \boldsymbol{\pi})^\top (\mathbf{W}_V \mathbf{S}(\boldsymbol{\pi} - \pi_j \boldsymbol{e}_j))}{\|\mathbf{W}_V \mathbf{S} \boldsymbol{\pi}\|^3})^{-1}} (\mathbf{S}(\boldsymbol{\pi} - \boldsymbol{e}_j))^\top] \\
= \mathbb{E}_{\mathcal{P}_{\mathrm{QA}}^{\mathrm{tr}}}[\sum_{j=1}^{M} \iota_{\mathbf{u}_{k_{\mathbf{y}}}, j} \cdot (\mathbf{W}_Q \mathbf{S} \boldsymbol{e}_L) \cdot (\mathbf{S}(\boldsymbol{\pi} - \boldsymbol{e}_j))^\top] \\
\mathbb{E}_{\mathcal{P}_{\mathrm{QA}}^{\mathrm{tr}}}[\partial_{\mathbf{W}_Q} L_{\mathcal{B}}(\boldsymbol{\theta})] = \mathbb{E}_{\mathcal{P}_{\mathrm{QA}}^{\mathrm{tr}}}[\sum_{j=1}^{M} \mathbf{W}_K \mathbf{S}(\boldsymbol{\pi} - \boldsymbol{e}_j) \cdot \\
\sum_{j=1}^{M} \frac{(((1-\omega_{k_{\mathbf{y}}})\mathbf{u}_{k_{\mathbf{y}}} - \sum_{k \neq k_{\mathbf{y}}} \omega_k \mathbf{u}_k)^\top (\mathbf{W}_V \mathbf{S} \pi_j \boldsymbol{e}_j))}{(\frac{(\mathbf{W}_V \mathbf{S} \boldsymbol{\pi})^\top (\mathbf{W}_V \mathbf{S}(\boldsymbol{\pi} - \boldsymbol{e}_j))}{\|\mathbf{W}_V \mathbf{S} \boldsymbol{\pi}\|^3})^{-1}} (\mathbf{S} \boldsymbol{e}_L)^\top] \\
= \mathbb{E}_{\mathcal{P}_{\mathrm{QA}}^{\mathrm{tr}}}[\sum_{j=1}^{M} \iota_{\mathbf{u}_{k_{\mathbf{y}}}, j} \cdot \mathbf{W}_K \mathbf{S}(\boldsymbol{\pi} - \boldsymbol{e}_j) \cdot (\mathbf{S} \boldsymbol{e}_L)^\top]
\end{aligned}
\tag{46}
$$

where the $\iota_{\mathbf{u}_{k_{\mathbf{y}}}, j}$ is defined as

$$
\iota_{\mathbf{u}_{k_{\mathbf{y}}}, j} := \frac{\{((1-\omega_{k_{\mathbf{y}}})\mathbf{u}_{k_{\mathbf{y}}} - \sum_{k \neq k_{\mathbf{y}}} \omega_k \mathbf{u}_k)^\top (\mathbf{W}_V \mathbf{S} \pi_j \boldsymbol{e}_j)\} \cdot \{(\mathbf{W}_V \mathbf{S} \boldsymbol{\pi})^\top (\mathbf{W}_V \mathbf{S}(\boldsymbol{\pi} - \pi_j \boldsymbol{e}_j))\}}{\|\mathbf{W}_V \mathbf{S} \boldsymbol{\pi}\|^3}
\tag{47}
$$

Similarly,

$$
\begin{aligned}
\mathbb{E}_{\mathcal{P}_{\mathrm{QA}}^{\mathrm{tr}}}[\partial_{\mathbf{W}_V} L_{\mathcal{B}}(\boldsymbol{\theta})] = -\mathbb{E}_{\mathcal{P}_{\mathrm{QA}}^{\mathrm{tr}}}[(1-\omega_{k_{\mathbf{y}}}) \sum_{j=1}^{M} (\boldsymbol{\Upsilon}_{\mathbf{u}_{k_{\mathbf{y}}}}^{j})^\top - \sum_{k \neq k_{\mathbf{y}}} \omega_k \sum_{j=1}^{M} (\boldsymbol{\Upsilon}_{\mathbf{u}_k}^{j})^\top] = \\
-(\mathbb{E}_{\mathcal{P}_{\mathrm{QA}}^{\mathrm{tr}}}[\frac{\sum_{\substack{j \in [L-1] \\ k \neq k_{\mathbf{y}}}} \{\omega_k \pi_j (\mathbf{y} - \mathbf{u}_k) \boldsymbol{e}_j^\top \mathbf{S}^\top - \frac{\omega_k \pi_j \sum_{i, l \in [L-1]} \pi_i \pi_l \mathbf{W}_V \mathbf{S} \boldsymbol{e}_i \boldsymbol{e}_j^\top \mathbf{S}^\top \mathbf{W}_V^\top (\mathbf{y} - \mathbf{u}_k) \boldsymbol{e}_l^\top \mathbf{S}^\top}{2\|\mathbf{W}_V \mathbf{S} \boldsymbol{\pi}\|^2}}{\|\mathbf{W}_V \mathbf{S} \boldsymbol{\pi}\|} \\
\frac{-\frac{\omega_k \pi_j \sum_{i, l \in [L-1]} \pi_i \pi_l \mathbf{W}_V \mathbf{S} \boldsymbol{e}_l (\mathbf{y} - \mathbf{u}_k)^\top \mathbf{W}_V \mathbf{S} \boldsymbol{e}_j \boldsymbol{e}_i^\top \mathbf{S}^\top}{2\|\mathbf{W}_V \mathbf{S} \boldsymbol{\pi}\|^2}\}}{\|\mathbf{W}_V \mathbf{S} \boldsymbol{\pi}\|}]) = - \\
\mathbb{E}_{\mathcal{P}_{\mathrm{QA}}^{\mathrm{tr}}}[\frac{((1-\omega_{k_{\mathbf{y}}})\mathbf{u}_{k_{\mathbf{y}}} - \sum_{k \neq k_{\mathbf{y}}} \omega_k \mathbf{u}_k)(\mathbf{S} \boldsymbol{\pi})^\top - \frac{((1-\omega_{k_{\mathbf{y}}})\mathbf{u}_{k_{\mathbf{y}}} - \sum_{k \neq k_{\mathbf{y}}} \omega_k \mathbf{u}_k)^\top (\mathbf{W}_V \mathbf{S} \boldsymbol{\pi}))(\mathbf{W}_V \mathbf{S} \boldsymbol{\pi})(\mathbf{S} \boldsymbol{\pi})^\top}{\|\mathbf{W}_V \mathbf{S} \boldsymbol{\pi}\|^2}}{\|\mathbf{W}_V \mathbf{S} \boldsymbol{\pi}\|}] \\
= -\mathbb{E}_{\mathcal{P}_{\mathrm{QA}}^{\mathrm{tr}}}[\frac{((1-\omega_{k_{\mathbf{y}}})\mathbf{u}_{k_{\mathbf{y}}} - \sum_{k \neq k_{\mathbf{y}}} \omega_k \mathbf{u}_k) - \frac{((1-\omega_{k_{\mathbf{y}}})\mathbf{u}_{k_{\mathbf{y}}} - \sum_{k \neq k_{\mathbf{y}}} \omega_k \mathbf{u}_k)^\top (\mathbf{W}_V \mathbf{S} \boldsymbol{\pi}))(\mathbf{W}_V \mathbf{S} \boldsymbol{\pi})}{\|\mathbf{W}_V \mathbf{S} \boldsymbol{\pi}\|^2}}{\|\mathbf{W}_V \mathbf{S} \boldsymbol{\pi}\|}(\mathbf{S} \boldsymbol{\pi})^\top] \\
= -\mathbb{E}_{\mathcal{P}_{\mathrm{QA}}^{\mathrm{tr}}}[\frac{\boldsymbol{\Pi}_{(\mathbf{W}_V \mathbf{S} \boldsymbol{\pi})^\perp}^{(1-\omega_{k_{\mathbf{y}}})\mathbf{u}_{k_{\mathbf{y}}} - \sum_{k \neq k_{\mathbf{y}}} \omega_k \mathbf{u}_k} (\mathbf{S} \boldsymbol{\pi})^\top}{\|\mathbf{W}_V \mathbf{S} \boldsymbol{\pi}\|}]
\end{aligned}
\tag{48}
$$

where the projection operator $\boldsymbol{\Pi}_{\boldsymbol{u}^\perp}^{\boldsymbol{v}}$ is defined as

$$
\boldsymbol{\Pi}_{\boldsymbol{u}^\perp}^{\boldsymbol{v}} := \boldsymbol{v} - \boldsymbol{v}^\top (\frac{\boldsymbol{u}}{\|\boldsymbol{u}\|}) \cdot \frac{\boldsymbol{u}}{\|\boldsymbol{u}\|}.
$$

$\square$

# F. Training Dynamics: QA data

In the following sections, we consider training is on QA data, and assume the results in Appendix D.2 all hold with high probability. Following Definition C.3, we denote:

- $\mathbf{S}_n \in \mathcal{P}_{\text{QA}}^{\text{tr}}$ as the $n$-th QA sentence sample in $\mathcal{P}_{\text{QA}}^{\text{tr}}$,

- $\mathbf{y}_n$ as the label vector (last column) of $\mathbf{S}_n$,

- $k_{\mathbf{S}_n} \in [K]$ as the co-task concept of $\mathbf{S}_n$,

- $y_{k_{\mathbf{S}_n}} = y_n$ as the task-specific low-level semantic real value label,

- $k_{\mathbf{y}_n} = 2k_{\mathbf{S}_n} + \frac{y_n+1}{2}$ as the dictionary index of $\mathbf{y}_n$,

- $m_{\mathbf{S}_n}$ as the position index of $\boldsymbol{a}_{k_{\mathbf{S}_n}}$ in sentence $\mathbf{S}_n$,

- $\pi_{n,m}^{(t)}$ as the attention score over the $m$-th position in $\mathbf{S}_n$ at the iteration $t$,

- $\omega_{n,k}^{(t)}$ as the weight of dictionary vector $\mathbf{u}_k$ in $\mathbf{S}_n$, defined in Eq. (30) at iteration $t$,

- $\iota_{\mathbf{u}_{k_{\mathbf{y}}},m}^{(t)}$ as the coefficients defined in Eq. (30) at iteration $t$,

- $\mathcal{N}_k^y \subset \mathcal{P}_{\text{QA}}^{\text{tr}}$ as the index set of QA samples with their high-level task concept $k_{\mathbf{S}} = k \in [K]$ and task-specific low-level semantic real value label $y_{k_{\mathbf{S}}} = y \in [\pm 1]$,

- $\mathcal{V}_{\mathcal{N}_k, k'}$ as the number of common token $\boldsymbol{\nu}_{k'}$ appearing in sample set $\mathcal{N}_k \subset \mathcal{P}_{\text{QA}}^{\text{tr}}$.

## F.1. Phase 1: Linear Growth of MLP and Accelerating Growth of Attention

In this phase, the MLP is upper and lower bounded by linear continuous counterparts. Meanwhile, the evolution of attention is upper and lower bounded by exponential and quadratic counterparts, respectively.

**Lemma F.1.** *Under Condition 3.1, during* $t \leq T_1 = \Theta((\eta q_V)^{-1}\sigma_1^2 dK)$, *we have*

$$\boldsymbol{a}_k^\top \mathbf{W}_V^{(t)} \boldsymbol{a}_k = o(\sigma_1 d^{1/2}),$$

$$|\boldsymbol{a}_k{}^\top \mathbf{W}_V^{(t)} \boldsymbol{u}|, \ |\boldsymbol{v}^\top \mathbf{W}_V^{(t)} \boldsymbol{v}|, \ |\boldsymbol{v}^\top \mathbf{W}_V^{(t)} \boldsymbol{v}'| = O(\sqrt{2\log(\frac{8(2K+K')^2}{\delta})}\sigma_1),$$

$$|(\mathbf{W}_Q^{(t)} \boldsymbol{a}_{k_{\mathbf{S}_n}})^\top \mathbf{W}_K^{(t)} (\boldsymbol{a}_{k_{\mathbf{S}_n}})| = O(\sigma_0^2 d e^{q_V^{-1}\sigma_1^2 d}),$$

$$|(\mathbf{W}_Q^{(t)} \mathbf{x}_n)^\top \mathbf{W}_K^{(t)} (\boldsymbol{\nu}_{n,\neg m_{\mathbf{S}_n}} + \boldsymbol{\xi}_{n,\neg m_{\mathbf{S}_n}})| = O(\sqrt{\log(\frac{16(2K+K')^2}{\delta})}\sigma_0^2 d^{1/2}),$$

(49)

*for* $\forall k \in [K], n \in [N], s \in [7], \neg m_{\mathbf{S}_n} \neq m_{\mathbf{S}_n} \in [M], \boldsymbol{u} \in \{\boldsymbol{a}_s\}_{s \neq k \in [K]} \cup \{\boldsymbol{b}_s\}_{s \in [K]} \cup \{\boldsymbol{\nu}_{k'}\}_{k' \in [K']} \cup \{\boldsymbol{\xi}_{n,m}\}_{n \in [N], m \in [M]}$ *and* $\boldsymbol{v} \neq \boldsymbol{v}' \in \{\boldsymbol{b}_s\}_{s \in [K]} \cup \{\boldsymbol{\xi}_{n,m}\}_{n \in [N], m \in [M]}$. *At the end of this stage, we have*

$$\boldsymbol{a}_k^\top \mathbf{W}_V^{(T_1)} \boldsymbol{a}_k \geq \frac{\bar{C}_1 \sigma_1 d^{1/2}}{M} - \sqrt{2\log(\frac{8(2K+K')^2}{\delta})}\sigma_1,$$

$$(\mathbf{W}_Q^{(T_1)} \boldsymbol{a}_{\hat{k}})^\top (\mathbf{W}_K^{(T_1)} \boldsymbol{a}_{\hat{k}}) \geq \bar{C}_2 \frac{\sigma_0^2 \sigma_1^2 d^2}{M^2 q_V} - \bar{C}_3 (\frac{\eta^2 q_V \sigma_0^2}{\sigma_1^2 M^2 K^2} + 4\sqrt{\log(16(2K+K')^2/\delta)}\sigma_0^2 d^{1/2}).$$

(50)

*for some positive constants* $\bar{C}_{1-3}$.

We first examine the initialization. At $t = 0$, the results in Eq.(49) can be directly derived based on the orders of the initialized products and the norms presented in Lemma D.8, Lemma D.9, as well as the small initialization $\sigma_0 = O(d^{-1/2})$, low noise $\sigma_p = O(\|\boldsymbol{u}\| d^{-1/2})$ and overparameterization condition $d = \Omega(M^2 \log(\frac{K'^2 N^2 M^2}{\delta}))$ in Condition 3.1:

$$|(\mathbf{W}_Q^{(0)} \mathbf{x}_n)^\top \mathbf{W}_K^{(0)} (\boldsymbol{a}_{k_{\mathbf{S}_n}} + \boldsymbol{\xi}_{n,m_{\mathbf{S}_n}})|, \ |(\mathbf{W}_Q^{(0)} \mathbf{x}_n)^\top \mathbf{W}_K^{(0)} (\boldsymbol{\nu}_{n,\neg m_{\mathbf{S}_n}} + \boldsymbol{\xi}_{n,\neg m_{\mathbf{S}_n}})| \leq 12\sqrt{\log(16(2K+K')^2/\delta)}\sigma_0^2 d^{1/2},$$

$$|\boldsymbol{a}_k^\top \mathbf{W}_V^{(0)} \boldsymbol{a}_k| \leq \sqrt{2\log(\frac{8(2K+K')^2}{\delta})}\sigma_1 < o(\sigma_1 d^{1/2}), \ |\boldsymbol{u}^\top \mathbf{W}_V^{(0)} \boldsymbol{u}|, \ |\boldsymbol{a}_k{}^\top \mathbf{W}_V^{(0)} \boldsymbol{u}| \leq \sqrt{2\log(\frac{8(2K+K')^2}{\delta})}\sigma_1,$$

for $\forall k \in [K], n \in [N], s \in [7], \neg m_{\mathbf{S}_n} \neq m_{\mathbf{S}_n} \in [M], \mathbf{u} \in \{\mathbf{a}_s\}_{s \neq k \in [K]} \cup \{\mathbf{b}_s\}_{s \in [K]} \cup \{\boldsymbol{\nu}_{k'}\}_{k' \in [K']} \cup \{\boldsymbol{\xi}_{n,m}\}_{n \in [N], m \in [M]}$. By direct calculations, it holds that

$$\pi^{(0)}_{n,m_{\mathbf{S}_n}} = \Theta(\frac{1}{M}), \ \pi^{(0)}_{n,\neg m_{\mathbf{S}_n}} = \Theta(\frac{1}{M}),$$

$$0.98\sigma_1 d^{1/2} \leq \|\mathbf{W}^{(0)}_V \mathbf{S}_n \boldsymbol{\pi}^{(0)}_n\| \leq 1.02\sigma_1 d^{1/2},$$

$$|\frac{\mathbf{u}^\top_{k_{\mathbf{y}_n}} \mathbf{W}^{(0)}_V \mathbf{S}_n \boldsymbol{\pi}^{(0)}_n}{\|\mathbf{W}^{(0)}_V \mathbf{S}_n \boldsymbol{\pi}^{(0)}_n\|}|, \ |\frac{\mathbf{u}^\top_{\neg k_{\mathbf{y}_n}} \mathbf{W}^{(0)}_V \mathbf{S}_n \boldsymbol{\pi}^{(0)}_n}{\|\mathbf{W}^{(0)}_V \mathbf{S}_n \boldsymbol{\pi}^{(0)}_n\|}|, \ |\frac{\boldsymbol{\xi}^\top_{n,m} \mathbf{W}^{(0)}_V \mathbf{S}_n \boldsymbol{\pi}^{(0)}_n}{\|\mathbf{W}^{(0)}_V \mathbf{S}_n \boldsymbol{\pi}^{(0)}_n\|}| \leq 8\sqrt{2\log(\frac{8(2K+K')^2}{\delta})}/\sqrt{d} = o(0.01),$$

$$|\frac{(\mathbf{u}_{k_{\mathbf{y}_n}} - \sum_{k \in [7K+K']} \omega_{n,k} \mathbf{u}_k)^\top (\mathbf{W}^{(0)}_V \mathbf{S}_n \boldsymbol{\pi}^{(0)}_n)}{\|\mathbf{W}^{(0)}_V \mathbf{S}_n \boldsymbol{\pi}^{(0)}_n\|}| \leq 16\sqrt{2\log(\frac{8(2K+K')^2}{\delta})}/\sqrt{d} = o(0.01),$$

$$\frac{(\mathbf{W}^{(0)}_V \mathbf{S}_n \boldsymbol{\pi}^{(0)}_n)^\top}{\|\mathbf{W}^{(0)}_V \mathbf{S}_n \boldsymbol{\pi}^{(0)}_n\|} \frac{\mathbf{W}^{(0)}_V \mathbf{S}_n \pi^{(0)}_{n,m} \mathbf{e}_m}{\|\mathbf{W}^{(0)}_V \mathbf{S}_n \boldsymbol{\pi}^{(0)}_n\|} \leq \frac{1.05}{0.95M^2} + \frac{(M-1)(4\sqrt{\log(16(2K+K')^2/\delta)}d^{-1/2})}{0.95M^2} \leq \frac{1.11}{M^2}.$$

Since these scales are crucial to our analyses of gradients, we first introduce the following results based on Eq. (49) and Eq.(50) at iteration $t \leq T_1$.

**Lemma F.2.** *Under Condition 3.1, suppose Eq. (49) and Eq.(50) hold at any iteration $t \leq T_1$, then*

$$\pi^{(T_1)}_{n,m_{\mathbf{S}_n}} \leq \Theta(\frac{1}{1 + (M-1)e^{-\sigma_0^2 de^{q_V^{-1}\sigma_1^2 d}}}) < 0.95, \ \pi^{(T_1)}_{n,\neg m_{\mathbf{S}_n}} \geq \Theta(\frac{e^{-\sigma_0^2 de^{q_V^{-1}\sigma_1^2 d}}}{1 + (M-1)e^{-\sigma_0^2 de^{q_V^{-1}\sigma_1^2 d}}}),$$

$$\|\mathbf{W}^{(t)}_V \mathbf{S}_n \boldsymbol{\pi}^{(t)}_n\| = \Theta(\sigma_1 d^{1/2}),$$

$$|\frac{\mathbf{u}^\top_{7k+s} \mathbf{W}^{(t)}_V \mathbf{S}_n \boldsymbol{\pi}^{(t)}_n}{\|\mathbf{W}^{(t)}_V \mathbf{S}_n \boldsymbol{\pi}^{(t)}_n\|}| \leq O(\frac{1}{1 + (M-1)e^{-\sigma_0^2 de^{q_V^{-1}\sigma_1^2 d}}}),$$

$$|\frac{\mathbf{u}^\top_{7 \neg k+s} \mathbf{W}^{(t)}_V \mathbf{S}_n \boldsymbol{\pi}^{(t)}_n}{\|\mathbf{W}^{(t)}_V \mathbf{S}_n \boldsymbol{\pi}^{(t)}_n\|}|, \ |\frac{\boldsymbol{\xi}^\top_{n,m} \mathbf{W}^{(t)}_V \mathbf{S}_n \boldsymbol{\pi}^{(t)}_n}{\|\mathbf{W}^{(t)}_V \mathbf{S}_n \boldsymbol{\pi}^{(t)}_n\|}| \leq \Theta(8\sqrt{2\log(\frac{8(2K+K')^2}{\delta})}d^{-1/2}), \quad (51)$$

$$|\frac{(\mathbf{u}_{k_{\mathbf{y}_n}} - \sum_{k \in [7K+K']} \omega_{n,k} \mathbf{u}_k)^\top (\mathbf{W}^{(t)}_V \mathbf{S}_n \boldsymbol{\pi}^{(t)}_n)}{\|\mathbf{W}^{(t)}_V \mathbf{S}_n \boldsymbol{\pi}^{(t)}_n\|}| \leq O(\frac{1}{1 + (M-1)e^{-\sigma_0^2 de^{q_V^{-1}\sigma_1^2 d}}}),$$

$$\frac{(\mathbf{W}^{(t)}_V \mathbf{S}_n \boldsymbol{\pi}^{(t)}_n)^\top}{\|\mathbf{W}^{(t)}_V \mathbf{S}_n \boldsymbol{\pi}^{(t)}_n\|} \frac{\mathbf{W}^{(t)}_V \mathbf{S}_n \pi^{(t)}_{n,m} \mathbf{e}_m}{\|\mathbf{W}^{(t)}_V \mathbf{S}_n \boldsymbol{\pi}^{(t)}_n\|} = o(1) < 0.993 < 1,$$

*for $\forall k \in [K], \neg k \neq k \in [K], n \in \mathcal{N}_k \subset [N], s \in [7], m_{\mathbf{S}_n} \neq m_{\mathbf{S}_n} \in [M]$.*

*Proof.* Suggest Eq.(49) and Eq.(50) holds at $t$, then $\pi^{(T_1)}_{n,m_{\mathbf{S}_n}} \leq \Theta(\frac{1}{1+(M-1)e^{-\sigma_0^2 de^{q_V^{-1}\sigma_1^2 d}}})$, $\Theta(\frac{e^{-\sigma_0^2 de^{q_V^{-1}\sigma_1^2 d}}}{1+(M-1)e^{-\sigma_0^2 de^{q_V^{-1}\sigma_1^2 d}}}) \leq$ $\pi^{(T_1)}_{n,\neg m_{\mathbf{S}_n}}$ by $q_V = \Omega(\frac{\sigma_1^2 d}{\log(\sigma_0^{-2}d^{-1}\log(\frac{M-1}{0.06}))})$. Furthermore, as we see that $|\mathbf{u}^\top \mathbf{W}^{(t)}_V \mathbf{u}|$, $|\mathbf{a}_k{}^\top \mathbf{W}^{(t)}_V \mathbf{u}|$ is feeble compared to the growths of $\mathbf{a}_k^\top \mathbf{W}^{(t)}_V \mathbf{a}_k$ suggested in Eq.(49) and Eq.(50), thus the $\mathbf{a}_k^\top \mathbf{W}^{(t)}_V \mathbf{S}_n \pi_{n,m_{\mathbf{S}_n}} \mathbf{e}_{m_{\mathbf{S}_n}}$ would be primarily responsible to the changes of $\|\mathbf{W}^{(t)}_V \mathbf{S}_n \boldsymbol{\pi}^{(t)}_n\|$, which is the change of the term

$$(\pi^{(t)}_{n,m_{\mathbf{S}_n}})^2 \cdot (\mathbf{a}_k \cdot \cos \langle \mathbf{W}^{(t)}_V \mathbf{a}_k, \mathbf{a}_k \rangle \|\mathbf{W}^{(t)}_V \mathbf{a}_k\|)^\top (\mathbf{W}^{(t)}_V \mathbf{a}_k) = (\pi^{(t)}_{n,m_{\mathbf{S}_n}})^2 \cdot (\mathbf{a}_k^\top \mathbf{W}^{(t)}_V \mathbf{a}_k)^2,$$

where $\cos \langle \mathbf{W}^{(t)}_V \mathbf{a}_k, \mathbf{a}_k \rangle = \frac{\mathbf{a}_k^\top \mathbf{W}^{(t)}_V \mathbf{a}_k}{\|\mathbf{W}^{(t)}_V \mathbf{a}_k\|}$. Note that if $\mathbf{a}_k^\top \mathbf{W}^{(t)}_V \mathbf{a}_k < 0$ at $t = 0$, $\mathbf{a}_k^\top \mathbf{W}^{(t)}_V \mathbf{a}_k$ will increase and thus $(\mathbf{a}_k^\top \mathbf{W}^{(t)}_V \mathbf{a}_k)^2$ would decrease, which might lead to the decrease of $\|\mathbf{W}^{(t)}_V \mathbf{S}_n \boldsymbol{\pi}^{(t)}_n\|$. The scale of $\|\mathbf{W}^{(t)}_V \mathbf{S}_n \boldsymbol{\pi}^{(t)}_n\|^2$ would remain at $\Theta(\sigma_1^2 d)$ during $(\pi^{(t)}_{n,m_{\mathbf{S}_n}})^2 \cdot (\mathbf{a}_k^\top \mathbf{W}^{(t)}_V \mathbf{a}_k)^2 \leq O(\|\mathbf{W}^{(t)}_V \mathbf{S}_n \boldsymbol{\pi}^{(t)}_n\|^2) \Rightarrow (\pi^{(t)}_{n,m_{\mathbf{S}_n}}) \cdot (\mathbf{a}_k^\top \mathbf{W}^{(t)}_V \mathbf{a}_k) \leq O(\sigma_1 d^{1/2})$. This apparently holds during $t \leq T_1$ by Eq.(49). Therefore, we can safely conclude that $0.95\sigma_1 d^{1/2} \leq \|\mathbf{W}^{(t)}_V \mathbf{S}_n \boldsymbol{\pi}^{(t)}_n\| = \Theta(\sigma_1 d^{1/2})$.

Then, based on the scales in Eq.49 as well as Condition 3.1,similarly we can directly compute the last three inequality by the definition of $\mathbf{u}_k$ ($\mathbf{u}_{k_{\mathbf{y}_{\hat{n}}}} = \mathrm{LN}(\boldsymbol{a}_{k_{\mathbf{S}_n}} + y_{k_{\mathbf{S}_n}} \boldsymbol{b}_{k_{\mathbf{S}_n}})$):

$$\left|\frac{\mathbf{u}_{7k+s}^{\top}\mathbf{W}_V^{(t)}\mathbf{S}_n\boldsymbol{\pi}_n^{(t)}}{\|\mathbf{W}_V^{(t)}\mathbf{S}_n\boldsymbol{\pi}_n^{(t)}\|}\right| \leq \Theta\left(\frac{1}{1+(M-1)e^{-\sigma_0^2 d e^{q_V^{-1}\sigma_1^2 d}}} \frac{o(\sigma_1 d^{1/2})}{\Theta(\sigma_1 d^{1/2})}\right)$$

$$= O\left(\frac{1}{1+(M-1)e^{-\sigma_0^2 d e^{q_V^{-1}\sigma_1^2 d}}}\right),$$

$$\left|\frac{\mathbf{u}_{7\neg k+s}^{\top}\mathbf{W}_V^{(t)}\mathbf{S}_n\boldsymbol{\pi}_n^{(t)}}{\|\mathbf{W}_V^{(t)}\mathbf{S}_n\boldsymbol{\pi}_n^{(t)}\|}\right|, \left|\frac{\boldsymbol{\xi}_{n,m}^{\top}\mathbf{W}_V^{(t)}\mathbf{S}_n\boldsymbol{\pi}_n^{(t)}}{\|\mathbf{W}_V^{(t)}\mathbf{S}_n\boldsymbol{\pi}_n^{(t)}\|}\right| \leq 8\sqrt{2\log\left(\frac{8(2K+K')^2}{\delta}\right)}d^{-1/2},$$

$$\left|\frac{(\mathbf{u}_{k_{\mathbf{y}_n}} - \sum_{k\in[7K+K']}\omega_{n,k}\mathbf{u}_k)^{\top}(\mathbf{W}_V^{(t)}\mathbf{S}_n\boldsymbol{\pi}_n^{(t)})}{\|\mathbf{W}_V^{(t)}\mathbf{S}_n\boldsymbol{\pi}_n^{(t)}\|}\right| \leq \Theta\left(o(1)\frac{o(\sigma_1 d^{1/2})}{\Theta(\sigma_1 d^{1/2})}\frac{1}{1+(M-1)e^{-\sigma_0^2 d e^{q_V^{-1}\sigma_1^2 d}}}\right)$$

$$= O\left(\frac{1}{1+(M-1)e^{-\sigma_0^2 d e^{q_V^{-1}\sigma_1^2 d}}}\right),$$

$$\frac{(\mathbf{W}_V^{(t)}\mathbf{S}_n\boldsymbol{\pi}_n^{(t)})^{\top}}{\|\mathbf{W}_V^{(t)}\mathbf{S}_n\boldsymbol{\pi}_n^{(t)}\|}\frac{\mathbf{W}_V^{(t)}\mathbf{S}_n\boldsymbol{\pi}_{n,m}^{(t)}\boldsymbol{e}_m}{\|\mathbf{W}_V^{(t)}\mathbf{S}_n\boldsymbol{\pi}_n^{(t)}\|} \leq \Theta\left(\frac{\Theta(1)}{(1+(M-1)e^{-\sigma_0^2 d e^{q_V^{-1}\sigma_1^2 d}})^2}\right)$$

$$+ \Theta\left(\frac{(M-1)(4\sqrt{\log(16(2K+K')^2/\delta)}d^{-1/2})}{0.95M(1+(M-1)e^{-\sigma_0^2 d e^{q_V^{-1}\sigma_1^2 d}})}\right)$$

$$\leq \Theta\left(\frac{1}{(1+(M-1)e^{-\sigma_0^2 d e^{q_V^{-1}\sigma_1^2 d}})^2}\right) < 1.$$

$\square$

The remaining is to prove Eq.(49) during $t \leq T_1$. To facilitate the proofs, we introduce the following auxiliary lemmas.

**Lemma F.3.** *Under Condition 3.1, for the whole iteration $t \leq T^{\star} = \Omega(\eta^{-1}q_V^{-1}\sigma_1^2 KMd^2\log(\frac{1}{\epsilon}))$, we have*

$$\omega_{n,k}^{(t)} = \Theta\left(\frac{1}{K'}\right) = o(1), \tag{52}$$

*for $\forall n \in [N], k \in [K]$.*

*Proof.* By the definition of $\omega_{n,k}^{(t)}$, we see that for the whole iteration $t \leq T^{\star}$, there exists some constants $C_1' > C_2' > 0$,

$$\omega_{n,k}^{(t)} = \frac{\exp(\mathbf{u}_k^{\top}(\mathbf{x}_n) + \mathbf{u}_k^{\top}\mathrm{LN}(\sum_{m=1}^{M}\pi_{n,m}\mathbf{W}_V^{(t)}\mathbf{S}\boldsymbol{e}_m))}{\sum_{k\in[7K+K']}\exp(\mathbf{u}_k^{\top}(\mathbf{x}_n) + \mathbf{u}_k^{\top}\mathrm{LN}(\sum_{j=1}^{M}\pi_{n,m}\mathbf{W}_V^{(t)}\mathbf{S}\boldsymbol{e}_m))}$$

$$\leq \Theta\left(\frac{\exp(1.5)}{(7K+K')\exp(-1)}\right) \leq \Theta\left(\frac{13}{7K+K'}\right) = \Theta\left(\frac{C_1'}{K'}\right) = o(1), \tag{53}$$

$$\omega_{n,k}^{(t)} \geq \frac{\exp(-1)}{(7K+K')\exp(1.5)} \geq \Theta\left(\frac{0.08}{7K+K'}\right) = \Theta\left(\frac{C_2'}{K'}\right) = o(1).$$

Here, the inequality is by the definitions of $\mathbf{u}_k$ and LN as well as the large number of common token condition $K' = \Omega(K)$ in Condition 3.1. This would suggest that $\omega_{n,k}^{(t)} = \Theta(\frac{1}{K'})$ always hold during the whole iteration $t \leq T^{\star}$. $\square$

Based on this results, now we examine the evolving formulas of $\boldsymbol{u}^{\top}\mathbf{W}_V^{(t)}\boldsymbol{v}, \forall \boldsymbol{u}, \boldsymbol{v} \in \{\boldsymbol{a}_s\}_{s\in[K]} \cup \{\boldsymbol{b}_s\}_{s\in[K]} \cup \{\boldsymbol{\nu}_{k'}\}_{k'\in[K']}, \boldsymbol{\xi}_l, \boldsymbol{\xi}_l' \in \{\boldsymbol{\xi}_{n,m}\}_{n\in[N],m\in[M]}$ in the period $t \leq T_1$ based on Eq. (49).

**Lemma F.4.** *Under Condition 3.1, suppose Eq. (49) holds at iteration $t \leq T_1$, then $\boldsymbol{a}_k^\top \mathbf{W}_V^{(t)} \boldsymbol{a}_k$ is increasing such that*

$$
\begin{aligned}
\boldsymbol{a}_{\hat{k}}^\top \mathbf{W}_V^{(t+1)} \boldsymbol{a}_{\hat{k}} - \boldsymbol{a}_{\hat{k}}^\top \mathbf{W}_V^{(t)} \boldsymbol{a}_{\hat{k}} &= \Theta([\sum_{\hat{n} \in \mathcal{N}_{\hat{k}}} \frac{\eta q_V(M\pi_{\hat{n},m\mathbf{S}_{\hat{n}}}^{(t)})}{2NM\|\mathbf{W}_V^{(t)}\mathbf{S}_{\hat{n}}\boldsymbol{\pi}_{\hat{n}}^{(t)}\|}]), \\
|\boldsymbol{a}_{\hat{k}}^\top \mathbf{W}_V^{(t+1)} \boldsymbol{u} - \boldsymbol{a}_{\hat{k}}^\top \mathbf{W}_V^{(t)} \boldsymbol{u}|, \ |\boldsymbol{v}^\top \mathbf{W}_V^{(t+1)} \boldsymbol{v} - \boldsymbol{v}^\top \mathbf{W}_V^{(t)} \boldsymbol{v}|, \\
|\boldsymbol{v}^\top \mathbf{W}_V^{(t+1)} \boldsymbol{v}' - \boldsymbol{v}^\top \mathbf{W}_V^{(t)} \boldsymbol{v}'| &< 0.01(\boldsymbol{a}_{\hat{k}}^\top \mathbf{W}_V^{(t+1)} \boldsymbol{a}_{\hat{k}} - \boldsymbol{a}_{\hat{k}}^\top \mathbf{W}_V^{(t)} \boldsymbol{a}_{\hat{k}}),
\end{aligned}
\tag{54}
$$

*for $\forall \hat{k} \in [K], k' \in [K'], \boldsymbol{u} \in \{\boldsymbol{a}_s\}_{s \neq \hat{k} \in [K]} \cup \{\boldsymbol{b}_s\}_{s \in [K]} \cup \{\boldsymbol{\xi}_{n,m}\}_{n \in [N], m \in [M]}$ and $\boldsymbol{v} \neq \boldsymbol{v}' \in \{\boldsymbol{b}_s\}_{s \in [K]} \cup \{\boldsymbol{\xi}_{n,m}\}_{n \in [N], m \in [M]}$.*

*Proof.* To examine the last inequality, we first examine the update of $\mathbf{W}_V^{(t)}$. By Lemma E.1, denote $\mathcal{N}_k^y \subset \mathcal{P}_{\text{QA}}^{\text{tr}}$ as the index set of QA samples with their high-level task concept $k_\mathbf{S} = k \in [K]$ and task-specific label real value $y_{k_\mathbf{S}} = y \in [\pm 1]$, for $\forall \hat{k} \in [K], k' \neq \hat{k}, n' \in \mathcal{N}_{k'}$, we have the following near-orthogonal relationship

$$
(\mathbf{S}_{n'}\boldsymbol{\pi}_{n'}^{(t)})^\top \boldsymbol{a}_{\hat{k}} \leq \Theta(\frac{\boldsymbol{a}_{\hat{k}}^\top(\sum_{m \neq m\mathbf{S}_{n'} \in [M]}(\boldsymbol{\xi}_{n',m} + \boldsymbol{\nu}_{n',m}) + \boldsymbol{\xi}_{n',m\mathbf{S}_{n'}} + \boldsymbol{a}_{k'})}{M}) \leq \Theta(\sigma_p \cdot \sqrt{2\log(\frac{3(2K+K')NM}{\delta})})
$$

$$
= o(0.001)
$$

$$
|\boldsymbol{a}_{\hat{k}}^\top(\mathbf{u}_{k_{\mathbf{y}_{n'}}} - (\frac{(\mathbf{u}_{k_{\mathbf{y}_{n'}}} - \sum_{k \in [7K+K']}\omega_{n',k}\mathbf{u}_k)^\top(\mathbf{W}_V^{(t)}\mathbf{S}_{n'}\boldsymbol{\pi}_{n'}^{(t)})}{\|\mathbf{W}_V^{(t)}\mathbf{S}_{n'}\boldsymbol{\pi}_{n'}^{(t)}\|})\frac{\mathbf{W}_V^{(t)}\mathbf{S}_{n'}\boldsymbol{\pi}_{n'}^{(t)}}{\|\mathbf{W}_V^{(t)}\mathbf{S}_{n'}\boldsymbol{\pi}_{n'}^{(t)}\|})| \leq |o(\frac{\boldsymbol{a}_{\hat{k}}^\top\mathbf{W}_V^{(t)}\mathbf{S}_{n'}\boldsymbol{\pi}_{n'}^{(t)}}{\|\mathbf{W}_V^{(t)}\mathbf{S}_{n'}\boldsymbol{\pi}_{n'}^{(t)}\|})|
$$

$$
= o(-0.01)
$$

$$
\tag{55}
$$

Here, the first result is by $\boldsymbol{a}_{\hat{k}} \perp \boldsymbol{a}_{k'}$ (Eq.(49)): $\Theta(\frac{1}{M}) \leq \pi_{n,m\mathbf{S}_n}^{(t)} \leq \Theta(\frac{1}{1+(M-1)e^{-\sigma_0^2 de^{q_V^{-1}\sigma_1^2 d}}})$, $\Theta(\frac{e^{-\sigma_0^2 de^{q_V^{-1}\sigma_1^2 d}}}{1+(M-1)e^{-\sigma_0^2 de^{q_V^{-1}\sigma_1^2 d}}}) \leq \pi_{n,\neg m\mathbf{S}_n}^{(t)} \leq \Theta(\frac{1}{M})$ as well as the low noise condition $\sigma_p \leq d^{-1/2}/C$ in Condition 3.1; the second result is due to Eq.(51). The first (0.001) can actually be smaller if we require a larger $C$ in Condition 3.1, but for the simplicity of presentation, we here choose a feasible one. We latter would show that $(\mathbf{S}\boldsymbol{\pi}_{n'}^{(t)})^\top \boldsymbol{a}_{\hat{k}}$ would maintain the scale during the whole iteration.

Then we have

$$
\begin{aligned}
\boldsymbol{a}_{\hat{k}}^\top \mathbf{W}_V^{(t+1)} \boldsymbol{a}_{\hat{k}} - \boldsymbol{a}_{\hat{k}}^\top \mathbf{W}_V^{(t)} \boldsymbol{a}_{\hat{k}} &= \eta q_V \mathbb{E}_{\mathcal{P}_{\mathrm{QA}}^{\mathrm{tr}}} \Big[ \frac{\boldsymbol{a}_{\hat{k}}^\top \boldsymbol{\Pi}_{(\mathbf{W}_V \mathbf{S}_n \boldsymbol{\pi}_n^{(t)})^\perp}^{\mathbf{u}_{k_{\mathbf{y}_n}} - \sum_{k \in [7K+K']} \omega_{n,k} \mathbf{u}_k} (\mathbf{S} \boldsymbol{\pi}_n^{(t)})^\top \boldsymbol{a}_{\hat{k}}}{\|\mathbf{W}_V^{(t)} \mathbf{S}_n \boldsymbol{\pi}_n^{(t)}\|} \Big] \\
&= \frac{\eta q_V}{N} \sum_{k \in [K]} \sum_{y \in [\pm 1]} \sum_{n \in \mathcal{N}_k^y} \Big[ \frac{\boldsymbol{a}_{\hat{k}}^\top \boldsymbol{\Pi}_{(\mathbf{W}_V \mathbf{S}_n \boldsymbol{\pi}_n^{(t)})^\perp}^{\mathbf{u}_{k_{\mathbf{y}_n}} - \sum_{k \in [7K+K']} \omega_{n,k} \mathbf{u}_k} (\mathbf{S} \boldsymbol{\pi}_n^{(t)})^\top \boldsymbol{a}_{\hat{k}}}{\|\mathbf{W}_V^{(t)} \mathbf{S}_n \boldsymbol{\pi}_n^{(t)}\|} \Big] \\
&= \frac{\eta q_V}{N} \sum_{y \in [\pm 1]} \sum_{\hat{n} \in \mathcal{N}_{\hat{k}}^y} \Theta \Big( \Big[ \frac{\boldsymbol{a}_{\hat{k}}^\top \boldsymbol{\Pi}_{(\mathbf{W}_V \mathbf{S}_{\hat{n}} \boldsymbol{\pi}_{\hat{n}}^{(t)})^\perp}^{\mathbf{u}_{k_{\mathbf{y}_{\hat{n}}}} - \sum_{k \in [7K+K']} \omega_{\hat{n},k} \mathbf{u}_k} (\mathbf{S} \boldsymbol{\pi}_{\hat{n}}^{(t)})^\top \boldsymbol{a}_{\hat{k}}}{\|\mathbf{W}_V^{(t)} \mathbf{S}_{\hat{n}} \boldsymbol{\pi}_{\hat{n}}^{(t)}\|} \Big] \Big) \\
&= \frac{\eta q_V}{N} \sum_{\substack{y \in [\pm 1] \\ \hat{n} \in \mathcal{N}_{\hat{k}}^y}} \Theta \Big( \Big[ \frac{\boldsymbol{a}_{\hat{k}}^\top \big( \mathbf{u}_{k_{\mathbf{y}_{\hat{n}}}} - \big( \frac{(\mathbf{u}_{k_{\mathbf{y}_{\hat{n}}}} - \sum_{k \in [7K+K']} \omega_{\hat{n},k} \mathbf{u}_k)^\top (\mathbf{W}_V^{(t)} \mathbf{S}_{\hat{n}} \boldsymbol{\pi}_{\hat{n}}^{(t)})}{\|\mathbf{W}_V^{(t)} \mathbf{S}_{\hat{n}} \boldsymbol{\pi}_{\hat{n}}^{(t)}\|} \big) \frac{\mathbf{W}_V^{(t)} \mathbf{S}_{\hat{n}} \boldsymbol{\pi}_{\hat{n}}^{(t)}}{\|\mathbf{W}_V^{(t)} \mathbf{S}_{\hat{n}} \boldsymbol{\pi}_{\hat{n}}^{(t)}\|} \big)}{\|\mathbf{W}_V^{(t)} \mathbf{S}_{\hat{n}} \boldsymbol{\pi}_{\hat{n}}^{(t)}\| ((\pi_{\hat{n},m_{\mathbf{S}_{\hat{n}}}}^{(t)} \boldsymbol{a}_{\hat{k}} + \sum_m (\pi_{\hat{n},m}^{(t)} \boldsymbol{\xi}_{\hat{n},m}))^\top \boldsymbol{a}_{\hat{k}})^{-1}} \Big] \Big) \\
&= \frac{\eta q_V}{N} \sum_{\hat{n} \in \mathcal{N}_{\hat{k}}} \Theta \Big( \Big[ \frac{(1/2)(M \pi_{\hat{n},m_{\mathbf{S}_{\hat{n}}}}^{(t)} + \sigma_p M \cdot (M \pi_{\hat{n},\neg m_{\mathbf{S}_{\hat{n}}}}^{(t)}) \sqrt{2 \log(3(2K+K')NM\delta^{-1})}}{M \|\mathbf{W}_V^{(t)} \mathbf{S}_{\hat{n}} \boldsymbol{\pi}_{\hat{n}}^{(t)}\|} \Big] \Big) \\
&= \Theta \Big( \Big[ \sum_{\hat{n} \in \mathcal{N}_{\hat{k}}} \frac{\eta q_V (M \pi_{\hat{n},m_{\mathbf{S}_{\hat{n}}}}^{(t)} \pm 0.001)}{2NM \|\mathbf{W}_V^{(t)} \mathbf{S}_{\hat{n}} \boldsymbol{\pi}_{\hat{n}}^{(t)}\|} \Big] \Big)
\end{aligned}
$$
$$(56)$$

Here, the first and second equalities are by definition; the third equality is by Eq.(55); the forth equality is by definition of $\boldsymbol{\Pi}_{\mathbf{u}^\perp}^v$ in Eq.(30); the fifth equality is by the definition of $\mathbf{u}_k$ ($\mathbf{u}_{k_{\mathbf{y}_{\hat{n}}}} = \mathrm{LN}(\boldsymbol{a}_{k_{\mathbf{S}_n}} + y_{k_{\mathbf{S}_n}} \boldsymbol{b}_{k_{\mathbf{S}_n}})$), Lemma D.8 as well as Eq.(51); the last equality is by the low noise condition in Condition 3.1. Note that $\eta \leq o(q_V^{-1} \sigma_1^2 d^{\frac{1}{2}} K \sqrt{\log(\frac{K'^2}{\delta})})$

Similarly, we have

$$
\begin{aligned}
|\boldsymbol{b}_{\hat{k}}^\top \mathbf{W}_V^{(t+1)} \boldsymbol{a}_{\hat{k}} - \boldsymbol{b}_{\hat{k}}^\top \mathbf{W}_V^{(t)} \boldsymbol{a}_{\hat{k}}| &= \eta q_V \Big| \mathbb{E}_{\mathcal{P}_{\mathrm{QA}}^{\mathrm{tr}}} \Big[ \frac{\boldsymbol{b}_{\hat{k}}^\top \boldsymbol{\Pi}_{(\mathbf{W}_V^{(t)} \mathbf{S}_n \boldsymbol{\pi}_n^{(t)})^\perp}^{\mathbf{u}_{k_{\mathbf{y}_n}} - \sum_{k \in [7K+K']} \omega_{n,k} \mathbf{u}_k} (\mathbf{S}_n \boldsymbol{\pi}_n^{(t)})^\top \boldsymbol{a}_{\hat{k}}}{\|\mathbf{W}_V^{(t)} \mathbf{S}_n \boldsymbol{\pi}_n^{(t)}\|} \Big] \Big| \\
&= \frac{\eta q_V}{N} \sum_{k \in [K]} \sum_{y \in [\pm 1]} \sum_{n \in \mathcal{N}_k^y} \Big| \Big[ \frac{\boldsymbol{b}_{\hat{k}}^\top \boldsymbol{\Pi}_{(\mathbf{W}_V \mathbf{S}_n \boldsymbol{\pi}_n^{(t)})^\perp}^{\mathbf{u}_{k_{\mathbf{y}_n}} - \sum_{k \in [7K+K']} \omega_{n,k} \mathbf{u}_k} (\mathbf{S} \boldsymbol{\pi}_n^{(t)})^\top \boldsymbol{a}_{\hat{k}}}{\|\mathbf{W}_V^{(t)} \mathbf{S}_n \boldsymbol{\pi}_n^{(t)}\|} \Big] \Big| \\
&= \frac{\eta q_V}{N} \sum_{y \in [\pm 1]} \sum_{\hat{n} \in \mathcal{N}_{\hat{k}}^y} \Theta \Big( \Big| \Big[ \frac{\boldsymbol{b}_{\hat{k}}^\top \boldsymbol{\Pi}_{(\mathbf{W}_V \mathbf{S}_{\hat{n}} \boldsymbol{\pi}_{\hat{n}}^{(t)})^\perp}^{\mathbf{u}_{k_{\mathbf{y}_{\hat{n}}}} - \sum_{k \in [7K+K']} \omega_{\hat{n},k} \mathbf{u}_k} (\mathbf{S} \boldsymbol{\pi}_{\hat{n}}^{(t)})^\top \boldsymbol{a}_{\hat{k}}}{\|\mathbf{W}_V^{(t)} \mathbf{S}_{\hat{n}} \boldsymbol{\pi}_{\hat{n}}^{(t)}\|} \Big] \Big| \Big) \\
&= \frac{\eta q_V}{N} \sum_{\substack{y \in [\pm 1] \\ \hat{n} \in \mathcal{N}_{\hat{k}}^y}} \Theta \Big( \Big| \Big[ \frac{\boldsymbol{b}_{\hat{k}}^\top \big( \mathbf{u}_{k_{\mathbf{y}_{\hat{n}}}} - \big( \frac{(\mathbf{u}_{k_{\mathbf{y}_{\hat{n}}}} - \sum_{k \in [7K+K']} \omega_{\hat{n},k} \mathbf{u}_k)^\top (\mathbf{W}_V^{(t)} \mathbf{S}_{\hat{n}} \boldsymbol{\pi}_{\hat{n}}^{(t)})}{\|\mathbf{W}_V^{(t)} \mathbf{S}_{\hat{n}} \boldsymbol{\pi}_{\hat{n}}^{(t)}\|} \big) \frac{\mathbf{W}_V^{(t)} \mathbf{S}_{\hat{n}} \boldsymbol{\pi}_{\hat{n}}^{(t)}}{\|\mathbf{W}_V^{(t)} \mathbf{S}_{\hat{n}} \boldsymbol{\pi}_{\hat{n}}^{(t)}\|} \big)}{\|\mathbf{W}_V^{(t)} \mathbf{S}_{\hat{n}} \boldsymbol{\pi}_{\hat{n}}^{(t)}\| ((\pi_{\hat{n},m_{\mathbf{S}_{\hat{n}}}}^{(t)} \boldsymbol{a}_{\hat{k}} + \sum_m (\pi_{\hat{n},m}^{(t)} \boldsymbol{\xi}_{\hat{n},m}))^\top \boldsymbol{a}_{\hat{k}})^{-1}} \Big] \Big| \Big) \\
&\leq \Theta \Big( \sum_{\hat{n} \in \mathcal{N}_{\hat{k}}} \Big[ \frac{\eta q_V ((1 + (0.001 + 0.01))/2 - (1 - (0.001 + 0.01))/2)(M \pi_{\hat{n},m_{\mathbf{S}_{\hat{n}}}}^{(t)} + 0.001)/2}{NM \|\mathbf{W}_V^{(t)} \mathbf{S}_{\hat{n}} \boldsymbol{\pi}_{\hat{n}}^{(t)}\|} \Big] \Big) \\
&= \Theta \Big( \Big[ \sum_{\hat{n} \in \mathcal{N}_{\hat{k}}} \frac{\eta q_V (0.01) \pi_{\hat{n},m_{\mathbf{S}_{\hat{n}}}}^{(t)}}{2NM \|\mathbf{W}_V^{(t)} \mathbf{S}_{\hat{n}} \boldsymbol{\pi}_{\hat{n}}^{(t)}\|} \Big] \Big) << \boldsymbol{a}_{\hat{k}}^\top \mathbf{W}_V^{(t+1)} \boldsymbol{a}_{\hat{k}} - \boldsymbol{a}_{\hat{k}}^\top \mathbf{W}_V^{(t)} \boldsymbol{a}_{\hat{k}}.
\end{aligned}
$$
$$(57)$$

Here, the deduction of the first-to-forth equality follows Eq.(56); the fifth inequality is by Lemma D.10, the definition of $\mathbf{u}_k$ ($\mathbf{u}_{k_{\mathbf{y}_{\hat{n}}}} = \mathrm{LN}(\boldsymbol{a}_{k_{\mathbf{S}_n}} + y_{k_{\mathbf{S}_n}} \boldsymbol{b}_{k_{\mathbf{S}_n}})$), Eq.(51) as well as overparameterization condition $d = \Omega(M^2 \log(K'^2 N^2 M^2 / \delta))$.

**The value** $0.001$ **can be further reduced by increasing** $C$ **in Condition** 3.1. **However, for simplicity, we opt for a illustrative choice here.** Similarly, for $\forall \hat{k} \in [K], k' \in [K'], \boldsymbol{u} \in \{\boldsymbol{a}_s\}_{s \neq \hat{k} \in [K]} \cup \{\boldsymbol{b}_s\}_{s \in [K]} \cup \{\boldsymbol{\xi}_{n,m}\}_{n \in [N], m \in [M]}$ and $\boldsymbol{v} \neq \boldsymbol{v}' \in \{\boldsymbol{b}_s\}_{s \in [K]} \cup \{\boldsymbol{\xi}_{n,m}\}_{n \in [N], m \in [M]}$, we would have the following

$$|\boldsymbol{a}_{\hat{k}}^{\top} \mathbf{W}_V^{(t+1)} \boldsymbol{u} - \boldsymbol{a}_{\hat{k}}^{\top} \mathbf{W}_V^{(t)} \boldsymbol{u}| \leq \Theta\Big( \sum_{\hat{n} \in \mathcal{N}_{\hat{k}}} [\frac{\eta q_V(0.001)}{2NM \|\mathbf{W}_V^{(t)} \mathbf{S}_{\hat{n}} \boldsymbol{\pi}_{\hat{n}}^{(t)}\|}]\Big),$$

$$|\boldsymbol{u}^{\top} \mathbf{W}_V^{(t+1)} \boldsymbol{a}_{\hat{k}} - \boldsymbol{u}^{\top} \mathbf{W}_V^{(t)} \boldsymbol{a}_{\hat{k}}| \leq \Theta\Big( \sum_{\hat{n} \in \mathcal{N}_{\hat{k}}} [\frac{\eta q_V(0.01) M \pi_{\hat{n}, m_{\mathbf{S}_{\hat{n}}}}^{(t)}}{2NM \|\mathbf{W}_V^{(t)} \mathbf{S}_{\hat{n}} \boldsymbol{\pi}_{\hat{n}}^{(t)}\|}]\Big),$$

$$|\boldsymbol{v}^{\top} \mathbf{W}_V^{(t+1)} \boldsymbol{v} - \boldsymbol{v}^{\top} \mathbf{W}_V^{(t)} \boldsymbol{v}| \leq \Theta\Big( \sum_{\hat{n} \in \mathcal{N}_{\hat{k}}} [\frac{\eta q_V(0.01)(0.001)}{2NM \|\mathbf{W}_V^{(t)} \mathbf{S}_{\hat{n}} \boldsymbol{\pi}_{\hat{n}}^{(t)}\|}]\Big),$$

$$|\boldsymbol{v}^{\top} \mathbf{W}_V^{(t+1)} \boldsymbol{v}' - \boldsymbol{v}^{\top} \mathbf{W}_V^{(t)} \boldsymbol{v}'| \leq \Theta\Big( \sum_{\hat{n} \in \mathcal{N}_{\hat{k}}} [\frac{\eta q_V(0.01)(0.001)}{2NM \|\mathbf{W}_V^{(t)} \mathbf{S}_{\hat{n}} \boldsymbol{\pi}_{\hat{n}}^{(t)}\|}]\Big),$$

$$(58)$$

Here, $(0.001)$ is by the term $(\mathbf{S} \boldsymbol{\pi}_{\hat{n}}^{(t)})^{\top} \boldsymbol{u} = \Theta((\pi_{\hat{n}, m_{\mathbf{S}_{\hat{n}}}}^{(t)} \boldsymbol{a}_{\hat{k}} + \sum_m (\pi_{\hat{n}, m}^{(t)} \boldsymbol{\xi}_{\hat{n}, m}))^{\top} \boldsymbol{u})$ that would appear if $\boldsymbol{u}$ is on the right side, Lemma D.8 and the low noise condition $\sigma_p = O(d^{-1/2})$ in Condition 3.1; $(0.01)$ is by the third and forth inner products in our Eq.(51) as well as overparameterization condition $d = \Omega(M^2 \log(K'^2 N^2 M^2/\delta))$, Lemma D.8 and the low noise condition $\sigma_p = O(d^{-1/2})$ in Condition 3.1, as well as the balanced property of the QA data deduced in Lemma D.10.

Besides, it holds that for $\forall k \in [K], k' \in [K']$, we have

$$|\boldsymbol{\nu}_{k'}^{\top} \mathbf{W}_V^{(t+1)} \boldsymbol{a}_k - \boldsymbol{\nu}_{k'}^{\top} \mathbf{W}_V^{(t)} \boldsymbol{a}_k| = \frac{\eta q_V}{N} \sum_{\substack{y \in [\pm 1] \\ \hat{n} \in \mathcal{N}_{\hat{k}}^y}} \Theta\Big(\Big|\Big[\frac{\boldsymbol{\nu}_{k'}^{\top}(\boldsymbol{u}_{k \boldsymbol{y}_{\hat{n}}} - (\frac{(\boldsymbol{u}_{k \boldsymbol{y}_{\hat{n}}} - \sum_{k \in [7K+K']} \omega_{\hat{n}, k} \boldsymbol{u}_k)^{\top} (\mathbf{W}_V^{(t)} \mathbf{S}_{\hat{n}} \boldsymbol{\pi}_{\hat{n}}^{(t)})}{\|\mathbf{W}_V^{(t)} \mathbf{S}_{\hat{n}} \boldsymbol{\pi}_{\hat{n}}^{(t)}\|}) \frac{\mathbf{W}_V^{(t)} \mathbf{S}_{\hat{n}} \boldsymbol{\pi}_{\hat{n}}^{(t)}}{\|\mathbf{W}_V^{(t)} \mathbf{S}_{\hat{n}} \boldsymbol{\pi}_{\hat{n}}^{(t)}\|})}{\|\mathbf{W}_V^{(t)} \mathbf{S}_{\hat{n}} \boldsymbol{\pi}_{\hat{n}}^{(t)}\|((\pi_{\hat{n}, m_{\mathbf{S}_{\hat{n}}}}^{(t)} \boldsymbol{a}_{\hat{k}} + \sum_m (\pi_{\hat{n}, m}^{(t)} \boldsymbol{\xi}_{\hat{n}, m}))^{\top} \boldsymbol{a}_{\hat{k}})^{-1}}\Big]\Big|\Big)$$

$$\leq \Theta\Big( \sum_{\hat{n} \in \mathcal{N}_{\hat{k}}} [\frac{\eta q_V(0.01)(M \pi_{\hat{n}, m_{\mathbf{S}_{\hat{n}}}}^{(t)} + 0.001)}{2NM \|\mathbf{W}_V^{(t)} \mathbf{S}_{\hat{n}} \boldsymbol{\pi}_{\hat{n}}^{(t)}\|}]\Big),$$

$$|\boldsymbol{\nu}_{k'}^{\top} \mathbf{W}_V^{(t+1)} \boldsymbol{b}_k - \boldsymbol{\nu}_{k'}^{\top} \mathbf{W}_V^{(t)} \boldsymbol{b}_k| \leq \Theta\Big( \sum_{\hat{n} \in \mathcal{N}_{\hat{k}}} [\frac{\eta q_V(0.01)(0.001)}{2NM \|\mathbf{W}_V^{(t)} \mathbf{S}_{\hat{n}} \boldsymbol{\pi}_{\hat{n}}^{(t)}\|}]\Big),$$

$$(59)$$

where the left $(0.01)$ is by (Eq.(51)) as well as the overparameterization condition $d = \Omega(M^2 \log(K'^2 N^2 M^2/\delta))$ in Condition 3.1. The right term, $(M \pi_{\hat{n}, m_{\mathbf{S}_{\hat{n}}}}^{(t)} + 0.001)$ or $(0.001)$, follows from the calculations of $(\mathbf{S} \boldsymbol{\pi}_{\hat{n}}^{(t)})^{\top} \boldsymbol{u} = \Theta((\pi_{\hat{n}, m_{\mathbf{S}_{\hat{n}}}}^{(t)} \boldsymbol{a}_{\hat{k}} + \sum_m (\pi_{\hat{n}, m}^{(t)} \boldsymbol{\xi}_{\hat{n}, m}))^{\top} \boldsymbol{u}), \boldsymbol{u} \in \{\boldsymbol{a}_k, \boldsymbol{b}_k\}_{k \in [K]}$, based on $\Theta(\frac{1}{M}) \leq \pi_{n, m_{\mathbf{S}_n}}^{(t)} \leq \Theta(\frac{1}{1 + (M-1)e^{-\sigma_0^2 de^q V^{-1} \sigma_1^2 d}})$, $\Theta(\frac{e^{-\sigma_0^2 de^q V^{-1} \sigma_1^2 d}}{1 + (M-1)e^{-\sigma_0^2 de^q V^{-1} \sigma_1^2 d}}) \leq \pi_{n, \neg m_{\mathbf{S}_n}}^{(t)} \leq \Theta(\frac{1}{M})$ in Eq.(49), as well as the low noise condition $\sigma_p = O(d^{-1/2})$ in Condition 3.1. In addition, denotes $\boldsymbol{a}_{\hat{k}} = \boldsymbol{\nu}_{n,m}$, by Lemma D.11 and $K' = \Omega(M)$ in Condition 3.1, as well as the strategy

above, the following holds

$$|\boldsymbol{a}_k^\top \mathbf{W}_V^{(t+1)} \boldsymbol{\nu}_{k'} - \boldsymbol{a}_k^\top \mathbf{W}_V^{(t)} \boldsymbol{\nu}_{k'}| = \frac{\eta q_V}{N} \sum_{\substack{y \in [\pm 1] \\ \hat{n} \in \mathcal{N}_{\hat{k}}^y}} \Theta(|[\frac{\boldsymbol{a}_k^\top (\mathbf{u}_{k_{\mathbf{y}_{\hat{n}}}} - (\frac{(\mathbf{u}_{k_{\mathbf{y}_{\hat{n}}}} - \sum_{k \in [7K+K']} \omega_{\hat{n},k} \mathbf{u}_k)^\top (\mathbf{W}_V^{(t)} \mathbf{S}_{\hat{n}} \boldsymbol{\pi}_{\hat{n}}^{(t)})}{\|\mathbf{W}_V^{(t)} \mathbf{S}_{\hat{n}} \boldsymbol{\pi}_{\hat{n}}^{(t)}\|}) \frac{\mathbf{W}_V^{(t)} \mathbf{S}_{\hat{n}} \boldsymbol{\pi}_{\hat{n}}^{(t)}}{\|\mathbf{W}_V^{(t)} \mathbf{S}_{\hat{n}} \boldsymbol{\pi}_{\hat{n}}^{(t)}\|})}{\|\mathbf{W}_V^{(t)} \mathbf{S}_{\hat{n}} \boldsymbol{\pi}_{\hat{n}}^{(t)}\|((\sum_m (\pi_{\hat{n},m}^{(t)} \boldsymbol{\nu}_{\hat{n},m} + \boldsymbol{\xi}_{\hat{n},m})^\top \boldsymbol{\nu}_{k'})^{-1}}]|)$$

$$\leq \Theta(\sum_{\hat{n} \in \mathcal{N}_{\hat{k}} \cap \mathcal{N}_{\boldsymbol{\nu}_{k'}}} [\frac{\eta q_V (M \pi_{\hat{n},m_n,\boldsymbol{\nu}_{k'}}^{(t)} + 0.001)}{2NM\|\mathbf{W}_V^{(t)} \mathbf{S}_{\hat{n}} \boldsymbol{\pi}_{\hat{n}}^{(t)}\|}]) \leq \Theta(\sum_{\hat{n} \in \mathcal{N}_{\hat{k}}} [\frac{\eta q_V M \pi_{\hat{n},m_n,\boldsymbol{\nu}_{k'}}^{(t)} (0.01)}{2NM\|\mathbf{W}_V^{(t)} \mathbf{S}_{\hat{n}} \boldsymbol{\pi}_{\hat{n}}^{(t)}\|}]),$$

$$|\boldsymbol{b}_k^\top \mathbf{W}_V^{(t+1)} \boldsymbol{\nu}_{k'} - \boldsymbol{b}_k^\top \mathbf{W}_V^{(t)} \boldsymbol{\nu}_{k'}| \leq \Theta(\sum_{\hat{n} \in \mathcal{N}_{\hat{k}} \cap \mathcal{N}_{\boldsymbol{\nu}_{k'}}} [\frac{\eta q_V (0.01)(M \pi_{\hat{n},m_n,\boldsymbol{\nu}_{k'}}^{(t)} + 0.001)}{2NM\|\mathbf{W}_V^{(t)} \mathbf{S}_{\hat{n}} \boldsymbol{\pi}_{\hat{n}}^{(t)}\|}])$$

$$\leq \Theta(\sum_{\hat{n} \in \mathcal{N}_{\hat{k}}} [\frac{\eta q_V M \pi_{\hat{n},m_n,\boldsymbol{\nu}_{k'}}^{(t)} (0.01)^2}{2NM\|\mathbf{W}_V^{(t)} \mathbf{S}_{\hat{n}} \boldsymbol{\pi}_{\hat{n}}^{(t)}\|}]),$$

(60)

where $\mathcal{N}_{\boldsymbol{\nu}_{k'}} \subset \mathcal{P}_{\mathrm{QA}}^{\mathrm{tr}}$ denote the index set of QA samples with $\boldsymbol{\nu}_{k'}$ in the sentence, and $m_{n,\boldsymbol{\nu}_{k'}}$ denotes the position index of $\boldsymbol{\nu}_{k'}$ when $n \in \mathcal{N}_{\boldsymbol{\nu}_{k'}}$.

By Eq.(56), (57), (58), (59), (60), $\Theta(\frac{1}{M}) \leq \pi_{n,m_{\mathbf{S}_n}}^{(t)} \leq \Theta(\frac{1}{1+(M-1)e^{-\sigma_0^2 de q_V^{-1}\sigma_1^2 d}})$, $\Theta(\frac{e^{-\sigma_0^2 de q_V^{-1}\sigma_1^2 d}}{1+(M-1)e^{-\sigma_0^2 de q_V^{-1}\sigma_1^2 d}}) \leq \pi_{n,\neg m_{\mathbf{S}_n}}^{(t)} \leq \Theta(\frac{1}{M})$ that for $\boldsymbol{v}^\top \mathbf{W}_V^{(t)} \boldsymbol{u}, \boldsymbol{v}, \boldsymbol{u} \in \{\boldsymbol{a}_s\}_{s \in [K]} \cup \{\boldsymbol{b}_s\}_{s \in [K]} \cup \{\boldsymbol{\nu}_{k'}\}_{k' \in [K']} \cup \{\boldsymbol{\xi}_{n,m}\}_{n \in [N], m \in [M]}$, only when $\boldsymbol{v} = \boldsymbol{u} = \boldsymbol{a}_k, \forall k \in [K]$ the products would grow in a non-negligible manner. Therefore, by definition of $m_{\mathbf{S}_n}$ and low noise condition $\sigma_p = O(d^{-1/2})$, the primary contributor to the change of $\|\mathbf{W}_V^{(t)} \mathbf{S}_n \boldsymbol{\pi}_n^{(t)}\|^2$ is the increase in $\pi_{n,m_{\mathbf{S}_n}}^{(t)}$ and $\boldsymbol{a}_k^\top \mathbf{W}_V^{(t)} \boldsymbol{a}_k$

$\square$

The following lemma examines two scenarios for the upper and lower bounds of the growth evolution of $\boldsymbol{a}_k^\top \mathbf{W}_V^{(t)} \boldsymbol{a}_k$.

**Lemma F.5.** *Under Condition 3.1, suppose Eq. (49) holds at iteration* $t \leq T_1 \leq t_2^\star$, *where* $t_2^\star := (\eta q_V)^{-1} C_2^{-1} \sigma_1 d^{1/2} K (d^{1/2} - \sqrt{2\log(\frac{8(2K+K')^2}{\delta})})$. *Then for* $\forall k \in [K]$ *and* $n \in \mathcal{N}_k$, *there exists some* $C_{1-2}, C_{1-2}', \hat{C}_{1-2} > 0$ *such that*

$$\frac{C_1 \eta q_V t}{\sigma_1 d^{1/2} MK} - \sqrt{2\log(\frac{8(2K+K')^2}{\delta})} \sigma_1 \leq \boldsymbol{a}_k^\top \mathbf{W}_V^{(t)} \boldsymbol{a}_k \leq \frac{C_2 \eta q_V t}{\sigma_1 d^{1/2} K} + \sqrt{2\log(\frac{8(2K+K')^2}{\delta})} \sigma_1,$$

$$\frac{\hat{C}_1 C_1 \eta q_V t}{\sigma_1^2 dM^2 K} - \frac{\hat{C}_1 \sqrt{2\log(\frac{8(2K+K')^2}{\delta})}}{Md^{1/2}} \leq \frac{\boldsymbol{a}_k^\top \mathbf{W}_V^{(t)} \mathbf{S}_n \pi_{n,m_{\mathbf{S}_n}}^{(t)} \boldsymbol{e}_{m_{\mathbf{S}_n}}}{\|\mathbf{W}_V^{(t)} \mathbf{S}_n \boldsymbol{\pi}_n^{(t)}\|} \leq \frac{\hat{C}_2 C_2 \eta q_V t}{\sigma_1^2 dK} + \frac{\hat{C}_2 \sqrt{2\log(\frac{8(2K+K')^2}{\delta})}}{\sigma_1 d^{1/2}}.$$

(61)

*Proof.* By Lemma D.8 and Condition 3.1.

At $t = 0$, by Lemma D.10 and Eq.(51) we have

$$\boldsymbol{a}_k^\top \mathbf{W}_V^{(t+1)} \boldsymbol{a}_k - \boldsymbol{a}_k^\top \mathbf{W}_V^{(t)} \boldsymbol{a}_k \geq \Theta(\frac{\eta q_V}{2MK(\sigma_1 d^{1/2})}),$$

(62)

Therefore, by Lemma D.1 and Lemma D.8, for some $C_1 > 0$ we have

$$\boldsymbol{a}_k^\top \mathbf{W}_V^{(t)} \boldsymbol{a}_k \geq C_1 \frac{\eta q_V t}{\sigma_1 d^{1/2} MK} - \sqrt{2\log(\frac{8(2K+K')^2}{\delta})} \sigma_1.$$

for $0 \leq t \leq T_1$. Define the positive estimate of the time point when the lower bound evolution of $\boldsymbol{a}_k^\top \mathbf{W}_V^{(t)} \boldsymbol{a}_k$ has potential to reach $O(\sigma_1 d^{1/2})$ as $t_1^\star := (\eta q_V)^{-1} C_1^{-1} \sigma_1^2 d^{1/2} MK (d^{1/2} + \sqrt{2\log(\frac{8(2K+K')^2}{\delta})})$. Note that $t_2^\star \leq t_1^\star$ thus we

have $T_1 \leq t_1^\star$. Collaborating with the non-decreasing nature of $\boldsymbol{a}_k^\top \mathbf{W}_V^{(t)} \boldsymbol{a}_k$, the non-increasing nature of $g(x) = 1/x$,

$$\Theta(\tfrac{1}{M}) \leq \pi_{n,m_{\mathbf{S}_n}}^{(t)} \leq \Theta(\frac{1}{1+(M-1)e^{-\sigma_0^2 de^{q_V^{-1}\sigma_1^2 d}}}), \ \Theta(\frac{e^{-\sigma_0^2 de^{q_V^{-1}\sigma_1^2 d}}}{1+(M-1)e^{-\sigma_0^2 de^{q_V^{-1}\sigma_1^2 d}}}) \leq \pi_{n,\neg m_{\mathbf{S}_n}}^{(t)} \leq \Theta(\tfrac{1}{M}), \text{ for some } \hat{C}_1 > 0, \text{ we}$$

directly have

$$\frac{\boldsymbol{a}_k^\top \mathbf{W}_V^{(t)} \mathbf{S}_n \pi_{n,m_{\mathbf{S}_n}}^{(t)} \boldsymbol{e}_{m_{\mathbf{S}_n}}}{\|\mathbf{W}_V^{(t)} \mathbf{S}_n \boldsymbol{\pi}_n^{(t)}\|} \geq \hat{C}_1 \pi_{n,m_{\mathbf{S}_n}}^{(t)} (C_1 \frac{\eta q_V t}{\sigma_1^2 dMK} - \sqrt{2\log(\frac{8(2K+K')^2}{\delta})}/(1.02d^{1/2})).$$

$$\geq \hat{C}_1 (C_1 \frac{\eta q_V t}{\sigma_1^2 dM^2 K} - \sqrt{2\log(\frac{8(2K+K')^2}{\delta})}/(Md^{1/2})).$$

In terms of the upper bound, by similar approaches, for some $C_2, C_2', \hat{C}_2 > 0$ we have

$$\boldsymbol{a}_k^\top \mathbf{W}_V^{(t)} \boldsymbol{a}_k \leq \frac{C_2 \eta q_V t}{\sigma_1 d^{1/2} K} + \sqrt{2\log(\frac{8(2K+K')^2}{\delta})}\sigma_1,$$

$$\frac{\boldsymbol{a}_k^\top \mathbf{W}_V^{(t)} \mathbf{S}_n \pi_{n,m_{\mathbf{S}_n}}^{(t)} \boldsymbol{e}_{m_{\mathbf{S}_n}}}{\|\mathbf{W}_V^{(t)} \mathbf{S}_n \boldsymbol{\pi}_n^{(t)}\|} \leq \frac{\hat{C}_2 C_2 \eta q_V t}{\sigma_1^2 dK} + \frac{\hat{C}_2 \sqrt{2\log(\frac{8(2K+K')^2}{\delta})}}{\sigma_1 d^{1/2}},$$

for $0 \leq t \leq t_2^\star := (\eta q_V)^{-1} C_2^{-1} \sigma_1^2 d^{1/2} K(d^{1/2} - \sqrt{2\log(\frac{8(2K+K')^2}{\delta})})$. Note that as $d = \Omega(M^2 \log(\frac{K'^2 N^2 M^2}{\delta}))$ by Condition 3.1, we have $t_2^\star >> 0$. We can thus appropriately choose $T_1 \leq t_2^\star$.

We also have

$$\boldsymbol{a}_k^\top \mathbf{W}_V^{(T_1)} \boldsymbol{a}_k \geq \frac{C_2^{-1} C_1 O(\sigma_1 d^{1/2})}{M} - \sqrt{2\log(\frac{8(2K+K')^2}{\delta})}\sigma_1$$

$$\geq \frac{\bar{C}_1 \sigma_1 d^{1/2}}{M} - \sqrt{2\log(\frac{8(2K+K')^2}{\delta})}\sigma_1,$$

$\square$

**Lemma F.6.** *Under Condition 3.1, suppose Eq. (49) holds at iteration $t \leq T_1$, then for $\forall k \in [K]$ and $n \in \mathcal{N}_k$, it holds that that*

$$\Theta(\frac{\sigma_p^2 d}{M^2}) \leq \frac{(\mathbf{W}_V^{(t)} \mathbf{S}_n \boldsymbol{\pi}_n^{(t)})^\top}{\|\mathbf{W}_V^{(t)} \mathbf{S}_n \boldsymbol{\pi}_n^{(t)}\|} \frac{\mathbf{W}_V^{(t)} \mathbf{S}_n \pi_{n,m_{\mathbf{S}_n}}^{(t)} \boldsymbol{e}_{n,m_{\mathbf{S}_n}}}{\|\mathbf{W}_V^{(t)} \mathbf{S}_n \boldsymbol{\pi}_n^{(t)}\|} \leq \Theta((\pi_{n,m_{\mathbf{S}_n}}^{(t)})^2) \tag{63}$$

*for $0 \leq t \leq T_1$.*

*Proof.* In terms of the lower bounds, in the scenario (1) and (2), for the numerator it holds that

$$(\mathbf{W}_V^{(t)} \mathbf{S}_n \boldsymbol{\pi}_n^{(t)})^\top \mathbf{W}_V^{(t)} \mathbf{S}_n \pi_{n,m_{\mathbf{S}_n}}^{(t)} \boldsymbol{e}_{n,m_{\mathbf{S}_n}} \geq \Theta((\mathbf{W}_V^{(t)} \mathbf{S}_n \pi_{n,m_{\mathbf{S}_n}}^{(t)} \boldsymbol{e}_{n,m_{\mathbf{S}_n}})^2)$$

$$\geq \Theta(\frac{1}{M^2}(\mathbf{W}_V^{(t)} \mathbf{S}_n \boldsymbol{e}_{n,m_{\mathbf{S}_n}})^2) = \Theta(\frac{1}{M^2}(\mathbf{W}_V^{(t)}(\boldsymbol{a}_k + \boldsymbol{\xi}_{n,m_{\mathbf{S}_n}}))^2),$$

where the inequalities are by $\mathbf{W}_V^{(t)} \mathbf{S}_n \boldsymbol{\pi}_n^{(t)} = \sum_{m \in [M]} \mathbf{W}_V^{(t)} \mathbf{S}_n \pi_{n,m}^{(t)} \boldsymbol{e}_{n,m}$, Eq.(51) and $\Theta(\tfrac{1}{M}) \leq \pi_{n,m_{\mathbf{S}_n}}^{(t)} \leq \Theta(\frac{1}{1+(M-1)e^{-\sigma_0^2 de^{q_V^{-1}\sigma_1^2 d}}})$, $\Theta(\frac{e^{-\sigma_0^2 de^{q_V^{-1}\sigma_1^2 d}}}{1+(M-1)e^{-\sigma_0^2 de^{q_V^{-1}\sigma_1^2 d}}}) \leq \pi_{n,\neg m_{\mathbf{S}_n}}^{(t)} \leq \Theta(\tfrac{1}{M})$. Therefore, the results are valid by considering the lower bounds of $\|\mathbf{W}_V^{(0)} \boldsymbol{\xi}_l\|^2 \leq 3\sigma_1^2 \sigma_p^2 d^2$, and upper bound $\|\mathbf{W}_V^{(t)} \mathbf{S}_n \boldsymbol{\pi}_n^{(t)}\| = \Theta(\sigma_1 d^{1/2})$ by Eq.(51).

Now consider the upper bound. Observe that

$$
\begin{aligned}
\frac{(\mathbf{W}_V^{(t)}\mathbf{S}_n\boldsymbol{\pi}_n^{(t)})^\top}{\|\mathbf{W}_V^{(t)}\mathbf{S}_n\boldsymbol{\pi}_n^{(t)}\|}\frac{\mathbf{W}_V^{(t)}\mathbf{S}_n\pi_{n,m_{\mathbf{S}_n}}^{(t)}\boldsymbol{e}_{n,m_{\mathbf{S}_n}}}{\|\mathbf{W}_V^{(t)}\mathbf{S}_n\boldsymbol{\pi}_n^{(t)}\|} = {} & \frac{(\mathbf{W}_V^{(t)}\mathbf{S}_n\pi_{n,m_{\mathbf{S}_n}}^{(t)}\boldsymbol{e}_{n,m_{\mathbf{S}_n}})^2}{\|\mathbf{W}_V^{(t)}\mathbf{S}_n\boldsymbol{\pi}_n^{(t)}\|^2} \\
& + \frac{\sum_{m\neq m_{\mathbf{S}_n}}(\mathbf{W}_V^{(t)}\mathbf{S}_n\pi_{n,m_{\mathbf{S}_n}}^{(t)}\boldsymbol{e}_{n,m_{\mathbf{S}_n}})^\top(\mathbf{W}_V^{(t)}\mathbf{S}_n\pi_{n,m}^{(t)}\boldsymbol{e}_{n,m})}{\|\mathbf{W}_V^{(t)}\mathbf{S}_n\boldsymbol{\pi}_n^{(t)}\|^2} \\
\leq {} & \frac{(\pi_{n,m_{\mathbf{S}_n}}^{(t)})^2\Theta(\sigma_1^2 d)+\pi_{n,m_{\mathbf{S}_n}}^{(t)}(\frac{M}{M})8\sqrt{\log(\frac{16(2K+K')^2}{\delta})}\sigma_1^2 d^{1/2}}{\Theta(\sigma_1^2 d)} \\
= {} & \Theta((\pi_{n,m_{\mathbf{S}_n}}^{(t)})^2).
\end{aligned}
$$

The first inequality is by the positive estimate of the products $(\mathbf{W}_V^{(t)}\mathbf{S}_n\pi_{n,m_{\mathbf{S}_n}}^{(t)}\boldsymbol{e}_{n,m_{\mathbf{S}_n}})^\top(\mathbf{W}_V^{(t)}\mathbf{S}_n\pi_{n,m}^{(t)}\boldsymbol{e}_{n,m})$, the lower and upper bounds of $\|\mathbf{W}_V^{(t)}\mathbf{S}_n\boldsymbol{\pi}_n^{(t)}\|$ as well as $\|\mathbf{W}_V^{(t)}\boldsymbol{a}_k\|$; the last inequality is by Eq.(51).

$\square$

**Lemma F.7.** *Under Condition 3.1, suppose Eq. (49) holds at iteration $t \leq T_1$, then for $\forall \hat{k} \in [K]$, $\boldsymbol{u} \in \{\boldsymbol{a}_s\}_{s\neq\hat{k}\in[K]} \cup \{\boldsymbol{b}_s\}_{s\in[K]} \cup \{\boldsymbol{\xi}_{n,m}\}_{n\in[N],m\in[M]}$ and $\boldsymbol{v} \neq \boldsymbol{v}' \in \{\boldsymbol{b}_s\}_{s\in[K]} \cup \{\boldsymbol{\xi}_{n,m}\}_{n\in[N],m\in[M]}$, there exist some positive constants $C_3, C_4, C_5, C_6$, such that*

- *If $\boldsymbol{a}_{\hat{k}}^\top\mathbf{W}_V^{(t)}\boldsymbol{a}_{\hat{k}} \leq 0$ at $t = 0$, after at most $t_3 = (\eta q_V)^{-1}C_1^{-1}\sigma_1^2 d^{1/2}MK(\sqrt{2\log(\frac{8(2K+K')^2}{\delta})})$ iterations, $\boldsymbol{a}_{\hat{k}}^\top\mathbf{W}_V^{(t)}\boldsymbol{a}_{\hat{k}}$ would get positive. During this period, we have*

$$
|(\mathbf{W}_Q^{(t)}\boldsymbol{a}_{\hat{k}})^\top(\mathbf{W}_K^{(t)}\boldsymbol{a}_{\hat{k}})| \leq 4\sqrt{\log(16(2K+K')^2/\delta)}\sigma_0^2 d^{1/2}. \tag{64}
$$

- *During $0 \leq \boldsymbol{a}_k^\top\mathbf{W}_V^{(t)}\boldsymbol{a}_k \leq O(\sigma_1 d^{1/2})$ within $[t_4, T_1]$, where $t_4$ satisfying $0 \leq t_4 \leq t_3$, it holds that*

$$
\begin{aligned}
(\mathbf{W}_Q^{(t)}\boldsymbol{a}_{\hat{k}})^\top(\mathbf{W}_K^{(t)}\boldsymbol{a}_{\hat{k}}) \geq {} & \frac{C_3\eta^2 q_V\sigma_0^2[(t-t_4-1)^2-1]}{\sigma_1^2 M^2 K^2} - 4\sqrt{\log(16(2K+K')^2/\delta)}\sigma_0^2 d^{1/2}, \\
(\mathbf{W}_Q^{(t)}\boldsymbol{a}_{\hat{k}})^\top(\mathbf{W}_K^{(t)}\boldsymbol{a}_{\hat{k}}) \leq {} & C_4((4\sqrt{\log(16(2K+K')^2/\delta)}\sigma_0^2 d^{1/2}+\frac{3\sigma_0^2 d}{2})+\frac{3\sigma_0^2 d}{2}\exp(\frac{\hat{C}_2 C_2\eta^2 q_V t^2}{\sigma_1^2 dK^2})), \\
(\mathbf{W}_Q^{(t)}\boldsymbol{a}_{\hat{k}})^\top(\mathbf{W}_K^{(t)}\boldsymbol{u}), (\mathbf{W}_Q^{(t)}\boldsymbol{v})^\top(\mathbf{W}_K^{(t)}\boldsymbol{v}), (\mathbf{W}_Q^{(t)}\boldsymbol{v})^\top(\mathbf{W}_K^{(t)}\boldsymbol{v}') = {} & O(\sqrt{\log(16(2K+K')^2/\delta)}\sigma_0^2 d^{1/2}).
\end{aligned} \tag{65}
$$

- *Furthermore, it holds that*

$$
\begin{aligned}
(\mathbf{W}_Q^{(T_1)}\boldsymbol{a}_{\hat{k}})^\top(\mathbf{W}_K^{(T_1)}\boldsymbol{a}_{\hat{k}}) \geq {} & C_6\frac{\sigma_0^2\sigma_1^2 d^2}{M^2 q_V} - \frac{C_3\eta^2 q_V\sigma_0^2}{\sigma_1^2 M^2 K^2} - 4C_3\sqrt{\log(16(2K+K')^2/\delta)}\sigma_0^2 d^{1/2}, \\
(\mathbf{W}_Q^{(T_1)}\boldsymbol{a}_{\hat{k}})^\top(\mathbf{W}_K^{(T_1)}\boldsymbol{a}_{\hat{k}}) \leq {} & C_5\sigma_0^2 d e^{q_V^{-1}\sigma_1^2 d}.
\end{aligned} \tag{66}
$$

*Proof.* For $\forall \hat{k} \in [K]$, observing that

$$
\begin{aligned}
\mathbf{W}_Q^{(t+1)}\boldsymbol{a}_{\hat{k}} - \mathbf{W}_Q^{(t)}\boldsymbol{a}_{\hat{k}} = {} & \eta\mathbb{E}_{\mathcal{P}_{\mathrm{QA}}^{\mathrm{tr}}}\big[\sum_{m\in[M]}\iota_{\mathbf{u}_{k_\mathbf{y}},m}\cdot\mathbf{W}_K^{(t)}\mathbf{S}_n(\boldsymbol{e}_m - \boldsymbol{\pi}_n^{(t)})(\mathbf{S}_n\boldsymbol{e}_{M+1})^\top\boldsymbol{a}_{\hat{k}}\big] \\
= {} & \frac{\eta}{N}\sum_{\substack{k\in[K] \\ y\in[\pm 1] \\ n\in\mathcal{N}_k^y}}\Big[\sum_{m\in[M]}\frac{(\mathbf{u}_{k_{\mathbf{y}_n}} - \sum_{k\in[7K+K']}\omega_{n,k}\mathbf{u}_k)^\top(\mathbf{W}_V^{(t)}\mathbf{S}_n\pi_{n,m}^{(t)}\boldsymbol{e}_m)}{\|\mathbf{W}_V^{(t)}\mathbf{S}_n\boldsymbol{\pi}_n^{(t)}\|} \\
& (1 - \frac{(\mathbf{W}_V^{(t)}\mathbf{S}_n\boldsymbol{\pi}_n^{(t)})^\top}{\|\mathbf{W}_V^{(t)}\mathbf{S}_n\boldsymbol{\pi}_n^{(t)}\|}\frac{\mathbf{W}_V^{(t)}\mathbf{S}_n\pi_{n,m}^{(t)}\boldsymbol{e}_{n,m}}{\|\mathbf{W}_V^{(t)}\mathbf{S}_n\boldsymbol{\pi}_n^{(t)}\|})\cdot\mathbf{W}_K^{(t)}\mathbf{S}_n(\boldsymbol{e}_m - \boldsymbol{\pi}_n^{(t)})(\mathbf{S}_n\boldsymbol{e}_{M+1})^\top\boldsymbol{a}_{\hat{k}}\Big],
\end{aligned}
$$

$$\tag{67}$$

and also it holds that

$$\mathbf{W}_K^{(t+1)}\boldsymbol{a}_{\hat{k}} - \mathbf{W}_K^{(t)}\boldsymbol{a}_{\hat{k}} = \frac{\eta}{N}\sum_{\substack{k\in[K]\\y\in[\pm 1]\\n\in\mathcal{N}_k^y}}\Big[\sum_{m\in[M]}\frac{(\mathbf{u}_{k_{\mathbf{y}_n}} - \sum_{k\in[7K+K']}\omega_{n,k}\mathbf{u}_k)^\top(\mathbf{W}_V^{(t)}\mathbf{S}_n\pi_{n,m}^{(t)}\boldsymbol{e}_m)}{\|\mathbf{W}_V^{(t)}\mathbf{S}_n\boldsymbol{\pi}_n^{(t)}\|}$$

$$(1 - \frac{(\mathbf{W}_V^{(t)}\mathbf{S}_n\boldsymbol{\pi}_n^{(t)})^\top}{\|\mathbf{W}_V^{(t)}\mathbf{S}_n\boldsymbol{\pi}_n^{(t)}\|}\frac{\mathbf{W}_V^{(t)}\mathbf{S}_n\pi_{n,m}^{(t)}\boldsymbol{e}_{n,m}}{\|\mathbf{W}_V^{(t)}\mathbf{S}_n\boldsymbol{\pi}_n^{(t)}\|})\cdot\mathbf{W}_Q^{(t)}\mathbf{S}_n\boldsymbol{e}_{M+1}(\mathbf{S}_n(\boldsymbol{e}_m - \boldsymbol{\pi}_n^{(t)}))^\top\boldsymbol{a}_{\hat{k}}\Big].$$

(68)

Therefore, we have

$$\begin{aligned}(\mathbf{W}_Q^{(t+1)}\boldsymbol{a}_{\hat{k}})^\top(\mathbf{W}_K^{(t+1)}\boldsymbol{a}_{\hat{k}}) - (\mathbf{W}_Q^{(t)}\boldsymbol{a}_{\hat{k}})^\top(\mathbf{W}_K^{(t)}\boldsymbol{a}_{\hat{k}}) =& (\mathbf{W}_Q^{(t+1)}\boldsymbol{a}_{\hat{k}} - \mathbf{W}_Q^{(t)}\boldsymbol{a}_{\hat{k}} + \mathbf{W}_Q^{(t)}\boldsymbol{a}_{\hat{k}})^\top(\mathbf{W}_K^{(t+1)}\boldsymbol{a}_{\hat{k}}-\\ & \mathbf{W}_K^{(t)}\boldsymbol{a}_{\hat{k}} + \mathbf{W}_K^{(t)}\boldsymbol{a}_{\hat{k}}) - (\mathbf{W}_Q^{(t)}\boldsymbol{a}_{\hat{k}})^\top(\mathbf{W}_K^{(t)}\boldsymbol{a}_{\hat{k}})\\ =& (\mathbf{W}_Q^{(t+1)}\boldsymbol{a}_{\hat{k}} - \mathbf{W}_Q^{(t)}\boldsymbol{a}_{\hat{k}})^\top(\mathbf{W}_K^{(t+1)}\boldsymbol{a}_{\hat{k}} - \mathbf{W}_K^{(t)}\boldsymbol{a}_{\hat{k}})\\ & + (\mathbf{W}_K^{(t)}\boldsymbol{a}_{\hat{k}})^\top(\mathbf{W}_Q^{(t+1)}\boldsymbol{a}_{\hat{k}} - \mathbf{W}_Q^{(t)}\boldsymbol{a}_{\hat{k}})\\ & + (\mathbf{W}_Q^{(t)}\boldsymbol{a}_{\hat{k}})^\top(\mathbf{W}_K^{(t+1)}\boldsymbol{a}_{\hat{k}} - \mathbf{W}_K^{(t)}\boldsymbol{a}_{\hat{k}}).\end{aligned}$$

(69)

Here,

$$\begin{aligned}(\mathbf{W}_Q^{(t+1)}\boldsymbol{a}_{\hat{k}} - \mathbf{W}_Q^{(t)}\boldsymbol{a}_{\hat{k}})^\top(\mathbf{W}_K^{(t)}\boldsymbol{a}_{\hat{k}}) =& \frac{\eta}{N}\sum_{\substack{k\in[K]\\y\in[\pm 1]\\n\in\mathcal{N}_k^y\\m\in[M]}}\Big[\frac{(\mathbf{u}_{k_{\mathbf{y}_n}} - \sum_{k\in[7K+K']}\omega_{n,k}\mathbf{u}_k)^\top(\mathbf{W}_V^{(t)}\mathbf{S}_n\pi_{n,m}^{(t)}\boldsymbol{e}_m)}{\|\mathbf{W}_V^{(t)}\mathbf{S}_n\boldsymbol{\pi}_n^{(t)}\|}(1 - \frac{(\mathbf{W}_V^{(t)}\mathbf{S}_n\boldsymbol{\pi}_n^{(t)})^\top}{\|\mathbf{W}_V^{(t)}\mathbf{S}_n\boldsymbol{\pi}_n^{(t)}\|}\\ & \frac{\mathbf{W}_V^{(t)}\mathbf{S}_n\pi_{n,m}^{(t)}\boldsymbol{e}_{n,m}}{\|\mathbf{W}_V^{(t)}\mathbf{S}_n\boldsymbol{\pi}_n^{(t)}\|})\cdot(\mathbf{W}_K^{(t)}\boldsymbol{a}_{\hat{k}})^\top\mathbf{W}_K^{(t)}\mathbf{S}_n(\boldsymbol{e}_m - \boldsymbol{\pi}_n^{(t)})(\mathbf{S}_n\boldsymbol{e}_{M+1})^\top\boldsymbol{a}_{\hat{k}}\Big]\\ =& \frac{\eta}{N}\sum_{\substack{y\in[\pm 1]\\n\in\mathcal{N}_{\hat{k}}^y\\m\in[M]}}\Big[\Theta(\frac{(\mathbf{u}_{k_{\mathbf{y}_n}} - \sum_{k\in[7K+K']}\omega_{n,k}\mathbf{u}_k)^\top(\mathbf{W}_V^{(t)}\mathbf{S}_n\pi_{n,m}^{(t)}\boldsymbol{e}_m)}{\|\mathbf{W}_V^{(t)}\mathbf{S}_n\boldsymbol{\pi}_n^{(t)}\|}(1 - \frac{(\mathbf{W}_V^{(t)}\mathbf{S}_n\boldsymbol{\pi}_n^{(t)})^\top}{\|\mathbf{W}_V^{(t)}\mathbf{S}_n\boldsymbol{\pi}_n^{(t)}\|}\\ & \frac{\mathbf{W}_V^{(t)}\mathbf{S}_n\pi_{n,m}^{(t)}\boldsymbol{e}_{n,m}}{\|\mathbf{W}_V^{(t)}\mathbf{S}_n\boldsymbol{\pi}_n^{(t)}\|})\cdot(\mathbf{W}_K^{(t)}\boldsymbol{a}_{\hat{k}})^\top\mathbf{W}_K^{(t)}\mathbf{S}_n(\boldsymbol{e}_m - \boldsymbol{\pi}_n^{(t)})(\mathbf{S}_n\boldsymbol{e}_{M+1})^\top\boldsymbol{a}_{\hat{k}})\Big]\\ =& \frac{\eta}{N}\sum_{\substack{y\in[\pm 1]\\n\in\mathcal{N}_{\hat{k}}^y}}\Big[\Theta(\frac{(\mathbf{u}_{k_{\mathbf{y}_n}} - \sum_{k\in[7K+K']}\omega_{n,k}\mathbf{u}_k)^\top(\mathbf{W}_V^{(t)}\mathbf{S}_n\pi_{n,m_{\mathbf{S}_n}}^{(t)}\boldsymbol{e}_{m_{\mathbf{S}_n}})}{\|\mathbf{W}_V^{(t)}\mathbf{S}_n\boldsymbol{\pi}_n^{(t)}\|}(1 - \frac{(\mathbf{W}_V^{(t)}\mathbf{S}_n\boldsymbol{\pi}_n^{(t)})^\top}{\|\mathbf{W}_V^{(t)}\mathbf{S}_n\boldsymbol{\pi}_n^{(t)}\|}\\ & \frac{\mathbf{W}_V^{(t)}\mathbf{S}_n\pi_{n,m_{\mathbf{S}_n}}^{(t)}\boldsymbol{e}_{n,m_{\mathbf{S}_n}}}{\|\mathbf{W}_V^{(t)}\mathbf{S}_n\boldsymbol{\pi}_n^{(t)}\|})(\mathbf{W}_K^{(t)}\boldsymbol{a}_{\hat{k}})^\top\cdot\mathbf{W}_K^{(t)}\mathbf{S}_n(\boldsymbol{e}_{m_{\mathbf{S}_n}} - \boldsymbol{\pi}_n^{(t)})(\mathbf{S}_n\boldsymbol{e}_{M+1})^\top\boldsymbol{a}_{\hat{k}})\Big],\end{aligned}$$

(70)

Here, the second equality is by the near-orthogonal relationship

$$(\mathbf{S}_{n'}\boldsymbol{e}_{M+1})^\top\boldsymbol{a}_{\hat{k}} = \boldsymbol{a}_{\hat{k}}^\top(\boldsymbol{a}_{k'} + \boldsymbol{\xi}_{n',M+1}) \le \Theta(\sigma_p\cdot\sqrt{2\log(\frac{3(2K+K')NM}{\delta})}) = o(0.001),$$

for $\forall\hat{k}\in[K], k'\ne\hat{k}, n'\in\mathcal{N}_{k'}$; the third equality follows from $\sigma_p M\cdot\sqrt{2\log\left(\frac{3(2K+K')NM}{\delta}\right)} \le O(0.001)$, as well as the non-increasing nature of $\frac{\boldsymbol{a}_{\hat{k}}^\top(\mathbf{W}_V^{(t)}\mathbf{S}_n\pi_{n,\neg m_{\mathbf{S}_n}}^{(t)}\boldsymbol{e}_{\neg m_{\mathbf{S}_n}})}{\|\mathbf{W}_V^{(t)}\mathbf{S}_n\boldsymbol{\pi}_n^{(t)}\|}, n\in\mathcal{N}_{\hat{k}}^y$, which is due to the significant increasing behavior of $\boldsymbol{a}_{\hat{k}}^\top(\mathbf{W}_V^{(t)}\boldsymbol{a}_{\hat{k}})$ as well as the scale differences of gradients characterized in Eq.(56), (57), (58), (59), (60).

Similarly,

$$
\begin{aligned}
(\mathbf{W}_K^{(t+1)}\boldsymbol{a}_{\hat{k}} - \mathbf{W}_K^{(t)}\boldsymbol{a}_{\hat{k}})^\top(\mathbf{W}_Q^{(t)}\boldsymbol{a}_{\hat{k}}) =& \frac{\eta}{N}\sum_{\substack{y\in[\pm 1]\\n\in\mathcal{N}_{\hat{k}}^y}} \Theta([\frac{(\mathbf{u}_{k_{\mathbf{y}_n}} - \sum_{k\in[7K+K']}\omega_{n,k}\mathbf{u}_k)^\top(\mathbf{W}_V^{(t)}\mathbf{S}_n\pi_{n,m_{\mathbf{S}_n}}^{(t)}\boldsymbol{e}_{m_{\mathbf{S}_n}})}{\|\mathbf{W}_V^{(t)}\mathbf{S}_n\boldsymbol{\pi}_n^{(t)}\|}(1- \\
& \frac{(\mathbf{W}_V^{(t)}\mathbf{S}_n\boldsymbol{\pi}_n^{(t)})^\top}{\|\mathbf{W}_V^{(t)}\mathbf{S}_n\boldsymbol{\pi}_n^{(t)}\|}\frac{\mathbf{W}_V^{(t)}\mathbf{S}_n\pi_{n,m_{\mathbf{S}_n}}^{(t)}\boldsymbol{e}_{n,m_{\mathbf{S}_n}}}{\|\mathbf{W}_V^{(t)}\mathbf{S}_n\boldsymbol{\pi}_n^{(t)}\|})\cdot(\mathbf{W}_Q^{(t)}\boldsymbol{a}_{\hat{k}})^\top\mathbf{W}_Q^{(t)}\mathbf{S}_n\boldsymbol{e}_{M+1} \\
& (\mathbf{S}_n(\boldsymbol{e}_{m_{\mathbf{S}_n}} - \boldsymbol{\pi}_n^{(t)}))^\top\boldsymbol{a}_{\hat{k}}]
\end{aligned}
\tag{71}
$$

By definition of $\mathbf{u}_{k_{\mathbf{y}_n}}$, small scale of $\frac{(\mathbf{u}_{k_{\mathbf{y}_n}} - \sum_{k\in[7K+K']}\omega_{n,k}\mathbf{u}_k)^\top(\mathbf{W}_V^{(t)}\mathbf{S}_n\pi_{n,m_{\mathbf{S}_n}}^{(t)}\boldsymbol{e}_{m_{\mathbf{S}_n}})}{\|\mathbf{W}_V^{(t)}\mathbf{S}_n\|}$, $\Theta(\frac{1}{M}) \leq \pi_{n,m_{\mathbf{S}_n}}^{(t)} \leq \Theta(\frac{1}{1+(M-1)e^{-\sigma_0^2 de^{q}\sigma_V^{-1}\sigma_1^2 d}})$, $\Theta(\frac{e^{-\sigma_0^2 de^{q}\sigma_V^{-1}\sigma_1^2 d}}{1+(M-1)e^{-\sigma_0^2 de^{q}\sigma_V^{-1}\sigma_1^2 d}}) \leq \pi_{n,\neg m_{\mathbf{S}_n}}^{(t)} \leq \Theta(\frac{1}{M})$, Lemma D.11 and $K' = \Omega(M)$, through similar derivation techniques in Eq.(57), we have

$$
\begin{aligned}
(\mathbf{W}_Q^{(t+1)}\boldsymbol{a}_{\hat{k}})^\top(\mathbf{W}_K^{(t+1)}\boldsymbol{a}_{\hat{k}}) - (\mathbf{W}_Q^{(t)}\boldsymbol{a}_{\hat{k}})^\top(\mathbf{W}_K^{(t)}\boldsymbol{a}_{\hat{k}}) =& \Theta((\mathbf{W}_Q^{(t)}\boldsymbol{a}_{\hat{k}})^\top(\mathbf{W}_K^{(t+1)}\boldsymbol{a}_{\hat{k}} - \mathbf{W}_K^{(t)}\boldsymbol{a}_{\hat{k}}) \\
& + (\mathbf{W}_K^{(t)}\boldsymbol{a}_{\hat{k}})^\top(\mathbf{W}_Q^{(t+1)}\boldsymbol{a}_{\hat{k}} - \mathbf{W}_Q^{(t)}\boldsymbol{a}_{\hat{k}})) \\
=& \frac{\eta}{N}\sum_{\substack{y\in[\pm 1]\\n\in\mathcal{N}_{\hat{k}}^y}} \Theta([\frac{(\mathbf{u}_{k_{\mathbf{y}_n}} - \sum_{k\in[7K+K']}\omega_{n,k}\mathbf{u}_k)^\top(\mathbf{W}_V^{(t)}\mathbf{S}_n\pi_{n,m_{\mathbf{S}_n}}^{(t)}\boldsymbol{e}_{m_{\mathbf{S}_n}})}{\|\mathbf{W}_V^{(t)}\mathbf{S}_n\boldsymbol{\pi}_n^{(t)}\|} \\
& (1 - \frac{(\mathbf{W}_V^{(t)}\mathbf{S}_n\boldsymbol{\pi}_n^{(t)})^\top}{\|\mathbf{W}_V^{(t)}\mathbf{S}_n\boldsymbol{\pi}_n^{(t)}\|}\frac{\mathbf{W}_V^{(t)}\mathbf{S}_n\pi_{n,m_{\mathbf{S}_n}}^{(t)}\boldsymbol{e}_{n,m_{\mathbf{S}_n}}}{\|\mathbf{W}_V^{(t)}\mathbf{S}_n\boldsymbol{\pi}_n^{(t)}\|})\cdot \\
& ((\mathbf{W}_K^{(t)}\boldsymbol{a}_{\hat{k}})^\top\mathbf{W}_K^{(t)}\mathbf{S}_n(\boldsymbol{e}_{m_{\mathbf{S}_n}} - \boldsymbol{\pi}_n^{(t)})(\mathbf{S}_n\boldsymbol{e}_{M+1})^\top\boldsymbol{a}_{\hat{k}} \\
& + (\mathbf{W}_Q^{(t)}\boldsymbol{a}_{\hat{k}})^\top\mathbf{W}_Q^{(t)}\mathbf{S}_n\boldsymbol{e}_{M+1}(\mathbf{S}_n(\boldsymbol{e}_{m_{\mathbf{S}_n}} - \boldsymbol{\pi}_n^{(t)}))^\top\boldsymbol{a}_{\hat{k}}]) \\
=& \Theta(\frac{\eta\sum_{\hat{n}\in\mathcal{N}_{\hat{k}}}}{N}\frac{\boldsymbol{a}_{\hat{k}}^\top\mathbf{W}_V^{(t)}\mathbf{S}_n\pi_{n,m_{\mathbf{S}_n}}^{(t)}\boldsymbol{e}_{m_{\mathbf{S}_n}}}{\|\mathbf{W}_V^{(t)}\mathbf{S}_n\boldsymbol{\pi}_n^{(t)}\|}(1 - \frac{(\mathbf{W}_V^{(t)}\mathbf{S}_n\boldsymbol{\pi}_n^{(t)})^\top}{\|\mathbf{W}_V^{(t)}\mathbf{S}_n\boldsymbol{\pi}_n^{(t)}\|} \\
& \frac{\mathbf{W}_V^{(t)}\mathbf{S}_n\pi_{n,m_{\mathbf{S}_n}}^{(t)}\boldsymbol{e}_{n,m_{\mathbf{S}_n}}}{\|\mathbf{W}_V^{(t)}\mathbf{S}_n\boldsymbol{\pi}_n^{(t)}\|})\cdot(1 - \pi_{n,m_{\mathbf{S}_n}}^{(t)})\boldsymbol{a}_{\hat{k}}^\top((\mathbf{W}_Q^{(t)})^\top\mathbf{W}_Q^{(t)} \\
& + (\mathbf{W}_K^{(t)})^\top\mathbf{W}_K^{(t)})\boldsymbol{a}_{\hat{k}}).
\end{aligned}
\tag{72}
$$

Therefore, we found $\boldsymbol{a}_{\hat{k}}^\top(\mathbf{W}_Q^{(t)})^\top\mathbf{W}_Q^{(t)}\boldsymbol{a}_{\hat{k}}, \boldsymbol{a}_{\hat{k}}^\top(\mathbf{W}_K^{(t)})^\top\mathbf{W}_K^{(t)}\boldsymbol{a}_{\hat{k}}$ terms in the gradients, which is nonnegative. Hence, $(\mathbf{W}_Q^{(t)}\boldsymbol{a}_{\hat{k}})^\top(\mathbf{W}_K^{(t)}\boldsymbol{a}_{\hat{k}})$ is non-decreasing after $\boldsymbol{a}_{\hat{k}}^\top\mathbf{W}_V^{(t)}\boldsymbol{a}_{\hat{k}}$ got strictly positive. Similarly, we have

$$
\begin{aligned}
(\mathbf{W}_Q^{(t+1)}\boldsymbol{a}_{\hat{k}})^\top(\mathbf{W}_Q^{(t+1)}\boldsymbol{a}_{\hat{k}}) - (\mathbf{W}_Q^{(t)}\boldsymbol{a}_{\hat{k}})^\top(\mathbf{W}_Q^{(t)}\boldsymbol{a}_{\hat{k}}) =& \Theta(\frac{\eta\sum_{\hat{n}\in\mathcal{N}_{\hat{k}}}}{N}\frac{\boldsymbol{a}_{\hat{k}}^\top\mathbf{W}_V^{(t)}\mathbf{S}_n\pi_{n,m_{\mathbf{S}_n}}^{(t)}\boldsymbol{e}_{m_{\mathbf{S}_n}}}{\|\mathbf{W}_V^{(t)}\mathbf{S}_n\boldsymbol{\pi}_n^{(t)}\|}(1 - \frac{(\mathbf{W}_V^{(t)}\mathbf{S}_n\boldsymbol{\pi}_n^{(t)})^\top}{\|\mathbf{W}_V^{(t)}\mathbf{S}_n\boldsymbol{\pi}_n^{(t)}\|} \\
& \frac{\mathbf{W}_V^{(t)}\mathbf{S}_n\pi_{n,m_{\mathbf{S}_n}}^{(t)}\boldsymbol{e}_{n,m_{\mathbf{S}_n}}}{\|\mathbf{W}_V^{(t)}\mathbf{S}_n\boldsymbol{\pi}_n^{(t)}\|})\cdot(1 - \pi_{n,m_{\mathbf{S}_n}}^{(t)})\boldsymbol{a}_{\hat{k}}^\top(2(\mathbf{W}_K^{(t)})^\top\mathbf{W}_K^{(t)})\boldsymbol{a}_{\hat{k}}), \\
(\mathbf{W}_K^{(t+1)}\boldsymbol{a}_{\hat{k}})^\top(\mathbf{W}_K^{(t+1)}\boldsymbol{a}_{\hat{k}}) - (\mathbf{W}_K^{(t)}\boldsymbol{a}_{\hat{k}})^\top(\mathbf{W}_K^{(t)}\boldsymbol{a}_{\hat{k}}) =& \Theta(\frac{\eta\sum_{\hat{n}\in\mathcal{N}_{\hat{k}}}}{N}\frac{\boldsymbol{a}_{\hat{k}}^\top\mathbf{W}_V^{(t)}\mathbf{S}_n\pi_{n,m_{\mathbf{S}_n}}^{(t)}\boldsymbol{e}_{m_{\mathbf{S}_n}}}{\|\mathbf{W}_V^{(t)}\mathbf{S}_n\boldsymbol{\pi}_n^{(t)}\|}(1 - \frac{(\mathbf{W}_V^{(t)}\mathbf{S}_n\boldsymbol{\pi}_n^{(t)})^\top}{\|\mathbf{W}_V^{(t)}\mathbf{S}_n\boldsymbol{\pi}_n^{(t)}\|} \\
& \frac{\mathbf{W}_V^{(t)}\mathbf{S}_n\pi_{n,m_{\mathbf{S}_n}}^{(t)}\boldsymbol{e}_{n,m_{\mathbf{S}_n}}}{\|\mathbf{W}_V^{(t)}\mathbf{S}_n\boldsymbol{\pi}_n^{(t)}\|})\cdot(1 - \pi_{n,m_{\mathbf{S}_n}}^{(t)})\boldsymbol{a}_{\hat{k}}^\top(2(\mathbf{W}_Q^{(t)})^\top\mathbf{W}_Q^{(t)})\boldsymbol{a}_{\hat{k}}).
\end{aligned}
\tag{73}
$$

Then we can control $(\mathbf{W}_Q^{(t)}\boldsymbol{a}_{\hat{k}})^\top(\mathbf{W}_K^{(t)}\boldsymbol{a}_{\hat{k}})$ by

$$(\mathbf{W}_Q^{(t+1)}\boldsymbol{a}_{\hat{k}})^\top(\mathbf{W}_K^{(t+1)}\boldsymbol{a}_{\hat{k}}) - (\mathbf{W}_Q^{(t)}\boldsymbol{a}_{\hat{k}})^\top(\mathbf{W}_K^{(t)}\boldsymbol{a}_{\hat{k}}) = \Theta\big(\frac{(\mathbf{W}_Q^{(t+1)}\boldsymbol{a}_{\hat{k}})^\top(\mathbf{W}_Q^{(t+1)}\boldsymbol{a}_{\hat{k}}) - (\mathbf{W}_Q^{(t)}\boldsymbol{a}_{\hat{k}})^\top(\mathbf{W}_Q^{(t)}\boldsymbol{a}_{\hat{k}})}{2}$$
$$+ \frac{(\mathbf{W}_K^{(t+1)}\boldsymbol{a}_{\hat{k}})^\top(\mathbf{W}_K^{(t+1)}\boldsymbol{a}_{\hat{k}}) - (\mathbf{W}_K^{(t)}\boldsymbol{a}_{\hat{k}})^\top(\mathbf{W}_K^{(t)}\boldsymbol{a}_{\hat{k}})}{2}\big) \tag{74}$$

By similar approaches in (57), (58), (59), (60), it holds that for $\forall \hat{k} \in [K], k' \in [K'], \boldsymbol{u} \in \{\boldsymbol{a}_s\}_{s \neq \hat{k} \in [K]} \cup \{\boldsymbol{b}_s\}_{s\in[K]} \cup \{\boldsymbol{\xi}_{n,m}\}_{n\in[N],m\in[M]}$ and $\boldsymbol{v} \neq \boldsymbol{v}' \in \{\boldsymbol{b}_s\}_{s\in[K]} \cup \{\boldsymbol{\xi}_{n,m}\}_{n\in[N],m\in[M]}$, the growing of $(\mathbf{W}_Q^{(t)}\boldsymbol{a}_{\hat{k}})^\top(\mathbf{W}_K^{(t)}\boldsymbol{u}), (\mathbf{W}_Q^{(t)}\boldsymbol{v})^\top(\mathbf{W}_K^{(t)}\boldsymbol{v}), (\mathbf{W}_Q^{(t)}\boldsymbol{v})^\top(\mathbf{W}_K^{(t)}\boldsymbol{v}')$ is relatively negligible.

Now we check the lower bound of $(\mathbf{W}_Q^{(t)}\boldsymbol{a}_{\hat{k}})^\top(\mathbf{W}_K^{(t)}\boldsymbol{a}_{\hat{k}})$, in the worse case before it starts to increase. By Lemma F.5, the $\boldsymbol{a}_{\hat{k}}^\top\mathbf{W}_V^{(t)}\boldsymbol{a}_{\hat{k}}$ would got positive after at most $t_3 = (\eta q_V)^{-1}C_1^{-1}\sigma_1^2 d^{1/2}MK(\sqrt{2\log(\frac{8(2K+K')^2}{\delta})})$ iterations. During the period $0 \leq t \leq t_3$, if (i). $(\mathbf{W}_Q^{(0)}\boldsymbol{a}_{\hat{k}})^\top(\mathbf{W}_K^{(0)}\boldsymbol{a}_{\hat{k}}) < 0$, $(\mathbf{W}_Q^{(t)}\boldsymbol{a}_{\hat{k}})^\top(\mathbf{W}_K^{(t)}\boldsymbol{a}_{\hat{k}})$ will increase, otherwise (ii) it will decrease. In situation (i), the lower bound of $(\mathbf{W}_Q^{(t_3)}\boldsymbol{a}_{\hat{k}})^\top(\mathbf{W}_K^{(t_3)}\boldsymbol{a}_{\hat{k}})$ would be its lower bound of initialization (i.e. $-4\sqrt{\log(16(2K+K')^2/\delta)}\sigma_0^2 d^{1/2}$ by Lemma D.9); in situation (ii), when the $(\mathbf{W}_Q^{(t)}\boldsymbol{a}_{\hat{k}})^\top(\mathbf{W}_K^{(t)}\boldsymbol{a}_{\hat{k}})$ decrease to be lower than 0, the situation would be back to (i). Therefore, the lower bound of $(\mathbf{W}_Q^{(t_3)}\boldsymbol{a}_{\hat{k}})^\top(\mathbf{W}_K^{(t_3)}\boldsymbol{a}_{\hat{k}})$ would be $-4\sqrt{\log(16(2K+K')^2/\delta)}\sigma_0^2 d^{1/2}$. Worth noting that the $(\mathbf{W}_Q^{(t)}\boldsymbol{a}_{\hat{k}})^\top(\mathbf{W}_Q^{(t)}\boldsymbol{a}_{\hat{k}})$ and $(\mathbf{W}_K^{(t)}\boldsymbol{a}_{\hat{k}})^\top(\mathbf{W}_K^{(t)}\boldsymbol{a}_{\hat{k}})$ also decrease in the same period. By Lemma D.4, the lower bound would be $(1 - bt_3)e_0$, where $b = 8\eta d^{-1/2}K^{-1}M^{-1}\sqrt{\log(\frac{8(2K+K')^2}{\delta})}$, $e_0 = \min_{X=Q,K}\{(\mathbf{W}_X^{(0)}\boldsymbol{a}_{\hat{k}})^\top(\mathbf{W}_X^{(0)}\boldsymbol{a}_{\hat{k}})\} = \sigma_0^2 d/2$. By $\sigma_1 \leq O(q_V^{\frac{1}{2}}(\log(\frac{8(2K+K')^2}{\delta}))^{\frac{1}{2}})$ in Condition 3.1, we would have $(\mathbf{W}_Q^{(t)}\boldsymbol{a}_{\hat{k}})^\top(\mathbf{W}_Q^{(t)}\boldsymbol{a}_{\hat{k}}), (\mathbf{W}_K^{(t)}\boldsymbol{a}_{\hat{k}})^\top(\mathbf{W}_K^{(t)}\boldsymbol{a}_{\hat{k}}) \geq \sigma_0^2 d/4$.

Next, we examine the increasing evolution of $(\mathbf{W}_Q^{(t)}\boldsymbol{a}_{\hat{k}})^\top(\mathbf{W}_K^{(t)}\boldsymbol{a}_{\hat{k}})$. During the period $\boldsymbol{a}_{\hat{k}}^\top\mathbf{W}_V^{(t)}\boldsymbol{a}_k \leq 0.01\sigma_1 d^{1/2}$ after $\boldsymbol{a}_{\hat{k}}^\top\mathbf{W}_V^{(t)}\boldsymbol{a}_{\hat{k}}$ becomes strictly positive at $0 \leq t_4 \leq t_3 = (\eta q_V)^{-1}C_1^{-1}\sigma_1^2 d^{1/2}MK(\sqrt{2\log(\frac{8(2K+K')^2}{\delta})})$ iterations (i.e., during $t_4 \leq t \leq T_1$), by Eq. (61), (74), Lemma D.5 as well as considering the positive estimate of the initial $(\mathbf{W}_Q^{(t_4)}\boldsymbol{a}_{\hat{k}})^\top(\mathbf{W}_K^{(t_4)}\boldsymbol{a}_{\hat{k}}), (\mathbf{W}_Q^{(t_4)}\boldsymbol{a}_{\hat{k}})^\top(\mathbf{W}_Q^{(t_4)}\boldsymbol{a}_{\hat{k}}), (\mathbf{W}_K^{(t_4)}\boldsymbol{a}_{\hat{k}})^\top(\mathbf{W}_K^{(t_4)}\boldsymbol{a}_{\hat{k}})$ at $t_4$, we have

$$(\mathbf{W}_Q^{(t)}\boldsymbol{a}_{\hat{k}})^\top(\mathbf{W}_K^{(t)}\boldsymbol{a}_{\hat{k}}) \leq C_4\big((4\sqrt{\log(16(2K+K')^2/\delta)}\sigma_0^2 d^{1/2} + \frac{3\sigma_0^2 d}{2}) + \frac{3\sigma_0^2 d}{2}\exp(\frac{\hat{C}_2 C_2 \eta^2 q_V t^2}{\sigma_1^2 dK^2})\big),$$

for some $C_4 > 0$. Now we extend $T_1 = \Theta((\eta q_V)^{-1}\sigma_1^2 dK)$ to this upper bound evolution, which is equivalent to solving the following system:

$$a + be^{ct^2},$$
$$\begin{cases} a &= C_4\big((4\sqrt{\log(16(2K+K')^2/\delta)}\sigma_0^2 d^{1/2} + \frac{3\sigma_0^2 d}{2})\big), \\ b &= C_4\frac{3\sigma_0^2 d}{2}, \\ c &= \frac{C_4\hat{C}_2 C_2 \eta^2 q_V}{\sigma_1^2 dK^2}, \\ t &= T_1. \end{cases}$$

We have

$$(\mathbf{W}_Q^{(T_1)}\boldsymbol{a}_{\hat{k}})^\top(\mathbf{W}_K^{(T_1)}\boldsymbol{a}_{\hat{k}}) \leq \Theta(\sigma_0^2 d(2 + e^{q_V^{-1}\sigma_1^2 d})) \leq C_5\sigma_0^2 de^{q_V^{-1}\sigma_1^2 d},$$

for some $C_5 > 0$. That is, by $q_V = O(\frac{\sigma_1^2 d}{\log(d^{-\frac{1}{2}}\sqrt{\log(\frac{K'^2}{\delta})})})$ and $q_V = \Omega(\frac{\sigma_1^2 d}{\log(\sigma_0^{-2}d^{-1}\log(\frac{M-1}{0.06}))})$, it holds that

$$\pi_{n,m_{\mathbf{S}_n}}^{(T_1)} \leq \Theta\big(\frac{e^{0.1C_5\sigma_0^2 de^{q_V^{-1}\sigma_1^2 d} + 9\cdot 4\sqrt{\log(\frac{16(2K+K')^2}{\delta})}\sigma_0^2 d^{1/2}}}{e^{C_5\sigma_0^2 de^{q_V^{-1}\sigma_1^2 d} + 9\cdot 4\sqrt{\log(\frac{16(2K+K')^2}{\delta})}\sigma_0^2 d^{1/2}} + (M-1)e^{-10\cdot 4\sqrt{\log(\frac{16(2K+K')^2}{\delta})}\sigma_0^2 d^{1/2}}}\big)$$
$$\leq \Theta\big(\frac{1}{1 + (M-1)e^{-\sigma_0^2 de^{q_V^{-1}\sigma_1^2 d}}}\big) < 0.95$$

Define $\triangle := (1 - (\frac{1}{1+(M-1)e^{-\sigma_0^2 de^{q_V^{-1}\sigma_1^2 d}}})^2)(1 - \frac{1}{1+(M-1)e^{-\sigma_0^2 de^{q_V^{-1}\sigma_1^2 d}}}) \geq (0.0975)(0.05) = 0.0048$, by Eq. (61), (63), (74), and Lemma D.5, it holds that

$$(\mathbf{W}_Q^{(t)}\boldsymbol{a}_{\hat{k}})^\top(\mathbf{W}_K^{(t)}\boldsymbol{a}_{\hat{k}}) \geq \Theta(\frac{\hat{C}_1 C_1 \eta^2 q_V \sigma_0^2[(t-t_4-1)^2-1]}{\sigma_1^2 M^2 K^2} - 4\sqrt{\log(16(2K+K')^2/\delta)}\sigma_0^2 d^{1/2})$$

$$\geq \frac{C_3 \eta^2 q_V \sigma_0^2[(t-t_4-1)^2-1]}{\sigma_1^2 M^2 K^2} - 4\sqrt{\log(16(2K+K')^2/\delta)}\sigma_0^2 d^{1/2}$$

for some $C_3 > 0$.

Similarly, we drag the $T_1$ to the lower bound via the following system:

$$a + bt^2,$$

$$\begin{cases} a &= -\frac{C_3 \eta^2 q_V \sigma_0^2}{\sigma_1^2 M^2 K^2} - 4C_3\sqrt{\log(16(2K+K')^2/\delta)}\sigma_0^2 d^{1/2}, \\ b &= \frac{C_3 \eta^2 q_V \sigma_0^2}{\sigma_1^2 M^2 K^2}, \\ t &= T_1 - 1. \end{cases}$$

It holds that

$$(\mathbf{W}_Q^{(t)}\boldsymbol{a}_{\hat{k}})^\top(\mathbf{W}_K^{(t)}\boldsymbol{a}_{\hat{k}}) \geq -\frac{C_3 \eta^2 q_V \sigma_0^2}{\sigma_1^2 M^2 K^2} - 4C_3\sqrt{\log(16(2K+K')^2/\delta)}\sigma_0^2 d^{1/2} + C_3\frac{\sigma_0^2 \sigma_1^2 d^2}{M^2 q_V}$$

$$\geq C_6 \frac{\sigma_0^2 \sigma_1^2 d^2}{M^2 q_V} - \frac{C_3 \eta^2 q_V \sigma_0^2}{\sigma_1^2 M^2 K^2} - 4C_3\sqrt{\log(16(2K+K')^2/\delta)}\sigma_0^2 d^{1/2},$$

for some $C_6 > 0$.

$\square$

*Proof.* *Proof of Lemma F.1.* From Lemma F.5 and the condition $T_1 \leq t_2^\star$, we derive the first result in Eq. (49) as well as the first lower bound in Eq. (50).

The second result follows from Lemma F.4, where the updates of $|\boldsymbol{a}_k^\top \mathbf{W}_V^{(t)}\boldsymbol{u}|$, $|\boldsymbol{v}^\top \mathbf{W}_V^{(t)}\boldsymbol{v}|$, and $|\boldsymbol{v}^\top \mathbf{W}_V^{(t)}\boldsymbol{v}'|$ are effectively constrained by the low-noise condition, the symmetry of the low-level real-value label distribution, and the large dictionary assumption detailed in Condition 3.1.

Finally, the third and fourth results in Eq. (49), as well as the second lower bound in Eq. (50) are established using Lemma F.7, together with the choice of $T_1$.

$\square$

## F.2. Phase 2: Decelerating Growth of MLP and Attention

In this section, we would witness the decelerating growth of MLP and Attention.

Recall that the main gains in the first section is the growth separations denoted in Eq.(49) and (50) in Lemma F.1:

$$\boldsymbol{a}_k^\top \mathbf{W}_V^{(T_1)}\boldsymbol{a}_k \geq \frac{\hat{C}\sigma_1 d^{1/2}}{M} - \sqrt{2\log(\frac{8(2K+K')^2}{\delta})}\sigma_1 = \Theta(M^{-1}\sigma_1 d^{1/2}),$$

$$|\boldsymbol{a}_k^\top \mathbf{W}_V^{(T_1)}\boldsymbol{u}|, \ |\boldsymbol{v}^\top \mathbf{W}_V^{(T_1)}\boldsymbol{v}|, \ |\boldsymbol{v}^\top \mathbf{W}_V^{(T_1)}\boldsymbol{v}'| = O(\sqrt{2\log(\frac{8(2K+K')^2}{\delta})}\sigma_1),$$

for $\forall k \in [K], \boldsymbol{u} \in \{\boldsymbol{a}_s\}_{s\neq k\in[K]} \cup \{\boldsymbol{b}_s\}_{s\in[K]} \cup \{\boldsymbol{\nu}_{k'}\}_{k'\in[K']} \cup \{\boldsymbol{\xi}_{n,m}\}_{n\in[N],m\in[M]}$ and $\boldsymbol{v} \neq \boldsymbol{v}' \in \{\boldsymbol{b}_s\}_{s\in[K]} \cup \{\boldsymbol{\xi}_{n,m}\}_{n\in[N],m\in[M]}$. By $d = \Omega(M^2 \log(\frac{K'^2 N^2 M^2}{\delta}))$ in Condition 3.1, the scale of $\boldsymbol{a}_k^\top \mathbf{W}_V^{(T_1)}\boldsymbol{a}_k$ greatly surpass others. However, as we see in Lemma F.2 that the role of $\boldsymbol{a}_k^\top \mathbf{W}_V^{(t)}\boldsymbol{a}_k$ in the growth of $\|\mathbf{W}_V^{(t)}\mathbf{S}_n \boldsymbol{\pi}_n^{(t)}\|$ mainly come from

$$(\pi_{n,m\mathbf{S}_n}^{(t)})^2 \cdot (\boldsymbol{a}_k \cdot \cos\langle \mathbf{W}_V^{(t)}\boldsymbol{a}_k, \boldsymbol{a}_k\rangle \|\mathbf{W}_V^{(t)}\boldsymbol{a}_k\|)^\top(\mathbf{W}_V^{(t)}\boldsymbol{a}_k) = (\pi_{n,m\mathbf{S}_n}^{(t)})^2 \cdot (\boldsymbol{a}_k^\top \mathbf{W}_V^{(t)}\boldsymbol{a}_k)^2.$$

Since $\boldsymbol{a}_k^\top \mathbf{W}_V^{(t)} \boldsymbol{a}_k = O(\sigma_1 d^{1/2})$ during $t \leq T_1$, it follows from the above formula that $\boldsymbol{a}_k^\top \mathbf{W}_V^{(t)} \boldsymbol{a}_k$ has only a mild influence on the initial scale of $\|\mathbf{W}_V^{(t)} \mathbf{S}_n \boldsymbol{\pi}_n^{(t)}\| = \Theta(\sigma_1 d^{-1/2})$, as shown in Lemma F.2. Consequently, the small cosine similarity in Eq.(51) holds:

$$\frac{\boldsymbol{a}_k^\top \mathbf{W}_V^{(t)} \mathbf{S}_n \boldsymbol{\pi}_n^{(t)}}{\|\mathbf{W}_V^{(t)} \mathbf{S}_n \boldsymbol{\pi}_n^{(t)}\|} \leq O\Big(\frac{1}{1 + (M-1)e^{-\sigma_0^2 d e^{q_V^{-1} \sigma_1^2 d}}}\Big),$$

for $\forall k \in [K], \neg k \neq k \in [K], n \in \mathcal{N}_k \subset [N], s \in [7], m_{\mathbf{S}_n} \neq m_{\mathbf{S}_n} \in [M]$. In this section we would witness the increasing role of $\mathbf{W}_V^{(t)} \boldsymbol{a}_k$ in the volume $\|\mathbf{W}_V^{(t)} \boldsymbol{a}_k\|$. To achieve this, based on the scale separations of gradients shown in Lemma F.4, we define

$$\boldsymbol{i}_{V,\boldsymbol{a}_k^\perp}^{(t)} := \mathbf{W}_V^{(t)} \boldsymbol{a}_k - \mathbf{W}_V^{(0)} \boldsymbol{a}_k - (\boldsymbol{a}_k^\top \mathbf{W}_V^{(t)} \boldsymbol{a}_k - \boldsymbol{a}_k^\top \mathbf{W}_V^{(0)} \boldsymbol{a}_k) \boldsymbol{a}_k,$$

where $\boldsymbol{i}_{V,\boldsymbol{a}_k^\perp}^{(t)}$ denotes the residual components of $\mathbf{W}_V^{(t)} \boldsymbol{a}_k - \mathbf{W}_V^{(0)} \boldsymbol{a}_k$ orthogonal to $\boldsymbol{a}_k$. That is,

$$\mathbf{W}_V^{(t)} \boldsymbol{a}_k = \mathbf{W}_V^{(0)} \boldsymbol{a}_k + (\boldsymbol{a}_k^\top \mathbf{W}_V^{(t)} \boldsymbol{a}_k - \boldsymbol{a}_k^\top \mathbf{W}_V^{(0)} \boldsymbol{a}_k) \boldsymbol{a}_k + \boldsymbol{i}_{V,\boldsymbol{a}_k^\perp}^{(t)}. \tag{75}$$

Similar to Lemma F.1, for all $k \in [K]$, the following lemma reveals that during the second phase $(T_1 \leq t \leq T^\star)$, the term $\boldsymbol{a}_k^\top \mathbf{W}_V^{(t)} \boldsymbol{a}_k$ exhibits substantially faster growth compared to other cross-terms of the form $\boldsymbol{u}^\top \mathbf{W}_V^{(t)} \boldsymbol{v}$, where $\{\boldsymbol{u}, \boldsymbol{v}\} \neq \{\boldsymbol{a}_s, \boldsymbol{a}_s\}_{s \in [K]}$. This growth pattern establishes $\boldsymbol{a}_k^\top \mathbf{W}_V^{(t)} \boldsymbol{a}_k$ as the dominant component in determining the order of $\|\mathbf{W}_V^{(t)} \boldsymbol{a}_k\|$. Furthermore, $\boldsymbol{i}_{V,\boldsymbol{a}_k^\perp}^{(t)}{}^\top \boldsymbol{a}_k = 0$, and the contributions from $\|\boldsymbol{i}_{V,\boldsymbol{a}_k^\perp}^{(t)}\|$ and $\boldsymbol{i}_{V,\boldsymbol{a}_k^\perp}^{(t)}{}^\top \mathbf{W}_V^{(0)} \boldsymbol{a}_k$ become asymptotically negligible, being bounded by the order of $\|\mathbf{W}_V^{(0)} \boldsymbol{a}_k\|$.

**Lemma F.8.** *Under Condition 3.1, during $t_1 \leq t \leq T^\star = \Omega(\eta^{-1} q_V^{-1} \sigma_1^2 K M d^2 \log(\frac{1}{\epsilon}))$ where $t_1 \leq T_1$, we have*

$$
\begin{aligned}
&|\boldsymbol{a}_k^\top \mathbf{W}_V^{(t)} \boldsymbol{u}|, \ |\boldsymbol{v}^\top \mathbf{W}_V^{(t)} \boldsymbol{v}|, \ |\boldsymbol{v}^\top \mathbf{W}_V^{(t)} \boldsymbol{v}'| = O\Big(\sqrt{2 \log\big(\frac{8(2K + K')^2}{\delta}\big)} \sigma_1\Big), \\
&\|\boldsymbol{i}_{V,\boldsymbol{a}_k^\perp}^{(t)}\|, \ \boldsymbol{i}_{V,\boldsymbol{a}_k^\perp}^{(t)}{}^\top \mathbf{W}_V^{(0)} \boldsymbol{a}_k = o(\|\mathbf{W}_V^{(0)} \boldsymbol{a}_k\|), \\
&\boldsymbol{a}_k^\top \mathbf{W}_V^{(t)} \boldsymbol{a}_k \geq \sqrt{C_7 \frac{\eta q_V (t - t_1)}{MK} + (\boldsymbol{a}_k^\top \mathbf{W}_V^{(t_1)} \boldsymbol{a}_k)^2}, \\
&\boldsymbol{a}_k^\top \mathbf{W}_V^{(t)} \boldsymbol{a}_k \leq \sqrt{C_8 \frac{\eta q_V (t - t_1)}{K} + (\boldsymbol{a}_k^\top \mathbf{W}_V^{(t_1)} \boldsymbol{a}_k)^2} + \frac{C_8 \eta q_V}{K (\boldsymbol{a}_k^\top \mathbf{W}_V^{(t_1)} \boldsymbol{a}_k)},
\end{aligned}
\tag{76}
$$

*for $\forall k \in [K], n \in [N], s \in [7], \neg m_{\mathbf{S}_n} \neq m_{\mathbf{S}_n} \in [M], \boldsymbol{u} \in \{\boldsymbol{a}_s\}_{s \neq k \in [K]} \cup \{\boldsymbol{b}_s\}_{s \in [K]} \cup \{\boldsymbol{\nu}_{k'}\}_{k' \in [K']} \cup \{\boldsymbol{\xi}_{n,m}\}_{n \in [N], m \in [M]}$ and $\boldsymbol{v} \neq \boldsymbol{v}' \in \{\boldsymbol{b}_s\}_{s \in [K]} \cup \{\boldsymbol{\xi}_{n,m}\}_{n \in [N], m \in [M]}.$*

*Furthermore, after $t \geq T^\star = \Omega(\eta^{-1} q_V^{-1} \sigma_1^2 K M d^2 \log(\frac{1}{\epsilon}))$, it holds that*

$$
\begin{aligned}
&\boldsymbol{a}_k^\top \mathbf{W}_V^{(t)} \boldsymbol{a}_k = \Omega\Big(3\big(1 + 15\sigma_p^\star \sqrt{\log(\tfrac{4}{\varepsilon})}\big) \sigma_1 d^{1/2}\Big) \\
&\|\mathbf{W}_V^{(t)}\|_F = \Theta\big(\sqrt{K} \boldsymbol{a}_k^\top \mathbf{W}_V^{(t)} \boldsymbol{a}_k\big)
\end{aligned}
\tag{77}
$$

*for $\forall k \in [K]$.*

*Proof.* The first two results can be derived through analogous reasoning as in Phase 1. Specifically, the updates of $\boldsymbol{a}_k^\top \mathbf{W}_V^{(t)} \boldsymbol{u}, \boldsymbol{v}^\top \mathbf{W}_V^{(t)} \boldsymbol{v}$, and $\boldsymbol{v}^\top \mathbf{W}_V^{(t)} \boldsymbol{v}'$ become progressively weaker. This weakening is attributed to the diminishing magnitude of $\boldsymbol{u}^\top \mathbf{W}_V^{(t)} \mathbf{S}_n \boldsymbol{\pi}_n^{(t)} / \|\mathbf{W}_V^{(t)} \mathbf{S}_n \boldsymbol{\pi}_n^{(t)}\|$ for $n \in \mathcal{N}_k$, which results from the increasing dominance of $\boldsymbol{a}_k^\top \mathbf{W}_V^{(t)} \mathbf{S}_n \boldsymbol{\pi}_n^{(t)}$. Additionally, the already weak updates of $\boldsymbol{a}_k^\top \mathbf{W}_V^{(t)} \boldsymbol{u}, \boldsymbol{v}^\top \mathbf{W}_V^{(t)} \boldsymbol{v}$, and $\boldsymbol{v}^\top \mathbf{W}_V^{(t)} \boldsymbol{v}'$ in Phase 1 further contribute to this effect. Consequently, the component norms and products $\|\boldsymbol{i}_{V,\boldsymbol{a}_k^\perp}^{(t)}\|, \boldsymbol{i}_{V,\boldsymbol{a}_k^\perp}^{(t)}{}^\top \mathbf{W}_V^{(0)} \boldsymbol{a}_k$ are comparatively feeble with initialization.

After $t \geq T_1$, the terms in Eq.(55) maintain negligible due to the increasing of $\pi^{(t)}_{n',m_{\mathbf{S}_{n'}}}$, $\boldsymbol{a}_{\hat{k}}^\top \mathbf{W}_V^{(t)} \boldsymbol{a}_{\hat{k}}$, $\|\mathbf{W}_V^{(t)} \mathbf{S}_{n'} \boldsymbol{\pi}_{n'}^{(t)}\| \geq \Theta(\|\mathbf{W}_V^{(0)} \mathbf{S}_{n'} \boldsymbol{\pi}_{n'}^{(0)}\|)$ as well as $\boldsymbol{a}_{\hat{k}}^\top \mathbf{W}_V^{(t)} \boldsymbol{a}_{k'} = O(\boldsymbol{a}_{\hat{k}}^\top \mathbf{W}_V^{(0)} \boldsymbol{a}_{k'})$.

Besides, as $\mathbf{W}_V^{(t)} \boldsymbol{u}, \forall \boldsymbol{u} \in \{\boldsymbol{a}_s\}_{s \in [K]} \cup \{\boldsymbol{b}_s\}_{s \in [K]} \cup \{\boldsymbol{\nu}_{k'}\}_{k' \in [K']} \cup \{\boldsymbol{\xi}_{n,m}\}_{n \in [N], m \in [M]}$ will always preserve some volume over some directions after initialization, it holds that $\cos\langle \boldsymbol{a}_{\hat{k}}, \mathbf{W}_V^{(t)} \boldsymbol{u} \rangle \leq$ constant $< 1$ during the entire iterations $T^\star$. Therefore, there exists some $c > 0$, such that

$$\boldsymbol{a}_{\hat{k}}^\top ((\mathbf{u}_{k_{\mathbf{y}_{\hat{n}}}} - \sum_{k \in [7K+K']} \omega_{\hat{n},k} \mathbf{u}_k) - (\frac{(\mathbf{u}_{k_{\mathbf{y}_{\hat{n}}}} - \sum_{k \in [7K+K']} \omega_{\hat{n},k} \mathbf{u}_k)^\top (\mathbf{W}_V^{(t)} \mathbf{S}_{\hat{n}} \boldsymbol{\pi}_{\hat{n}}^{(t)})}{\|\mathbf{W}_V^{(t)} \mathbf{S}_{\hat{n}} \boldsymbol{\pi}_{\hat{n}}^{(t)}\|}) \frac{\mathbf{W}_V^{(t)} \mathbf{S}_{\hat{n}} \boldsymbol{\pi}_{\hat{n}}^{(t)}}{\|\mathbf{W}_V^{(t)} \mathbf{S}_{\hat{n}} \boldsymbol{\pi}_{\hat{n}}^{(t)}\|}) \geq c > 0,$$

for all the iterations $t \leq T^\star$. Similar to Eq.(56), we then have

$$
\begin{aligned}
\boldsymbol{a}_{\hat{k}}^\top \mathbf{W}_V^{(t+1)} \boldsymbol{a}_{\hat{k}} - \boldsymbol{a}_{\hat{k}}^\top \mathbf{W}_V^{(t)} \boldsymbol{a}_{\hat{k}} &= \frac{\eta q_V}{N} \sum_{y \in [\pm 1]} \sum_{\hat{n} \in \mathcal{N}_{\hat{k}}^y} \Theta([\frac{\boldsymbol{a}_{\hat{k}}^\top \boldsymbol{\Pi}_{(\mathbf{W}_V \mathbf{S}_{\hat{n}} \boldsymbol{\pi}_{\hat{n}}^{(t)})^\perp}^{\mathbf{u}_{k_{\mathbf{y}_{\hat{n}}}} - \sum_{k \in [7K+K']} \omega_{\hat{n},k} \mathbf{u}_k} (\mathbf{S}\boldsymbol{\pi}_{\hat{n}}^{(t)})^\top \boldsymbol{a}_{\hat{k}}}{\|\mathbf{W}_V^{(t)} \mathbf{S}_{\hat{n}} \boldsymbol{\pi}_{\hat{n}}^{(t)}\|}]) \\
&= \frac{\eta q_V}{N} \sum_{\substack{y \in [\pm 1] \\ \hat{n} \in \mathcal{N}_{\hat{k}}^y}} \Theta([\frac{((\pi^{(t)}_{\hat{n},m_{\mathbf{S}_{\hat{n}}}} \boldsymbol{a}_{\hat{k}} + \sum_m (\pi^{(t)}_{\hat{n},m} \boldsymbol{\xi}_{\hat{n},m}))^\top \boldsymbol{a}_{\hat{k}}}{\|\mathbf{W}_V^{(t)} \mathbf{S}_{\hat{n}} \boldsymbol{\pi}_{\hat{n}}^{(t)}\|}]) \\
&= \frac{\eta q_V}{N} \sum_{\hat{n} \in \mathcal{N}_{\hat{k}}} \Theta([\frac{M\pi^{(t)}_{\hat{n},m_{\mathbf{S}_{\hat{n}}}} + \sigma_p M \cdot (M\pi^{(t)}_{\hat{n},\neg m_{\mathbf{S}_{\hat{n}}}}) \sqrt{2\log(3(2K+K')NM\delta^{-1})}}{M\|\mathbf{W}_V^{(t)} \mathbf{S}_{\hat{n}} \boldsymbol{\pi}_{\hat{n}}^{(t)}\|}]) \\
&= \Theta([\sum_{\hat{n} \in \mathcal{N}_{\hat{k}}} \frac{\eta q_V (M\pi^{(t)}_{\hat{n},m_{\mathbf{S}_{\hat{n}}}})}{2NM\|\mathbf{W}_V^{(t)} \mathbf{S}_{\hat{n}} \boldsymbol{\pi}_{\hat{n}}^{(t)}\|}]).
\end{aligned}
\tag{78}
$$

In this phase $t \geq T_1$, the $\boldsymbol{a}_k^\top \mathbf{W}_V^{(t)} \boldsymbol{a}_k$ is comparable to the potential maximum norm of the initialized $\|\mathbf{W}_V^{(0)} \mathbf{S}_n \boldsymbol{\pi}_n^{(0)}\|$, and its order is larger than those $\boldsymbol{\xi}_{n,m}^\top \mathbf{W}_V^{(t)} \boldsymbol{a}_k, \boldsymbol{v}_{n,m}^\top \mathbf{W}_V^{(t)} \boldsymbol{a}_k$. Therefore,

$$
\begin{aligned}
\|\mathbf{W}_V^{(t)} \mathbf{S}_{\hat{n}} \boldsymbol{\pi}_{\hat{n}}^{(t)}\|^2 &= (\pi^{(t)}_{\hat{n},m_{\mathbf{S}_{\hat{n}}}})^2 (\boldsymbol{a}_{\hat{k}}^\top \mathbf{W}_V^{(t)} \boldsymbol{a}_{\hat{k}})^2 + 2(\pi^{(t)}_{\hat{n},m_{\mathbf{S}_{\hat{n}}}})(\boldsymbol{a}_{\hat{k}}^\top \mathbf{W}_V^{(t)} \boldsymbol{a}_{\hat{k}})\|\mathbf{W}_V^{(0)} \mathbf{S}_{\hat{n}} \boldsymbol{\pi}_{\hat{n}}^{(0)}\| + \Theta(\|\mathbf{W}_V^{(0)} \mathbf{S}_{\hat{n}} \boldsymbol{\pi}_{\hat{n}}^{(0)}\|^2) \\
&= \Theta((\boldsymbol{a}_{\hat{k}}^\top \mathbf{W}_V^{(t)} \boldsymbol{a}_{\hat{k}})^2).
\end{aligned}
\tag{79}
$$

Then by $\pi^{(t)}_{n,m_{\mathbf{S}_n}} \geq \Theta(\frac{1}{M})$, it holds that

$$\boldsymbol{a}_k^\top \mathbf{W}_V^{(t+1)} \boldsymbol{a}_k - \boldsymbol{a}_k^\top \mathbf{W}_V^{(t)} \boldsymbol{a}_k \geq \Theta(\frac{\eta q_V}{2MK(\boldsymbol{a}_k^\top \mathbf{W}_V^{(t)} \boldsymbol{a}_k)}),$$

for $\forall k \in [K]$. By the positiveness and monotonicity of $\boldsymbol{a}_k^\top \mathbf{W}_V^{(t)} \boldsymbol{a}_k$ as well as Lemma D.3, for $\forall k \in [K]$ and some $C_7 > 0$, we have

$$\boldsymbol{a}_k^\top \mathbf{W}_V^{(t)} \boldsymbol{a}_k \geq \sqrt{C_7 \frac{\eta q_V (t - t_1)}{MK} + (\boldsymbol{a}_k^\top \mathbf{W}_V^{(t_1)} \boldsymbol{a}_k)^2},$$

during $t_1 \leq T_1 \leq t \leq T^\star$. Define $T_2 = C_7^{-1}(\eta q_V)^{-1} \sigma_1 K(\sigma_1 Md + 15\sigma_1 Md\sigma_p \sqrt{\log(\frac{4}{\varepsilon})} + M\sqrt{2\log(\frac{8(2K+K')^2}{\delta})} + \sigma_1 d) \leq T^\star = \Omega(\eta^{-1} q_V^{-1} \sigma_1^2 KMd^2 \log(\frac{1}{\epsilon}))$, we can choose $t \geq T_2$. Subsequently, extending $t_1 = T_1$ and $\boldsymbol{a}_k^\top \mathbf{W}_V^{(T_1)} \boldsymbol{a}_k \geq \frac{\bar{C}_1 \sigma_1 d^{1/2}}{M} - \sqrt{2\log(\frac{8(2K+K')^2}{\delta})}\sigma_1$ in this formula, such that for some $t \geq T^\star$, we have

$$\boldsymbol{a}_k^\top \mathbf{W}_V^{(t)} \boldsymbol{a}_k \geq \Omega(3(1 + 15\sigma_p^\star \sqrt{\log(\frac{4}{\varepsilon})})\|\mathbf{W}_V^{(0)} \boldsymbol{a}_{k_{\mathbf{T}}}\|) = \Omega(3(1 + 15\sigma_p^\star \sqrt{\log(\frac{4}{\varepsilon})})\sigma_1 d^{1/2}),$$

for $\forall k \in [K]$. Similarly, we define $T_3 = C_7^{-1}(\eta q_V)^{-1}\sigma_1 K(\sigma_1 d^2 K^{-3} + M\sqrt{2\log(\frac{8(2K+K')^2}{\delta})} + \sigma_1 d)$, and choose $T^\star \geq T_3 \leq T^\star = \Omega(\eta^{-1}q_V^{-1}\sigma_1^2 KMd^2 \log(\frac{1}{\epsilon}))$. By $\|\mathbf{W}_V^{(0)}\|_F^2 \leq 2d^2\sigma_1^2$ in Lemma D.8, for $\forall t \geq T_3$, it holds that

$$\boldsymbol{a}_k^\top \mathbf{W}_V^{(t)}\boldsymbol{a}_k \geq \Omega(\frac{\sigma_1 d}{\sqrt{K}}) \Rightarrow \sum_{k\in[K]}(\boldsymbol{a}_k^\top \mathbf{W}_V^{(t)}\boldsymbol{a}_k)^2 = \Omega(\|\mathbf{W}_V^{(0)}\|_F^2).$$

As we see that $\boldsymbol{u}\mathbf{W}_V^{(t)}\boldsymbol{v}$ only grow if $\boldsymbol{u} = \boldsymbol{v} = \boldsymbol{a}_k, \forall k \in [K]$, we have $\|\mathbf{W}_V^{(t)}\|_F = \Theta(\sqrt{K}\boldsymbol{a}_k^\top \mathbf{W}_V^{(t)}\boldsymbol{a}_k), \forall k \in [K]$. Similarly, by Lemma D.3, for some $C_8 > 0$ we also have

$$\boldsymbol{a}_k^\top \mathbf{W}_V^{(t)}\boldsymbol{a}_k \leq \sqrt{C_8 \frac{\eta q_V(t - t_1)}{K} + (\boldsymbol{a}_k^\top \mathbf{W}_V^{(t_1)}\boldsymbol{a}_k)^2} + \frac{C_8 \eta q_V}{K(\boldsymbol{a}_k^\top \mathbf{W}_V^{(t_1)}\boldsymbol{a}_k)},$$

for $t_1 \leq T_1 \leq t \leq T^\star$. $\qquad \square$

**Lemma F.9.** *Under Condition 3.1, during* $t_1 \leq t \leq T^\star = \Omega(\eta^{-1}q_V^{-1}\sigma_1^2 KMd^2 \log(\frac{1}{\epsilon}))$ *where* $t_1 \leq T_1$, *we denote* $\triangle = \min_t \pi_{n,m_{\mathbf{S}_n}}^{(t)} = \min\{\frac{1}{M}, \frac{1}{1+(M-1)e^{-\sigma_0^2 de^{q_V^{-1}\sigma_1^2 d}(d\log(\frac{1}{\epsilon}))^{\frac{M}{16q_V}}}}\}$). *Then it holds that*

$$(\mathbf{W}_Q^{(t)}\boldsymbol{a}_{\hat{k}})^\top(\mathbf{W}_K^{(t)}\boldsymbol{u}), (\mathbf{W}_Q^{(t)}\boldsymbol{v})^\top(\mathbf{W}_K^{(t)}\boldsymbol{v}), (\mathbf{W}_Q^{(t)}\boldsymbol{v})^\top(\mathbf{W}_K^{(t)}\boldsymbol{v}') = O(\sqrt{\log(16(2K+K')^2/\delta)}\sigma_0^2 d^{1/2})$$

$$(\mathbf{W}_Q^{(t)}\boldsymbol{a}_{\hat{k}})^\top(\mathbf{W}_K^{(t)}\boldsymbol{a}_{\hat{k}}) \leq C_9 \sigma_0^2 de^{q_V^{-1}\sigma_1^2 d}[\frac{\eta q_V t}{\sigma_1^2 dKM}]^{\frac{M}{16q_V}},$$

$$(\mathbf{W}_Q^{(t)}\boldsymbol{a}_{\hat{k}})^\top(\mathbf{W}_K^{(t)}\boldsymbol{a}_{\hat{k}}) \geq C_{10} \frac{\sigma_0^2 \sigma_1^2 d^2}{M^2 q_V}e^{2q_V^{-1}K\triangle^2(1-\triangle)^2(\frac{-1}{\frac{\eta q_V t}{2\sigma_1^2 dK}+1}+1)} \tag{80}$$
$$- \frac{C_3 \eta^2 q_V \sigma_0^2}{\sigma_1^2 M^2 K^2} - 4C_3 \sqrt{\log(16(2K+K')^2/\delta)}\sigma_0^2 d^{1/2},$$

*for* $\forall k \in [K], n \in [N], s \in [7], \neg m_{\mathbf{S}_n} \neq m_{\mathbf{S}_n} \in [M], \boldsymbol{u} \in \{\boldsymbol{a}_s\}_{s\neq k\in[K]} \cup \{\boldsymbol{b}_s\}_{s\in[K]} \cup \{\boldsymbol{\nu}_{k'}\}_{k'\in[K']} \cup \{\boldsymbol{\xi}_{n,m}\}_{n\in[N],m\in[M]}$ *and* $\boldsymbol{v} \neq \boldsymbol{v}' \in \{\boldsymbol{b}_s\}_{s\in[K]} \cup \{\boldsymbol{\xi}_{n,m}\}_{n\in[N],m\in[M]}$.

*Furthermore, after* $t \geq T^\star = \Omega(\eta^{-1}q_V^{-1}\sigma_1^2 KMd^2 \log(\frac{1}{\epsilon}))$, *it holds that*

$$(\mathbf{W}_Q^{(t)}\boldsymbol{a}_{\hat{k}})^\top(\mathbf{W}_K^{(t)}\boldsymbol{a}_{\hat{k}}) \leq O(\sigma_0^2 de^{q_V^{-1}\sigma_1^2 d}(d\log(\frac{1}{\varepsilon}))^{\frac{M}{16q_V}}),$$
$$\pi_{n,m_{\mathbf{S}_n}}^{(T^\star)} \leq \Theta(\frac{1}{1+(M-1)e^{-\sigma_0^2 de^{q_V^{-1}\sigma_1^2 d}(d\log(\frac{1}{\varepsilon}))^{\frac{M}{16q_V}}}}), \tag{81}$$

*for* $\forall k \in [K]$.

*Proof.* First, observing that

$$(1 - \frac{(\mathbf{W}_V^{(t)}\mathbf{S}_n\boldsymbol{\pi}_n^{(t)})^\top}{\|\mathbf{W}_V^{(t)}\mathbf{S}_n\boldsymbol{\pi}_n^{(t)}\|} \frac{\mathbf{W}_V^{(t)}\mathbf{S}_n\pi_{n,m_{\mathbf{S}_n}}^{(t)}\boldsymbol{e}_{n,m_{\mathbf{S}_n}}}{\|\mathbf{W}_V^{(t)}\mathbf{S}_n\boldsymbol{\pi}_n^{(t)}\|}) = \frac{\sum_{m\neq m_{\mathbf{S}_n}}(\mathbf{W}_V^{(t)}\mathbf{S}_n\pi_{n,m}^{(t)}\boldsymbol{e}_{n,m})^\top(\mathbf{W}_V^{(t)}\mathbf{S}_n\pi_{n,m_{\mathbf{S}_n}}^{(t)}\boldsymbol{e}_{n,m_{\mathbf{S}_n}})}{\|\mathbf{W}_V^{(t)}\mathbf{S}_n\boldsymbol{\pi}_n^{(t)}\|^2}$$

$$= \frac{\sum_{m\neq m_{\mathbf{S}_n}}(\mathbf{W}_V^{(t)}\mathbf{S}_n\pi_{n,m}^{(t)}\boldsymbol{e}_{n,m})^\top(\mathbf{W}_V^{(t)}\mathbf{S}_n\pi_{n,m_{\mathbf{S}_n}}^{(t)}\boldsymbol{e}_{n,m_{\mathbf{S}_n}})}{\Theta((\boldsymbol{a}_k^\top \mathbf{W}_V^{(t)}\boldsymbol{a}_k)^2)}$$

$$= \frac{\Theta((\sum_{m\neq m_{\mathbf{S}_n}}\pi_{n,m}^{(t)})(\pi_{n,m_{\mathbf{S}_n}}^{(t)})(\sigma_1^2 d))}{\Theta((\boldsymbol{a}_k^\top \mathbf{W}_V^{(t)}\boldsymbol{a}_k)^2)} = \frac{\Theta((1 - \pi_{n,m_{\mathbf{S}_n}}^{(t)})(\pi_{n,m_{\mathbf{S}_n}}^{(t)})(\sigma_1^2 d))}{\Theta((\boldsymbol{a}_k^\top \mathbf{W}_V^{(t)}\boldsymbol{a}_k)^2)} \tag{82}$$

for $\forall n \in \mathcal{N}_k$. Here, the first equality is by definition; the second equality is by Eq.(79); the third equality is by the feeble growths of task-irrelevant components denoted in Lemma F.8 and $\|\mathbf{W}_V^{(0)}\boldsymbol{a}_k\|^2 = \Theta(\sigma_1^2 d)$. Therefore, similar to Eq.(72),

for $\forall \hat{k} \in [K]$ we have

$$
\begin{aligned}
(\mathbf{W}_Q^{(t+1)}\boldsymbol{a}_{\hat{k}})^\top(\mathbf{W}_K^{(t+1)}\boldsymbol{a}_{\hat{k}}) - (\mathbf{W}_Q^{(t)}\boldsymbol{a}_{\hat{k}})^\top(\mathbf{W}_K^{(t)}\boldsymbol{a}_{\hat{k}}) =& \Theta\Big(\frac{\eta\sum_{\hat{n}\in\mathcal{N}_{\hat{k}}}}{N}\frac{\boldsymbol{a}_{\hat{k}}^\top\mathbf{W}_V^{(t)}\mathbf{S}_n\pi_{n,m\mathbf{S}_n}^{(t)}\boldsymbol{e}_{m\mathbf{S}_n}}{\|\mathbf{W}_V^{(t)}\mathbf{S}_n\boldsymbol{\pi}_n^{(t)}\|}(1-\frac{(\mathbf{W}_V^{(t)}\mathbf{S}_n\boldsymbol{\pi}_n^{(t)})^\top}{\|\mathbf{W}_V^{(t)}\mathbf{S}_n\boldsymbol{\pi}_n^{(t)}\|}\\
& \frac{\mathbf{W}_V^{(t)}\mathbf{S}_n\pi_{n,m\mathbf{S}_n}^{(t)}\boldsymbol{e}_{n,m\mathbf{S}_n}}{\|\mathbf{W}_V^{(t)}\mathbf{S}_n\boldsymbol{\pi}_n^{(t)}\|})\cdot(1-\pi_{n,m\mathbf{S}_n}^{(t)})\boldsymbol{a}_{\hat{k}}^\top((\mathbf{W}_Q^{(t)})^\top\mathbf{W}_Q^{(t)}\\
& +(\mathbf{W}_K^{(t)})^\top\mathbf{W}_K^{(t)})\boldsymbol{a}_{\hat{k}}\\
=& \Theta\Big(\frac{\eta(1-\pi_{n,m\mathbf{S}_n}^{(t)})^2(\pi_{n,m\mathbf{S}_n}^{(t)})^2}{K(\boldsymbol{a}_{\hat{k}}^\top\mathbf{W}_V^{(t)}\boldsymbol{a}_{\hat{k}})^2})\boldsymbol{a}_{\hat{k}}^\top((\mathbf{W}_Q^{(t)})^\top\mathbf{W}_Q^{(t)}+\\
& (\mathbf{W}_K^{(t)})^\top\mathbf{W}_K^{(t)})\boldsymbol{a}_{\hat{k}}\\
=& \Theta\Big(\frac{(\mathbf{W}_Q^{(t+1)}\boldsymbol{a}_{\hat{k}})^\top(\mathbf{W}_Q^{(t+1)}\boldsymbol{a}_{\hat{k}})-(\mathbf{W}_Q^{(t)}\boldsymbol{a}_{\hat{k}})^\top(\mathbf{W}_Q^{(t)}\boldsymbol{a}_{\hat{k}})}{2}\\
& +\frac{(\mathbf{W}_K^{(t+1)}\boldsymbol{a}_{\hat{k}})^\top(\mathbf{W}_K^{(t+1)}\boldsymbol{a}_{\hat{k}})-(\mathbf{W}_K^{(t)}\boldsymbol{a}_{\hat{k}})^\top(\mathbf{W}_K^{(t)}\boldsymbol{a}_{\hat{k}})}{2}\Big),
\end{aligned}
$$
(83)

where the first equality is by Eq.(72); the second inequality is by Eq.(82) and Eq.(82); the last equality is by Eq.(74). Note that $g(x) = x^2(1-x)^2, x \in (0,1)$ has a pole at $x = 1/2, y = 1/16$, and is symmetry around the axis $x = 1/2$. Therefore, by Lemma D.2, Lemma F.8 as well as the upper bound of $(\mathbf{W}_Q^{(t)}\boldsymbol{a}_{\hat{k}})^\top(\mathbf{W}_K^{(t)}\boldsymbol{a}_{\hat{k}})$ in Lemma F.7, the upper bound of the evolution of $(\mathbf{W}_Q^{(t)}\boldsymbol{a}_{\hat{k}})^\top(\mathbf{W}_K^{(t)}\boldsymbol{a}_{\hat{k}}), \forall \hat{k} \in [K]$ is the following system:

$$
(\mathbf{W}_Q^{(t)}\boldsymbol{a}_{\hat{k}})^\top(\mathbf{W}_K^{(t)}\boldsymbol{a}_{\hat{k}}) \leq \Theta\Big((\frac{b_{t_1}}{(t_1+\frac{d}{c})^{\frac{a}{c}}})(t+\frac{d}{c})^{\frac{a}{c}}\Big),
$$

$$
\begin{cases}
a & := (\frac{\eta}{16K}),\\
b_{t_1} & = C_5\sigma_0^2 de^{q_V^{-1}\sigma_1^2 d},\\
c & = C_7\frac{\eta q_V}{MK},\\
d & := C_7\frac{\eta q_V((\eta q_V)^{-1}\sigma_1^2 dK)}{MK}+\sigma_1^2 d\\
t_1 & := (\eta q_V)^{-1}\sigma_1^2 dK.
\end{cases}
$$

Here, obviously $a < c$ by $M \geq 2$ and $q_V \leq \sigma_1 d^{1/2} < 1$ by Condition 3.1. we take $b_{t_1}$ as the upper bound of $(\mathbf{W}_Q^{(t)}\boldsymbol{a}_{\hat{k}})^\top(\mathbf{W}_K^{(t)}\boldsymbol{a}_{\hat{k}}), \forall \hat{k} \in [K]$ during the first phase; $d$ is defined by the lower bound of $\boldsymbol{a}_{\hat{k}}^\top\mathbf{W}_V^{(t)}\boldsymbol{a}_{\hat{k}}$ in the second phase by Lemma F.8, with $\boldsymbol{a}_{\hat{k}}^\top\mathbf{W}_V^{(t_1)}\boldsymbol{a}_{\hat{k}} := \sigma_1 d^{1/2}$ be the upper bound. Therefore, we have

$$
\begin{aligned}
(\mathbf{W}_Q^{(t)}\boldsymbol{a}_{\hat{k}})^\top(\mathbf{W}_K^{(t)}\boldsymbol{a}_{\hat{k}}) &\leq C_9\sigma_0^2 de^{q_V^{-1}\sigma_1^2 d}\big[\frac{\eta q_V t}{\sigma_1^2 dK(M+1)}+\frac{M+1}{M+2}\big]^{\frac{M}{16q_V}},\\
&\leq C_9\sigma_0^2 de^{q_V^{-1}\sigma_1^2 d}\big[\frac{\eta q_V t}{\sigma_1^2 dKM}\big]^{\frac{M}{16q_V}},
\end{aligned}
$$

for $\hat{k} \in [K]$. Here $C_9 > 0$ is some constant. Extend $T^\star = \Omega(\eta^{-1}q_V^{-1}\sigma_1^2 KMd^2\log(\frac{1}{\epsilon}))$ in the formula, we obtain

$$
(\mathbf{W}_Q^{(t)}\boldsymbol{a}_{\hat{k}})^\top(\mathbf{W}_K^{(t)}\boldsymbol{a}_{\hat{k}}) \leq O(\sigma_0^2 de^{q_V^{-1}\sigma_1^2 d}(d\log(\frac{1}{\varepsilon}))^{\frac{M}{16q_V}}).
$$

As such, it holds that

$$
\pi_{n,m\mathbf{S}_n}^{(T^\star)} \leq \Theta\Big(\frac{1}{1+(M-1)e^{-\sigma_0^2 de^{q_V^{-1}\sigma_1^2 d}(d\log(\frac{1}{\varepsilon}))^{\frac{M}{16q_V}}}}\Big).
$$

To examine the lower bound, we first denote $\triangle = \min_t \pi_{n,m\mathbf{S}_n}^{(t)} = \min\{\frac{1}{M}, \frac{1}{1+(M-1)e^{-\sigma_0^2 de^{q_V^{-1}\sigma_1^2 d}(d\log(\frac{1}{\varepsilon}))^{\frac{M}{16q_V}}}}\})$. Secondly, to alleviate the complexity, we choose a linear upper bound of $\boldsymbol{a}_k^\top\mathbf{W}_V^{(t)}\boldsymbol{a}_k \leq \sqrt{C_8\frac{\eta q_V(t-t_1)}{K}+(\boldsymbol{a}_k^\top\mathbf{W}_V^{(t_1)}\boldsymbol{a}_k)^2}+$

$\frac{C_8 \eta q_V}{K(\boldsymbol{a}_k^\top \mathbf{W}_V^{(t_1)} \boldsymbol{a}_k)}$. By the concavity of $g(x) = \sqrt{ax+b} + c, x \geq x_1$, the linear upper bound after $x_1$ is $\hat{g}(x) = \frac{1}{2}(ax_1 + b)^{-1/2}(x - x_1) + g(x_1)$. That is

$$
\begin{aligned}
\boldsymbol{a}_k^\top \mathbf{W}_V^{(t)} \boldsymbol{a}_k &\leq \frac{C_8 \eta q_V \sigma_1^{-1} d^{-1/2}(t - t_1)}{2K} + \sigma_1 d^{\frac{1}{2}} \\
&\leq \frac{C_8 \eta q_V \sigma_1^{-1} d^{-1/2} t}{2K} + \sigma_1 d^{\frac{1}{2}}.
\end{aligned}
\tag{84}
$$

Therefore, by Lemma D.6, Eq.(65), (73), (74), (83), the lower bound of the evolution of $(\mathbf{W}_Q^{(t)} \boldsymbol{a}_{\hat{k}})^\top (\mathbf{W}_K^{(t)} \boldsymbol{a}_{\hat{k}}), \forall \hat{k} \in [K]$ is the following system:

$$
(\mathbf{W}_Q^{(t)} \boldsymbol{a}_{\hat{k}})^\top (\mathbf{W}_K^{(t)} \boldsymbol{a}_{\hat{k}}) \geq \Theta\left(g_{t_1} e^{-\frac{a}{b(bt+c)} + \frac{a}{b(bt_1+c)}}\right) - \frac{C_3 \eta^2 q_V \sigma_0^2}{\sigma_1^2 M^2 K^2} - 4C_3 \sqrt{\log(16(2K + K')^2/\delta)} \sigma_0^2 d^{1/2},
$$

$$
\begin{cases}
a &:= \eta \triangle^2 (1 - \triangle)^2, \\
b &= \frac{C_8 \eta q_V \sigma_1^{-1} d^{-1/2}}{2K}, \\
c &= \sigma_1 d^{\frac{1}{2}}, \\
g_{t_1} &:= C_6 \frac{\sigma_0^2 \sigma_1^2 d^2}{M^2 q_V}, \\
t_1 &:= 0.
\end{cases}
$$

Here, $a < 2c$ is by $\triangle^2 (1 - \triangle)^2 \leq \frac{(M-1)^2}{M^4}$ and the minor gradient step $\eta = o\left(\frac{\sigma_1 d^{\frac{1}{2}} M^4}{(M-1)^2}\right)$ by Condition 3.1. Therefore, it holds that

$$
\begin{aligned}
(\mathbf{W}_Q^{(t)} \boldsymbol{a}_{\hat{k}})^\top (\mathbf{W}_K^{(t)} \boldsymbol{a}_{\hat{k}}) \geq & C_{10} \frac{\sigma_0^2 \sigma_1^2 d^2}{M^2 q_V} e^{\frac{-\eta \triangle^2 (1-\triangle)^2}{\frac{\eta q_V \sigma_1^{-1} d^{-1/2}}{2K}\left(\frac{\eta q_V \sigma_1^{-1} d^{-1/2}}{2K} t + \sigma_1 d^{\frac{1}{2}}\right)} + \frac{\eta \triangle^2 (1-\triangle)^2}{\frac{\eta q_V \sigma_1^{-1} d^{-1/2}}{2K} \sigma_1 d^{\frac{1}{2}}}} \\
& - \frac{C_3 \eta^2 q_V \sigma_0^2}{\sigma_1^2 M^2 K^2} - 4C_3 \sqrt{\log(16(2K + K')^2/\delta)} \sigma_0^2 d^{1/2} \\
\geq & C_{10} \frac{\sigma_0^2 \sigma_1^2 d^2}{M^2 q_V} e^{\frac{-2 q_V^{-1} K \triangle^2 (1-\triangle)^2}{\frac{\eta q_V t}{2\sigma_1^2 dK} + 1} + 2 q_V^{-1} K \triangle^2 (1-\triangle)^2} \\
& - \frac{C_3 \eta^2 q_V \sigma_0^2}{\sigma_1^2 M^2 K^2} - 4C_3 \sqrt{\log(16(2K + K')^2/\delta)} \sigma_0^2 d^{1/2},
\end{aligned}
$$

for some positive constant $C_{10}$. $\qquad\qquad\square$

# G. Training Dynamics: ICL-type Data

This section also assumes the results in Appendix D.2 all hold with high probability. This section examines scenarios involving training on ICL data, using notations for $\mathbf{T}_n$ consistent with Section F. The proof strategies for QA-ICL data are identical to those in this section. Therefore, we focus solely on the training dynamics for $\mathbf{T}_n, n \in [N]$.

The projection evolution dynamics remain largely the same as described in Section F, with the key distinction being the evolution of $\boldsymbol{b}_k^\top \mathbf{W}_V^{(t)} \boldsymbol{b}_k$. This difference arises inherently from the gradient forms, where the unavoidable imbalance between positive and negative data for some concept $k \in [K]$ can drive the growth of either $\boldsymbol{b}_k^\top \mathbf{W}_V^{(t)} \boldsymbol{b}_k$ or $-\boldsymbol{b}_k^\top \mathbf{W}_V^{(t)} \boldsymbol{b}_k$, as suggested by Lemma D.10.

**Lemma G.1.** *Under Condition 3.1, during $t \leq T_1 = \Theta((\eta q_V)^{-1} \sigma_1^2 dK)$, for $\forall y \in \{\pm 1\}$, there exists $k_y \in [K]$, it holds*

*that*

$$|\boldsymbol{b}_{k_+}^\top \mathbf{W}_V^{(t)} \boldsymbol{b}_{k_+}|, \; |\boldsymbol{b}_{k_-}^\top \mathbf{W}_V^{(t)} \boldsymbol{b}_{k_-}|, \; \boldsymbol{a}_k^\top \mathbf{W}_V^{(t)} \boldsymbol{a}_k = o(\sigma_1 d^{1/2}),$$

$$|\boldsymbol{b}_{k_+}{}^\top \mathbf{W}_V^{(t)} \boldsymbol{b}_{k_-}|, \; |\boldsymbol{b}_{k_+}{}^\top \mathbf{W}_V^{(t)} \boldsymbol{a}_k|, \; |\boldsymbol{b}_{k_-}{}^\top \mathbf{W}_V^{(t)} \boldsymbol{a}_k| = O(\sqrt{2\log(\frac{8(2K+K')^2}{\delta})}\sigma_1),$$

$$|\boldsymbol{b}_{k_+}{}^\top \mathbf{W}_V^{(t)} \boldsymbol{u}|, \; |\boldsymbol{b}_{k_-}{}^\top \mathbf{W}_V^{(t)} \boldsymbol{u}|, \; |\boldsymbol{a}_k{}^\top \mathbf{W}_V^{(t)} \boldsymbol{u}|, \; |\boldsymbol{v}^\top \mathbf{W}_V^{(t)} \boldsymbol{v}|, \; |\boldsymbol{v}^\top \mathbf{W}_V^{(t)} \boldsymbol{v}'| = O(\sqrt{2\log(\frac{8(2K+K')^2}{\delta})}\sigma_1), \quad (85)$$

$$|(\mathbf{W}_Q^{(t)} \boldsymbol{a}_{k_{\mathbf{T}_n}})^\top \mathbf{W}_K^{(t)}(\boldsymbol{a}_{k_{\mathbf{T}_n}})| = O(\sigma_0^2 d e^{q_V^{-1}\sigma_1^2 d}),$$

$$|(\mathbf{W}_Q^{(t)} \mathbf{x}_n)^\top \mathbf{W}_K^{(t)}(\boldsymbol{\nu}_{n,\neg m_{\mathbf{T}_n}} + \boldsymbol{\xi}_{n,\neg m_{\mathbf{T}_n}})| = O(\sqrt{\log(\frac{16(2K+K')^2}{\delta})}\sigma_0^2 d^{1/2}),$$

*for* $\forall k \in [K], n \in [N], s \in [7], \neg m_{\mathbf{T}_n} \neq m_{\mathbf{T}_n} \in [M], \boldsymbol{u} \in \{\boldsymbol{a}_s\}_{s\neq k\in[K]} \cup \{\boldsymbol{b}_s\}_{s\neq k_\pm \in[K]} \cup \{\boldsymbol{\nu}_{k'}\}_{k'\in[K']} \cup \{\boldsymbol{\xi}_{n,m}\}_{n\in[N],m\in[M]}$ *and* $\boldsymbol{v} \neq \boldsymbol{v}' \in \{\boldsymbol{b}_s\}_{s\neq k_\pm\in[K]} \cup \{\boldsymbol{\xi}_{n,m}\}_{n\in[N],m\in[M]}$. *At the end of this stage, we have*

$$\boldsymbol{a}_k^\top \mathbf{W}_V^{(T_1)} \boldsymbol{a}_k \geq \frac{\underline{C}_1 \sigma_1 d^{1/2}}{M} - \sqrt{2\log(\frac{8(2K+K')^2}{\delta})}\sigma_1,$$

$$(\mathbf{W}_Q^{(T_1)} \boldsymbol{a}_{\hat{k}})^\top (\mathbf{W}_K^{(T_1)} \boldsymbol{a}_{\hat{k}}) \geq \underline{C}_2 \frac{\sigma_0^2 \sigma_1^2 d^2}{M^2 q_V} - \underline{C}_3 (\frac{\eta^2 q_V \sigma_0^2}{\sigma_1^2 M^2 K^2} + 4\sqrt{\log(16(2K+K')^2/\delta)}\sigma_0^2 d^{1/2}),$$

$$|\boldsymbol{b}_{k_+}^\top \mathbf{W}_V^{(T_1)} \boldsymbol{b}_k| \geq \frac{\underline{C}_4 \sigma_1 d^{1/2}}{M} - \sqrt{2\log(\frac{8(2K+K')^2}{\delta})}\sigma_1, \quad (86)$$

$$|\boldsymbol{b}_{k_-}^\top \mathbf{W}_V^{(T_1)} \boldsymbol{b}_{k_-}| \geq \frac{\underline{C}_5 \sigma_1 d^{1/2}}{M} - \sqrt{2\log(\frac{8(2K+K')^2}{\delta})}\sigma_1,$$

*for some positive constants* $\underline{C}_{1-5}$.

*Proof.* The proof strategies follows Lemma F.1, despite differences in the growing of $|\boldsymbol{b}_{k_+}^\top \mathbf{W}_V^{(T_1)} \boldsymbol{b}_k|$ and $|\boldsymbol{b}_{k_-}^\top \mathbf{W}_V^{(T_1)} \boldsymbol{b}_{k_-}|$ as follows.

By Lemma D.10, we see that for $\forall y \in \{\pm 1\}$, there exists $k_y \in [K]$, such that $|\mathcal{N}_{k_y}^y - N/(2K)| \geq 5 \cdot 10^{-3} N/(2K)$. Therefore, similar to Eq.(57), we have

$$|\boldsymbol{b}_{k_y}^\top \mathbf{W}_V^{(t+1)} \boldsymbol{b}_{k_y} - \boldsymbol{b}_{k_y}^\top \mathbf{W}_V^{(t)} \boldsymbol{b}_{k_y}| = \eta q_V |\mathbb{E}_{\mathcal{P}_{\mathrm{QA}}^{\mathrm{tr}}}[\frac{\boldsymbol{b}_{k_y}^\top \mathbf{\Pi}^{\mathbf{u}_{k_{\mathbf{y}_n}} - \sum_{k\in[7K+K']} \omega_{n,k} \mathbf{u}_k}_{(\mathbf{W}_V^{(t)} \mathbf{S}_n \boldsymbol{\pi}_n^{(t)})^\perp}(\mathbf{S}_n \boldsymbol{\pi}_n^{(t)})^\top \boldsymbol{b}_{k_y}}{\|\mathbf{W}_V^{(t)} \mathbf{S}_n \boldsymbol{\pi}_n^{(t)}\|}]|$$

$$= \frac{\eta q_V}{N} \sum_{y\in[\pm 1]} \sum_{\hat{n}\in\mathcal{N}_{k_y}^y} \Theta(|[\frac{\boldsymbol{b}_{k_y}^\top \mathbf{\Pi}^{\mathbf{u}_{k_{\mathbf{y}_{\hat{n}}}} - \sum_{k\in[7K+K']} \omega_{\hat{n},k} \mathbf{u}_k}_{(\mathbf{W}_V \mathbf{S}_{\hat{n}} \boldsymbol{\pi}_{\hat{n}}^{(t)})^\perp}(\mathbf{S} \boldsymbol{\pi}_{\hat{n}}^{(t)})^\top \boldsymbol{b}_{k_y}}{\|\mathbf{W}_V^{(t)} \mathbf{S}_{\hat{n}} \boldsymbol{\pi}_{\hat{n}}^{(t)}\|}]|) \quad (87)$$

$$\leq \Theta([\sum_{\hat{n}\in\mathcal{N}_{k_y}^y \setminus \mathcal{N}_{k_y}^{-y}} \frac{\eta q_V \pi_{\hat{n},m_{\mathbf{S}_{\hat{n}}}}^{(t)}}{2NM\|\mathbf{W}_V^{(t)} \mathbf{S}_{\hat{n}} \boldsymbol{\pi}_{\hat{n}}^{(t)}\|}]) = \Theta(\boldsymbol{a}_{k_y}^\top \mathbf{W}_V^{(t+1)} \boldsymbol{a}_{k_y} - \boldsymbol{a}_{k_y}^\top \mathbf{W}_V^{(t)} \boldsymbol{a}_{k_y}).$$

The inequality is inherently due to the existence of $\boldsymbol{b}_{k_y}$ in $\mathbf{T}_{\hat{n}}$, while QA data did not possess $\boldsymbol{b}_{k_y}$ in its sentence. Since we see that $\mathcal{N}_{k_y}^y$ is guaranteed to not balanced with probability $1 - \delta$, the growing of $|\boldsymbol{b}_{k_y}^\top \mathbf{W}_V^{(t+1)} \boldsymbol{b}_{k_y} - \boldsymbol{b}_{k_y}^\top \mathbf{W}_V^{(t)} \boldsymbol{b}_{k_y}|$ is inevitable. The remaining proofs follow Lemma F.2. $\square$

**Lemma G.2.** *Under Condition 3.1, suppose Eq. (49) holds at iteration* $t \leq T_1$, *then for* $\forall \hat{k} \in [K], \boldsymbol{u} \in \{\boldsymbol{a}_s\}_{s\neq\hat{k}\in[K]} \cup \{\boldsymbol{b}_s\}_{s\in[K]} \cup \{\boldsymbol{\xi}_{n,m}\}_{n\in[N],m\in[M]}$ *and* $\boldsymbol{v} \neq \boldsymbol{v}' \in \{\boldsymbol{b}_s\}_{s\in[K]} \cup \{\boldsymbol{\xi}_{n,m}\}_{n\in[N],m\in[M]}$, *there exist some positive constants* $\underline{C}_3, \underline{C}_4, \underline{C}_5, \underline{C}_6$, *such that*

- *If* $\boldsymbol{a}_{\hat{k}}^\top \mathbf{W}_V^{(t)} \boldsymbol{a}_{\hat{k}} \leq 0$ *at* $t = 0$, *after at most* $t_3 = (\eta q_V)^{-1} \underline{C}_1^{-1} \sigma_1^2 d^{1/2} M K(\sqrt{2\log(\frac{8(2K+K')^2}{\delta})})$ *iterations,* $\boldsymbol{a}_{\hat{k}}^\top \mathbf{W}_V^{(t)} \boldsymbol{a}_{\hat{k}}$ *would get positive. During this period, we have*

$$|(\mathbf{W}_Q^{(t)} \boldsymbol{a}_{\hat{k}})^\top (\mathbf{W}_K^{(t)} \boldsymbol{a}_{\hat{k}})| \leq 4\sqrt{\log(16(2K+K')^2/\delta)}\sigma_0^2 d^{1/2}. \quad (88)$$

- *During $0 \leq \boldsymbol{a}_k^\top \mathbf{W}_V^{(t)} \boldsymbol{a}_k \leq O(\sigma_1 d^{1/2})$ within $[t_4, T_1]$, where $t_4$ satisfying $0 \leq t_4 \leq t_3$, it holds that*

$$
\begin{aligned}
(\mathbf{W}_Q^{(t)} \boldsymbol{a}_{\hat{k}})^\top (\mathbf{W}_K^{(t)} \boldsymbol{a}_{\hat{k}}) &\geq \frac{\underline{C_3} \eta^2 q_V \sigma_0^2 [(t - t_4 - 1)^2 - 1]}{\sigma_1^2 M^2 K^2} - 4\sqrt{\log(16(2K + K')^2/\delta)}\sigma_0^2 d^{1/2}, \\
(\mathbf{W}_Q^{(t)} \boldsymbol{a}_{\hat{k}})^\top (\mathbf{W}_K^{(t)} \boldsymbol{a}_{\hat{k}}) &\leq \underline{C_4}((4\sqrt{\log(16(2K + K')^2/\delta)}\sigma_0^2 d^{1/2} + \frac{3\sigma_0^2 d}{2}) + \frac{3\sigma_0^2 d}{2}\exp(\frac{\hat{C}_2 C_2 \eta^2 q_V t^2}{\sigma_1^2 dK^2})), \quad (89) \\
(\mathbf{W}_Q^{(t)} \boldsymbol{a}_{\hat{k}})^\top (\mathbf{W}_K^{(t)} \boldsymbol{u}), (\mathbf{W}_Q^{(t)} \boldsymbol{v})^\top (\mathbf{W}_K^{(t)} \boldsymbol{v}), \ (\mathbf{W}_Q^{(t)} \boldsymbol{v})^\top (\mathbf{W}_K^{(t)} \boldsymbol{v}') &= O(\sqrt{\log(16(2K + K')^2/\delta)}\sigma_0^2 d^{1/2}).
\end{aligned}
$$

- *Furthermore, it holds that*

$$
\begin{aligned}
(\mathbf{W}_Q^{(T_1)} \boldsymbol{a}_{\hat{k}})^\top (\mathbf{W}_K^{(T_1)} \boldsymbol{a}_{\hat{k}}) &\geq \underline{C_6} \frac{\sigma_0^2 \sigma_1^2 d^2}{M^2 q_V} - \frac{\underline{C_3} \eta^2 q_V \sigma_0^2}{\sigma_1^2 M^2 K^2} - 4C_3\sqrt{\log(16(2K + K')^2/\delta)}\sigma_0^2 d^{1/2}, \\
(\mathbf{W}_Q^{(T_1)} \boldsymbol{a}_{\hat{k}})^\top (\mathbf{W}_K^{(T_1)} \boldsymbol{a}_{\hat{k}}) &\leq \underline{C_5} \sigma_0^2 d e^{q_V^{-1} \sigma_1^2 d}.
\end{aligned} \quad (90)
$$

*Proof.* The proof strategies follows Lemma F.7. $\qquad\square$

**Lemma G.3.** *Under Condition 3.1, during $t_1 \leq t \leq T^\star = \Omega(\eta^{-1} q_V^{-1} \sigma_1^2 KMd^2 \log(\frac{1}{\epsilon}))$ where $t_1 \leq T_1$, we have*

$$
\begin{aligned}
|\boldsymbol{b}_{k_+}^\top \mathbf{W}_V^{(t)} \boldsymbol{b}_{k_-}|, \ |\boldsymbol{b}_{k_+}^\top \mathbf{W}_V^{(t)} \boldsymbol{a}_k|, \ |\boldsymbol{b}_{k_-}^\top \mathbf{W}_V^{(t)} \boldsymbol{a}_k| &= O(\sqrt{2\log(\frac{8(2K + K')^2}{\delta})}\sigma_1), \\
|\boldsymbol{b}_{k_+}^\top \mathbf{W}_V^{(t)} \boldsymbol{u}|, \ |\boldsymbol{b}_{k_-}^\top \mathbf{W}_V^{(t)} \boldsymbol{u}|, \ |\boldsymbol{a}_k^\top \mathbf{W}_V^{(t)} \boldsymbol{u}|, \ |\boldsymbol{v}^\top \mathbf{W}_V^{(t)} \boldsymbol{v}|, \ |\boldsymbol{v}^\top \mathbf{W}_V^{(t)} \boldsymbol{v}'| &= O(\sqrt{2\log(\frac{8(2K + K')^2}{\delta})}\sigma_1), \\
\|\boldsymbol{i}_{V,\boldsymbol{a}_k^\perp}^{(t)}\|, \ {\boldsymbol{i}_{V,\boldsymbol{a}_k^\perp}^{(t)}}^\top \mathbf{W}_V^{(0)} \boldsymbol{a}_k &= o(\|\mathbf{W}_V^{(0)} \boldsymbol{a}_k\|), \quad (91) \\
\boldsymbol{a}_k^\top \mathbf{W}_V^{(t)} \boldsymbol{a}_k &\geq \sqrt{\underline{C_7} \frac{\eta q_V (t - t_1)}{MK} + (\boldsymbol{a}_k^\top \mathbf{W}_V^{(t_1)} \boldsymbol{a}_k)^2}, \\
\boldsymbol{a}_k^\top \mathbf{W}_V^{(t)} \boldsymbol{a}_k &\leq \sqrt{\underline{C_8} \frac{\eta q_V (t - t_1)}{K} + (\boldsymbol{a}_k^\top \mathbf{W}_V^{(t_1)} \boldsymbol{a}_k)^2} + \frac{C_8 \eta q_V}{K(\boldsymbol{a}_k^\top \mathbf{W}_V^{(t_1)} \boldsymbol{a}_k)},
\end{aligned}
$$

*for $\forall k \in [K], n \in [N], s \in [7], \neg m_{\mathbf{T}_n} \neq m_{\mathbf{T}_n} \in [M], \boldsymbol{u} \in \{\boldsymbol{a}_s\}_{s \neq k \in [K]} \cup \{\boldsymbol{b}_s\}_{s \neq k_\pm \in [K]} \cup \{\boldsymbol{\nu}_{k'}\}_{k' \in [K']} \cup \{\boldsymbol{\xi}_{n,m}\}_{n \in [N], m \in [M]}$ and $\boldsymbol{v} \neq \boldsymbol{v}' \in \{\boldsymbol{b}_s\}_{s \in [K]} \cup \{\boldsymbol{\xi}_{n,m}\}_{n \in [N], m \in [M]}$.*

*Furthermore, after $t \geq T^\star = \Omega(\eta^{-1} q_V^{-1} \sigma_1^2 KMd^2 \log(\frac{1}{\epsilon}))$, it holds that*

$$
\begin{aligned}
\boldsymbol{a}_k^\top \mathbf{W}_V^{(t)} \boldsymbol{a}_k &= \Omega(3(1 + 15\sigma_p^\star \sqrt{\log(\frac{4}{\varepsilon})})\sigma_1 d^{1/2}) \\
\|\mathbf{W}_V^{(t)}\|_F &= \Theta(\sqrt{K}\boldsymbol{a}_k^\top \mathbf{W}_V^{(t)} \boldsymbol{a}_k), \quad (92) \\
|\boldsymbol{b}_{k_+}^\top \mathbf{W}_V^{(t)} \boldsymbol{b}_k|, \ |\boldsymbol{b}_{k_-}^\top \mathbf{W}_V^{(t)} \boldsymbol{b}_{k_-}| &= \Theta(\boldsymbol{a}_k^\top \mathbf{W}_V^{(t)} \boldsymbol{a}_k).
\end{aligned}
$$

*for $\forall k \in [K]$.*

*Proof.* Based on Lemma G.1, the proof strategy follows Lemma G.3. $\qquad\square$

**Lemma G.4.** *Under Condition 3.1, during $t_1 \leq t \leq T^\star = \Omega(\eta^{-1} q_V^{-1} \sigma_1^2 KMd^2 \log(\frac{1}{\epsilon}))$ where $t_1 \leq T_1$, we denote*

$\triangle = \min_t \pi_{n,m_{\mathbf{T}_n}}^{(t)} = \min\{\frac{1}{M}, \frac{1}{1+(M-1)e^{-\sigma_0^2 de^{q_V^{-1}\sigma_1^2 d}(d\log(\frac{1}{\varepsilon}))^{\frac{M}{16q_V}}}}\}$). *Then it holds that*

$$(\mathbf{W}_Q^{(t)}\boldsymbol{a}_{\hat{k}})^\top(\mathbf{W}_K^{(t)}\boldsymbol{u}), (\mathbf{W}_Q^{(t)}\boldsymbol{v})^\top(\mathbf{W}_K^{(t)}\boldsymbol{v}), (\mathbf{W}_Q^{(t)}\boldsymbol{v})^\top(\mathbf{W}_K^{(t)}\boldsymbol{v}') = O(\sqrt{\log(16(2K+K')^2/\delta)}\sigma_0^2 d^{1/2})$$

$$(\mathbf{W}_Q^{(t)}\boldsymbol{a}_{\hat{k}})^\top(\mathbf{W}_K^{(t)}\boldsymbol{a}_{\hat{k}}) \le \underline{C}_9\sigma_0^2 de^{q_V^{-1}\sigma_1^2 d}[\frac{\eta q_V t}{\sigma_1^2 dKM}]^{\frac{M}{16q_V}},$$

$$(\mathbf{W}_Q^{(t)}\boldsymbol{a}_{\hat{k}})^\top(\mathbf{W}_K^{(t)}\boldsymbol{a}_{\hat{k}}) \ge \underline{C}_{10}\frac{\sigma_0^2\sigma_1^2 d^2}{M^2 q_V}e^{2q_V^{-1}K\triangle^2(1-\triangle)^2(\frac{-1}{\frac{\eta q_V t}{2\sigma_1^2 dK}+1}+1)} \tag{93}$$

$$-\frac{\underline{C}_3\eta^2 q_V\sigma_0^2}{\sigma_1^2 M^2 K^2} - 4C_3\sqrt{\log(16(2K+K')^2/\delta)}\sigma_0^2 d^{1/2},$$

*for* $\forall k \in [K], n \in [N], s \in [7], \neg m_{\mathbf{T}_n} \ne m_{\mathbf{T}_n} \in [M], \boldsymbol{u} \in \{\boldsymbol{a}_s\}_{s\ne k\in[K]} \cup \{\boldsymbol{b}_s\}_{s\in[K]} \cup \{\boldsymbol{\nu}_{k'}\}_{k'\in[K']} \cup \{\boldsymbol{\xi}_{n,m}\}_{n\in[N],m\in[M]}$
*and* $\boldsymbol{v} \ne \boldsymbol{v}' \in \{\boldsymbol{b}_s\}_{s\in[K]} \cup \{\boldsymbol{\xi}_{n,m}\}_{n\in[N],m\in[M]}$.

*Furthermore, after* $t \ge T^\star = \Omega(\eta^{-1}q_V^{-1}\sigma_1^2 KMd^2\log(\frac{1}{\epsilon}))$, *it holds that*

$$(\mathbf{W}_Q^{(t)}\boldsymbol{a}_{\hat{k}})^\top(\mathbf{W}_K^{(t)}\boldsymbol{a}_{\hat{k}}) \le O(\sigma_0^2 de^{q_V^{-1}\sigma_1^2 d}(d\log(\frac{1}{\varepsilon}))^{\frac{M}{16q_V}}),$$

$$\pi_{n,m_{\mathbf{T}_n}}^{(T^\star)} \le \Theta(\frac{1}{1+(M-1)e^{-\sigma_0^2 de^{q_V^{-1}\sigma_1^2 d}(d\log(\frac{1}{\varepsilon}))^{\frac{M}{16q_V}}}}), \tag{94}$$

*for* $\forall k \in [K]$.

*Proof.* The proof strategies follow Lemma F.9. $\qquad\square$

# H. Test Loss Convergence

This section also assumes the results in Appendix D.2 all hold with high probability.

*Proof. Proof of Theorem 3.2 and Theorem 3.3.* For the ICL task, given the prompt $\mathbf{T} := [\mathbf{x}_1^{\mathbf{T}}, \mathbf{y}_1^{\mathbf{T}}, \cdots, \mathbf{x}_J^{\mathbf{T}}, \mathbf{y}_J^{\mathbf{T}}, \mathbf{x}_{J+1}^{\mathbf{T}}] \in \mathbb{R}^{d\times(2J+1)}$, we denote $k_{\mathbf{y}_{J+1}^{\mathbf{T}}} = 7k_{\mathbf{T}} + \frac{y_{\mathbf{T},J+1}+1}{2}$ as the dictionary index for $\mathbf{y}_{J+1}^{\mathbf{T}}$, $k_{\mathbf{T}}$ as the task concept index, $y_{\mathbf{T},j}, j \in [J+1]$ as the label indicator of $\mathbf{y}_j^{\mathbf{T}}$. Also we define

$$\pi_{j,\mathbf{T}}^{(t)} = \frac{\exp(\boldsymbol{e}_{J+1}^\top\mathbf{T}^\top(\mathbf{W}_Q^{(t)})^\top\mathbf{W}_K^{(t)}\mathbf{T}\boldsymbol{e}_j)}{\sum_{j\in[2J]}\exp(\boldsymbol{e}_{J+1}^\top\mathbf{T}^\top(\mathbf{W}_Q^{(t)})^\top\mathbf{W}_K^{(t)}\mathbf{T}\boldsymbol{e}_j)}, \forall j \in [2J],$$

If $\mathcal{P}^\star = \mathcal{P}_{\mathbf{T}}$, then

$$L_{\mathcal{P}^\star} = \mathbb{E}_{\mathcal{P}^\star}[\mathbf{1}(k_{\mathbf{y}_{J+1}} \ne \underset{k}{\arg\max}(\frac{\exp(\mathbf{u}_k^\top\mathbf{h}_{\boldsymbol{\theta}})}{\sum_{k\in[7K+K']}\exp(\mathbf{u}_k^\top\mathbf{h}_{\boldsymbol{\theta}})}))]$$

$$= \mathbb{E}_{\mathcal{P}^\star}[\mathbf{1}(k_{\mathbf{y}_{J+1}^{\mathbf{T}}} \ne \underset{k}{\arg\max}\,\mathbf{u}_k^\top\mathbf{h}_{\boldsymbol{\theta}})],$$

$$= \mathbb{E}_{\mathcal{P}^\star}[\mathbf{1}(k_{\mathbf{y}_{J+1}^{\mathbf{T}}} \ne \underset{k}{\arg\max}\,\mathbf{u}_k^\top(\mathbf{x}_{J+1}^{\mathbf{T}} + \frac{\sum_{j\in[2J]}\mathbf{W}_V^{(t)}\mathbf{T}\pi_{j,\mathbf{T}}^{(t)}\boldsymbol{e}_j}{\|\sum_{j\in[2J]}\mathbf{W}_V^{(t)}\mathbf{T}\pi_{j,\mathbf{T}}^{(t)}\boldsymbol{e}_j\|}))]$$

$$= \mathbb{E}_{\mathcal{P}^\star}[\mathbf{1}(k_{\mathbf{y}_{J+1}^{\mathbf{T}}} \ne \underset{k}{\arg\max}\,\mathbf{u}_k^\top(\mathbf{x}_{J+1}^{\mathbf{T}} + \frac{\sum_{j\in[J]}\mathbf{W}_V^{(t)}(\pi_{2j-1,\mathbf{T}}^{(t)}\mathbf{x}_j^{\mathbf{T}} + \pi_{2j,\mathbf{T}}^{(t)}\mathbf{y}_j^{\mathbf{T}})}{\|\sum_{j\in[J]}\mathbf{W}_V^{(t)}(\pi_{2j-1,\mathbf{T}}^{(t)}\mathbf{x}_j^{\mathbf{T}} + \pi_{2j,\mathbf{T}}^{(t)}\mathbf{y}_j^{\mathbf{T}})\|}))] \tag{95}$$

$$= \mathbb{E}_{\mathcal{P}^\star}[\mathbf{1}(7k_{\mathbf{T}} + \frac{y_{\mathbf{T},J+1}+1}{2} \ne \underset{k}{\arg\max}(\mathbf{u}_k)^\top((\sum_{k_{J+1}\in\mathcal{X}_{\mathbf{T},J+1}} x_a \cdot \boldsymbol{a}_{k_{J+1}} + y_{k_{J+1}} \cdot \boldsymbol{b}_{k_{J+1}}) + \boldsymbol{\xi}_{J+1,\mathbf{T}}$$

$$+ \frac{\sum_{j\in[J]}\mathbf{W}_V^{(t)}(\pi_{2j-1,\mathbf{T}}^{(t)}\mathbf{x}_j^{\mathbf{T}} + \pi_{2j,\mathbf{T}}^{(t)}\mathbf{y}_j^{\mathbf{T}})}{\|\sum_{j\in[J]}\mathbf{W}_V^{(t)}(\pi_{2j-1,\mathbf{T}}^{(t)}\mathbf{x}_j^{\mathbf{T}} + \pi_{2j,\mathbf{T}}^{(t)}\mathbf{y}_j^{\mathbf{T}})\|}))]$$

where the equalities are by definition.

By the isotropic probability property in Eq.(14) and the definition of $\mathbf{U}$ in Eq.(16), we have

$$
\begin{aligned}
L_{\mathcal{P}^\star} &= \mathbb{E}_{\mathcal{P}^\star}[\mathbf{1}(1 + \frac{(x_a \boldsymbol{a}_{k_{\mathbf{T}}} + y_{\mathbf{T},J+1}\boldsymbol{b}_{k_{\mathbf{T}}})^\top (\boldsymbol{\xi}_{J+1,\mathbf{T}})}{\sqrt{1.01}} + \frac{(x_a \boldsymbol{a}_{k_{\mathbf{T}}} + y_{\mathbf{T},J+1}\boldsymbol{b}_{k_{\mathbf{T}}})^\top}{\sqrt{1.01}} \frac{\sum_{j\in[J]} \mathbf{W}_V^{(t)}(\pi_{2j-1,\mathbf{T}}^{(t)}\mathbf{x}_j^{\mathbf{T}} + \pi_{2j,\mathbf{T}}^{(t)}\mathbf{y}_j^{\mathbf{T}})}{\|\sum_{j\in[J]} \mathbf{W}_V^{(t)}(\pi_{2j-1,\mathbf{T}}^{(t)}\mathbf{x}_j^{\mathbf{T}} + \pi_{2j,\mathbf{T}}^{(t)}\mathbf{y}_j^{\mathbf{T}})\|} \\
&> \frac{1.1}{\sqrt{2}} + \frac{(\boldsymbol{a}_{k_{\mathbf{T}}} + y_{\mathbf{T},J+1}\boldsymbol{b}_{k_{\mathbf{T}}})^\top (\boldsymbol{\xi}_{J+1,\mathbf{T}})}{\sqrt{2}} + \frac{(\boldsymbol{a}_{k_{\mathbf{T}}} + y_{\mathbf{T},J+1}\boldsymbol{b}_{k_{\mathbf{T}}})^\top}{\sqrt{2}} \frac{\sum_{j\in[J]} \mathbf{W}_V^{(t)}(\pi_{2j-1,\mathbf{T}}^{(t)}\mathbf{x}_j^{\mathbf{T}} + \pi_{2j,\mathbf{T}}^{(t)}\mathbf{y}_j^{\mathbf{T}})}{\|\sum_{j\in[J]} \mathbf{W}_V^{(t)}(\pi_{2j-1,\mathbf{T}}^{(t)}\mathbf{x}_j^{\mathbf{T}} + \pi_{2j,\mathbf{T}}^{(t)}\mathbf{y}_j^{\mathbf{T}})\|})] \\
&\leq \mathbb{E}_{\mathcal{P}^\star}[\mathbf{1}(1.42 + \boldsymbol{\xi}_{J+1,\mathbf{T}}^\top(-0.585\boldsymbol{a}_{k_{\mathbf{T}}} + 0.415 y_{\mathbf{T},J+1}\boldsymbol{b}_{k_{\mathbf{T}}}) \\
&\quad + 1.41(x_a \boldsymbol{a}_{k_{\mathbf{T}}} + y_{\mathbf{T},J+1}\boldsymbol{b}_{k_{\mathbf{T}}})^\top \frac{\sum_{j\in[J]} \mathbf{W}_V^{(t)}(\pi_{2j-1,\mathbf{T}}^{(t)}\mathbf{x}_j^{\mathbf{T}} + \pi_{2j,\mathbf{T}}^{(t)}\mathbf{y}_j^{\mathbf{T}})}{\|\sum_{j\in[J]} \mathbf{W}_V^{(t)}(\pi_{2j-1,\mathbf{T}}^{(t)}\mathbf{x}_j^{\mathbf{T}} + \pi_{2j,\mathbf{T}}^{(t)}\mathbf{y}_j^{\mathbf{T}})\|} \\
&\quad > 1 + (\boldsymbol{a}_{k_{\mathbf{T}}} + y_{\mathbf{T},J+1}\boldsymbol{b}_{k_{\mathbf{T}}})^\top \frac{\sum_{j\in[J]} \mathbf{W}_V^{(t)}(\pi_{2j-1,\mathbf{T}}^{(t)}\mathbf{x}_j^{\mathbf{T}} + \pi_{2j,\mathbf{T}}^{(t)}\mathbf{y}_j^{\mathbf{T}})}{\|\sum_{j\in[J]} \mathbf{W}_V^{(t)}(\pi_{2j-1,\mathbf{T}}^{(t)}\mathbf{x}_j^{\mathbf{T}} + \pi_{2j,\mathbf{T}}^{(t)}\mathbf{y}_j^{\mathbf{T}})\|})] \\
&= \mathbb{E}_{\mathcal{P}^\star}[\mathbf{1}(0.42 + \boldsymbol{\xi}_{J+1,\mathbf{T}}^\top(-0.585\boldsymbol{a}_{k_{\mathbf{T}}} + 0.415 y_{\mathbf{T},J+1}\boldsymbol{b}_{k_{\mathbf{T}}}) \\
&\quad > [0.86\boldsymbol{a}_{k_{\mathbf{T}}} - 0.41 y_{\mathbf{T},J+1}\boldsymbol{b}_{k_{\mathbf{T}}}]^\top \frac{\sum_{j\in[J]} \mathbf{W}_V^{(t)}(\pi_{2j-1,\mathbf{T}}^{(t)}\mathbf{x}_j^{\mathbf{T}} + \pi_{2j,\mathbf{T}}^{(t)}\mathbf{y}_j^{\mathbf{T}})}{\|\sum_{j\in[J]} \mathbf{W}_V^{(t)}(\pi_{2j-1,\mathbf{T}}^{(t)}\mathbf{x}_j^{\mathbf{T}} + \pi_{2j,\mathbf{T}}^{(t)}\mathbf{y}_j^{\mathbf{T}})\|})], \\
&\leq \mathbb{E}_{\mathcal{P}^\star}[\mathbf{1}(0.5 + \boldsymbol{\xi}_{J+1,\mathbf{T}}^\top(-0.7\boldsymbol{a}_{k_{\mathbf{T}}} + 0.5 y_{\mathbf{T},J+1}\boldsymbol{b}_{k_{\mathbf{T}}}) \\
&\quad > [\boldsymbol{a}_{k_{\mathbf{T}}} - 0.5 y_{\mathbf{T},J+1}\boldsymbol{b}_{k_{\mathbf{T}}}]^\top \frac{\sum_{j\in[J]} \mathbf{W}_V^{(t)}(\pi_{2j-1,\mathbf{T}}^{(t)}\mathbf{x}_j^{\mathbf{T}} + \pi_{2j,\mathbf{T}}^{(t)}\mathbf{y}_j^{\mathbf{T}})}{\|\sum_{j\in[J]} \mathbf{W}_V^{(t)}(\pi_{2j-1,\mathbf{T}}^{(t)}\mathbf{x}_j^{\mathbf{T}} + \pi_{2j,\mathbf{T}}^{(t)}\mathbf{y}_j^{\mathbf{T}})\|})],
\end{aligned}
$$
(96)

where the first equality is by the definition of dictionary $\mathbf{U}$ as well as the mutual-orthogonality of $\{\boldsymbol{a}_s\}_{s\neq k\in[K]} \cup \{\boldsymbol{b}_s\}_{s\in[K]} \cup \{\boldsymbol{\nu}_{k'}\}_{k'\in[K']}$, which implies that the main competitor of the label-related dictionary token, namely $\frac{\boldsymbol{a}_{k_{\mathbf{T}}} + y_{\mathbf{T},J+1}\boldsymbol{b}_{k_{\mathbf{T}}}}{\|\boldsymbol{a}_{k_{\mathbf{T}}} + y_{\mathbf{T},J+1}\boldsymbol{b}_{k_{\mathbf{T}}}\|} = \frac{\boldsymbol{a}_{k_{\mathbf{T}}} + y_{\mathbf{T},J+1}\boldsymbol{b}_{k_{\mathbf{T}}}}{\sqrt{2}}$, is the word-related dictionary token, namely $\frac{x_a \boldsymbol{a}_{k_{\mathbf{T}}} + y_{\mathbf{T},J+1}\boldsymbol{b}_{k_{\mathbf{T}}}}{\|x_a \boldsymbol{a}_{k_{\mathbf{T}}} + y_{\mathbf{T},J+1}\boldsymbol{b}_{k_{\mathbf{T}}}\|} = \frac{x_a \boldsymbol{a}_{k_{\mathbf{T}}} + y_{\mathbf{T},J+1}\boldsymbol{b}_{k_{\mathbf{T}}}}{\sqrt{2}}$; the second inequality is by $\sqrt{2} \leq 1.415, \sqrt{2(1.01)} \leq 1.43, \sqrt{2/(1.01)} \leq 1.41, \sqrt{1.01} > 1, 1.1\sqrt{1.01}, \sqrt{1.01} < 1$; the last inequality and equality are by direct calculations.

Then denotes $\pi_{\boldsymbol{a}_{k_{\mathbf{T}}}}^{(t)} := \sum_{j\in[J]}(0.1\pi_{2j-1,\mathbf{T}}^{(t)} + \pi_{2j,\mathbf{T}}^{(t)})$ as the accumulated softmax weights assigned on $\boldsymbol{a}_{k_{\mathbf{T}}}$ in $\mathbf{T}$, $\mathcal{X}_{\mathbf{T}} = \bigcup_{j\in[J]} \mathcal{X}_{\mathbf{T},j}$, $\pi_{\boldsymbol{a}_{k_l}}^{(t)}$ as the accumulated softmax weights assigned on those current prompts' co-task-irrelevant task vectors $\boldsymbol{a}_{k_l}$ in $\mathbf{T}$, and $i_{\mathbf{T}} := \sum_{j\in[J]}((\sum_{k_j\in\mathcal{X}_{\mathbf{T},j}} y_{k_j} \cdot \boldsymbol{b}_{k_j}) + y_{k_{\mathbf{T}},j} \cdot \boldsymbol{b}_{k_{\mathbf{T}}})$, we have

$$
\sum_{j\in[J]}(\pi_{2j-1,\mathbf{T}}^{(t)}\mathbf{x}_j^{\mathbf{T}} + \pi_{2j,\mathbf{T}}^{(t)}\mathbf{y}_j^{\mathbf{T}}) = \pi_{\boldsymbol{a}_{k_{\mathbf{T}}}}^{(t)}\boldsymbol{a}_{k_{\mathbf{T}}} + (\sum_{k_l\in\mathcal{X}_{\mathbf{T}}\setminus\{k_{\mathbf{T}}\}} \pi_{\boldsymbol{a}_{k_l}}^{(t)}\boldsymbol{a}_{k_l}) + i_{\mathbf{T}} + \boldsymbol{\xi}_T.
$$
(97)

Here, $\boldsymbol{\xi}_T := \sum_{j\in[J]}(\pi_{2j-1,\mathbf{T}}^{(t)}\boldsymbol{\xi}_{j,\mathbf{x}} + \pi_{2j,\mathbf{T}}^{(t)}\boldsymbol{\xi}_{j,\mathbf{y}}) \sim \mathcal{N}(\mathbf{0}, \sigma_p^{\star 2}\sum_{j\in[J]}\left((\pi_{2j-1,\mathbf{T}}^{(t)})^2 + (\pi_{2j,\mathbf{T}}^{(t)})^2\right)\mathbb{I})$. Let $\sigma_{\mathbf{T}}^2 := \sigma_p^{\star 2}\sum_{j\in[J]}\left((\pi_{2j-1,\mathbf{T}}^{(t)})^2 + (\pi_{2j,\mathbf{T}}^{(t)})^2\right)$. It is straightforward to observe that the upper and lower bounds of $\sigma_{\mathbf{T}}^2$ are 1 and $\frac{1}{2J}$, respectively, corresponding to the cases of extreme and uniform weights.

Consider the training distribution is $\mathcal{P}_{\mathrm{QA}}$, then we have

$$
\begin{aligned}
L_{\mathcal{P}^\star} &\leq \mathbb{E}_{\mathcal{P}^\star}[\mathbf{1}(\boldsymbol{\xi}_{J+1,\mathbf{T}}^\top(-0.7\boldsymbol{a}_{k_\mathbf{T}} + 0.5y_{\mathbf{T},J+1}\boldsymbol{b}_{k_\mathbf{T}}) \\
&\qquad > [\boldsymbol{a}_{k_\mathbf{T}} - 0.5y_{\mathbf{T},J+1}\boldsymbol{b}_{k_\mathbf{T}}]^\top \frac{\mathbf{W}_V^{(t)}(\pi_{\boldsymbol{a}_{k_\mathbf{T}}}^{(t)}\boldsymbol{a}_{k_\mathbf{T}} + (\sum_{k_l \in \mathcal{X}_\mathbf{T}\setminus\{k_\mathbf{T}\}} \pi_{\boldsymbol{a}_{k_l}}^{(t)}\boldsymbol{a}_{k_l}) + i_\mathbf{T} + \boldsymbol{\xi}_T)}{\|\mathbf{W}_V^{(t)}(\pi_{\boldsymbol{a}_{k_\mathbf{T}}}^{(t)}\boldsymbol{a}_{k_\mathbf{T}} + (\sum_{k_l \in \mathcal{X}_\mathbf{T}\setminus\{k_\mathbf{T}\}} \pi_{\boldsymbol{a}_{k_l}}^{(t)}\boldsymbol{a}_{k_l}) + i_\mathbf{T} + \boldsymbol{\xi}_T)\|}) - 0.5], \\
&\leq \mathbb{E}_{\mathcal{P}^\star}[\mathbf{1}(\boldsymbol{\xi}_{J+1,\mathbf{T}}^\top(-0.7\boldsymbol{a}_{k_\mathbf{T}} + 0.5y_{\mathbf{T},J+1}\boldsymbol{b}_{k_\mathbf{T}}) \\
&\qquad > \frac{\pi_{\boldsymbol{a}_{k_\mathbf{T}}}^{(t)}\boldsymbol{a}_{k_\mathbf{T}}^\top\mathbf{W}_V^{(t)}\boldsymbol{a}_{k_\mathbf{T}} - |\boldsymbol{a}_{k_\mathbf{T}}^\top\mathbf{W}_V^{(t)}\boldsymbol{\xi}_T|}{\Theta(\pi_{\boldsymbol{a}_{k_\mathbf{T}}}^{(t)}\|\mathbf{W}_V^{(t)}\boldsymbol{a}_{k_\mathbf{T}}\| + \sum_{k_l \in \mathcal{X}_\mathbf{T}\setminus\{k_\mathbf{T}\}} \pi_{\boldsymbol{a}_{k_l}}^{(t)}\|\mathbf{W}_V^{(t)}\boldsymbol{a}_{k_l}\|) + \|\mathbf{W}_V^{(t)}\boldsymbol{\xi}_T\|} - 0.5] \\
&\leq \mathbb{E}_{\mathcal{P}^\star}[\mathbf{1}(\boldsymbol{\xi}_{J+1,\mathbf{T}}^\top(-0.7\boldsymbol{a}_{k_\mathbf{T}} + 0.5y_{\mathbf{T},J+1}\boldsymbol{b}_{k_\mathbf{T}}) \\
&\qquad > \Theta(\frac{\pi_{\boldsymbol{a}_{k_\mathbf{T}}}^{(t)}\boldsymbol{a}_{k_\mathbf{T}}^\top\mathbf{W}_V^{(t)}\boldsymbol{a}_{k_\mathbf{T}} - |\boldsymbol{a}_{k_\mathbf{T}}^\top\mathbf{W}_V^{(t)}\boldsymbol{\xi}_T|}{(\pi_{\boldsymbol{a}_{k_\mathbf{T}}}^{(t)} + \sum_{k_l \in \mathcal{X}_\mathbf{T}\setminus\{k_\mathbf{T}\}} \pi_{\boldsymbol{a}_{k_l}}^{(t)})\Theta(\|\mathbf{W}_V^{(t)}\boldsymbol{a}_{k_\mathbf{T}}\|) + \|\mathbf{W}_V^{(t)}\boldsymbol{\xi}_T\|}) - 0.5] \\
&\leq \mathbb{E}_{\mathcal{P}^\star}[\mathbf{1}(\boldsymbol{\xi}_{J+1,\mathbf{T}}^\top(-0.7\boldsymbol{a}_{k_\mathbf{T}} + 0.5y_{\mathbf{T},J+1}\boldsymbol{b}_{k_\mathbf{T}}) \\
&\qquad > \frac{\pi_{\boldsymbol{a}_{k_\mathbf{T}}}^{(t)}\boldsymbol{a}_{k_\mathbf{T}}^\top\mathbf{W}_V^{(t)}\boldsymbol{a}_{k_\mathbf{T}} - |\boldsymbol{a}_{k_\mathbf{T}}^\top\mathbf{W}_V^{(t)}\boldsymbol{\xi}_T|}{(\pi_{\boldsymbol{a}_{k_\mathbf{T}}}^{(t)} + \sum_{k_l \in \mathcal{X}_\mathbf{T}\setminus\{k_\mathbf{T}\}} \pi_{\boldsymbol{a}_{k_l}}^{(t)})(\Theta(\|\mathbf{W}_V^{(0)}\boldsymbol{a}_{k_\mathbf{T}}\|) + \boldsymbol{a}_{k_\mathbf{T}}^\top\mathbf{W}_V^{(t)}\boldsymbol{a}_{k_\mathbf{T}}) + \|\mathbf{W}_V^{(t)}\boldsymbol{\xi}_T\|} - 0.5] \\
&\leq \mathbb{E}_{\mathcal{P}^\star}[\mathbf{1}(\boldsymbol{\xi}_{J+1,\mathbf{T}}^\top(-0.7\boldsymbol{a}_{k_\mathbf{T}} + 0.5y_{\mathbf{T},J+1}\boldsymbol{b}_{k_\mathbf{T}}) \\
&\qquad > \frac{1 - \frac{|\boldsymbol{a}_{k_\mathbf{T}}^\top\mathbf{W}_V^{(t)}\boldsymbol{\xi}_T|}{\pi_{\boldsymbol{a}_{k_\mathbf{T}}}^{(t)}\boldsymbol{a}_{k_\mathbf{T}}^\top\mathbf{W}_V^{(t)}\boldsymbol{a}_{k_\mathbf{T}}}}{(1 + \frac{\sum_{k_l \in \mathcal{X}_\mathbf{T}\setminus\{k_\mathbf{T}\}} \pi_{\boldsymbol{a}_{k_l}}^{(t)}}{\pi_{\boldsymbol{a}_{k_\mathbf{T}}}^{(t)}})(\Theta(\frac{\|\mathbf{W}_V^{(0)}\boldsymbol{a}_{k_\mathbf{T}}\|}{\boldsymbol{a}_{k_\mathbf{T}}^\top\mathbf{W}_V^{(t)}\boldsymbol{a}_{k_\mathbf{T}}}) + 1) + \frac{\|\mathbf{W}_V^{(t)}\boldsymbol{\xi}_T\|}{\pi_{\boldsymbol{a}_{k_\mathbf{T}}}^{(t)}\boldsymbol{a}_{k_\mathbf{T}}^\top\mathbf{W}_V^{(t)}\boldsymbol{a}_{k_\mathbf{T}}}} - 0.5]
\end{aligned}
\tag{98}
$$

Here, the first inequality is by Eq.(96) and (97); the second inequality is by triangle inequality of norm as well as Lemma F.8 denoting the dominant growing of $\boldsymbol{a}_{k_\mathbf{T}}^\top\mathbf{W}_V^{(t)}\boldsymbol{a}_{k_\mathbf{T}}$ compared to other projections; the third inequality is by the balanced growing of all $\|\mathbf{W}_V^{(t)}\boldsymbol{a}_k\|$ denoted in Lemma F.8; the forth inequality is also by traingle inequality as well as Lemma F.8 showing the main contributors of norm. Subsequently, by Gaussian tail bounds as well as $\sigma_p^\star = O((K\log(\frac{2J+1}{\varepsilon}))^{-\frac{1}{2}})$ we see that

$$
\mathbb{P}_{\mathcal{P}^\star}(\boldsymbol{\xi}_{J+1,\mathbf{T}}^\top(-0.7\boldsymbol{a}_{k_\mathbf{T}} + 0.5y_{\mathbf{T},J+1}\boldsymbol{b}_{k_\mathbf{T}}) \geq 1.5\sigma_p^\star\sqrt{\log(\frac{4}{\varepsilon})}) \leq \varepsilon/4.
\tag{99}
$$

Additionally, recall that $\pi_{\boldsymbol{a}_{k_\mathbf{T}}}^{(t)} = \sum_{j\in[J]}(0.1\pi_{2j-1,\mathbf{T}}^{(t)} + \pi_{2j,\mathbf{T}}^{(t)})$, and $B_i \sim \mathrm{Ber}(K^{-1})$. We first analyze the worst-case scenario, where the number of task concepts outside the prompt's co-task concept may exceed $1.1J$ (the number of $\boldsymbol{a}_{k_\mathbf{T}}$ in the prompt). Given $J = \Omega\left(\log\left(\frac{1}{\varepsilon}\right)/(2\log(K))\right)$, the probability of this event is bounded by $\varepsilon/8$. By Lemma F.7 and Lemma F.9, which characterize the progressive learning of task vectors and the limited learning of non-task vectors, we have $\sigma_p^\star = O\left(\left(K\log\left(\frac{2J+1}{\varepsilon}\right)\right)^{-\frac{1}{2}}\right)$. This implies that the probability of noise vectors exhibiting significant components along the axes of any $\boldsymbol{a}_{k_l}$ or $-\boldsymbol{a}_{k_\mathbf{T}}$ is constrained to less than $\varepsilon/8$. It follows that:

$$
\mathbb{P}_{\mathcal{P}^\star}\left(\frac{\sum_{k_l \in \mathcal{X}_\mathbf{T}\setminus\{k_\mathbf{T}\}} \pi_{\boldsymbol{a}_{k_l}}^{(t)}}{\pi_{\boldsymbol{a}_{k_\mathbf{T}}}^{(t)}} = o(0.1)\right) \geq 1 - \varepsilon/4.
\tag{100}
$$

Besides, by Eq. (99), a sufficient condition for the event in the last inequality of Eq. (98) is that the right-hand side

$$
\frac{1 - \frac{|\boldsymbol{a}_{k_\mathbf{T}}^\top\mathbf{W}_V^{(t)}\boldsymbol{\xi}_T|}{\pi_{\boldsymbol{a}_{k_\mathbf{T}}}^{(t)}\boldsymbol{a}_{k_\mathbf{T}}^\top\mathbf{W}_V^{(t)}\boldsymbol{a}_{k_\mathbf{T}}}}{1.1 + 1.1\Theta\left(\frac{\|\mathbf{W}_V^{(0)}\boldsymbol{a}_{k_\mathbf{T}}\|}{\boldsymbol{a}_{k_\mathbf{T}}^\top\mathbf{W}_V^{(t)}\boldsymbol{a}_{k_\mathbf{T}}}\right) + \frac{\|\mathbf{W}_V^{(t)}\boldsymbol{\xi}_T\|}{\pi_{\boldsymbol{a}_{k_\mathbf{T}}}^{(t)}\boldsymbol{a}_{k_\mathbf{T}}^\top\mathbf{W}_V^{(t)}\boldsymbol{a}_{k_\mathbf{T}}}} < 0.5 + 1.5\sigma_p^\star\sqrt{\log\left(\frac{4}{\varepsilon}\right)}.
\tag{101}
$$

This requires the denominator to be no less than $1 - \frac{|a_{k_\mathbf{T}}^\top \mathbf{W}_V^{(t)} \xi_T|}{\pi_{a_{k_\mathbf{T}}}^{(t)} a_{k_\mathbf{T}}^\top \mathbf{W}_V^{(t)} a_{k_\mathbf{T}}} / \left(0.5 + 1.5\sigma_p^\star \sqrt{\log\left(\frac{4}{\varepsilon}\right)}\right)$. By Lemma F.8, as well

as $\sigma_\mathbf{T}^2 \leq \sigma_p^\star$ we have $\frac{|a_{k_\mathbf{T}}^\top \mathbf{W}_V^{(t)} \xi_T|}{\pi_{a_{k_\mathbf{T}}}^{(t)} a_{k_\mathbf{T}}^\top \mathbf{W}_V^{(t)} a_{k_\mathbf{T}}} \leq \frac{\|\mathbf{W}_V^{(t)} \xi_T\|}{a_{k_\mathbf{T}}^\top \mathbf{W}_V^{(t)} a_{k_\mathbf{T}}} \leq \frac{\|\mathbf{W}_V^{(t)}\|_F \|\xi_T\|}{a_{k_\mathbf{T}}^\top \mathbf{W}_V^{(t)} a_{k_\mathbf{T}}} \sim \mathcal{N}(\mathbf{0}, \Theta(\sigma_p^{\star 2} K \mathbf{I}_{d\times d}))$. Again, by Gaussian tail

bounds as well as $\sigma_p^{\star\star} = O((K\log(\frac{2J+1}{\varepsilon}))^{-\frac{1}{2}})$ we have

$$\mathbb{P}_{\mathcal{P}^\star}\left(\frac{\|\mathbf{W}_V^{(t)} \xi_T\|}{a_{k_\mathbf{T}}^\top \mathbf{W}_V^{(t)} a_{k_\mathbf{T}}} \leq 2\sigma_p^\star \sqrt{K\log(\frac{4}{\varepsilon})} < \frac{1}{2}\frac{0.9 - 7.3\sigma_p^\star \sqrt{K\log(\frac{4}{\varepsilon})}}{1 + 3\sigma_p^\star \sqrt{K\log(\frac{4}{\varepsilon})}}\right) \leq \varepsilon/4. \tag{102}$$

Note that $\frac{1 - 2\sigma_p^\star \sqrt{K\log(\frac{4}{\varepsilon})}}{\left(0.5 + 1.5\sigma_p^\star \sqrt{\log(\frac{4}{\varepsilon})}\right)} - 1.1 = \frac{0.9 - 3.3\sigma_p^\star \sqrt{\log(\frac{4}{\varepsilon})} - 4\sigma_p^\star \sqrt{K\log(\frac{4}{\varepsilon})}}{1 + 3\sigma_p^\star \sqrt{\log(\frac{4}{\varepsilon})}} \geq \frac{0.9 - 7.3\sigma_p^\star \sqrt{K\log(\frac{4}{\varepsilon})}}{1 + 3\sigma_p^\star \sqrt{K\log(\frac{4}{\varepsilon})}}$. Therefore, by Eq.(102)

we bound the $\frac{|a_{k_\mathbf{T}}^\top \mathbf{W}_V^{(t)} \xi_T|}{\pi_{a_{k_\mathbf{T}}}^{(t)} a_{k_\mathbf{T}}^\top \mathbf{W}_V^{(t)} a_{k_\mathbf{T}}}$ in the numerator to be less than $2\sigma_p^\star \sqrt{K\log(\frac{4}{\varepsilon})}$ and the $\frac{\|\mathbf{W}_V^{(t)} \xi_T\|}{a_{k_\mathbf{T}}^\top \mathbf{W}_V^{(t)} a_{k_\mathbf{T}}}$ in the domina-

tor to be less than $\frac{1}{2}\left(\frac{1 - 2\sigma_p^\star \sqrt{K\log(\frac{4}{\varepsilon})}}{\left(0.5 + 1.5\sigma_p^\star \sqrt{\log(\frac{4}{\varepsilon})}\right)} - 1.1\right)$. Therefore, our last job could be showing $1.1\Theta\left(\frac{\|\mathbf{W}_V^{(0)} a_{k_\mathbf{T}}\|}{a_{k_\mathbf{T}}^\top \mathbf{W}_V^{(t)} a_{k_\mathbf{T}}}\right) \leq$

$\frac{1}{2}\left(\frac{1 - 2\sigma_p^\star \sqrt{K\log(\frac{4}{\varepsilon})}}{\left(0.5 + 1.5\sigma_p^\star \sqrt{\log(\frac{4}{\varepsilon})}\right)} - 1.1\right)$, which can then ensure Eq.(101) holds with probability less than $\varepsilon$.

By Lemma F.8, $\sigma_p^\star = O((K\log(\frac{2J+1}{\varepsilon}))^{-\frac{1}{2}})$ and the Taylor expansion of $g(x) = \frac{1+3x}{0.9-7.3x} = \frac{10}{9} + \frac{1000}{81}x + O(x)$ at zero, we conclude that

$$a_{k_\mathbf{T}}^\top \mathbf{W}_V^{(t)} a_{k_\mathbf{T}} = \Omega(3(1 + 15\sigma_p^{\star 2} \sqrt{\log(\frac{4}{\varepsilon})})\|\mathbf{W}_V^{(0)} a_{k_\mathbf{T}}\|)$$

$$\geq 2.2(\frac{10}{9} + \frac{1000}{81}(1.5\sigma_p^\star \sqrt{\log(\frac{4}{\varepsilon})}))\|\mathbf{W}_V^{(0)} a_{k_\mathbf{T}}\| \tag{103}$$

$$\Rightarrow (1.1)\frac{\|\mathbf{W}_V^{(0)} a_{k_\mathbf{T}}\|}{a_{k_\mathbf{T}}^\top \mathbf{W}_V^{(t)} a_{k_\mathbf{T}}} \leq \frac{1}{2}\left(\frac{1 - 2\sigma_p^\star \sqrt{K\log(\frac{4}{\varepsilon})}}{\left(0.5 + 1.5\sigma_p^\star \sqrt{\log\left(\frac{4}{\varepsilon}\right)}\right)} - 1.1\right).$$

By Eq.(98), (99), (100), (102), (103), by union bound we have

$$L_{\mathcal{P}^\star} \leq \varepsilon.$$

Similarly, for QA Sentence distribution $\mathbf{S} \sim \mathcal{P}_{\mathrm{QA}}$ as well as QA Sentence based Prompt Dsitribution $\mathbf{T}_{\mathrm{QA}} \sim \mathcal{P}_{\mathrm{QA}}^\mathbf{T}$, the population loss convergence could be shown with the same strategy

$$L_{\mathcal{P}_{\mathrm{QA}}}, \; L_{\mathcal{P}_{\mathrm{QA}}^\mathbf{T}} \leq \varepsilon.$$

Separately, when the training distribution is on $\mathcal{P}_\mathbf{T}$ or $\mathcal{P}_{\mathrm{QA}}^\mathbf{T}$, for prompts whose task concepts are $k_y$, with probability $1/2 >> \delta$, $\mathbf{T}$ has more words with $b_{k_y}$ than $b_{-k_y}$. This would lead $b_{k_\mathbf{T}}^\top \mathbf{W}_V^{(t)} i_\mathbf{T}$ ignorable by Lemma G.3. In addition, $\|i_\mathbf{T}\|$'s contribution to $\|\sum_{j\in[J]} \mathbf{W}_V^{(t)}(\pi_{2j-1,\mathbf{T}}^{(t)} \mathbf{x}_j^\mathbf{T} + \pi_{2j,\mathbf{T}}^{(t)} \mathbf{y}_j^\mathbf{T})\|$ is also ignorable by Lemma G.3. Therefore, it's safe to draw the conclusion that with probability at least $1 - \delta$,

$$L_{\mathcal{P}_\mathbf{T}}(\theta^{(t)}) = \Theta(1).$$

Through similar approaches we have $L_{\mathcal{P}_{\mathrm{QA}}^\mathbf{T}}(\theta^{(t)}) = \Theta(1)$.

*Proof.* **Proof of Proposition 3.4**. The proof follows proof of Theorem 3.2 and Theorem 3.3, grounded on the learned knowledge on task vector $a_k, \forall k \in [K]$. We here omit the proofs for brevity. $\square$

$\square$

