# OpenReview forum: "Provable In-Context Vector Arithmetic via Retrieving Task Concepts"
_ICML.cc/2025/Conference — ICML 2025 poster_

### Official Review · Reviewer_UbXD · 2025-03-10

**Overall Recommendation:** 2

**Summary:**

This work analyzes the optimization dynamics of transformer networks trained on in-context learning tasks via Gradient-descent on an L2-regularized cross-entropy loss. The analysis relies on a simple specific data-generating process which provides a way to formalize the notion of task concept vector. The results shows that the internal representation of a Transformer can recover the the task concept vector. Experiments validating the theory are performed.

**Claims And Evidence:**

From what I understood of the paper, I believe the claims are supported.

**Essential References Not Discussed:**

The authors might be interested in https://arxiv.org/abs/2410.23501.

**Experimental Designs Or Analyses:**

No.

**Methods And Evaluation Criteria:**

N/A

**Other Comments Or Suggestions:**

-

**Other Strengths And Weaknesses:**

Strengths:

- I think the theoretical questions tackled here are interesting.
- The analysis seems non-trivial. That being said, I do not think I am qualified to judge its novelty and relevance.

Weakness:

- Clarity is a big issue (see comments about theory). I provide some examples below:
    - Eq (1) is weird… $a^f_\theta$ is a deterministic function of T? Why is there a distribution p(a | T)? Same question for b… Also, are we saying a + b = f here? If so it should be written explicitly.
    - Line 74 (left): What does the acronym MLM stand for?
    - w^i_k is introduced without explaining what i is. Is there multiple low-level binary concepts for a given high-level concept? Is this what i is supposed to mean?
    - Line 161 (right): Notation is unnecessarily heavy, see for example $a_{k_{l_y}}$ which has too many nested indices. This makes everything difficult to parse…
    - Line 156 (right): I don’t understand, an expression is given for the expected label of the query but that expression contains a random element $y_{k_T,J+1}$. How is that possible? Also, I computed the expected value of y_j with j<=J and I don’t get the same expectation as for j = J+1. That’s a bit weird no? I was expecting the distribution of y given x to be the same across all examples j in the task T. Looking at (12) in the appendix didn’t help me understand.
    - Line 189: The number $M$ appears for the first time, without explanations.
    - Line 201: This illustrative example is unclear.
    - Line 181 (right): Something’s off with that equation. You are multiplying a vector with a vector, and it yields a vector. I think the softmax should be indexed by $\ell$.
    - I don’t understand how you obtain Eq. (2)
    - Eq. 3, what’s K’? Where does this "7’’ come from? What is the matrix U? It seems many important details are left in the appendix. The contribution should be understandable from the main text. Another example is the algorithm shown only in Appendix C.
- As mentioned above, many important details necessary to understand the contribution seem to be hidden in the Appendix.
- I not sure the data-generating process is realistic or interesting. I understand that this kind of analysis is challenging without simplifying assumptions, but overall I believe the data-generating process could be motivated further. That being said, it is possible I did not understand it properly due to clarity issues.

**Questions For Authors:**

See clarity issues above.

**Relation To Broader Scientific Literature:**

I am not well versed in this literature.

**Theoretical Claims:**

I read the theoretical statements contained in the main paper. I list some of my confusions which may be due to the fact that I am not very familiar with learning theory and bounds holding with high probability and so maybe this is commonly used jargon.

In Theorem 3.2, at first I was confused since the statement does not explicitly mention which optimizer is used on which loss, but this is mentioned earlier in the text (not in an assumption environment nor anything like that). I think it should be more explicit in the theorem statement.

I was  very confused by the use of the $O$ and $\Omega$ notation in Theorem 3.2 and Prop 3.3. It's unclear to me what is varying/increasing here. Is it 1/epsilon? K? t? The representation dimensionality? This should be more explicit.

Also in Theorem 3.2, I was confused by a statement of the form "there exists t = O(g(epsilon, L, M, ...))". Are we saying that there exists a sequence t's that belongs to O(...) ? And again, what's varying?

These points are examples of the general lack of clarity in this work.

---

> ### Author Rebuttal · Authors · 2025-03-27
>
> Thank you for acknowledging our theoretical contribution and recognizing our analysis non-trivial.
>
> **Q**:Confusion over Theorem 1
>
> - We'd make it in the theorem statement explicitly by referring Algorithm 1.
> - As we claim for any ε>0, the varying item here is ε, with other parameters satisfying Condition 3.1 and the conditions in our statement. The existence of $t=O(g(ε,...))$ means for every (varying) ε, there exist a corresponding t upper bounded by $C\log(1/ε)·(\eta^{-1}q_{V}^{-1}σ_{1}^{2}d^{2}KM)$ for some constant $C$.
>
> **Q**:Room for presentation
>
> We highly appreciate the feedback:
>
> - Eq.(1): We’d note $f = a^f + b$ as the label vector representation. Per [1], one task $f$ corresponds to one task vector $a^f$ in latent space, yielding $f(x_{\text{query}})$ when added to $b(x_{\text{query}})$, an encoded residual stream independent of demo-pairs in prompt $T$. Model $\theta$ infers $a_θ^f$ from $T$, with $p_θ(a|T)$ reflecting model's confidence in recognizing $a$ as the task vector.
> - MLM: MLM loss refers Masked Language Model loss.
> - $w^i_k$: It should be replaced by $w_k$ with $i$ as a typo.
> - Line 189-201: M represent question length excluding query **x**. 201 explains our modeling intuition—QA includes irrelevant tokens, task vector, word, label—similar to [2] and modeled after [3].
> - Line 156-161 & Eq.(2): Every **x_l** - **y_l** pair in the prompt $T$ share a co-task $k_{T}\in[K]$ but each pair can have its own low-level real label $y_{k_T,l}$ regarding $k_T$. The label vector of the prompt *only* depends on $k_T$ and the $y_{k_T,J+1}$ in query **x_{J+1}**. An illustrative example showing this intuition is $T$=[Japan,Tokyo,China,Beijing,France], **y_3**=Paris. The expected **y_3** only depends on the co-task “capital” and the semantic of France (not Japan or China). Under our modeling, **x_{J+1}** ≈$0.1a_{k_{T}}+y_{k_T,l}b_{k_{T}}$, and **y_{J+1}** ≈$a_{k_{T}}+y_{k_T, l}b_{k_{T}}$, modeled after Figure 1[1]. We'd avoid nested indices by let $l_x=l_y=l,l\in[J]$.
> - Line 181: We’d replace $W_KT$ by $W_KT_l$.
> - Eq.(3): K’ is the number of irrelevant token in U, U is the dictionary matrix for cross-entropy training akin to [4] described in our 'Training Setups' in Section 2.2, 7 denotes K sets of {a±b, 0.1a±b, ±b, a} in eq.(14).
>
> While key details, such as Algorithm 1's procedure in Section 2.2, are included in the main text, we understand the importance of making our contributions more accessible—we'll add a notation table in a new appendix to summarize key definitions and intuitions for easier reference.
>
> **Q**:arXiv2410.23501
>
> Thanks for sharing the paper—aligns with [5-6], backing up our data modeling, and we’ll cite it accordingly.
>
> **Q**:Merit of the problem & Real-world impact
>
> We'd like to emphasize that our models, grounded in empirical observations, offer significant theoretical merit:
>
> - Modeled after [1] per Figure 1 as well as the concept latent geometry [5-6], our models are indeed sparse coding approaches suitable for capturing language polysemy [7-8] (see our 1st and 3rd responses to Reviewer mnid)—we successfully show the **OOD edge** of transformer over word2vec—addresses *Question 5.1.4* in [9] in theory.
> - Indeed, our empirically-grounded theoretical approach is common in theory (see our 1st response to Reviewer R7sM)—akin to [10-11] analyzing on feature-noise vision data, which is justified by the properties of ResNet’s latent space [10]. While [11] states the harmful overfitting over *vision* data is due to noise’s memorization, our harmful overfitting over ICL data is by falsely memorize co-occurrence of low-level features—akin to [12]'s result—feature co-occurrence biases gradients toward memorization.
> - We offer the **first** optimization theory for a *softmax-layernorm-residual* realistic transformer on empirical-motivated QA data trained by cross-entropy loss for factual recall ICL per [1], going beyond prior theory with idealized assumption like residual-free [13], QA-combined attention [14] with unrealistic loss functions (square or hinge loss) and oversimplified data. High nonlinearities of our problem induces complex gradient, for which we introduce *six continuous flows* in Appendix D.1 to capture its property in different phases, contributing to theory community as a method for treating complex optimization dynamics beyond [15].
>
> **Summary**: We thank the reviewer for acknowledging the theoretical questions tackled here interesting and our analysis non-trivial. We highly value the feedback, and should the reviewer have any further advice or wish to discuss any point further, we would be more than delighted to continue our productive exchange. Once again, we deeply appreciate the reviewer’s valuable time and comments.
>
> *Reference (arXiv identifier)*
>
> [1]2305.16130
> [2]2412.06538
> [3]2309.14316
> [4]2305.16380
> [5]2406.01506
> [6]2403.03867
> [7]1601.03764
> [8]2105.15134
> [9]2405.01964
> [10]2012.09816
> [11]2202.05928
> [12]2410.09605
> [13]2402.15607
> [14]2402.01258
> [15]2310.01975

---

### Official Review · Reviewer_mnid · 2025-03-13

**Overall Recommendation:** 4

**Summary:**

The authors study task vectors in the context of single-token factual-recall ICL tasks. In this context, they show that training on QA data enables learning a task vector which can effectively solve ICL problems for Word-Label and QA tasks. They additionally analyze this phenomenon theoretically.

**Claims And Evidence:**

The theoretical claims are well-supported in the paper, albeit in a limited context and not for real-world LLMs, and the empirical claims have some support.

**Essential References Not Discussed:**

None

**Experimental Designs Or Analyses:**

The experiments seem sound, but limited in scope.

**Methods And Evaluation Criteria:**

The methods, e.g. evaluating the test loss under different training settings, make sense but are difficult to understand concretely. It is unclear if the QA and Word-Label tasks in this context are natural language, or a simplification. It appears to be the latter, but this makes the example provided in the QA Sentence Distribution paragraph a bit confusing. The paper would benefit from showing some examples of the exact task sequences in each case.

**Other Comments Or Suggestions:**

The authors discuss the geometric relationship between words and labels, and similar discuss the relationship between task vectors and task-related words in the context of pretrained LLMs. I would like to see some analysis of whether the same relationships exist in their simplified settings. When training on the QA-ICL dataset, do you notice similar relationships emerge?

**Other Strengths And Weaknesses:**

Strengths:

* The authors provide a controlled and precise setting to study a model of task vectors in ICL
* The theoretical conclusions are well-supported and there is some additional empirical evidence provided

Weaknesses:

* The study of geometric relationships seems to disappear when discussing the empirical results with the trained models
* Broader impact may be limited, as it is unclear how well this transfers to the motivating settings of task vectors in pretrained LLMs

**Questions For Authors:**

Is there a way to vary and study the complexity of QA and Word-Label tasks? In the natural language settings you draw from for motivation, we have a clearer sense of complexity for different QA pairs and, similarly, a clearer sense of concept hierarchy.

**Relation To Broader Scientific Literature:**

The paper relates to prior work identifying task vectors in the context of ICL, showing that demonstrations allow LLMs to internally construct a vector that expresses the task concept, as well as theoretical study of ICL.

**Theoretical Claims:**

I checked the proofs of Linear Growth and Decelerating Growth and did not notice any glaring errors.

---

> ### Author Rebuttal · Authors · 2025-03-28
>
> Thank you for acknowledging our theory as empirically-supported and sound. We appreciate your professional review and address your concerns below.
>
> **Q**:Are QA and Word-Label tasks natural language or simplifications? The example in QA Sentence Distribution is confusing. The paper would benefit from showing some examples of the exact task sequences in each case.
>
> **A**:In our theoretical context, the QA and Word-Label ICL tasks are empirically-motivated abstracted simplifications. The example in lines 201-202 backs our modeling intuition—QA comprises irrelevant tokens, a task vector, a word, and a label—akin to [1]’s model (in its Figure 2) and inspired by [2]. Per your advice, we’d add illustrative examples to further clarify our ICL data's intuition: e.g., $T_1$=[Japan,Sakura,France,Rooster,China] with co-task “National Symbol” yields **y_3**=Panda, while $T_2$=[Japan,Sakura,France,Iris,China] with co-task “National Flower” yields **y_3**=Peony. These highlight the multi-task nature of our multi-concept modeling (e.g., “**x,y**=Japan,Sakura” fits ≥2 tasks), reflecting this sparse coding-type model suited for capturing *language polysemy*[3-4], more realistic than prior ICL theories over unrealistic data [5-6].
>
> **Q**:The study of geometric relationships of the trained models seems to disappear
>
> **A**:We would like to remark:
> - **LLM Geometry (Figure 1)**: Task vectors align differently with words and labels in trained LLM prerequisite layers, forming our data modeling for optimization theory, grounded in LLM concept geometry [7-8].
> - **Trained Model Geometry (Theorem 3.2 and the discussions below, Figures 2-4)**: the first layer’s output $h_{\theta,0}$, formed by $W_V,W_Q,W_K$ before adding residual vector, varies by training data. QA training aligns $h_{\theta,0}$ with the true task vector $a_{k^{\star}}$, ensuring correct prediction when added to any task-specific **x** via residuals. However, when trained via ICL-type data, the $W_V,W_Q,W_K$ would *non-negligibly* memorize some low-level features—akin to [9]’s results. That is, $h_{\theta,0}$ aligns with both $a_{k^{\star}}$ and low-level $±b_{k^{\star}}$ to some non-negligible extent, leading to constant test error w.h.p.. Detailed descriptions would be added to the figures' captions.
>
> **Q**:Broader impact may be limited, as it is unclear how well this transfers to the motivating settings of task vectors in pretrained LLMs
>
> **A**:We would like to remark our impact on elucidating observed LLM phenomena:
> - We elucidate why gradient methods yield task vector arithmetic in LLM[10], why QA boosts retrieval[2], and why transformers outshine word2vec—grounded in an empirically-motivated theoretical modeling akin to [11-12]’s analyses on feature-noise vision data (justified by ResNet’s latent space [12]), common due to the intractability of analyzing multi-layer dynamics.
> - Beyond this, our theory supports recent LLM task vector-arithmetic work (e.g., application of task vector in editing, unlearning, merging [13-14]), which assume vector arithmetic between pretrained and modified models *without explaining its origins in language model*. Though focused on single-token recall, our optimization theory, grounded in concept geometry, provides a foundational step. Section 6 outlines future work on complex mechanisms to further enhance this domain’s theoretical merit.
>
> **Q**:Is there a way to vary and study the complexity of QA and Word-Label tasks? In the natural language settings you draw from for motivation, we have a clearer sense of complexity for different QA pairs and, similarly, a clearer sense of concept hierarchy.
>
> **A**:In our context regarding retrieval over hierarchical concept graph, one way to measure a task’s complexity is the difficulty a model faces in achieving high confidence for its argmax-sampled prediction. A simple metric could be $C(T)=1/\max_{y}p_θ(y|T)$, where $\max_{y}p_θ(y|T)$ is model θ’s top answer confidence, and a higher $C(T)$ denotes greater complexity. QA tasks terms to be simpler: a keyword (e.g., “capital” in “What is the capital of Japan?”) guides θ to the task collaborating with query word, more-likely keeping $C(T)$ low. Word-Label task, in contrast, lacking this cue, requires the θ to infer the task behind the pair—given prompt $T_3$=[Japan,Sakura,China], the model might have non-trivial confidence over both Panda and Peony (e.g. the answers of $T_1$ and $T_2$ defined before). This stems from the polysemy-induced challenge due to the hierarchical concept knowledge encoded in the tokens.
>
> Should the reviewer wish to discuss any point further, we would be more than delighted to continue our productive exchange! Once again, we deeply appreciate the reviewer’s time and valuable comments!
>
> *Reference (arXiv identifier)*
>
> [1]2412.06538
> [2]2309.14316
> [3]1601.03764
> [4]2105.15134
> [5]2402.15607
> [6]2402.01258
> [7]2406.01506
> [8]2403.03867
> [9]2410.09605
> [10]2305.16130
> [11]2012.09816
> [12]2202.05928
> [13]2212.04089
> [14]forum?id=vRvVVb0NAz

---

> > ### Comment · Reviewer_mnid · 2025-04-09
> >
> > Thank you for your responses. I have increased my score to a 4

---

> > > ### Author Response · Authors · 2025-04-09
> > >
> > > Dear Reviewer mnid,
> > >
> > > Thank you for raising your score to 4. We're glad that your concerns have been adressed. We deeply value your thoughtful engagement and keen insight into our approach, particularly your endorsement of its empirical relevance and soundness.
> > >
> > > We’ll ensure your suggestions strengthen our broader impact in the camera-ready version. Thank you again for your valuable time and consideration.
> > >
> > > Best regards,
> > >
> > > Authors of Submission 15363

---

### Official Review · Reviewer_R7sM · 2025-03-14

**Overall Recommendation:** 4

**Summary:**

To study retrieval of task vectors in ICL, the authors perform a careful gradient descent analysis on residual self-attention modules (with nonlinearities and normalization) under a synthetic (but empirically-motivated) data distribution. They find that when pre-training on QA distribution (and testing on word-pair ICL distribution) the model achieves near zero test error w.h.p.; however, the model fails to accurately retrieve the task vector when pre-trained on the pair ICL distribution or a hybrid distribution. The transformer learns to extract the task vector from the context, then adds it to the final token embedding at the residual step, thus completing the ICL task in word2vec fashion. Accurate task vector retrieval explains the model's ability to perform ICL on low-level concepts unseen during training. The results are strongly dependent on the distribution of the prompt embeddings, which is assumed to have some particular structure (motivated by empirical observations) but the theory does not explain the process by which the prompt embeddings obtain this structure.

**Claims And Evidence:**

The claims are clear and the theoretical evidence appears convincing (I did not carefully check the proofs, though). The experiments could definitely be more convincing (see Experimental Designs section).

I have some concerns about chicken-and-egg reasoning. In particular, the theoretical setup here assumes that the latent features that are inputs to the transformer layer *already have* the prerequisite structure which makes word2vec-style ICL possible. This structure then enables the model to solve ICL by simply extracting the task vector. However, it seems plausible to me that, instead, the model first learns orthogonal task and concept directions in the weight matrices, potentially by some other mechanism; these task and concept singular vectors might then be carried to previous layer weights via backpropagation, which pushes the latent representations towards having the observed structure. In other words, it's possible that W_Q, W_K, and W_V obtain the structure depicted in Fig 2 through a *different* mechanism, which *later* causes the embeddings to have the hierarchical structure observed by Park et al. If this were the case, it seems that it would invalidate the gradient descent analysis performed in this work. This possibility is not currently ruled out by the proposed theory. (Most likely, what is actually happening is something in between -- the embeddings and the attention weights gain structure in tandem, each reinforcing the other, throughout training. But I understand that such an analysis would be very difficult to do.)

**Essential References Not Discussed:**

I don't know this area well enough to comment on this.

**Experimental Designs Or Analyses:**

Are the prompt embeddings synthetic, or are they taken from the learned embeddings from open-source models? If it is the former, then it may be useful to perform an experiment with real data, to show more convincingly that the prompt embedding assumptions are satisfied in practice.

An even more convincing experiment would be to pretrain a small language model from scratch, on a small dataset, and showing that the input embeddings converge to the desired structure quickly, and that the attention weights then learn the structure indicated by the theorems. Such an empirical result would also address the chicken-and-egg issue raised earlier.

**Methods And Evaluation Criteria:**

N/A

**Other Comments Or Suggestions:**

In Matplotlib, it's straightforward to use LaTeX formatting in the titles/axes/labels. It would greatly enhance the readability of the plots. Minor point, but some of the y-axes are labelled "projection length" but lengths must be non-negative. Maybe simply "projection" is appropriate?

In eq. 14, I believe there should be an ellipsis between \nu_1 and \nu_{K'}. Also, the meaning of K' is inferrible from eq. 14, but I can't find it stated in the main text.

**Other Strengths And Weaknesses:**

This paper makes strong assumptions on the prompt embedding distribution; in return, the analysis is able to handle the full architectural complexities of transformers, including realistic prompt structure, softmax nonlinearities, layernorm, and cross-entropy loss. This seems to me to be a major strength of the paper (that the authors obtain analytical results in this complicated and highly nonlinear regime).

Probably the greatest weakness is that the proofs are very difficult to follow. I understand that it may not be possible to simplify them. However, if that's the case, I think it will be very beneficial to provide a less rigorous derivation of the main results (for example, eq. 11, ineq. 9) in a simplified setting. This will likely help readers build intuition and make the key components of the result more transparent. I think the intuition behind the proofs should be provided in the main text as well, especially explaining eq. 11 in more detail. Why do the weight matrices learn singular vectors aligned with b in ICL and QA-ICL pretraining, but not in QA pretraining?

**Questions For Authors:**

I'd rate the current submission as a very weak reject -- the proof technique appears impressive, especially in its ability to handle realistic transformer architectures, but there are a few key weaknesses in the approach, as far as I understand. I would be happy to increase my score, pending some clarification or addendums.

My biggest concern is the chicken-and-egg dilemma raised in the Claims And Evidence section. A compelling explanation that rules out the alternative hypothesis would convince me to increase my score. Ideally, I'd love to see an experiment that addresses this, but I understand that it's probably a lot of extra work.

I would also like to better understand the main ideas behind the major results. If it's possible to add an appendix where versions of ineq. 9 and eq. 11 are derived in a simplified setting (a nonrigorous derivation is fine, in my opinion), I would increase my score. I would also appreciate if the authors provide further explanation/intuition of this part of the result in the main text.

The last minor point would be to improve the readability of the plots and provide more description of the plots in the figure captions.

**Relation To Broader Scientific Literature:**

I don't know this area well enough to comment on this.

**Theoretical Claims:**

The proofs are rather cumbersome and outside my wheelhouse, so I did not check them carefully. Sorry!

---

> ### Author Rebuttal · Authors · 2025-03-26
>
> Thank you for your thoughtful review!
>
> **Q**:Chicken-and-Egg Dilemma & real-world impact
>
> We'd like to emphasize that theories often relies on abstract, empirical-motivated models to enable tractable analysis and explore a model’s potential—an approach we adopt. However, we appreciate the opportunity to discuss how our theory connects to real-world multi-layer dynamics.
> 1. **Concept Geometry & Vector Retrieval**: Next-token prediction (NTP) shapes concept geometry in prerequisite layers[1]. [2] show that task vectors emerge in earlier layers during ICL, while arithmetic retrieval occurs in later layers.
> 2. **Theoretical Simplification & Justification**: The analysis of multi-layer dynamics is typically intractable. Akin to [3-4], which model simple feature-noise vision data justified by *ResNet’s deep latent space* properties[3], we simplify by directly modeling the resulting concept geometry from prerequisite layers. This is further supported by *layer convergence bias*—shallower layers typically converge faster[5].
> 3. **Alignment to Real World**: We offer the first optimization theory for a realistic transformer on QA data for vector-retrieval ICL and showing its *OOD edge* over word2vec, potentially addressing Question 5.1.4 in [6]. Yet, like [3-4], our outcome does not fully match real-world multi-layer dynamics. Empirical evidence suggests that shallow and deep layers most-likely *co-evolve*: even in a toy setting for approximating polynomial functions, [7] shows that mechanisms across layers must evolve simultaneously to reinforce each other—layer-by-layer training induces unwanted errors.
>
> By studying this non-trivial learning problem in a **comparatively realistic** setting, we take an important step forward. A key future direction is to extend beyond [7] by analyzing a 2-3 block transformer on random spherical features, integrating NTP and QA training to model the *self-reinforcing* co-evolution of concept geometry and arithmetic retrieval based on empirical observations.
>
> **Q**:Experimental details
>
> Our experiments are conducted on synthetic data defined in Definition C.1-3;our model is defined in Section 2.2(no layers frozen);our hyperparameters are provided in Section 5&B;our Algorithm procedure is in Section C.1.
>
> **Q**:Room for presentation
>
> We highly appreciate the feedback and would add:
> - **Ineq.(9)**: Simplified flows bounding projection updates are listed in Appendix D.1(Lemmas D.1-D.6), abstracting complex updates into sequences with simple constants (e.g., $a,b,c,d$). The idea is that since analyzing original complex gradient update in eq.(27) is too hard due to the nonlinearities of our problem, we instead split training into phases identifying key components that dictate update growth rates, avoiding tackling original dynamics directly. For example, in the first phase, $||W_VS_n\pi||=\Theta(\sigma_1d)$ dominates, while $a_kW_Va_k$ starts small and the grows’ rate is then $a_{t+1}=a_t+b$ (eq.(60)), bounded by Lemma D.1’s continuous flow,yielding ineq.(9). In the next phase, once $a_kW_Va_k$ controls $||W_VS_n\pi||$’s order, the latter, as a gradient denominator, slows $a_kW_Va_k$’s update (Lemma 4.3, simplified in Lemma D.3). Unlike [8], our problem’s pronounced nonlinearities demand more complex treatment. Per your advice, we’d add intuitive links between simplified flows and actual updates in both main text and appendices.
> - **Eq.(11)**: To explain our *harmful overfitting* in eq.(11), we contrast it with [6], where vision-inspired data(features+Gaussian noise) leads to noise memorization. There, high noise-to-feature ratios make the inner product of weights with noise significant in the gradient, causing harmful overfitting[6]. Our case involves memorizing low-level features: in ICL prompts, some demo pair's low-level features w.h.p. co-occur with query word's ones due to imbalanced frequencies in **finite** training sets, which produces small but non-negligible products in projection's updates, driving harmful memorization. Akin to [9]'s result—feature co-occurrence biases gradients toward memorization—in our case ICL pairs introduce unexpected co-occurrence. QA data, lacking low-level features before query words, would not memorize this co-occurrence and focus on task vector. [10]’s "relation" token mirrors task vector, though their artificial model (their eq.(13)) and data lack realism. Empirical backing includes [11], showing QA aids (multi-token) recall.
> - **Plots & K’**: We’ll update y-axes to "projection", use LaTeX formatting, and enhance captions. We’d fix eq.(14)’s ellipsis and explain $K'$ (number of task-irrelevant tokens) around eq.(13) with intuition in the main text.
>
> Thanks again for your feedback! We welcome further discussion and appreciate your time!
>
> *Reference (arXiv identifier)*
>
> [1]2403.03867
> [2]2305.16130
> [3]2012.09816
> [4]2202.05928
> [5]iclr.cc/virtual/2023/poster/11533
> [6]2405.01964
> [7]2001.04413
> [8]2310.01975
> [9]2410.09605
> [10]2412.06538
> [11]2309.14316

---

> > ### Comment · Reviewer_R7sM · 2025-04-03
> >
> > I have increased my score to 4, contingent on the proposed changes being made to the camera-ready version.

---

> > > ### Author Response · Authors · 2025-04-04
> > >
> > > Dear Reviewer R7sM,
> > >
> > > Thank you for raising your score to a 4 — your recognition is truly encouraging. We greatly appreciate your thoughtful engagement throughout the review process. In particular, your clear understanding of our approach and your acknowledgment of our non-trivial theoretical contributions mean a great deal to us. Your insightful suggestions will help improve our broader impact.
> > >
> > > We deeply value your confidence in our work and will ensure the camera-ready version reflects the proposed changes. Thank you again for your time and trust!
> > >
> > > Best regards,
> > >
> > > Author of Submission 15363

---

### Decision · Program_Chairs · 2025-05-01

**Decision:**

Accept (poster)

**Comment:**

This paper presents a theoretical framework explaining how large language models perform in-context learning (ICL) for factual recall through task vector arithmetic, grounded in hierarchical concept modeling and optimization theory. The authors show that nonlinear residual transformers trained via gradient descent can generalize well under data shifts and concept recombinations, offering both provable guarantees and empirical validation.

The motivation of the paper is interesting and the analysis is good with solid theory support. The reviewers have a few concerns about the clarity and accessibility of the theoretical proofs, particularly equations (9) and (11), suggesting that simplified derivations and stronger intuitive explanations in the main text would help, and they also raise issues regarding the empirical validation of theoretical claims, the limited discussion of geometric relationships in trained models, and the potential narrow scope of broader impact. During the rebuttal, the authors made an effort to address these concerns, and the reviewers were generally satisfied with their responses.